# PEPBENCHMARK: A STANDARDIZED BENCHMARK FOR PEPTIDE MACHINE LEARNING

**Jiahui Zhang**[1,2,*]**, Rouyi Wang**[1,3,*]**, Kuangqi Zhou**[1,*]**, Tianshu Xiao**[1,4]**,**

**Lingyan Zhu**[3]**, Yaosen Min**[1,†]**, Yang Wang**[2,†]

[1] Zhongguancun Academy     [2] University of Science and Technology of China
[3] Nankai University     [4] Tianjin University
[*] Equal contribution     [†] Corresponding authors

## ABSTRACT

Peptide therapeutics are widely regarded as the "third generation" of drugs, yet progress in peptide Machine Learning (ML) are hindered by the absence of standardized benchmarks. Here we present **PepBenchmark**, which unifies datasets, preprocessing, and evaluation protocols for peptide drug discovery. PepBenchmark comprises three components: (1) **PepBenchData**, a well-curated collection comprising 29 canonical-peptide and 6 non-canonical-peptide datasets across 7 groups, systematically covering key aspects of peptide drug development—representing, to the best of our knowledge, the most comprehensive AI-ready dataset resource to date; (2) **PepBenchPipeline**, a standardized preprocessing pipeline that ensures consistent dataset cleaning, construction, splitting, and feature transformation, mitigating quality issues common in ad hoc pipelines; and (3) **PepBenchLeaderboard**, a unified evaluation protocol and leaderboard with strong baselines across 4 major methodological families: Fingerprint-based, GNN-based, PLM-based, and SMILES-based models. Together, PepBenchmark provides the first standardized and comparable foundation for peptide drug discovery, facilitating methodological advances and translation into real-world applications. The data and code are publicly available at
`https://github.com/ZGCI-AI4S-Pep/PepBenchmark/`.

## 1 INTRODUCTION

Peptides—short chains of amino acids—are rapidly emerging as therapeutics, following small molecules and monoclonal antibodies as the "third generation" of drugs (Zheng et al., 2025). They offer advantages such as synthetic accessibility, high biological specificity, and favorable safety profiles. With the growing availability of peptide data, Machine Learning (ML) has recently begun to transform peptide drug discovery, achieving early successes in anticancer (Arif et al., 2024) and antimicrobial (Wan et al., 2024; Du et al., 2023).

Despite recent successes, the development of peptide ML faces three major challenges. (1) Data sources are scattered across different databases and publications (Ma et al., 2025; Li et al., 2023; Aguilera-Mendoza et al., 2023; Minkiewicz et al., 2008; Singh et al., 2016; Cabas-Mora et al., 2024), forcing researchers to curate their own datasets, which is a labor-intensive process that often introduces inconsistency across studies. The problem is compounded by the heterogeneous representations of non-canonical peptides from different sources, which further limit progress in non-canonical peptide modeling (Singh et al., 2025; Li et al., 2023). (2) There are currently no standardized preprocessing pipelines for peptide data. As a result, steps such as redundancy removal, negative sampling, and dataset splitting vary considerably even when applied to the same raw data. Inappropriate preprocessing strategies may result in overly simplified datasets, which can artificially inflate model performance and thus fail to accurately reflect the true predictive capability of the models (Wang et al., 2025b; Wan et al., 2024). (3) Evaluation metrics also differ substantially across studies. Collectively, these issues make it difficult to fairly compare different ML methods and to discern genuine methodological advances in this field.

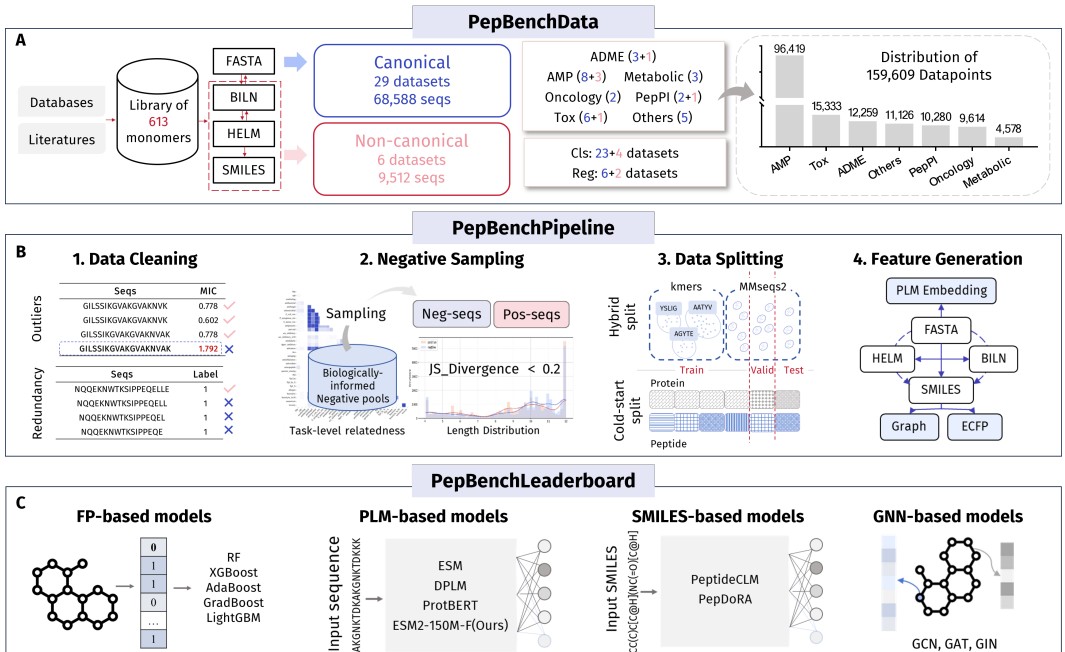

Figure 1: Overview of PepBenchmark, illustrating its three core components: PepBenchData, PepBenchPipeline, and PepBenchLeaderboard.

To address these challenges, we introduce **PepBenchmark**, the most comprehensive peptide ML benchmark, offering a standardized and reproducible framework for unified datasets and model evaluation (Figure 1). It comprises three core components:

- **PepBenchData.** We comprehensively collect and process 35 datasets, providing the *largest* AI-ready peptide database, which includes 29 canonical datasets (68,588 sequences) and 6 non-canonical datasets (9,512 sequences) across 7 groups related to the pharmacological properties of peptides. For non-canonical peptides, we collected 613 amino acid monomers and developed a tool that enables mutual conversion among commonly used representations, including BILN (Fox et al., 2022), HELM (Zhang et al., 2012), and SMILES, thereby providing a unified representation framework for non-canonical peptide modeling.

- **PepBenchPipeline.** We propose a novel preprocessing pipeline, named *PepBenchPipeline*, which standardizes the data curation process through four essential steps: data cleaning, negative sampling, dataset splitting, and feature generation. In **data cleaning**, we remove redundancy and filter outliers to ensure both data quality and representativeness. For **negative sampling**, we propose a **B**iologically-informed and **D**istribution-controlled strategy (**BDNegSamp**), which ensures that positive and negative samples are indistinguishable by shallow features while reducing the risk of pseudo-negatives. During **dataset splitting**, we address the overlooked issue of *kmer leakage*, where frequent kmers disproportionately appear in positive samples and act as shortcut features, potentially inflating model performance. Finally, in **feature generation**, we provide a versatile toolkit for constructing diverse feature sets, enabling compatibility with different modeling paradigms and facilitating fair and comprehensive benchmarking.

- **PepBenchLeaderboard.** Building on PepBenchData and PepBenchPipeline, we perform a large-scale comparison of four model families—fingerprint-based models, Graph Neural Networks (GNNs) (Wu et al., 2020), Protein Language Models (PLMs) (Elnaggar et al., 2020), and SMILES-based models—under a unified set of evaluation metrics. The comprehensive analysis not only benchmarks the current state of peptide ML models but also highlights directions for future development. Furthermore, by fine-tuning ESM2-150M on peptide-specific pretraining data, we develop **ESM-150M-F**, which achieves state-of-the-art performance.

Finally, to facilitate the use of our benchmark and future extensions, we release a Python package, also called `PepBenchmark`, which encapsulates all the three components. With just a few lines of code, users can load preprocessed datasets, perform standardized splitting, train models, and conduct evaluations (see Appendix J).

Our paper is organized as follows. Section 2 reviews related work on peptide machine learning. Sections 3, 4, and 5 present the three components of our benchmark—**PepBenchData**, **PepBench-Pipeline**, and **PepBenchLeaderboard**, respectively. Finally, Section 6 concludes the paper.

## 2    RELATED WORK

**ML in Peptide Prediction.**    ML models has been widely applied to predict therapeutically relevant properties of peptides, especially for canonical peptides. For example, amPEPpy (Lawrence et al., 2021) trains a Random Forest (RF) classifier on physicochemical descriptors to identify antimicrobial peptides, while AMPredictor (Dong et al., 2025) uses a graph convolutional network (GCN), incorporating both ESM embeddings and Morgan fingerprints as node features, for the prediction of minimum inhibitory concentration (MIC) of antimicrobial peptides. HemoPI2 (Rathore et al., 2025) combines PLM embeddings and traditional ML to predict the hemolytic activity of peptides. Large language models (LLMs) are also widely applied to representation learning of canonical peptides. For example, AMPDesigner (Wang et al., 2025a) trains a GPT model on peptide sequences, whose embeddings can be used to achieve good MIC prediction performance. PepBERT (Du & Li, 2025), built on the BERT architecture, outperforms larger PLMs on a variety of downstream tasks. PeptideBERT (Guntuboina et al., 2023) fine-tunes ProtBert (Elnaggar et al., 2020) on peptide sequences and achieves state-of-the-art (SOTA) performance in predicting hemolysis and nonfouling properties. While these advances highlight the effectiveness of ML approaches for canonical peptides, modeling non-canonical peptides remains far less explored, largely due to the lack of unified and well-curated datasets. Current efforts mainly focus on transmembrane activity. For example, PeptideCLM (Feller & Wilke, 2025), a BERT-style transformer pretrained on large-scale synthetic non-canonical peptides, accurately predicts membrane diffusion ability of peptides. The Pepland (Zhang et al., 2025) pretraining framework uses a multi-view heterogeneous GNNs to generate representations of both canonical and non-canonical peptides, enabling prediction of transmembrane activity, Peptide-Protein Interaction (PepPI) and solubility. PepDoRA (Wang et al., 2024a) fine-tunes ChemBERTa to achieve accurate prediction of diverse therapeutic properties and PepPI.

**AI for Life Science Benchmarks**    Benchmarks play a crucial role in advancing machine learning by enabling standardized evaluation and comparison of models. In the field of AI for life science, numerous benchmarks have been established, with a focu s on protein and small molecules (Wu et al., 2018; Rao et al., 2019). For example, ProteinGym (Notin et al., 2023) provides a large-scale, comprehensive benchmark for protein fitness prediction and design. For small molecules, TDC (Huang et al., 2022) offers a unified platform to access and evaluate machine learning across the entire drug development pipeline. In contrast, benchmarking efforts for peptide data remain scarce. To the best of our knowledge, only three attempts have been made to establish peptide benchmarks, none of which have seen wide adoption due to inherent limitations. The earliest effort, UniDL4BioPep (Du et al., 2023), covers 20 different peptide bioactivity data collected from a plethora of prior works, but is still limited in data scale and lacks standardized data processing pipelines. Peptipedia (Cabas-Mora et al., 2024), collects data from over 70 databases covering 11 major biological activities, forming the largest collection of peptide activity data to our knowedge. However, the datasets in it are mostly not ready for ML models due to lack of preprocessing. More relevant to our work is AutoPeptideML (Fernández-Díaz et al., 2024), which offers automated data-processing tools and a platform for building ML models for peptide bioactivity prediction. In comparison, we provide a larger-scale benchmark covering a broader set of ML tasks, and our data preprocessing pipelines enable more meaningful comparisons among models (as discussed in Section H).

## 3    PEPBENCHDATA

PepbenchData represents the most comprehensive and systematic effort to date to organize machine learning (ML) tasks for peptide prediction. In total, 35 datasets have been collected and standardized (see Table 1). Each dataset has undergone unified cleaning and preprocessing, and is split into

training, validation, and test sets according to a fixed ratio (8:1:1) to ensure comparability of model evaluation. The number of data points ranges from 277 to 52,941. Based on their roles in peptide development, the datasets are organized into seven groups across three stages of the pipeline.

- *Activity Modeling:* This includes 5 groups related to peptide therapeutic activities, which can be further grouped into 2 types. The first comprises target-independent activities, including four groups: 11 antimicrobial peptide (AMP) datasets, two oncology-related datasets, four metabolic-disease-related datasets, and five datasets covering other bioactivities. The second category involves target-specific activities, represented by three peptide–protein interaction (PepPI) datasets that support the design of peptides targeting specific proteins.

- *Pharmacokinetics Profiling:* This type includes a single group comprising 4 datasets that describes the pharmacokinetics-related properties of peptide drugs. In our paper, we follow the convention of the drug discovery field and call this type "ADME", which stands for Absorption, Distribution, Metabolism and Excretion.

- *Safety Assessment:* This also consists of a single group, Tox, which includes 6 datasets capturing peptide toxicity-related properties for drug safety evaluation.

For varying requirements of various applications, we categorize the datasets from three orthogonal perspectives, (1) **Input Type**: Single-input (32 datasets) for peptide-only dataset, and Multi-input (3 datasets) for peptide–protein interaction dataset; (2) **Peptide Type**: canonical (29 datasets), consisting of peptides composed solely of the 20 canonical amino acids, and non-canonical (6 datasets), consisting of peptides incorporate cyclic structures or non-canonical amino acid monomers; (3) **Task type**: classification tasks (27 datasets) or regression tasks (8 datasets).

In PepbenchData, canonical peptide datasets are mainly collected from benchmark datasets curated in prior literatures for specific tasks, supplemented with additional entries from the Peptipedia database (Cabas-Mora et al., 2024). Non-canonical peptide datasets are primarily obtained from CycPeptMPDB (Li et al., 2023) and Hemolytic2 (Singh et al., 2025), where raw non-canonical peptide sequences are originally encoded in HELM and MAP formats, respectively. The two formats employ distinct monomer representations and cannot be directly converted into one another. To unify these sources, we merge their monomer libraries and obtain a consolidated library of 613 unique monomers. All non-canonical peptide sequences are then converted into the SMILES format, which facilitates featurization.

Following Peptipedia, only sequences of length $\leq 150$ are retained. We note that only 5 datasets contain $\geq 10\%$ of sequences longer than 50. Therefore, given that a maximum length of 50 residues is commonly adopted for peptide drug candidates(Wang et al., 2025c;b), we provide two versions: **PepbenchData-50** and **PepbenchData-150**.

## 4  PEPBENCHPIPELINE

A standardized data preprocessing pipeline is essential for constructing reliable benchmark datasets. Here, we outline the 4 key components and the main design choices of each step, while detailed implementation can be found in Appendix D, Appendix E, and Appendix F.

**Data Cleaning**  The data cleaning stage consists of 2 main procedures: *outlier removal* for regression datasets and *redundancy removal* for classification datasets.

(1) Outlier Removal: Many peptide datasets are derived from heterogeneous literature sources, where identical sequences often yield multiple experimental results due to variations in experiment conditions. Existing studies typically average all available measurements, while outlier detection is often overlooked (Huang et al., 2023), and thus the resulted label may be contaminated by outliers. To address this issue, we employ an InterQuartile Range (IQR)-based method to achieve robust outlier removal in regression datasets.

(2) Near-duplicate Redundancy Removal: In peptide design, new candidates are often generated by introducing minor mutations into a known active peptide, which leads to substantial redundancy as many positive samples are nearly identical. If not properly addressed, model may simply memorize the frequent patterns rather than learn a meaningful mapping. To mitigate this issue, `MMseqs2` is employed to remove sequences with more than 90% similarity.

Table 1: Overview of datasets in **PepBenchData**. Here, "bc" denotes binary classification and "reg" denotes regression; "ca" denotes canonical and "n-ca" denotes non-canonical. Detailed data sources are provided in Table 6.

| Application | Dataset Name | Task Type | Peptide Type | Input Type | Unit | Size | Split |
|---|---|---|---|---|---|---|---|
| ADME | nonfouling | bc | ca | single | | 7200 | Hybrid |
| | cpp | bc | ca | single | | 2296 | ECFP |
| | bbp | bc | ca | single | | 665 | Hybrid |
| | nc-cpp_pampa | reg | n-ca | single | log(cm/s) | 6970 | Hybrid |
| AMP | antimicrobial | bc | ca | single | | 52941 | Hybrid |
| | antibacterial | bc | ca | single | | 28591 | Hybrid |
| | antifungal | bc | ca | single | | 12887 | Hybrid |
| | antiparasitic | bc | ca | single | | 6755 | Hybrid |
| | antiviral | bc | ca | single | | 7785 | Hybrid |
| | nc-antimicrobial | bc | n-ca | single | | 2495 | ECFP |
| | nc-antibacterial | bc | n-ca | single | | 1668 | ECFP |
| | nc-antifungal | bc | n-ca | single | | 407 | ECFP |
| | E.coli_mic | reg | ca | single | $\lg(\mu M)$ | 3204 | Hybrid |
| | S.aureus_mic | reg | ca | single | $\lg(\mu M)$ | 2822 | Hybrid |
| | P.aeruginosa_mic | reg | ca | single | $\lg(\mu M)$ | 1490 | Hybrid |
| Metabolic | ace_inhibitory | bc | ca | single | | 3537 | Hybrid |
| | antidiabetic | bc | ca | single | | 3028 | Hybrid |
| | dppiv_inhibitors | bc | ca | single | | 1268 | Hybrid |
| | ace_inhibitory_ic50 | reg | ca | single | $\lg(\mu M)$ | 337 | Random |
| Oncology | anticancer | bc | ca | single | | 12013 | Hybrid |
| | ttca | bc | ca | single | | 1182 | Hybrid |
| Others | neuropeptide | bc | ca | single | | 8687 | Hybrid |
| | antiinflamatory | bc | ca | single | | 7665 | Hybrid |
| | antioxidant | bc | ca | single | | 390 | Hybrid |
| | antiaging | bc | ca | single | | 548 | Hybrid |
| | quorum_sensing | bc | ca | single | | 490 | Hybrid |
| PepPI | PpI | bc | ca | multi | | 44148 | cold-start |
| | PpI_ba | reg | ca | multi | -lg(M) | 1433 | cold-start |
| | nc-PpI_ba | reg | n-ca | multi | -lg(M) | 277 | cold-start |
| Tox | hemolytic | bc | ca | single | | 4306 | Hybrid |
| | toxicity | bc | ca | single | | 4056 | Hybrid |
| | neurotoxin | bc | ca | single | | 3159 | Hybrid |
| | allergen | bc | ca | single | | 2538 | Hybrid |
| | nc-hemolytic | bc | n-ca | single | | 3971 | ECFP |
| | hemolytic_hc50 | reg | ca | single | $\lg(\mu M)$ | 1926 | Hybrid |

**Negative Sampling (BDNegSamp)** In peptide-related studies, datasets typically contain only positive samples, since negative samples are extremely scarce. Consequently, negative samples must be manually constructed, and devising a principled sampling strategy remains a major challenge. Prior studies have mainly adopted two strategies: (1) randomly sampling sequences (Agrawal et al., 2021; Pinacho-Castellanos et al., 2021; Bin et al., 2020), or (2) using peptides from another activity dataset as negatives (Agrawal et al., 2021; Fernández-Díaz et al., 2025). The first strategy risks models exploiting superficial differences between random sequences and active peptides, while the second may lead models to simply learn activity-specific differences between the target property and the property used for negative sampling. Moreover, both of them might introduce significant distributional shifts (e.g., in sequence length or charge), thereby simplifying the learning task and potentially leading to overly optimistic performance estimates. To address this issue, AutoPeptideML (Fernández-Díaz et al., 2024) proposes sampling negatives from a negative sampling pool that consists of peptides with diverse bioactivities, while keeping the sequence length distribution close to that of the positive samples. However, this method does not consider task-to-task correlations. For instance, antimicrobial peptides often exhibit anticancer activity; thus, including them in the negative sampling pool for anticancer peptide prediction may introduce a substantial risk of false negatives. Based on above reasoning, we argue that a high-quality negative set must meet the two criteria below:

1. There should not be dataset-specific artifacts (e.g., clear distributional differences, inherent contrasts between active and inactive peptides) between positive and negative data
2. False negatives should be avoided as much as possible.

This motivates us to propose the **B**iologically-informed and **D**istribution-controlled **N**egative Sampling (BDNegSamp), which includes 4 key steps:

1. **Sampling pool construction.** Build a sampling pool by combining all the bioactivity data we collect.

2. **Task correlation filtering.** Use expert prior knowledge and statistical analysis to estimate correlation among tasks, and excluding data from tasks closely related to the target task.

3. **Similarity filtering and deduplication.** Remove sequences in the sampling pool that are highly similar to positive samples, and eliminate duplicate sequences.

4. **Distribution-controlled sampling.** Perform sampling from the filtered sampling pool while ensuring that multiple sequence-level features (such as length, net charge, and hydrophobicity) of the sampled negative data closely matches those of the positive samples.

The above 4 steps work well for canonical petides, but would fail for non-canonical peptide data, for which there are insufficient sequences to form a negative sampling pool. To address this limitation, we train a generative model that takes a canonical peptide as input and generates a chemically modified non-canonical peptide. This strategy effectively transforms the negative sampling problem for non-canonical peptides into one defined in the canonical peptide space (see Appendix I).

**Data Splitting**  Data splitting strategy critically affects the credibility of model evaluation. Conventional random splitting fail to capture the distribution shifts encountered in real-world drug discovery. Inspired by practices in protein modeling, recent studies (Fernández-Díaz et al., 2024; Pinacho-Castellanos et al., 2021) attempt to lower similarity between training and test sets by similarity-based splitting using `MMseqs2` (Steinegger & Söding, 2017) or CD-HIT. However, our analysis reveals that relying solely on sequence similarity is insufficient — we identify an important but previously overlooked issue: **kmer leakage**.

In particular, in most peptide datasets, some kmers show significant distributional difference betwen positive and negative samples (which we refer to as *representative kmers*). Such kmers may correspond to activity-related motifs that experts intentionally introduce during peptide design. This would allow models to simply memorize these local shortcuts rather than learning biologically meaningful patterns, resulting in poor generalization to new active motifs. Importantly, `MMseqs2` cannot resolve this issue, since two sequences sharing the same kmer may exhibit low overall sequence similarity, and thus would not be clustered together. To address this issue, we introduce two splitting strategies. The first is **kmer-split**, which contains three steps: (i) apply Fisher's exact test to identify *representative kmers*.; (ii) group peptides sharing the same *representative kmer* into the same cluster; and (iii) assign clusters to the training, validation, and test sets.

The second strategy is **hybrid-split**, which combines kmer-split with `MMseqs2`-based similarity clustering. This is achieved by first applying kmer-split and then using similarity-based cluster to further split sequences that does not contain *representative kmers*. Hybrid-split simultaneously prevents **kmer leakage** and enforces similrity-aware grouping, yielding a more rigorous evaluation protocol. We recommend hybrid-split as the default splitting strategy.

So far, we have discussed dataset splitting for single-input canonical peptide datasets. We next consider the `PepPI` task involving multi-input data, as well as datasets of non-canonical peptides. For the `PepPI` task, we adopt a **cold-start split** following the paradigm used in previous studies, where proteins serve as the basis for data splitting: proteins are clustered based on similarity using `MMseqs2`, and the cold-start setting is enforced, meaning that proteins appearing in the training set are excluded from the test set. For non-canonical peptides, where FASTA sequences are not available, we adopt a splitting strategy based on ECFP fingerprints. Specifically, we compute pairwise similarities between peptides using molecular fingerprints, construct a similarity graph where edges connect pairs exceeding a similarity threshold, and then extract connected components as clusters. Each component is subsequently assigned to training, validation, or test sets.

**Feature Generation**  Feature generation plays a crucial role in machine learning. To facilitate related research, we provide a variety of feature generation and transformation methods, including ECFP-based molecular fingerprint construction, pretrained language model (PLM) embeddings, as well as conversion tools among BILN, HELM, and SMILES representations.

## 5 PEPBENCHLEADERBOARD

We introduce **PepBenchLeaderboard**, a standardized benchmark designed for the systematic evaluation of peptide property prediction models. The main body of our experiments is conducted on PepBenchData-50, while results on PepBenchData-150, which includes longer peptide sequences, are presented in Appendix H. To facilitate the organization of experiments, we categorize all tasks based on two dimensions: input type (*single peptide* or *peptide–protein*) and peptide type (*canonical* or *non-canonical*). Accordingly, the experiments are grouped into three major parts: **S**ingle-Input **C**anonical **P**eptide **P**rediction (**SCPP**), **S**ingle-Input **N**on-**C**anonical **P**eptide **P**rediction (**SNCPP**), and **Pep**tide–**P**rotein **I**nteraction (**PepPI**). In the following sections, we first describe the experimental setup and then present the results and analyses for each of these three parts.

### 5.1 EXPERIMENTAL SETUP

**Datasets Description.** SCPP includes 22 canonical peptide classification datasets and 4 canonical regression datasets. SNCPP includes 4 non-canonical peptide classification datasets and 1 non-canonical regression dataset. PepPI consists of three datasets: non-canonical regression (nc-PPI_ba), canonical regression (PPI_ba), and canonical classification (PPI). Detailed dataset descriptions are provided in Table 1.

**Dataset Split.** For each task, data are divided into training, validation, and test sets in an 8:1:1 ratio. For **SCPP**, to ensure a rigorous evaluation of generalization, we employ the *hybrid-split* strategy. Sequences containing enriched $k$-mers are first assigned to the training set, and the remaining sequences are then clustered using MMseqs2 with a 30% sequence identity threshold. For **SNCPP**, we use an ECFP-split strategy, where peptides are clustered based on fingerprint similarity (threshold set to 0.95). For **PepPI**, we adopt a strict protein-based cold-start split, ensuring that no protein appears across training, validation, and test sets. Additionally, to enforce sequence-level diversity and support robust generalization assessment, protein clusters (30% identity) are assigned entirely to one partition using MMseqs2.

**Benchmarking Model Families.** For **SCPP**, we benchmark four families of models, each defined by a distinct peptide representation. **(1) Fingerprint-based (FP-based):** Traditional machine learning models—including Random Forest, AdaBoost, Gradient Boosting, XGBoost, and LightGBM– are trained on 1024-bit ECFP6 molecular fingerprints. **(2) GNN-based:** Graph Neural Networks (GNNs) represent peptides as atom-level molecular graphs. We evaluate classical architectures such as GCN (Kipf & Welling, 2017), GAT (Veličković et al., 2018), and GIN (Xu et al., 2019), as well as *Pepland* (Zhang et al., 2025), a GNN framework pretrained on large-scale peptide data. **(3) SMILES-based:** This family encodes peptides as chemical language sequences in SMILES format. ChemBERTa-77M-MLM extends BERT with chemistry-specific adaptations to better capture the structural features of SMILES strings. PepDoRA fine-tunes ChemBERTa-77M-MLM with a masked language modeling objective, producing optimized embeddings for downstream property prediction involving both modified and unmodified peptides. PeptideCLM is a peptide-oriented chemical language model capable of encoding peptides with chemical modifications and cyclizations. **(4) PLM-based:** Protein Language Models (PLMs) leverage large-scale pretraining to generate sequence-based embeddings. We evaluate models from the ESM2 family (8M–650M parameters), the DPLM family (150M, 650M) (Wang et al., 2024c), and ProtBERT (Elnaggar et al., 2020). To further investigate the effect of pretraining, we include two variants of ESM2: ESM2-8M-Scratch (ESM-8M-S), which is trained from random initialization based on ESM2-8M, and ESM2-150M-Finetune (ESM-150M-F), which is obtained by further pretraining ESM2-150M on 1.9M short peptides (lengths < 50) from UniRef50 (Suzek et al., 2015) (see Appendix G.2). For **SNCPP**, since non-canonical peptides lack FASTA representations, PLM-based approaches are inapplicable. Thus, evaluation is limited to FP-, GNN-, and SMILES-based models. For **PepPI**, peptide and protein embeddings are concatenated and fed into a multilayer perceptron (MLP). To isolate the contribution of the peptide encoder, the protein encoder is fixed as a frozen ESM2-150M model (see Appendix H for results with a trainable protein encoder). The peptide encoders under evaluation cover the same four model families introduced above, with one distinction: for the FP-based family, both ECFP4 and ECFP6 fingerprints are employed as peptide representations.

**Metrics.** We report the mean and standard deviation over five independent dataset splits. ROC-AUC is used for classification tasks, and Mean Absolute Error (MAE) for regression tasks.

## 5.2 RESULTS AND ANALYSIS

We next present experimental analyses for **SCPP**, **SNCPP**, and **PepPI**. Due to space constraints, only representative results are reported in the main text, while the complete results are provided in Appendix H.

### 5.2.1 RESULTS AND ANALYSIS FOR SCPP

Tables 2 and 3 summarize the performance of SCPP on classification and regression tasks, respectively. The results reveal several noteworthy observations.

Table 2: Performance of models on canonical peptide classification (ROC-AUC↑, %) with hybrid-split. Dataset sizes are shown separately; results are mean$_{\pm std}$. Best and second-best scores per row are in **bold** and gray shadow.

| Dataset | Size | FP-based models | | | GNN-based models | | SMILES-based models | | PLM-based models | | | | | | |
|---|---|---|---|---|---|---|---|---|---|---|---|---|---|---|---|
| | | RF | XGBoost | LightGBM | GIN | Pepland | ChemBERTa | PepDoRA | DPLM-150M | ESM2-8M | ESM2-35M | ESM2-150M | ESM2-650M | ESM2-8M-S | ESM2-150M-F |
| ace_inhibitory | 3537 | 82.2±1.3 | 80.4±1.8 | 81.1±1.8 | 78.8±1.4 | 72.4±3.6 | 77.3±2.5 | 70.3±2.0 | 79.7±1.0 | 79.2±1.4 | 80.7±1.9 | 80.2±0.5 | 79.6±1.9 | 79.1±1.6 | 80.2±2.3 |
| allergen | 2538 | 84.0±4.2 | 85.6±3.4 | 86.1±3.6 | 77.5±1.7 | 58.7±3.6 | 81.5±3.8 | 61.8±1.5 | 88.3±1.4 | 88.3±2.6 | 87.2±3.6 | 90.2±1.2 | 87.4±3.8 | 83.2±2.5 | 86.8±1.8 |
| antiaging | 548 | 66.5±1.9 | 63.3±5.6 | 65.3±2.4 | 53.2±11.7 | 64.0±12.2 | 50.6±6.2 | 55.6±3.5 | 61.4±4.6 | 67.2±5.2 | 62.5±3.3 | 63.6±5.8 | 58.1±2.2 | 61.9±3.9 | |
| antibacterial | 28591 | 88.4±0.9 | 88.3±0.8 | 87.6±0.7 | 82.2±1.6 | 62.7±3.4 | 88.9±2.2 | 73.8±2.6 | 92.3±1.2 | 92.0±0.9 | 92.5±0.8 | 93.0±1.2 | 93.0±0.6 | 90.8±0.6 | 93.2±0.5 |
| anticancer | 12013 | 87.7±1.0 | 87.6±0.9 | 87.1±0.6 | 78.9±2.6 | 69.5±4.6 | 87.5±1.2 | 71.4±2.8 | 92.2±0.9 | 92.3±0.9 | 92.3±0.9 | 92.3±0.8 | 92.4±0.6 | 90.7±0.8 | 92.2±0.6 |
| antidiabetic | 3028 | 73.8±3.4 | 71.0±1.6 | 71.5±3.7 | 66.2±1.3 | 62.2±2.6 | 66.3±3.6 | 62.9±2.0 | 71.9±4.5 | 72.9±2.0 | 74.1±4.1 | 70.5±4.1 | 74.6±3.4 | 69.8±2.5 | 72.7±2.5 |
| antifungal | 12887 | 87.3±0.8 | 87.3±0.8 | 87.1±0.8 | 78.5±3.6 | 53.5±3.5 | 87.0±1.1 | 70.3±1.2 | 90.7±1.4 | 90.6±0.6 | 90.7±1.1 | 91.0±1.0 | 90.6±0.8 | 87.8±2.2 | 91.1±0.7 |
| antiinflamatory | 7665 | 74.3±1.4 | 72.4±1.0 | 73.7±1.2 | 71.1±3.0 | 57.7±2.7 | 73.5±2.4 | 62.4±1.8 | 77.4±2.3 | 77.3±1.9 | 78.0±2.0 | 76.8±2.3 | 77.3±1.9 | 74.6±1.7 | 79.9±2.0 |
| antimicrobial | 52941 | 87.6±0.4 | 87.8±0.5 | 87.5±0.6 | 80.5±1.9 | 67.5±1.6 | 89.9±0.8 | 75.4±0.9 | 92.4±0.2 | 92.3±0.3 | 92.4±0.4 | 92.3±0.6 | 92.6±1.1 | 92.0±0.6 | 93.2±0.3 |
| antioxidant | 390 | 68.1±2.7 | 65.9±3.2 | 65.6±2.8 | 59.7±4.3 | 61.3±3.0 | 58.9±3.3 | 53.8±3.7 | 64.8±6.1 | 67.4±4.0 | 66.5±5.7 | 66.3±2.7 | 65.1±5.0 | 61.5±4.3 | 68.0±3.5 |
| antiparasitic | 6755 | 86.8±1.6 | 87.3±1.0 | 87.3±0.9 | 75.6±2.2 | 58.7±2.5 | 85.0±1.3 | 67.2±3.5 | 91.8±0.8 | 90.7±1.4 | 90.8±1.1 | 91.1±0.9 | 91.4±1.0 | 88.0±1.4 | 91.6±1.5 |
| antiviral | 7785 | 84.2±0.9 | 83.6±1.0 | 84.5±1.3 | 74.6±4.9 | 71.0±3.1 | 81.2±2.9 | 59.7±2.2 | 86.3±2.2 | 86.0±1.1 | 85.2±1.6 | 86.2±1.5 | 86.8±0.4 | 84.1±2.8 | 87.1±1.1 |
| bbp | 665 | 69.3±6.4 | 68.2±8.4 | 69.2±7.5 | 61.2±3.4 | 56.1±9.3 | 61.2±7.1 | 54.7±7.2 | 64.6±8.9 | 66.8±9.5 | 61.8±15.7 | 62.4±12.2 | 64.6±9.0 | 71.6±5.7 | 64.6±12.5 |
| cpp | 2296 | 82.1±2.9 | 82.3±4.9 | 82.1±3.4 | 57.7±5.5 | 68.2±7.2 | 72.2±8.2 | 58.9±4.3 | 81.9±2.9 | 80.5±3.2 | 82.2±2.3 | 83.2±3.1 | 80.1±6.0 | 79.6±8.0 | 81.9±3.1 |
| dppiv_inhibitors | 1268 | 82.4±3.0 | 79.8±1.5 | 81.2±3.4 | 71.3±4.8 | 65.6±4.6 | 76.0±3.1 | 63.4±8.3 | 84.4±2.4 | 81.5±2.2 | 80.5±3.8 | 79.9±2.3 | 80.5±1.9 | 73.2±4.0 | 81.8±2.9 |
| hemolytic | 4306 | 82.7±2.7 | 81.8±2.6 | 82.8±2.0 | 70.3±4.2 | 67.8±3.8 | 80.2±2.0 | 58.3±2.9 | 85.2±1.6 | 84.5±2.4 | 85.1±1.9 | 85.3±1.4 | 85.7±1.7 | 82.9±2.9 | 84.3±2.7 |
| neuropeptide | 8687 | 84.1±1.7 | 84.0±1.8 | 84.3±1.7 | 72.5±1.2 | 67.8±3.2 | 78.8±2.6 | 66.8±1.8 | 87.3±3.2 | 85.2±3.6 | 86.7±2.3 | 86.3±2.3 | 88.0±1.1 | 84.0±1.9 | 85.6±1.8 |
| neurotoxin | 3159 | 63.7±4.1 | 60.7±3.0 | 61.9±1.7 | 56.9±2.7 | 53.2±2.0 | 56.1±5.6 | 51.0±5.0 | 71.0±4.1 | 66.5±3.7 | 67.7±4.8 | 69.9±3.4 | 69.4±3.4 | 62.4±3.0 | 73.0±4.7 |
| nonfouling | 7200 | 76.4±1.5 | 75.4±2.1 | 76.4±1.4 | 76.7±1.4 | 75.3±1.3 | 76.2±1.0 | 70.7±2.1 | 77.3±1.3 | 77.6±1.1 | 77.9±1.5 | 77.4±1.0 | 77.0±1.2 | 78.2±1.0 | |
| quorum_sensing | 490 | 85.2±5.1 | 81.3±2.9 | 81.4±3.1 | 60.5±9.6 | 67.9±3.4 | 55.3±5.2 | 50.7±5.7 | 83.7±6.5 | 85.2±8.7 | 82.6±8.1 | 85.4±5.8 | 84.8±6.2 | 74.5±11.1 | 86.6±6.4 |
| toxicity | 4056 | 64.3±1.4 | 63.2±2.1 | 64.1±1.2 | 55.4±4.0 | 53.4±3.9 | 59.5±5.8 | 53.1±2.6 | 72.9±2.0 | 69.6±3.0 | 71.6±2.7 | 72.3±2.6 | 72.8±3.3 | 65.8±1.8 | 78.0±3.0 |
| ttca | 1182 | 80.6±4.5 | 78.1±8.6 | 78.8±7.0 | 70.5±8.1 | 71.4±4.1 | 67.0±4.9 | 54.8±4.7 | 79.2±7.1 | 79.7±6.2 | 79.0±5.7 | 79.2±4.4 | 75.7±7.3 | 66.8±6.9 | 81.7±5.8 |
| avg | - | 79.2 | 78.0 | 78.5 | 69.5 | 63.9 | 73.2 | 62.1 | 80.9 | 80.3 | 80.6 | 80.6 | 80.6 | 77.2 | 81.5 |

Table 3: Performance of models on canonical peptide regression (MAE↓, %) with hybrid-split. Dataset sizes are shown separately; results are mean$_{\pm std}$. Best and second-best scores per row are in **bold** and gray shadow.

| Dataset | Size | FP-based models | | | GNN-based models | | SMILES-based models | | PLM-based models | | | | | | |
|---|---|---|---|---|---|---|---|---|---|---|---|---|---|---|---|
| | | RF | XGBoost | LightGBM | GIN | Pepland | ChemBERTa | PepDoRA | DPLM-150M | ESM2-8M | ESM2-35M | ESM2-150M | ESM2-650M | ESM2-8M-S | ESM2-150M-F |
| E.coli_mic | 3204 | 0.593±0.015 | 0.604±0.019 | 0.584±0.019 | 0.631±0.015 | 0.650±0.062 | 0.603±0.007 | 0.650±0.013 | 0.525±0.006 | 0.539±0.023 | 0.527±0.004 | 0.509±0.010 | 0.488±0.008 | 0.548±0.024 | 0.517±0.023 |
| P.aeruginosa_mic | 1490 | 0.540±0.030 | 0.559±0.025 | 0.550±0.026 | 0.537±0.032 | 0.566±0.037 | 0.520±0.041 | 0.546±0.029 | 0.524±0.036 | 0.506±0.046 | 0.497±0.042 | 0.496±0.046 | 0.471±0.043 | 0.516±0.031 | 0.483±0.024 |
| S.aureus_mic | 2822 | 0.569±0.035 | 0.588±0.022 | 0.572±0.025 | 0.635±0.032 | 0.648±0.016 | 0.627±0.011 | 0.657±0.025 | 0.563±0.025 | 0.545±0.030 | 0.549±0.022 | 0.527±0.022 | 0.522±0.022 | 0.585±0.016 | 0.533±0.025 |
| hemolytic_hc50 | 1926 | 0.517±0.042 | 0.527±0.038 | 0.523±0.035 | 0.528±0.036 | 0.516±0.033 | 0.498±0.040 | 0.535±0.027 | 0.404±0.023 | 0.422±0.024 | 0.400±0.021 | 0.413±0.047 | 0.394±0.017 | 0.463±0.011 | 0.412±0.026 |
| avg | - | 0.555 | 0.569 | 0.557 | 0.583 | 0.595 | 0.562 | 0.597 | 0.504 | 0.503 | 0.493 | 0.486 | 0.469 | 0.528 | 0.486 |

1. **Choice of Representation.** We observe that PLM-based models consistently achieve the best performance, followed by FP-based models, whereas SMILES-based and GNN-based approaches perform the worst. The superior performance of PLM-based models underscores the critical role of large-scale pretraining on protein sequences, which can be effectively transferred to peptide property prediction. FP-based models, particularly RF, also exhibit competitive results. Although ECFP6 was originally designed for small molecules and is not necessarily optimal for peptides, its strong performance here suggests that developing peptide-specific fingerprint representations could be a promising research direction. In contrast, the relatively poor results of GNN- and SMILES-based models indicate that, for canonical peptides composed of the 20 standard amino acids, atom-level representations may introduce unnecessary redundancy.

2. **Effect of Pretraining.** The pretrained `ESM2-8M` (80.3 ROC-AUC) outperforms its randomly initialized counterpart, `ESM2-8M-S` (77.2 ROC-AUC), confirming the benefits of large-scale protein pretraining. Moreover, the established performance hierarchy among protein PLMs (`DPLM > ESM2 > ProtBert`) is consistently maintained on peptide prediction tasks, indicating effective cross-domain knowledge transfer. In contrast, `Pepland` and `PepDoRA`, which are pretrained on peptide data using graph and SMILES representations respectively, perform worse than their non-pretrained baselines (e.g., GIN, ChemBERTa). This suggests that naive, domain-specific pretraining with such representations can lead to negative transfer, the mitigation of which remains an open research question.

3. **Scaling Laws of PLMs.** For classification tasks, larger models (e.g., `ESM2-8M → ESM2-650M`) correspond to a general, albeit not uniform, increase in average performance. For regression tasks, the effect of scaling is more pronounced, suggesting that larger model capacity is particularly advantageous for quantitative prediction.

4. **Necessity of Peptide-Aware Finetuning for PLMs.** PLMs are typically pretrained on large protein databases such as UniRef, in which short sequences ($\leq 50$ AAs) are severely underrepresented ( 2.8%). As a result, finetuning on a peptide-specific corpus becomes essential. Our finetuned model, `ESM2-150M-F`, achieves substantial gains over its parent model `ESM2-150M` on classification tasks, attaining state-of-the-art performance across all models. The improvements on regression tasks, however, are relatively modest. This suggests that peptide-aware finetuning is particularly critical for classification, while regression performance is more constrained by the intrinsic capacity of the parent PLM.

5. **FP-based Methods in Low-Data Regimes.** On classification datasets, RF with ECFP6 fingerprints achieves an average ROC-AUC of 79.2, second only to PLMs. It frequently ranks first or second on smaller datasets such as *ace_inhibitory*, *antiaging*, *antioxidant*, *bbp*, and *dp-piv_inhibitors*. These results demonstrate that fingerprint-based methods remain a strong baseline in low-data regimes and further highlight the need to design peptide-specific descriptors, given that ECFP6 was originally developed for small molecules and is not ideally suited for peptides.

### 5.2.2 RESULTS AND ANALYSIS FOR SNCPP

Table 4: Performance of models on non-canonical peptide classification (ROC-AUC↑, %) and regression (MAE↓, %) with ECFP-split. Dataset sizes are shown separately; results are mean$_{\pm std}$. Best and second-best scores per row are in **bold** and gray shadow.

| Task | Metric | Dataset | Size | FP-based models | | | | | GNN-based models | | | | SMILES-based models | | |
|---|---|---|---|---|---|---|---|---|---|---|---|---|---|---|---|
| | | | | RF | XGBoost | LightGBM | GradBoost | AdaBoost | GCN | GAT | GIN | Pepland | ChemBERTa | PeptideCLM | PepDoRA |
| cls | AUC ROC | nc-antibacterial | 1668 | 94.4$_{\pm1.4}$ | 93.6$_{\pm2.2}$ | 93.8$_{\pm1.8}$ | 92.5$_{\pm2.8}$ | 90.9$_{\pm3.4}$ | 93.0$_{\pm2.6}$ | 81.5$_{\pm3.5}$ | 90.7$_{\pm4.9}$ | 84.9$_{\pm3.9}$ | 91.9$_{\pm2.7}$ | 71.4$_{\pm1.8}$ | 72.4$_{\pm7.4}$ |
| | | nc-antifungal | 407 | 95.4$_{\pm3.2}$ | 96.5$_{\pm2.9}$ | 94.8$_{\pm3.1}$ | 95.2$_{\pm3.8}$ | 95.4$_{\pm2.4}$ | 70.1$_{\pm12.0}$ | 78.5$_{\pm6.2}$ | 86.1$_{\pm2.8}$ | 83.5$_{\pm4.4}$ | 78.0$_{\pm18.3}$ | 65.4$_{\pm7.0}$ | 65.9$_{\pm12.0}$ |
| | | nc-antimicrobial | 2465 | 97.6$_{\pm0.9}$ | 97.7$_{\pm0.9}$ | 97.8$_{\pm0.6}$ | 97.3$_{\pm0.7}$ | 95.3$_{\pm1.8}$ | 94.9$_{\pm1.6}$ | 79.2$_{\pm4.4}$ | 91.5$_{\pm2.3}$ | 88.0$_{\pm0.9}$ | 95.2$_{\pm1.9}$ | 68.3$_{\pm4.7}$ | 80.0$_{\pm1.9}$ |
| | | nc-hemolytic | 3971 | 96.1$_{\pm1.1}$ | 96.2$_{\pm0.5}$ | 96.3$_{\pm0.7}$ | 95.3$_{\pm0.8}$ | 93.7$_{\pm1.6}$ | 89.5$_{\pm3.7}$ | 85.7$_{\pm2.6}$ | 89.0$_{\pm4.1}$ | 82.8$_{\pm2.1}$ | 91.4$_{\pm1.5}$ | 76.1$_{\pm4.5}$ | 72.1$_{\pm7.3}$ |
| | | avg | - | 95.9 | 96.0 | 95.7 | 95.1 | 93.8 | 86.9 | 81.2 | 89.3 | 84.8 | 89.1 | 70.3 | 72.6 |
| reg | MAE | nc-cpp | 6970 | 0.649$_{\pm0.006}$ | 0.705$_{\pm0.026}$ | 0.651$_{\pm0.016}$ | 0.683$_{\pm0.007}$ | 0.829$_{\pm0.019}$ | 0.754$_{\pm0.027}$ | 0.767$_{\pm0.025}$ | 0.701$_{\pm0.027}$ | 0.742$_{\pm0.029}$ | 0.712$_{\pm0.033}$ | 0.822$_{\pm0.035}$ | 0.879$_{\pm0.013}$ |

As reported in Table 4, FP-based models achieve the best performance, while GNN- and SMILES-based models continue to lag behind. Intuitively, the chemical diversity provided by over 600 non-canonical monomers should favor atom-level representations; however, the empirical results do not support this hypothesis. We highlight two key directions for future research. First, finetuning existing PLMs may prove effective, as many non-canonical amino acids share structural backbones with the 20 canonical residues, suggesting that pretrained knowledge could transfer. Second, novel pretraining frameworks are needed for both canonical and non-canonical peptides. For sequence-level modeling, notations such as BILN or HELM present promising options, whereas for atom-level modeling the central challenge lies in capturing the modular structure of peptides. The main obstacle remains the scarcity of large-scale non-canonical peptide datasets, which we are beginning to address through rule-based generation of synthetic databases.

### 5.2.3 RESULTS AND ANALYSIS FOR PEPPI

Table 5: Performance of models on PepPI classification (ROC-AUC↑, %) and regression (MAE↓, %) with cold-start-split. Dataset sizes are shown separately; results are mean$_{\pm std}$. Best and second-best scores per row are in **bold** and gray shadow.

| Task | Metric | Dataset | Size | FP-based models | | GNN-based models | | SMILES-based models | | | PLM-based models | | | | |
|---|---|---|---|---|---|---|---|---|---|---|---|---|---|---|---|
| | | | | ECFP4 | ECFP6 | GIN | Pepland | ChemBERTa | PeptideCLM | PepDoRA | ESM2-8M | ESM2-35M | ESM2-150M | ESM2-650M | ESM2-150M-F |
| cls | AUC ROC | ppi | 44148 | 54.4$_{\pm2.3}$ | 53.7$_{\pm2.3}$ | 61.3$_{\pm7.2}$ | 59.6$_{\pm2.7}$ | 52.0$_{\pm4.4}$ | 51.4$_{\pm3.6}$ | 59.4$_{\pm2.8}$ | 60.2$_{\pm6.3}$ | 57.6$_{\pm7.2}$ | 55.4$_{\pm3.5}$ | 51.9$_{\pm2.4}$ | 56.0$_{\pm3.1}$ |
| reg | MAE | ppi_ba | 1433 | 1.043$_{\pm0.050}$ | 1.043$_{\pm0.060}$ | 1.189$_{\pm0.140}$ | 1.176$_{\pm0.152}$ | 1.034$_{\pm0.013}$ | 1.128$_{\pm0.042}$ | 1.084$_{\pm0.044}$ | 1.061$_{\pm0.043}$ | 1.056$_{\pm0.092}$ | 1.079$_{\pm0.038}$ | 1.051$_{\pm0.072}$ | 1.038$_{\pm0.030}$ |
| | | nc_ppi_ba | 278 | 1.665$_{\pm0.260}$ | 1.647$_{\pm0.286}$ | 1.741$_{\pm0.359}$ | 1.705$_{\pm0.366}$ | 1.613$_{\pm0.189}$ | 1.580$_{\pm0.142}$ | 1.465$_{\pm0.234}$ | - | - | - | - | - |

As shown in Table 5, both GNN- and SMILES-based models perform competitively on PepPI tasks. This outcome, which contrasts with their underperformance in single-peptide property prediction, suggests that atom-level representations are better suited for modeling peptide–protein interactions. PLM-based models achieve the second-best overall performance; however, on the larger *ppi* dataset,

they exhibit a counterintuitive trend—scaling up model size does not improve performance and in fact leads to degradation.

## 5.3 SUMMARY OF CORE EXPERIMENTAL INSIGHTS

Across rigorously partitioned peptide benchmarks, PLMs emerge as the most effective approaches for single-peptide property prediction, while FP-based methods provide strong and data-efficient baselines. In particular, they dominate in non-canonical settings where PLMs are inapplicable. By contrast, SMILES- and GNN-based models generally lag behind. Considering that ECFP6, despite not being peptide-specific, already delivers solid performance, the development of peptide-tailored fingerprints appears highly promising. Pretraining and scale prove critical: pretrained ESM2 variants consistently outperform randomly initialized counterparts, and larger PLMs yield steadier improvements—especially for regression—while peptide-aware continued pretraining primarily benefits classification. In PepPI tasks, atom-level representations (GNN/SMILES) become competitive, suggesting that fine-grained molecular detail is more advantageous for modeling cross-molecular interactions than for single-peptide properties.

## 6 CONCLUSION

**PepBenchmark** establishes a unified, reproducible framework for peptide ML, resolving fragmentation in data curation, preprocessing, and evaluation. Empirically, PLM-based models achieve the strongest overall performance, FP-based methods remain competitive in low-data regimes, whereas current GNN/SMILES approaches lag behind; our *hybrid-split* and *ECFP-split* strategies mitigate leakage and yield more discriminative benchmarks. The accompanying Python package operationalizes standardized datasets, splits, and metrics, enabling fair comparison and rapid iteration across canonical and non-canonical peptide tasks. Together, these resources provide a practical foundation for advancing robust, generalizable peptide therapeutics.

ACKNOWLEDGMENTS

This work is supported by the Zhongguancun Academy, Beijing, China.

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

## A  LIMITATIONS

A key limitation of the current work is the lack of structural datasets and the corresponding evaluation of structure-based prediction and generation tasks. At present, our benchmark focuses primarily on sequence-level data. Incorporating structural information, however, is essential for advancing realistic peptide modeling. The major challenge lies in data availability: existing peptide structure datasets are extremely scarce, with only about 2,000 peptide entries in the Protein Data Bank (PDB), among which fewer than 100 correspond to non-natural peptides. Consequently, structural expansion must rely heavily on computational simulation. Fortunately, compared with proteins, peptides are much smaller molecules, typically consisting of only a few hundred atoms. This size advantage makes high-accuracy simulations feasible, including advanced approaches such as quantum mechanics/molecular mechanics (QM/MM) hybrid methods, enhanced molecular dynamics (MD), or other physics-informed simulations. We are currently developing an efficient and accurate peptide structural simulation pipeline to generate reliable structural data, which will support structure-related tasks within *PepBenchmark*. In future iterations, we plan to extend the benchmark to include the evaluation of peptide structure prediction and generative modeling methods.

## B  DETAILS OF PEPBENCHDATA

*PepBenchData* compiles datasets spanning 7 tasks that correspond to 3 stages of the drug discovery pipeline: activity modeling (i.e. AMP, PepPI, Oncology, Metabolic, Others), pharmacokinetic profiling (i.e. ADME), and safety assessment (i.e. Tox). The detailed descriptions of each dataset are presented in this section. The sources of our datasets are provided in Table 6. Overviews of classification and regression datasets are provided in  7 and  8, respectively.

Table 6: Dataset sources.

| Application | Dataset Name | Data Source |
|---|---|---|
| ADME | nonfouling | Guntuboina et al. (2023) |
| | cpp | Zhang et al. (2025); Peptipedia (Cabas-Mora et al., 2024) |
| | bbp | Dai et al. (2021); Peptipedia (Cabas-Mora et al., 2024) |
| | nc-cpp_pampa | CycPeptMPDB (Li et al., 2023) |
| AMP | antimicrobial | Wang et al. (2025c); Peptipedia (Cabas-Mora et al., 2024) |
| | antibacterial | Pinacho-Castellanos et al. (2021); Peptipedia (Cabas-Mora et al., 2024) |
| | antifungal | Pinacho-Castellanos et al. (2021); Peptipedia (Cabas-Mora et al., 2024) |
| | antiparasitic | Zhang et al. (2022); Peptipedia (Cabas-Mora et al., 2024) |
| | antiviral | Pinacho-Castellanos et al. (2021); Peptipedia (Cabas-Mora et al., 2024) |
| | nc-antimicrobial | Hemolytik 2.0 (Singh et al., 2025) |
| | nc-antibacterial | Hemolytik 2.0 (Singh et al., 2025) |
| | nc-antifungal | Hemolytik 2.0 (Singh et al., 2025) |
| | E.coli_mic | Grampa (Witten & Witten, 2019) |
| | S.aureus_mic | Grampa (Witten & Witten, 2019) |
| | P.aeruginosa_mic | Grampa (Witten & Witten, 2019) |
| Metabolic | ace_inhibitory | Manavalan et al. (2019); Peptipedia (Cabas-Mora et al., 2024) |
| | antidiabetic | Yue et al. (2024); Hikida et al. (2013); Peptipedia (Cabas-Mora et al., 2024) |
| | dppiv_inhibitors | Charoenkwan et al. (2020a); Hikida et al. (2013) |
| | ace_inhibitory_ic50 | Kumar et al. (2015b) |
| Oncology | anticancer | Agrawal et al. (2021); Peptipedia (Cabas-Mora et al., 2024) |
| | ttca | Charoenkwan et al. (2020b) |
| Others | neuropeptide | Peptipedia (Cabas-Mora et al., 2024); Bin et al. (2020) |
| | antiinflamatory | Peptipedia (Cabas-Mora et al., 2024) |
| | antioxidant | Olsen et al. (2020); Peptipedia (Cabas-Mora et al., 2024) |
| | antiaging | Peptipedia (Cabas-Mora et al., 2024) |
| | quorum_sensing | Peptipedia (Cabas-Mora et al., 2024); Rajput et al. (2015) |
| PepPI | PpI | Bhat et al. (2025); Abdin et al. (2022) |
| | PpI_ba | Zhang et al. (2025) |
| | nc-PpI_ba | Zhang et al. (2025) |
| Tox | hemolytic | Hemolytik 2.0 (Singh et al., 2025); Peptipedia (Cabas-Mora et al., 2024) |
| | toxicity | Peptipedia (Cabas-Mora et al., 2024); Wei et al. (2021) |
| | neurotoxin | Peptipedia (Cabas-Mora et al., 2024) |
| | allergen | Peptipedia (Cabas-Mora et al., 2024) |

Table 7: Overview of classification datasets.

| Application | Dataset Name | Task Type | Peptide Type | Origin_pos | Filt_pos | Exp_neg | Filt_neg | Total_Size | min_len | max_len | mean_len | Len > 50% |
|---|---|---|---|---|---|---|---|---|---|---|---|---|
| ADME | bbp | bc | ca | 358 | 336 | 13 | 13 | 672 | 2 | 82 | 14.14 | 1.04 |
| | cpp | bc | ca | 1318 | 1162 | 0 | - | 2324 | 3 | 61 | 17.24 | 1.20 |
| | nonfouling | bc | ca | 3600 | 3600 | 0 | - | 7200 | 5 | 11 | 6.06 | 0.00 |
| | solubility | bc | ca | 5096 | 5096 | 5384 | 5384 | 10480 | 19 | 150 | 112.34 | 97.86 |
| AMP | antibacterial | bc | ca | 21220 | 15838 | 0 | - | 31676 | 2 | 150 | 25.33 | 9.74 |
| | antifungal | bc | ca | 11087 | 8349 | 0 | - | 16698 | 2 | 148 | 34.17 | 22.82 |
| | antimicrobial | bc | ca | 42800 | 30752 | 0 | - | 61504 | 2 | 150 | 28.55 | 13.92 |
| | antiparasitic | bc | ca | 6041 | 4316 | 0 | - | 8632 | 2 | 140 | 35.97 | 21.74 |
| | antiviral | bc | ca | 5210 | 4134 | 0 | - | 8268 | 2 | 138 | 22.75 | 5.84 |
| | nc-antifungal | bc | n-ca | 207 | - | 0 | - | 410 | 4 | 76 | 18.18 | 1.93 |
| | nc-antibacterial | bc | n-ca | 845 | - | 0 | - | 1666 | 4 | 76 | 16.82 | 0.95 |
| | nc-antimicrobial | bc | n-ca | 1269 | - | 0 | - | 2496 | 4 | 130 | 17.79 | 0.24 |
| Metabolic | ace_inhibitory | bc | ca | 1833 | 1780 | 0 | - | 3560 | 4 | 81 | 8.08 | 0.65 |
| | antidiabetic | bc | ca | 1599 | 1514 | 75 | 75 | 3028 | 2 | 46 | 10.41 | 0.00 |
| | dppiv_inhibitors | bc | ca | 650 | 634 | 85 | 85 | 1268 | 2 | 33 | 6.15 | 0.00 |
| Oncology | anticancer | bc | ca | 9022 | 6926 | 0 | - | 13852 | 2 | 145 | 27.83 | 13.28 |
| | ttca | bc | ca | 592 | 591 | 0 | - | 1182 | 8 | 20 | 9.36 | 0.00 |
| Others | antiaging | bc | ca | 282 | 279 | 0 | - | 558 | 2 | 80 | 10.93 | 1.79 |
| | antiinflamatory | bc | ca | 3902 | 3875 | 0 | - | 7750 | 2 | 107 | 16.95 | 1.10 |
| | antioxidant | bc | ca | 1146 | 195 | 195 | 195 | 2242 | 2 | 11 | 4.10 | 0.00 |
| | neuropeptide | bc | ca | 5336 | 4627 | 0 | - | 9254 | 2 | 150 | 19.57 | 6.13 |
| | quorum_sensing | bc | ca | 265 | 245 | 0 | - | 490 | 3 | 48 | 10.83 | 0.00 |
| PepPI | PpI | bc | ca | 7358 | - | 0 | - | 44148 | 5 | 25 | 12.8 | 0 |
| Tox | allergen | bc | ca | 2405 | 1677 | 0 | - | 3354 | 4 | 150 | 39.97 | 24.33 |
| | hemolytic | bc | ca | 3096 | 2256 | 544 | 475 | 4512 | 2 | 144 | 23.69 | 4.57 |
| | nc-hemolytic | bc | n-ca | 2002 | - | 342 | - | 3937 | 4 | 130 | 17.99 | 0.5 |
| | neurotoxin | bc | ca | 2509 | 1753 | 0 | - | 3506 | 7 | 138 | 39.24 | 9.90 |
| | toxicity | bc | ca | 2509 | 2204 | 0 | - | 4408 | 7 | 138 | 38.41 | 7.99 |

Table 8: Overview of regression datasets.

| Application | Dataset Name | Peptide Type | Unit | Total_Size | min_len | max_len | mean_len | Len > 50% |
|---|---|---|---|---|---|---|---|---|
| ADME | nc-cpp_pampa | n-ca | log(cm/s) | 6970 | 2 | 15 | 8.04 | 0 |
| AMP | E.coli_mic | ca | lg($\mu M$) | 3312 | 2 | 140 | 23.24 | 3.23 |
| | P.aeruginosa_mic | ca | lg($\mu M$) | 1531 | 2 | 140 | 22.01 | 2.61 |
| | S.aureus_mic | ca | lg($\mu M$) | 2900 | 2 | 140 | 22.70 | 2.66 |
| Metabolic | ace_inhibitory_ic50 | ca | lg($\mu M$) | 337 | 2 | 3 | 2.61 | 0.00 |
| PepPI | nc-PpI_ba | n-ca | -lg(M) | 277 | 5 | 19 | 8.93 | 0 |
| | PpI_ba | ca | -lg(M) | 1433 | 5 | 50 | 16.14 | 0 |
| Tox | hemolytic_hc50 | ca | lg($\mu M$) | 1926 | 6 | 39 | 18.40 | 0.00 |

## B.1 ADME

> **Definition.** Datasets in this group primarily focus on the pharmacokinetics-related properties of peptide drugs. In our paper, we follow the convention of the drug discovery field and call this type "ADME", which stands for Absorption, Distribution, Metabolism and Excretion.
> **Impact.** ADME properties are closely related to the stability, permeability, and bioavailability of peptide drugs, and therefore determine whether a therapeutically effective concentration of the peptide can reach its target site after administration, which is critical for both efficacy and safety.
> **Pipeline.** Pharmacokinetic Profiling

### B.1.1 NONFOULING

• **Property and Application**: Nonfouling characterizes resistance to nonspecific interactions between peptides and other biomolecules. Nonfouling peptides are less likely to aggregate or trigger off-target effects as a result of nonspecific binding, which is essential for ensuring the specific bioactivity of peptides in complex biological environments.

• **Data Sources**: Nonfouling peptides are sourced from White et al. (2012). The dataset is constructed by manually designing peptides resistance to nonspecific interaction by mimicking the amino acid distributions observed on protein surfaces and the inner surfaces of molecular chaperones. The original dataset contains 3,600 nonfouling peptides.

• **Dataset Statistics**: The dataset contains 7,200 datapoints with sequences ranging from 5 to 11 amino acids (average length 6.06) in length.

**Task: Classification; Split: Hybrid; Evaluation: ROC-AUC**

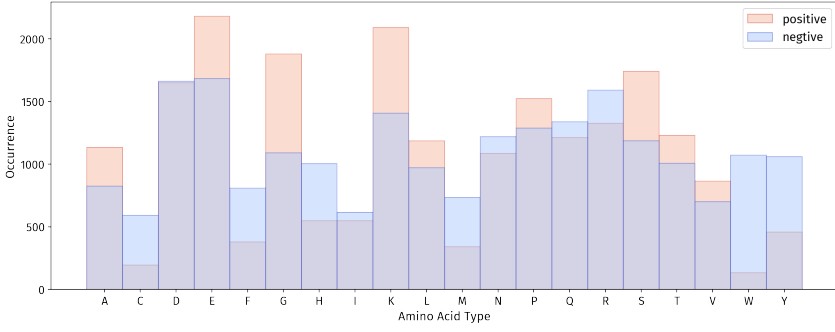

Figure 2: Amino acid distribution comparison between positive and negative samples for nonfouling dataset.

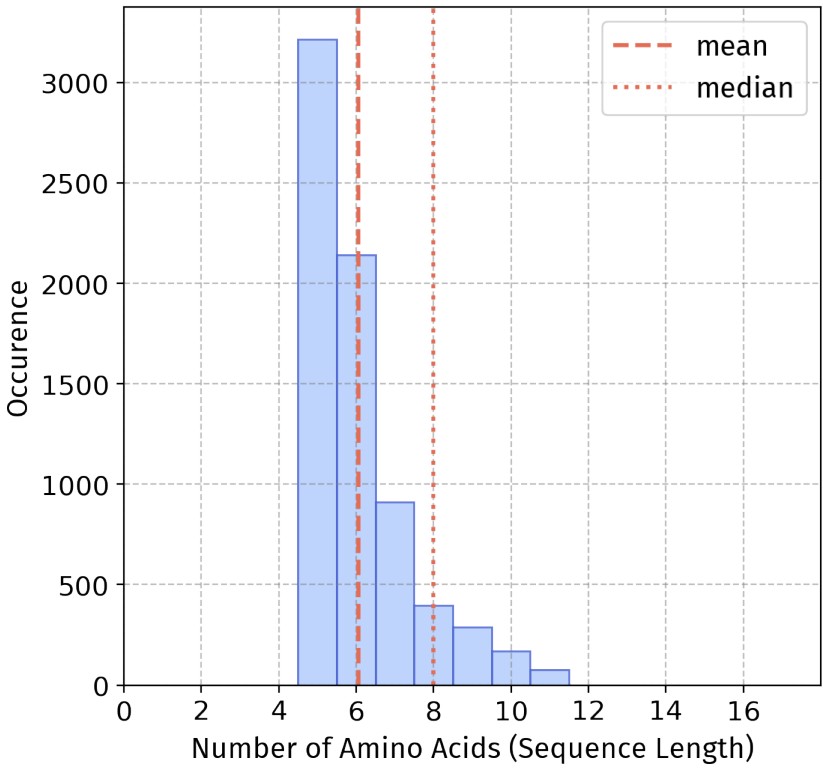

Figure 3: Length distribution of nonfouling dataset.

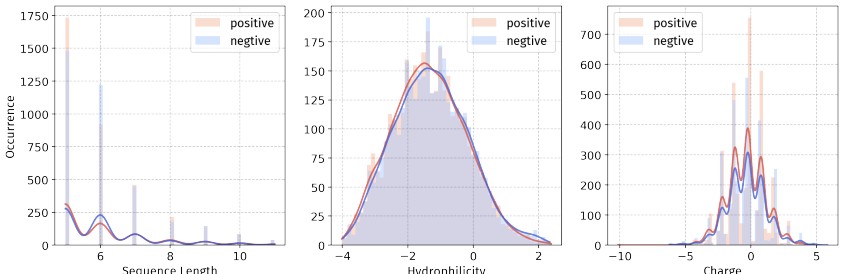

Figure 4: Property comparison between positive and negative samples for nonfouling dataset.

### B.1.2  CELL-PENETRATING PEPTIDES(CPP)

• **Property and Application**:  The cpp dataset characterizes the ability of canonical peptides to penetrate cell membranes, a property crucial for designing efficient drug delivery systems targeting intracellular sites.

• **Data Source**:  The dataset is collected from Pepland (Zhang et al., 2025), which extracts 1,162 transmembrane peptides from cell-penetrating peptide databases CPPsite and CPPsite2.0 (Kardani & Bolhassani, 2021; Gautam et al., 2012), excluding any sequences with non-canonical amino acids. In addition, 156 canonical transmembrane peptides are supplemented from Peptipedia v2.0 (Cabas-Mora et al., 2024). Peptipedia v2.0 is constructed by integrating peptide records from literature, databases, public repositories and major protein databases such as UniProtKB and PDB using key-word searches. For each peptide, detailed information—including biological activities, sequence descriptions, experimental details, and associated publications or patents are extracted from all sources. Peptides are filtered by sequence length (3–150 residues) and annotated for biological activity through semantic analysis. The database comprises data from 76 sources, and it can be con-

sidered to comprehensively cover sequences with specific reported properties across all collected sources.

• **Dataset Statistics**: The dataset contains 2,324 datapoints with sequences ranging from 3 to 61 amino acids (average length 17.24) in length.

**Task: Classification; Split: Hybrid; Evaluation: ROC-AUC**

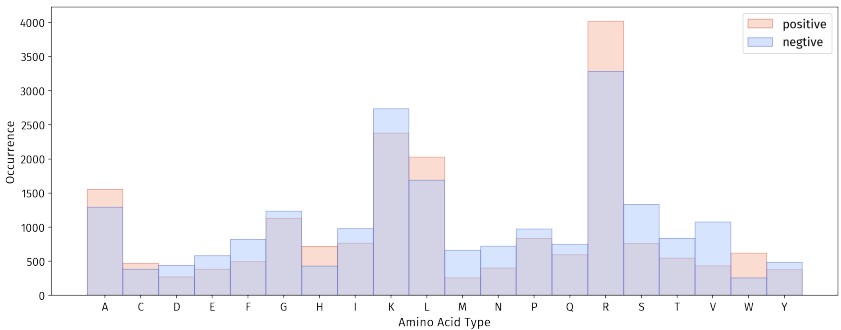

Figure 5: Amino acid distribution comparison between positive and negative samples for cpp dataset.

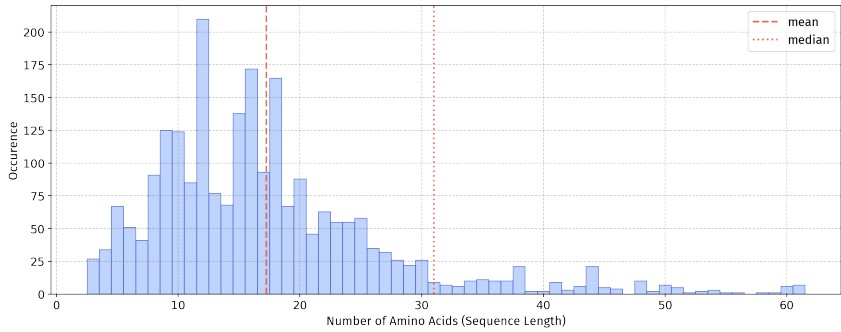

Figure 6: Length distribution of cpp dataset.

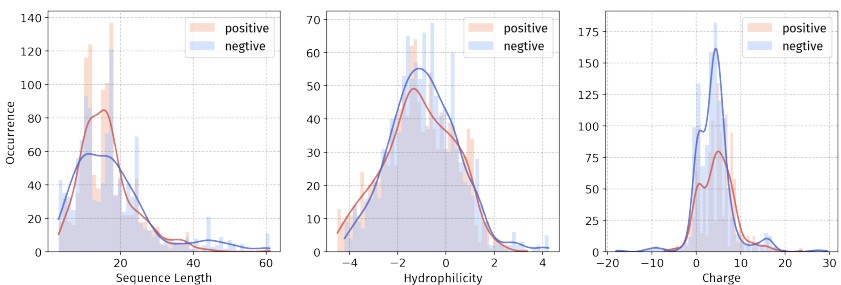

Figure 7: Property comparison between positive and negative samples for cpp dataset.

### B.1.3 NON-CANONICAL CELL-PENETRATING PEPTIDES (NC-CPP_PAMPA)

• **Property and Application**: The nc-cpp_pampa dataset focuses on the membrane penetration ability of non-canonical cyclic peptides, providing insights into the design of synthetic peptides for drug delivery. This dataset enables the evaluation of how chemical modifications affect membrane permeability.

• **Data Source**: The dataset is a subset of CycPeptMPDB (Li et al., 2023), including only molecules that are measured using PAMPA (Parallel Artificial Membrane Permeability Assay). Data points categorized as "undetectable" (marked as -10) are removed to avoid potential errors due to aggregation or measurement artifacts. Labels are reported as log Pexp (log cm/s).

• **Dataset Statistics**:   The dataset contains 6,970 datapoints with sequences ranging from 2 to 15 amino acids (average length 8.04) in length.

**Task: Regression; Split: ECFP-based; Evaluation: MAE**

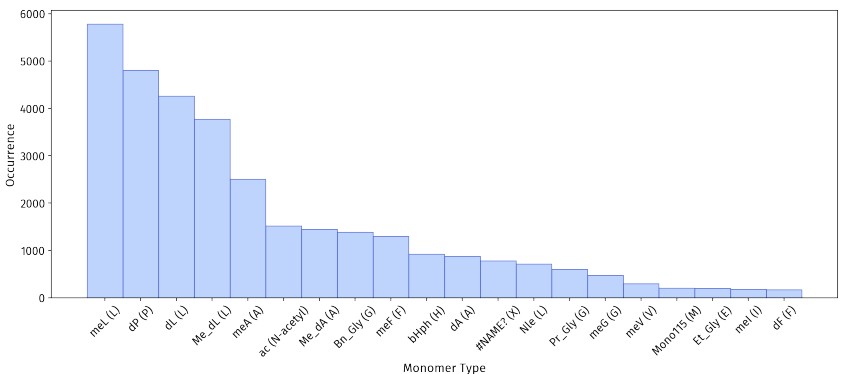

Figure 8: Amino acid distribution for nc-cpp_pampa dataset.

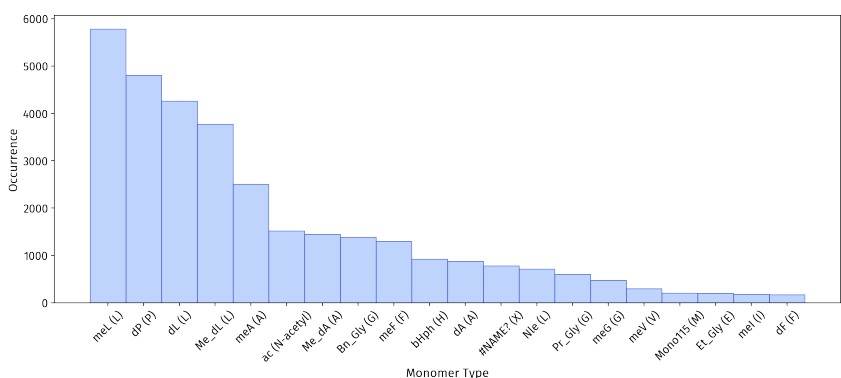

Figure 9: non-canonical monomers distribution for nc-cpp_pampa dataset.

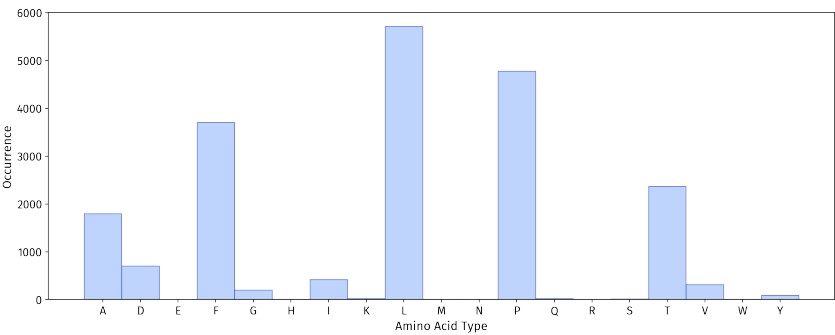

Figure 10: Length distribution of nc-cpp_pampa dataset.

### B.1.4   BLOOD-BRAIN BARRIER PEPTIDES (BBP)

• **Property and Application**:   The bbp dataset focuses on peptide's ability to penetrate the blood-brain barrier, a critical property for developing neuroactive peptides suitable for central nervous system diseases, such as Alzheimer's disease.

• **Data Source**:   The dataset is sourced from BBPpred (Dai et al., 2021), which collects BBP from Brainpeps (Van Dorpe et al., 2012), SATPdb (Singh et al., 2016), PepBank (Shtatland et al., 2007), and other literatures with experimental validation.  Redundancy is removed using CD-HIT with a

sequence identity threshold below 90%, and obtains total 119 BBP. Additionally, 240 BBP from Peptipedia are merged into the dataset.

**Experiment Negative Samples**   Negative samples are generated through the official negative sampling procedure plus manually collected 14 dipeptide negative samples from  Tanaka et al. (2019).

• **Dataset Statistics**:   The dataset contains 672 datapoints with sequences ranging from 2 to 82 amino acids (average length 14.14) in length.

**Task: Classification; Split: Hybrid; Evaluation: ROC-AUC**

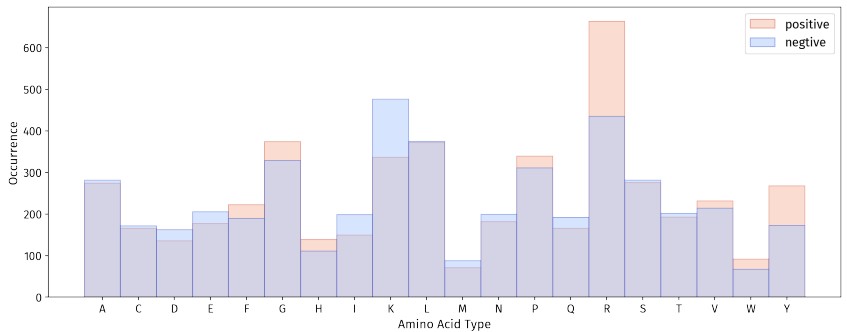

Figure 11:  Amino acid distribution comparison between positive and negative samples for bbp dataset.

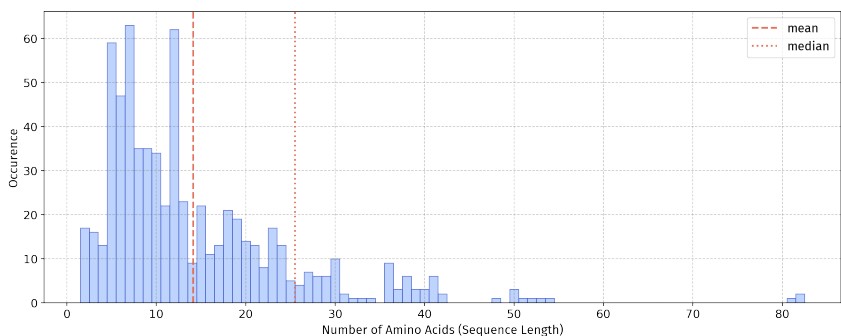

Figure 12: Length distribution of bbp dataset.

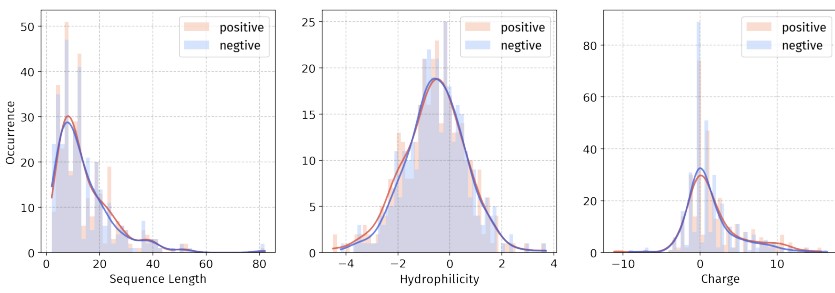

Figure 13: Property comparison between positive and negative samples for bbp dataset.

## B.2 AMP

> **Definition.** Datasets in this group focus on experimentally validated Antimicrobial peptides (AMP), covering peptides with either broad-spectrum antimicrobial activity or activity against specific types of microorganisms, such as bacteria, viruses, and fungi.
> **Impact.** Given the global health crisis posed by antimicrobial resistance, AMP represent promising alternatives or adjuncts to conventional antibiotics, offering broad-spectrum and evolutionarily conserved defense mechanisms.
> **Pipeline.** Activity Modeling

### B.2.1 ANTIMICROBIAL

• **Property and Application**: The antimicrobial dataset encompasses peptides with broad-spectrum activity against bacteria, fungi, viruses, and parasites. These peptides represent a diverse class of natural defense molecules with potential therapeutic applications against resistant pathogens.

• **Data Source**: This dataset originates from Wang et al. (2025c), which collects 22176 antimicrobial peptides from dbAMP (Yao et al., 2025), DRAMP (Ma et al., 2025), GRAMPA (Witten & Witten, 2019), and starPep (Aguilera-Mendoza et al., 2023), after removing duplicates. Additionally, another 16213 antimicrobial peptides from Peptipedia are merged, covering antibacterial, antifungal, antiviral, and antiparasitic peptides.

• **Dataset Statistics**: The dataset contains 61,504 datapoints with sequences ranging from 2 to 150 amino acids (average length 28.55) in length.

**Task: Classification; Split: Hybrid; Evaluation: ROC-AUC**

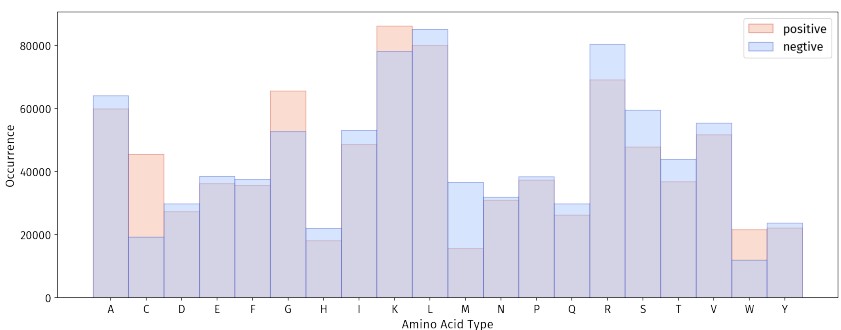

Figure 14: Amino acid distribution comparison between positive and negative samples for antimicrobial dataset.

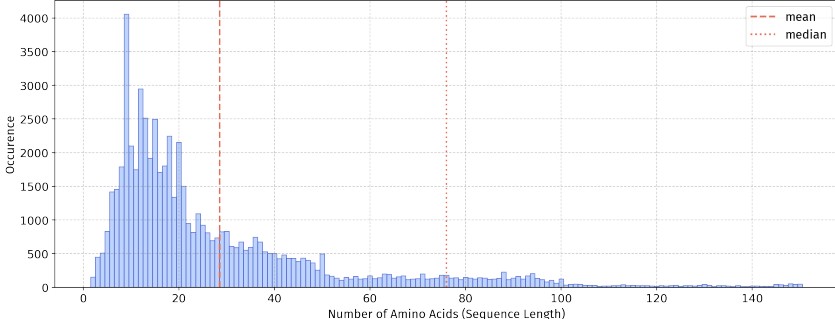

Figure 15: Length distribution of antimicrobial dataset.

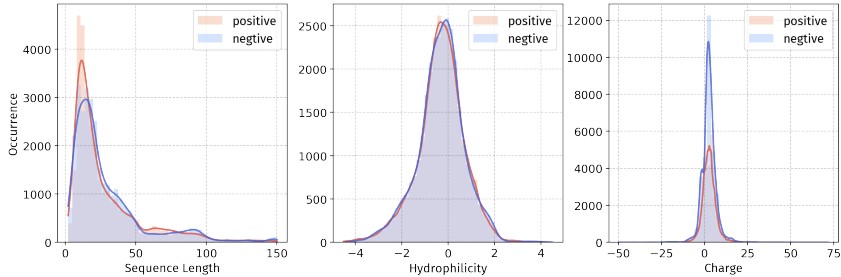

Figure 16: Property comparison between positive and negative samples for antimicrobial dataset.

### B.2.2 ANTIBACTERIAL

• **Property and Application**: The antibacterial dataset focuses on peptides with activity against bacterial pathogens. These peptides are crucial for developing novel antibiotics to combat bacterial resistance.

• **Data Source**: This dataset originates from Pinacho-Castellanos et al. (2021) which extracts a set of active AMP from starPepDB, comprising 8,278, 993, 130, and 2,944 AMPs with antibacterial, antifungal, antiparasitic, and antiviral activities, respectively. After retaining AMP with length between 5 and 100 natural amino acids and removing those listed as having more than one of the activities mentioned above, only 6010 peptides with single antibacterial activity are included in the training dataset. After adding 15210 peptides from Peptipedia, the dataset is expanded to 21220 peptides, which also includes antibacterial peptides with activity against other types of microorganisms.

• **Dataset Statistics**: The dataset contains 31,676 datapoints with sequences ranging from 2 to 150 amino acids (average length 25.33) in length.

**Task: Classification; Split: Hybrid; Evaluation: ROC-AUC**

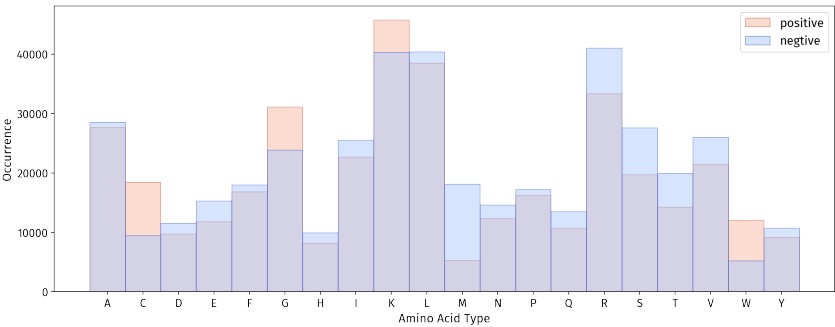

Figure 17: Amino acid distribution comparison between positive and negative samples for antibacterial dataset.

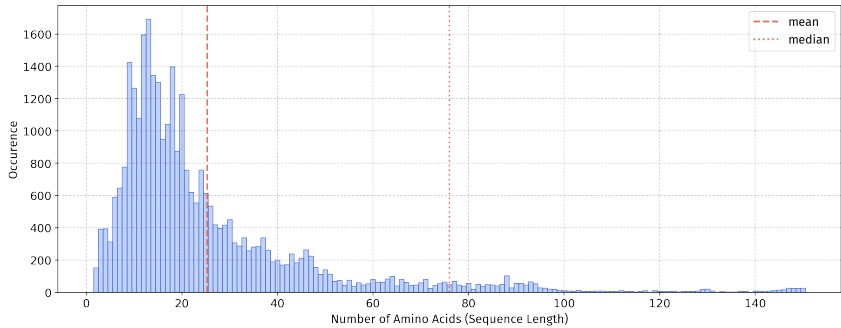

Figure 18: Length distribution of antibacterial dataset.

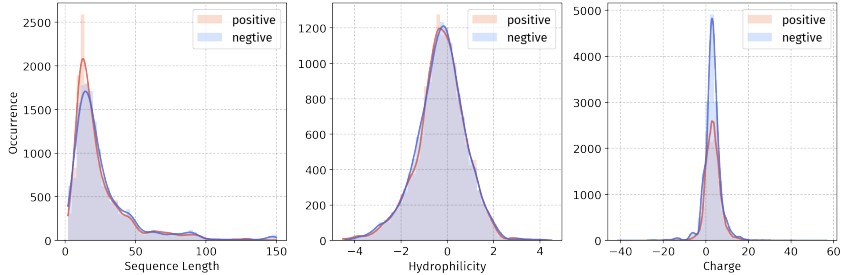

Figure 19: Property comparison between positive and negative samples for antibacterial dataset.

### B.2.3 ANTIFUNGAL

• **Property and Application**: The antifungal dataset collects peptides with activity against fungal pathogens. These peptides are important for developing treatments against fungal infections, which are increasingly problematic in immunocompromised patients.

• **Data Source**: 993 peptides are extracted from Pinacho-Castellanos et al. (2021) and 10094 antifungal peptides are merged from Peptipedia.

• **Dataset Statistics**: The dataset contains 16,698 datapoints with sequences ranging from 2 to 148 amino acids (average length 34.17) in length.

**Task: Classification; Split: Hybrid; Evaluation: ROC-AUC**

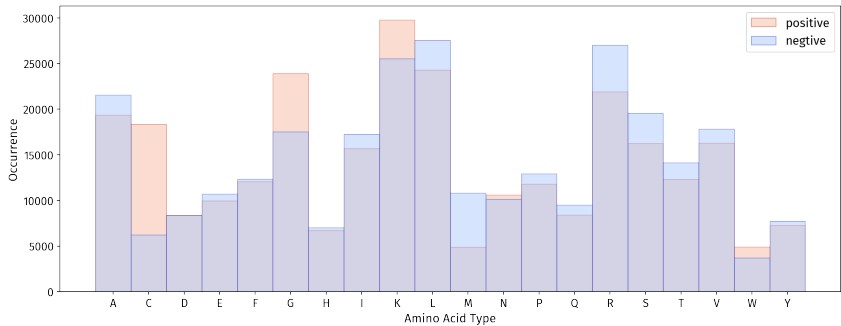

Figure 20: Amino acid distribution comparison between positive and negative samples for antifungal dataset.

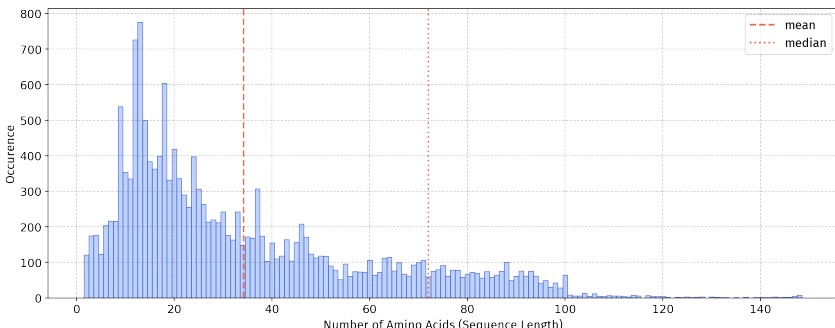

Figure 21: Length distribution of antifungal dataset.

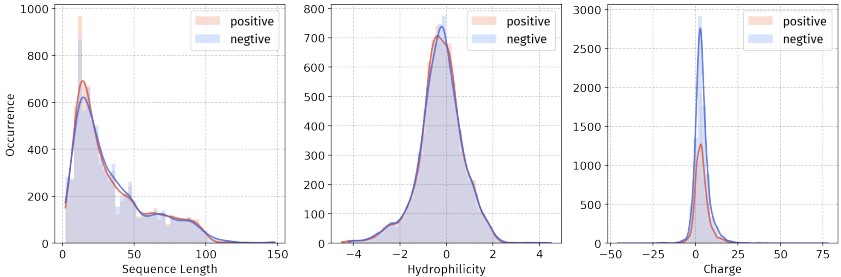

Figure 22: Property comparison between positive and negative samples for antifungal dataset.

### B.2.4 ANTIPARASITIC

• **Property and Application**: The antiparasitic dataset contains peptides active against parasites. These peptides represent potential therapeutics for parasitic diseases that affect millions worldwide.

• **Data Source**: The dataset originates from Zhang et al. (2022), which collects peptides from two resources: (1) APP and AMP databases including ParaPep (Mehta et al., 2014), APD3 (Wang et al., 2015), dbAMP (Jhong et al., 2018), CAMP (Thomas et al., 2009), DRAMP (Fan et al., 2016), and ADAM (Nothaft et al., 2015); (2) APP-related articles during 2015-01-01 to 2019-10-31 in PubMed. Homologous sequences at a threshold of 90% are removed for positive samples, obtaining 301 peptides in total. After adding 5741 antiparasitic peptides from Peptipedia, the dataset is expanded to 6042 peptides.

• **Dataset Statistics**: The dataset contains 8,632 datapoints with sequences ranging from 2 to 140 amino acids (average length 35.97) in length.

**Task: Classification; Split: Hybrid; Evaluation: ROC-AUC**

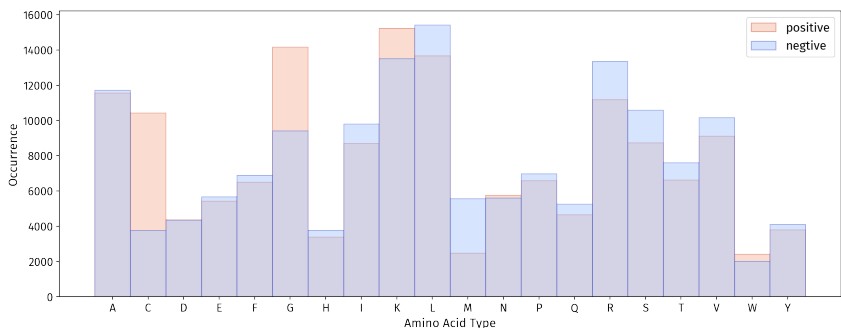

Figure 23: Amino acid distribution comparison between positive and negative samples for antiparasitic dataset.

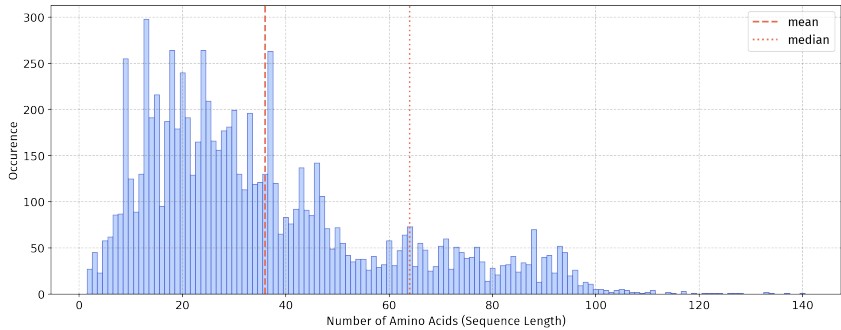

Figure 24: Length distribution of antiparasitic dataset.

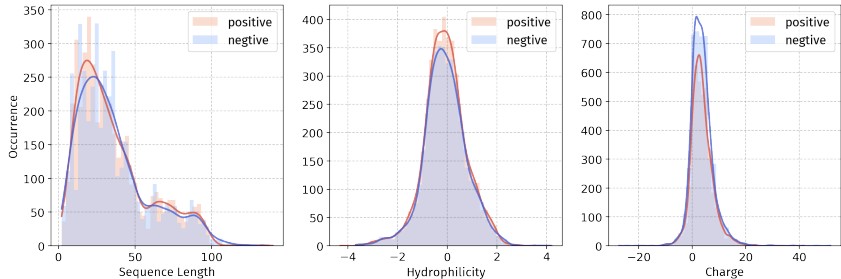

Figure 25: Property comparison between positive and negative samples for antiparasitic dataset.

### B.2.5 ANTIVIRAL

• **Property and Application**: The antiviral dataset focuses on peptides with activity against viral pathogens. These peptides are crucial for developing novel antiviral therapeutics, especially important given the emergence of new viral threats.

• **Data Source**: 2994 antifungal peptides are extracted from Pinacho-Castellanos et al. (2021) and 2266 antifungal peptides are merged from Peptipedia.

• **Dataset Statistics**: The dataset contains 8,268 datapoints with sequences ranging from 2 to 138 amino acids (average length 22.75) in length.

**Task: Classification; Split: Hybrid; Evaluation: ROC-AUC**

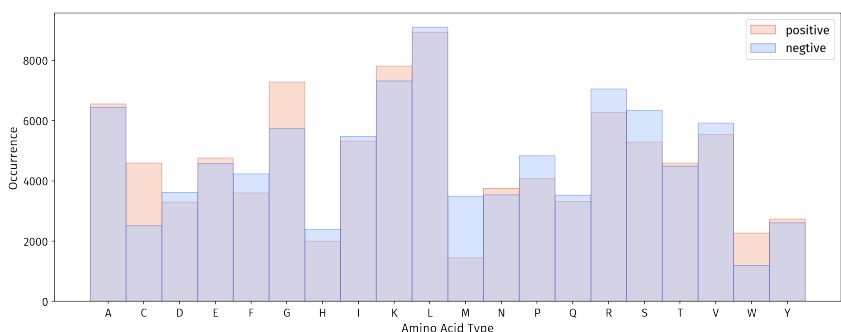

Figure 26: Amino acid distribution comparison between positive and negative samples for antiviral dataset.

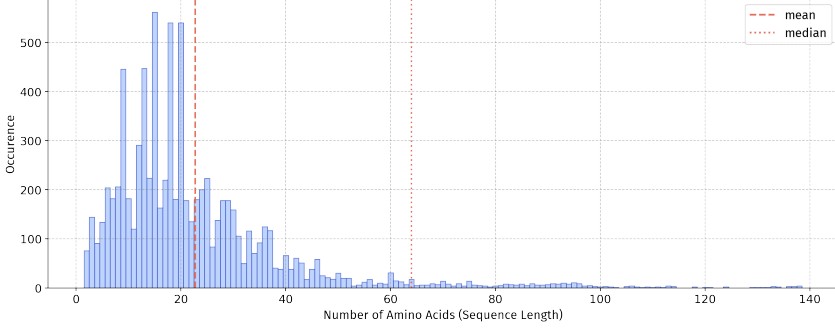

Figure 27: Length distribution of antiviral dataset.

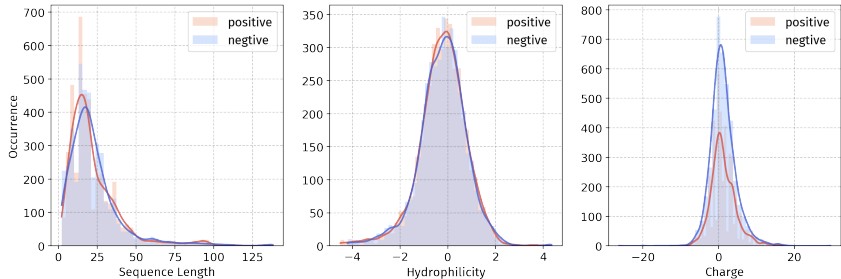

Figure 28: Property comparison between positive and negative samples for antiviral dataset.

### B.2.6 E. COLI MIC

• **Property and Application**: The E. coli MIC dataset reports the minimum inhibitory concentrations (MICs) of peptides against *Escherichia coli*. MIC, defined as the lowest drug concentration that inhibits pathogen growth *in vitro* after 18-24 hours of bacterial culture, provides a quantitative measure of antibacterial potency.

• **Data Source**: The Dataset is sourced from GRAMPA (Witten & Witten, 2019), which records peptide MIC values against different bacterial strains. To minimize experimental variation caused by different experimental environments and batches, data points outside the range [Q1-1.5×IQR, Q3+1.5×IQR] are identified as outliers and removed, with the average of remaining values used as the final label. The MIC values are reported as $\log(\mu M)$.

• **Dataset Statistics**: The dataset contains 3,312 datapoints with sequences ranging from 2 to 140 amino acids (average length 23.24) in length.

**Task: Regression; Split: Hybrid; Evaluation: MAE**

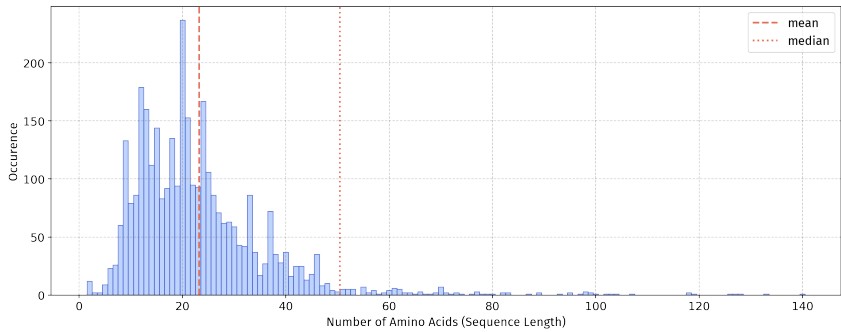

Figure 29: Length distribution of E. coli MIC dataset.

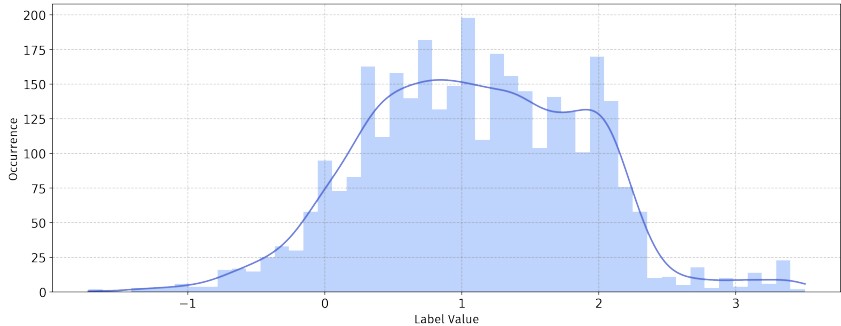

Figure 30: Label distribution of E. coli MIC dataset.

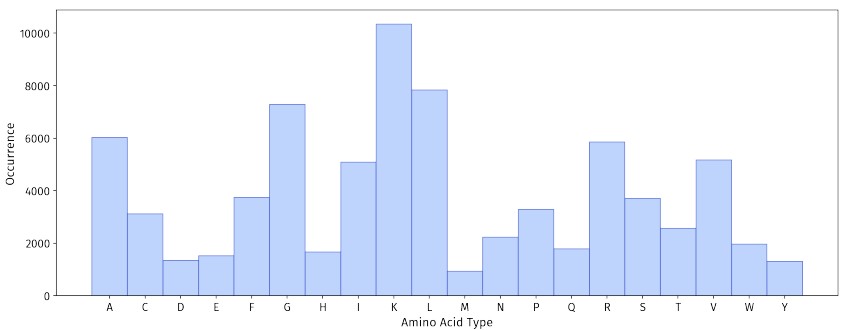

Figure 31: Amino acid distribution of E. coli MIC dataset.

### B.2.7 P. AERUGINOSA MIC

• **Property and Application**: This dataset provides quantitative measurements of peptide activity against *Pseudomonas aeruginosa*, a clinically important Gram-negative pathogen known for its resistance to many antibiotics.

• **Data Source**: The Dataset is sourced from GRAMPA (Witten & Witten, 2019), following the same data processing methodology as the E. coli MIC dataset to ensure consistency and reliability.

• **Dataset Statistics**: The dataset contains 1,531 datapoints with sequences ranging from 2 to 140 amino acids (average length 22.01) in length.

**Task: Regression; Split: Hybrid; Evaluation: MAE**

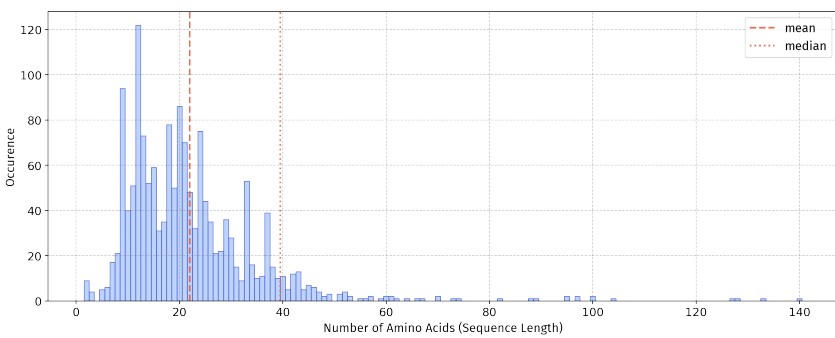

Figure 32: Length distribution of P. aeruginosa MIC dataset.

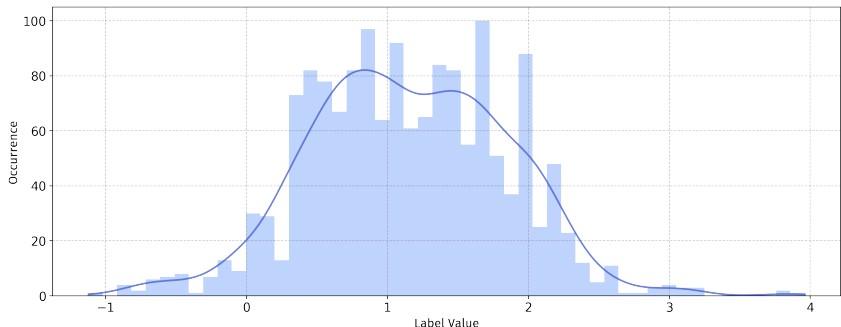

Figure 33: Label distribution of P. aeruginosa MIC dataset.

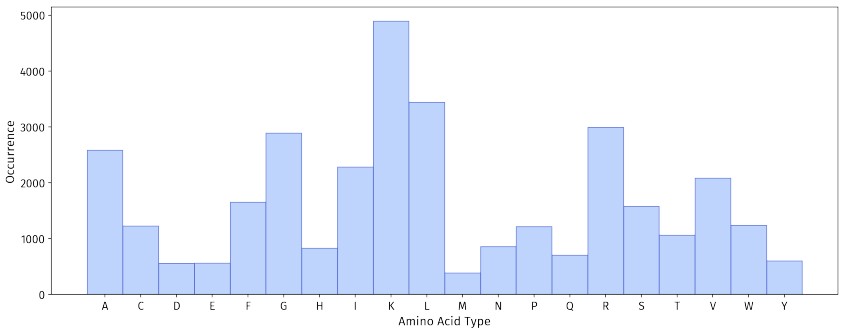

Figure 34: Amino acid distribution of P. aeruginosa MIC dataset.

### B.2.8    S. AUREUS MIC

• **Property and Application**:   This dataset provides quantitative measurements of peptide activity against *Staphylococcus aureus*, a representative Gram-positive pathogen that causes various infections.

• **Data Source**:   The Dataset is sourced from GRAMPA (Witten & Witten, 2019), following the same data processing methodology as other MIC datasets to ensure consistency and reliability.

• **Dataset Statistics**:   The dataset contains 2,900 datapoints with sequences ranging from 2 to 140 amino acids (average length 22.70) in length.

**Task: Regression; Split: Hybrid; Evaluation: MAE**

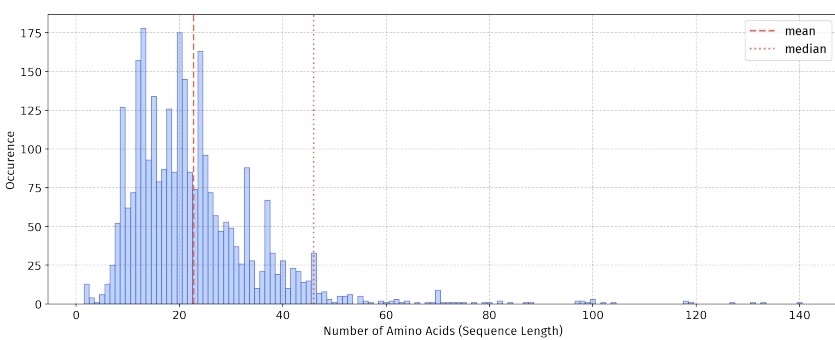

Figure 35: Length distribution of S. aureus MIC dataset.

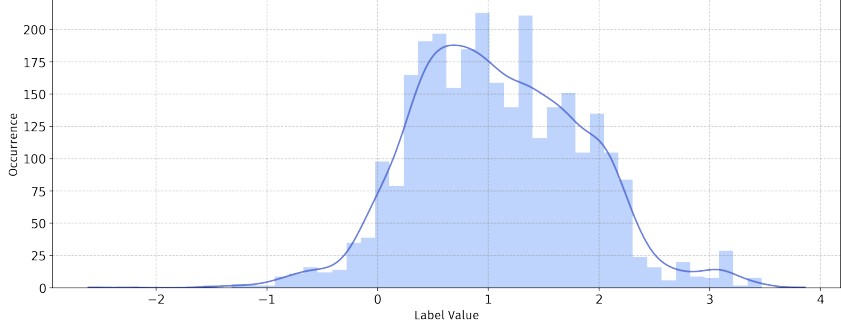

Figure 36: Label distribution of S. aureus MIC dataset.

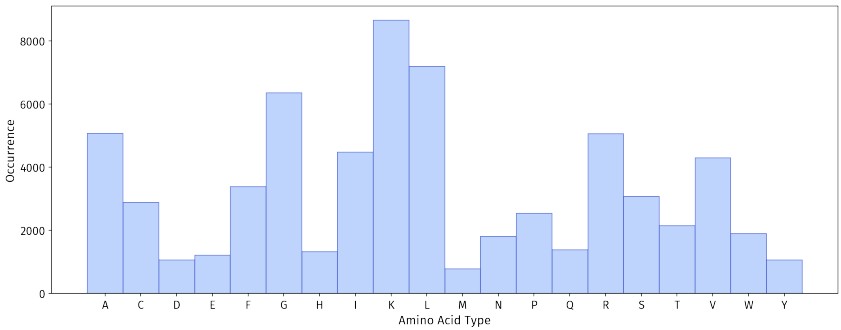

Figure 37: Amino acid distribution of S. aureus MIC dataset.

### B.2.9 NON-CANONICAL ANTIMICROBIAL (NC-ANTIMICROBIAL)

• **Property and Application**: The nc-antimicrobial dataset includes non-canonical peptides with broad-spectrum antimicrobial activity. These peptides represent the potential of synthetic modifications to enhance antimicrobial properties.

• **Data Source**: The Dataset is sourced from Hemolytik 2.0 (Singh et al., 2025), which contains manually collected data from published literature and various databases. Hemolytik 2.0 is subset to only include sequences that have antimicrobial records. Sequences in Modified Amino acid Peptide (MAP) format stored in the original database are converted into BILN format (Fox et al., 2022) supported by this project.

• **Dataset Statistics**: The dataset contains 2,496 datapoints with sequences ranging from 4 to 130 amino acids (average length 17.79) in length.

**Task: Classification; Split: ECFP-based; Evaluation: ROC-AUC**

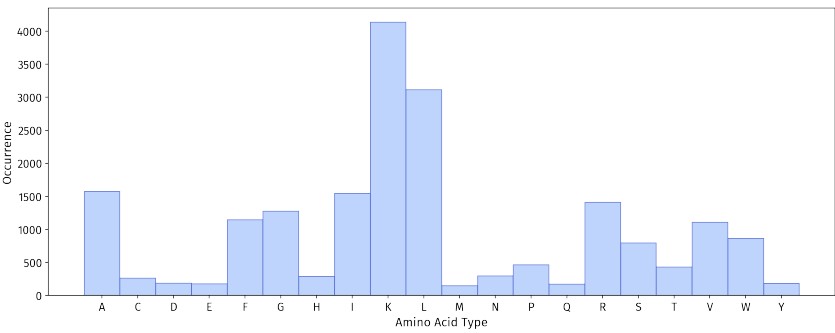

Figure 38: Canonical Amino acid distribution comparison between positive and negative samples for nc-antimicrobial dataset.

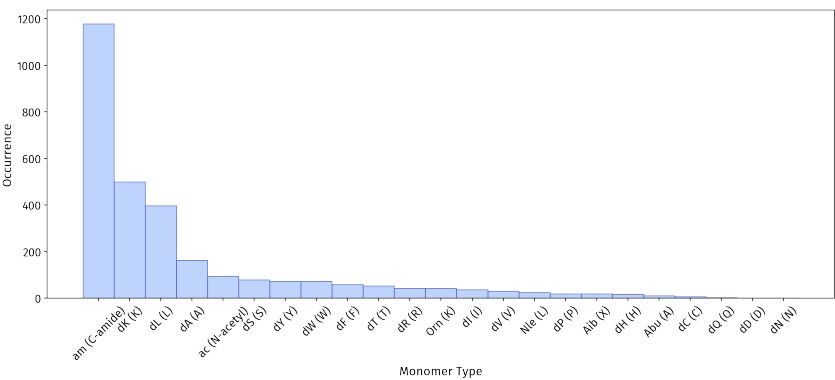

Figure 39: non-canonical Amino acid distribution comparison between positive and negative samples for nc-antimicrobial dataset.

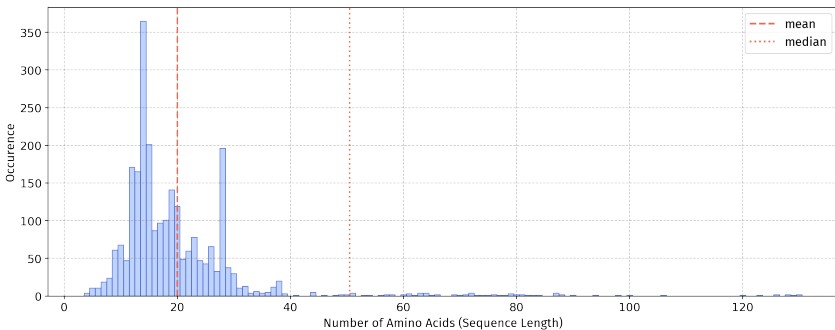

Figure 40: Length distribution of nc-antimicrobial dataset.

### B.2.10 NON-CANONICAL ANTIBACTERIAL (NC-ANTIBACTERIAL)

• **Property and Application**: The nc-antibacterial dataset includes non-canonical peptides with activity against bacterial pathogens. This dataset enables evaluation of how synthetic modifications affect antibacterial activity.

• **Data Source**: The Dataset is sourced from Hemolytik 2.0 (Singh et al., 2025). Hemolytik 2.0 is subset to only include sequences that have antibacterial records. Sequences in MAP format stored in the original database are converted into BILN format supported by this project.

• **Dataset Statistics**: The dataset contains 1,666 datapoints with sequences ranging from 4 to 76 amino acids (average length 16.82) in length.

**Task: Classification; Split: ECFP-based; Evaluation: ROC-AUC**

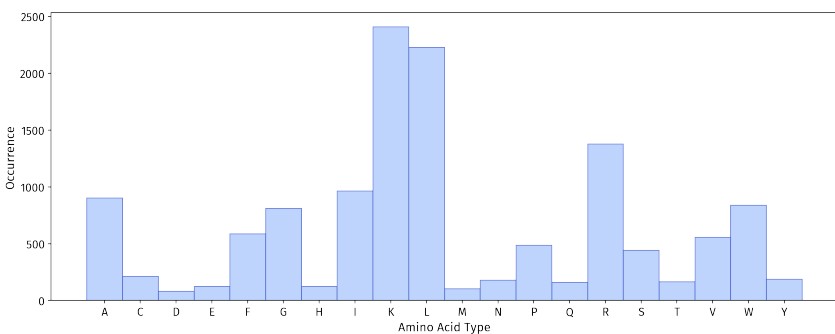

Figure 41: Canonical Amino acid distribution comparison between positive and negative samples for nc-antibacterial dataset.

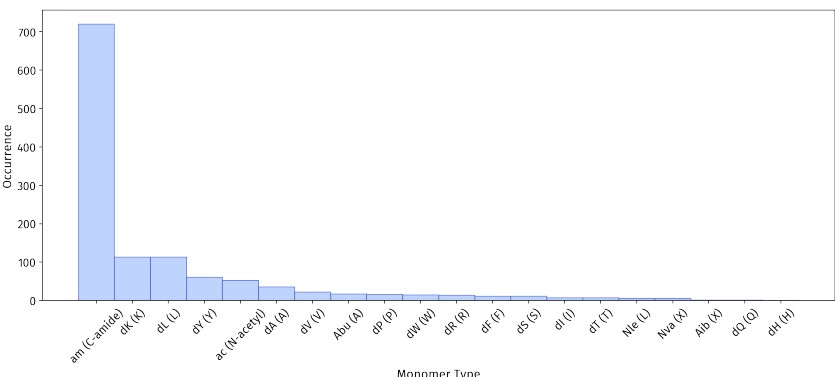

Figure 42: non-canonical Amino acid distribution comparison between positive and negative samples for nc-antibacterial dataset.

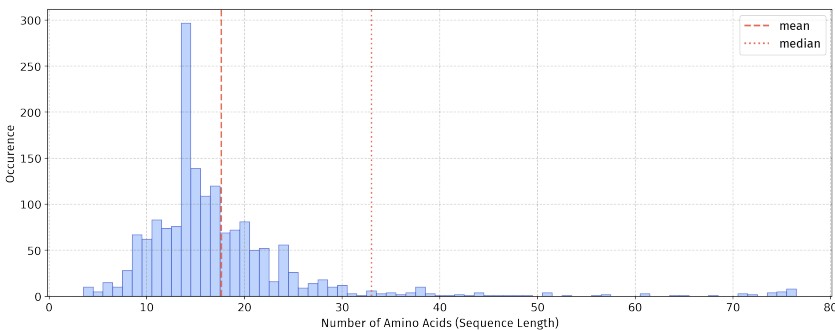

Figure 43: Length distribution of nc-antibacterial dataset.

### B.2.11 NON-CANONICAL ANTIFUNGAL (NC-ANTIFUNGAL)

• **Property and Application**: The nc-antifungal dataset collects non-canonical peptides targeting fungal pathogens. This dataset enables evaluation of synthetic modifications for antifungal activity.

• **Data Source**: The Dataset is sourced from Hemolytik 2.0 (Singh et al., 2025). Hemolytik 2.0 is subset to only include sequences that have antifungal records. Sequences in MAP format stored in the original database are converted into BILN format supported by this project.

• **Dataset Statistics**: The dataset contains 410 datapoints with sequences ranging from 4 to 76 amino acids (average length 18.18) in length.

**Task: Classification; Split: ECFP-based; Evaluation: ROC-AUC**

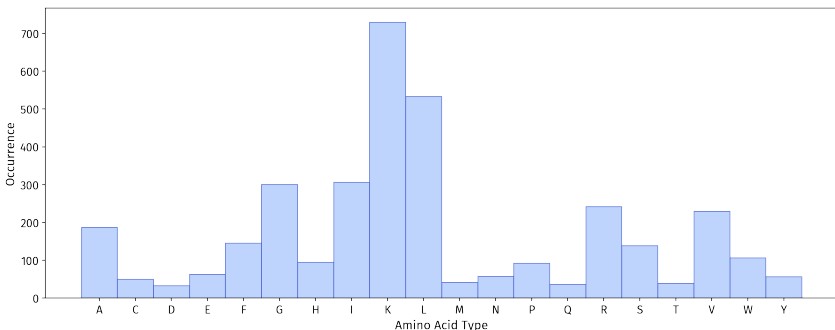

Figure 44: Canonical Amino acid distribution comparison between positive and negative samples for nc-antifungal dataset.

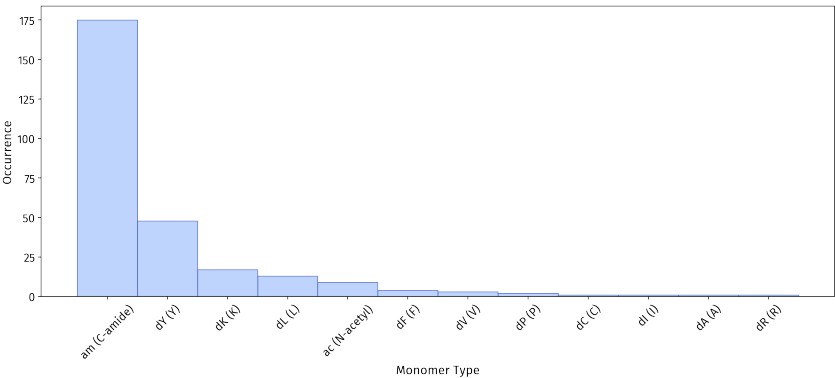

Figure 45: non-canonical Amino acid distribution comparison between positive and negative samples for nc-antifungal dataset.

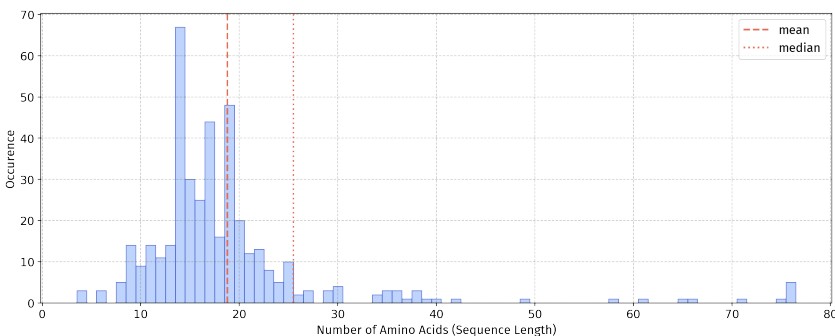

Figure 46: Length distribution of nc-antifungal dataset.

## B.3 ONCOLOGY

> **Definition.** Datasets in this group include peptides with experimentally validated anticancer activity. The mechanisms of the activity include cytotoxic effects and immune regulatory functions.
> **Impact.** Peptides offer unique advantages in cancer therapy due to their high target specificity, low off-target toxicity, and potential to modulate the immune system. These features make them promising candidates for both direct anticancer treatments and peptide-based vaccines, addressing limitations of conventional chemotherapies.
> **Pipelines.** Activity Modeling

### B.3.1 ANTICANCER

• **Property and Application**: The anticancer dataset includes peptides that exert anticancer activity by targeting tumor cells through mechanisms such as membrane disruption, interference with specific intracellular processes, or induction of apoptosis and programmed cell death. Anticancer Peptides (ACPs) represent promising therapeutic agents for cancer treatment.

• **Data Source**: The dataset originates from AntiCP 2.0 (Agrawal et al., 2021).The original literature obtained experimentally validated anticancer peptides from datasets of previous studies including ACP-DL (Yi et al., 2019), ACPP (Vijayakumar & Ptv, 2015), ACPred-FL (Wei et al., 2018), and iACP (Aziz et al., 2022). In addition, data are also extracted from the ACP database CancerPPD (Tyagi et al., 2015). After removing small, long, identical and non-natural peptides, 970 unique ACPs having 4 or more residues and 50 or fewer residues were obtained. Additionally, 8161 ACPs from Peptipedia are merged.

• **Dataset Statistics**: The dataset contains 13,852 datapoints with sequences ranging from 2 to 145 amino acids (average length 27.83) in length.

**Task: Classification; Split: Hybrid; Evaluation: ROC-AUC**

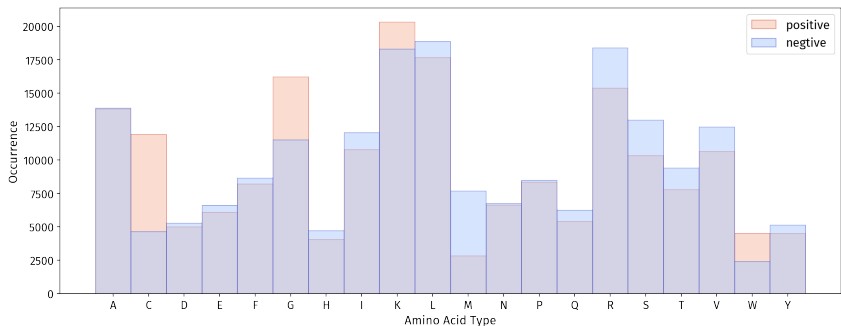

Figure 47: Amino acid distribution comparison between positive and negative samples for anticancer dataset.

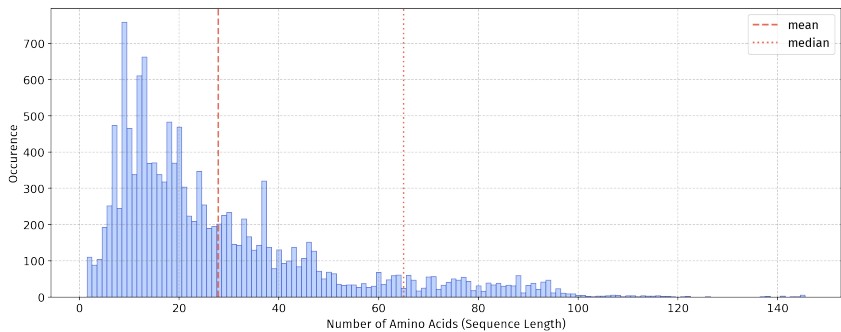

Figure 48: Length distribution of anticancer dataset.

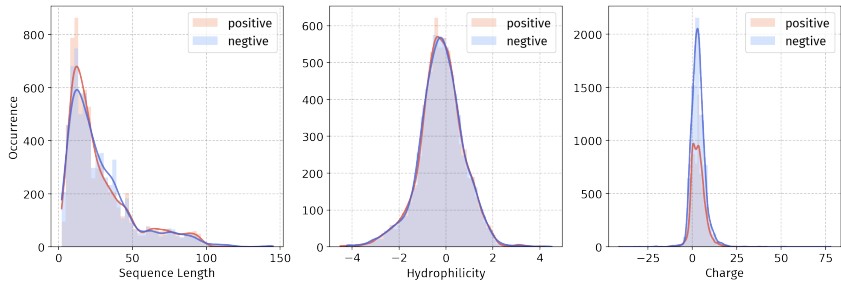

Figure 49: Property comparison between positive and negative samples for anticancer dataset.

### B.3.2  TUMOR T-CELL ANTIGENS (TTCA)

• **Property and Application**:  The ttca dataset comprises tumor T-cell antigen (ttca) peptides, which are capable of stimulating antitumor immune responses.  These peptides function by presenting tumor-associated epitopes to T cells, thereby activating cellular immunity against cancer cells.

• **Data Source**:  The dataset originates from  Charoenkwan et al. (2020b). The literature constructs a benchmark dataset, where ttca peptides are obtained from TANTIGEN (Olsen et al., 2017) and TANTIGEN 2.0 (Zhang et al., 2021), whereby a total of 529 unique MHC class I peptides are collected and considered as positive samples.

• **Dataset Statistics**:  The dataset contains 1,182 datapoints with sequences ranging from 8 to 20 amino acids (average length 9.36) in length.

**Task: Classification; Split: Hybrid; Evaluation: ROC-AUC**

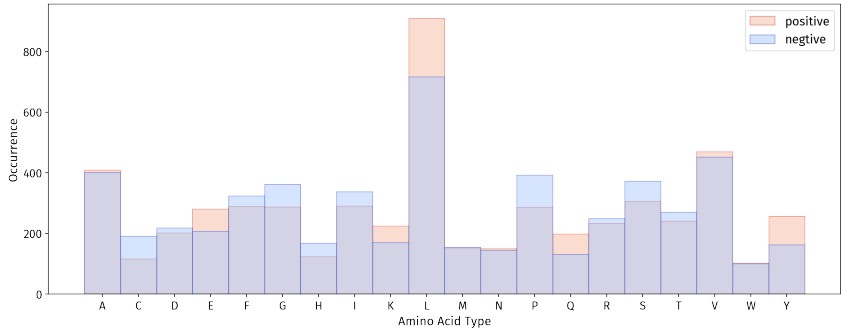

Figure 50: Amino acid distribution comparison between positive and negative samples for tumor T-cell antigen dataset.

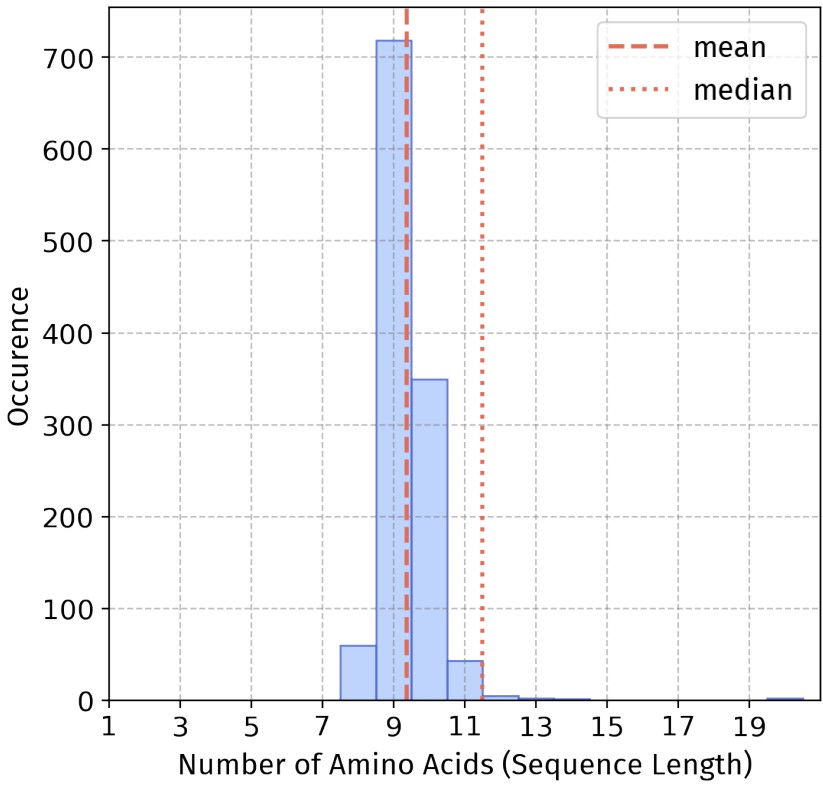

Figure 51: Length distribution of tumor T-cell antigen dataset.

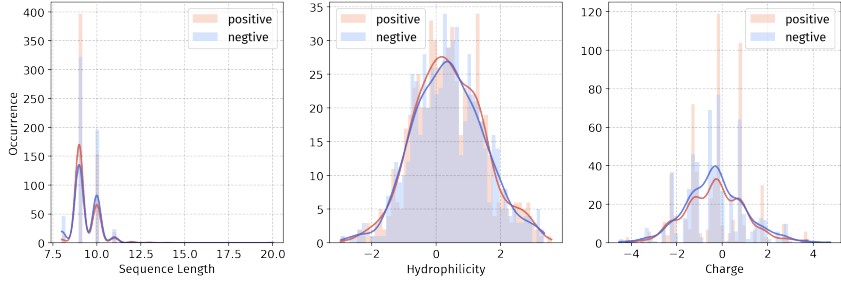

Figure 52: Property comparison between positive and negative samples for tumor T-cell antigen dataset.

## B.4 METABOLIC GROUP

> **Definition.** Datasets in this group comprise peptides with therapeutic potential for metabolic disorders, including diabetes and hypertension. These peptides act via mechanisms such as enzyme inhibition (e.g., DPP-IV or ACE inhibitors) or modulation of hormone-like signaling pathways.
> **Impact.** Peptides are particularly well-suited for the long-term management of metabolic diseases, as they possess high target specificity, favorable biocompatibility, and reduced off-target effects, which can help minimize cumulative toxicity and improve patient adherence over prolonged treatment periods.
> **Pipeline.** Activity Modeling

### B.4.1 ACE INHIBITORY

• **Property and Application**: The ace inhibitory dataset comprises peptides that inhibit Angiotensin-Converting Enzyme (ACE), which normally converts angiotensin I into angiotensin II, a potent vasoconstrictor. By preventing this conversion, ACE inhibitory peptides help lower blood pressure and reduce cardiovascular risk.

• **Data Source**: The dataset originates from Manavalan et al. (2019), which extracts ACE inhibitory peptides from literature and publicly available databases like AHTPDB (Kumar et al., 2015a), BIOPEP (Minkiewicz et al., 2008), and PDB (Kumar et al., 2015a). The original literature provides inhibitory activity (IC50) for dipeptides and tripeptides, while peptides with more than 3 residues are only provided classification labels. Since classification labels of longer peptide are collected from different literature sources, making it difficult to unify the positive/negative sample division threshold for dipeptides and tripeptides. Therefore, peptides with lengths ¡ 5 amino acid residues are excluded from the classification task, leaving 1,053 peptides. Additionally, 780 ACE inhibitory peptides from Peptipedia are merged.

• **Dataset Statistics**: The dataset contains 3,560 datapoints with sequences ranging from 4 to 81 amino acids (average length 8.08) in length.

**Task: Classification; Split: Hybrid; Evaluation: ROC-AUC**

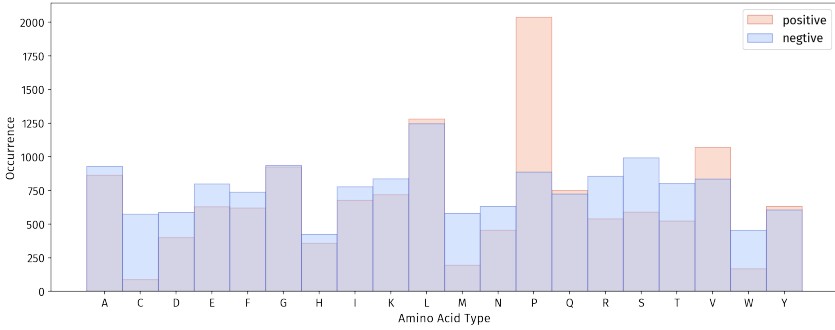

Figure 53: Amino acid distribution comparison between positive and negative samples for ace inhibitory dataset.

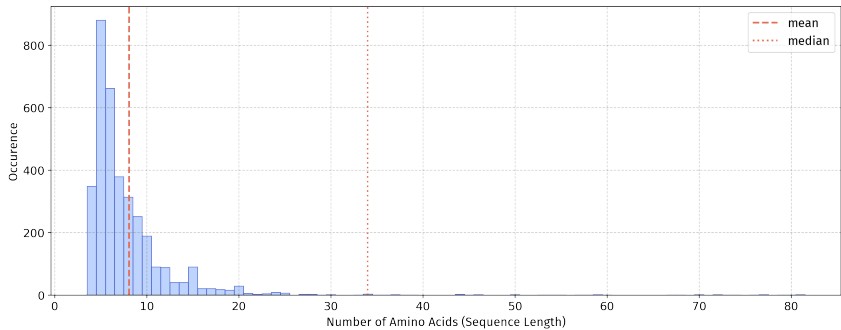

Figure 54: Length distribution of ace inhibitory dataset.

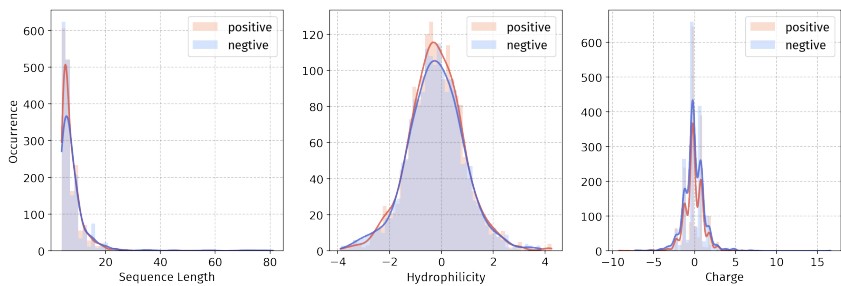

Figure 55: Property comparison between positive and negative samples for ace inhibitory dataset.

### B.4.2  ACE INHIBITORY IC50

• **Property and Application**:   This dataset comprises dipeptides and tripeptides with ACE inhibitory ability. Developing predictive models specifically for these short peptides can help identify structural motifs that are most relevant for ACE inhibition, providing valuable leads for peptide-based drug development.

• **Data Source**:   The dataset originates from  Manavalan et al. (2019). The dataset contains 131 dipeptides having inhibitory activity (IC50) between 0.92 to 17,000 $\mu$M; 205 tripeptides having IC50 between 0.04 to 2,700 $\mu$M. IC50 values were converted into normalized pIC50 values =-log ($\mu$M) to narrow down the scale.

• **Dataset Statistics**:   The dataset contains 337 datapoints with sequences ranging from 2 to 3 amino acids (average length 2.61) in length.

**Task: Regression; Split: Hybrid; Evaluation: MAE**

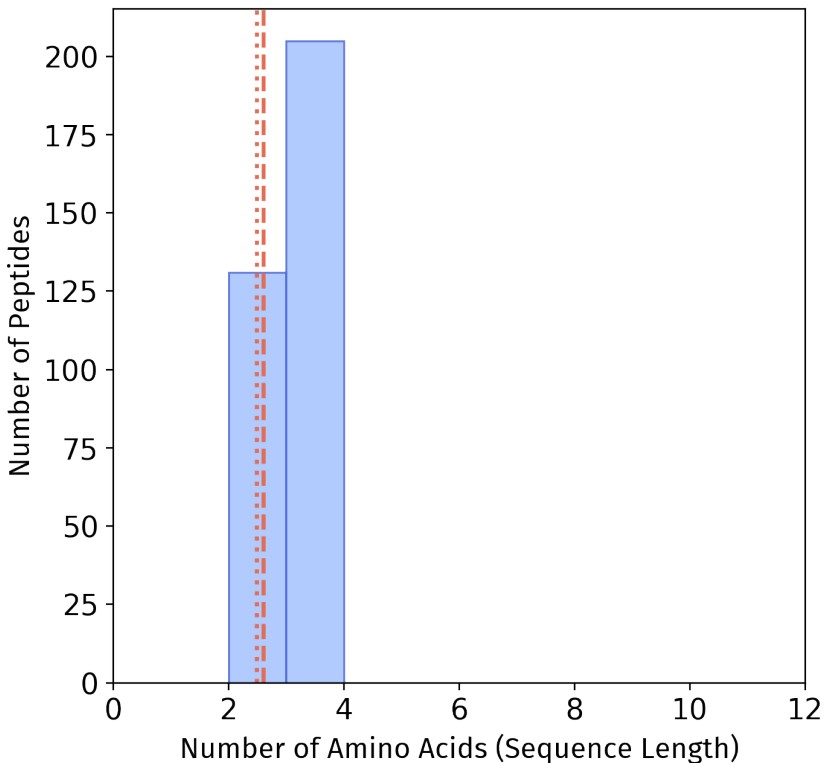

Figure 56: Length distribution of ace inhibitory ic50 dataset.

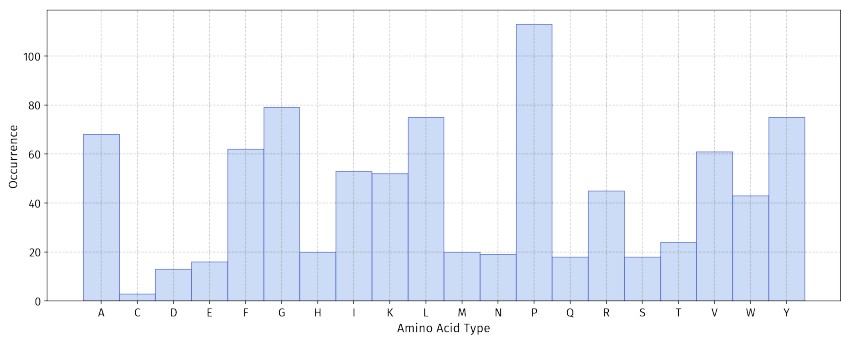

Figure 57: Label distribution of ACE inhibitory IC50 dataset.

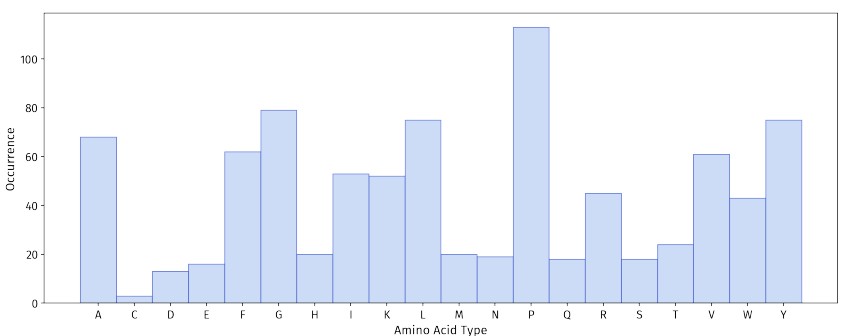

Figure 58: Amino acid distribution of ace inhibitory ic50 dataset.

### B.4.3 DPP-IV INHIBITORS

• **Property and Application**: The dpp-iv inhibitor dataset includes peptides that inhibit dipeptidyl peptidase-4 (DPP-4), thereby preventing the degradation of incretins such as Glucagon-Like Peptide 1 (GLP-1) and Gastric Inhibitory Polypeptide (GIP). This inhibition enhances insulin secretion and improves glycemic control in diabetic patients. Clinically, this mechanism helps manage blood glucose levels while minimizing the risk of hypoglycemia and weight gain.

• **Data Source**: The positive samples are collected from Charoenkwan et al. (2020a), Which extractes 665 unique DPP-IV inhibitors from literatures and publically available databases (i.e., BIOPEP-UWM).

**Experiment Negative Samples** Dipeptides with DPPIV inhibition rate ¡5% at 0.5 mM from Hikida et al. (2013) are added as negative samples to the dataset. After removing 13 duplicate sequences between positive and negative samples in the merged dataset, the final dataset contains 86 dipeptide negative samples.

• **Dataset Statistics**: The dataset contains 1,268 datapoints with sequences ranging from 2 to 33 amino acids (average length 6.15) in length.

**Task: Classification; Split: Hybrid; Evaluation: ROC-AUC**

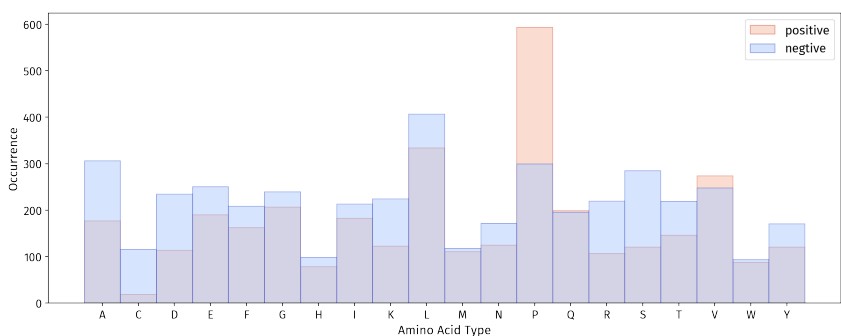

Figure 59: Amino acid distribution comparison between positive and negative samples for dpp-iv inhibitor dataset.

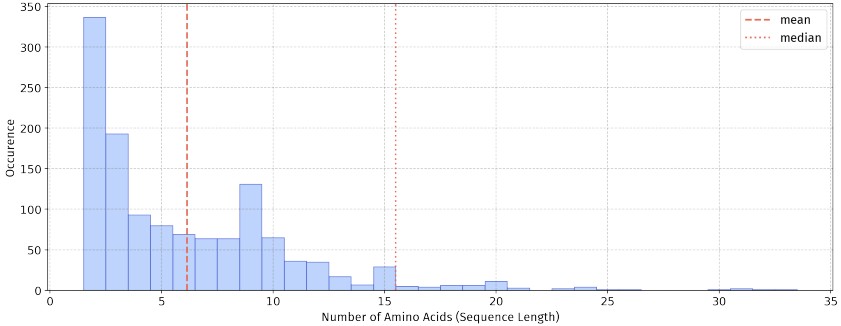

Figure 60: Length distribution of dpp-iv inhibitor dataset.

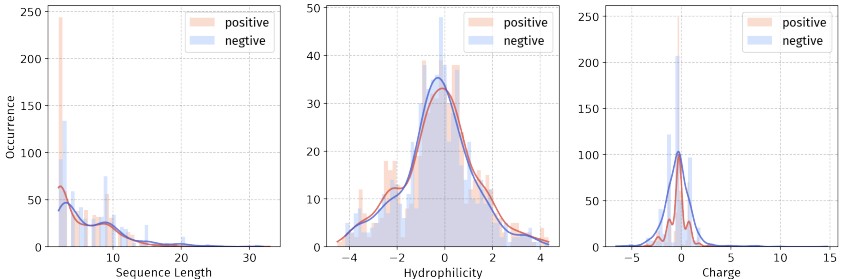

Figure 61: Property comparison between positive and negative samples for dpp-iv inhibitor dataset.

### B.4.4 ANTIDIABETIC

• **Property and Application**: The antidiabetic peptide dataset captures peptides that modulate insulin signaling pathways and other mechanisms involved in glucose homeostasis. These peptides contribute to lowering blood glucose levels, improving insulin sensitivity, and inhibiting diabetes-related metabolic dysfunctions.

• **Data Source**: The positive samples are collected from Yue et al. (2024), combined with antidiabetic peptides recorded in Peptipedia. The literature collected a total of 1,786 antidiabetic peptides related to Type 1 Diabetes Mellitus (T1DM) and 756 related to Type 2 Diabetes Mellitus (T2DM) from the BioDADPep database (Roy & Teron, 2019). Peptides containing non-standard residues are removed, and redundancy is addressed using the CD-HIT are eliminated.

**Experiment Negative Samples** In dipeptides, the main activity related to diabetes is DPP-IV inhibition. Therefore, dipeptides with DPPIV inhibition rate ¡5% from Hikida et al. (2013) were added as negative samples to the dataset. After removing 23 duplicate sequences between positive and negative samples in the merged dataset, the final dataset contains 76 dipeptide negative samples.

• **Dataset Statistics**: The dataset contains 3,028 datapoints with sequences ranging from 2 to 46 amino acids (average length 10.41) in length.

**Task: Classification; Split: Hybrid; Evaluation: ROC-AUC**

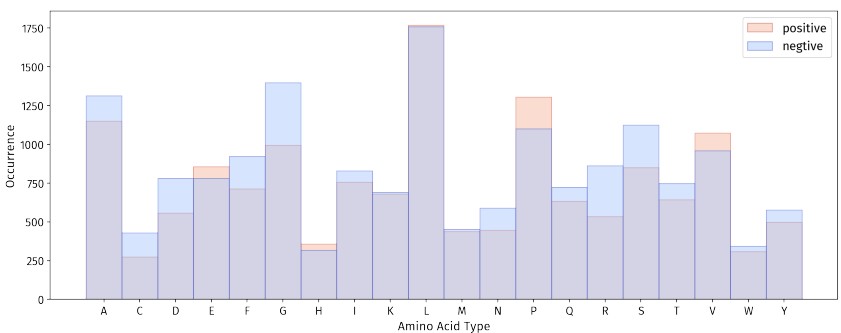

Figure 62: Amino acid distribution comparison between positive and negative samples for antidiabetic dataset.

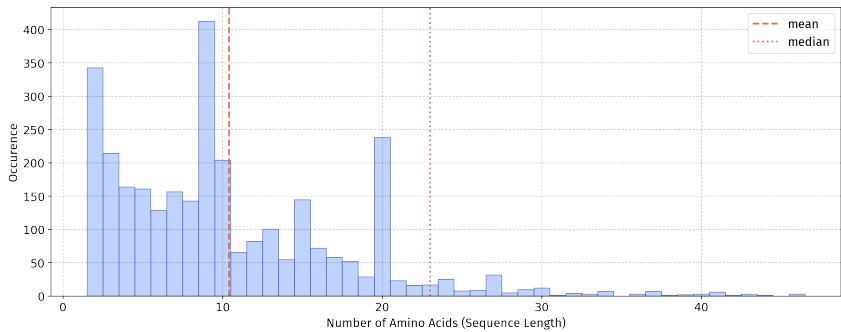

Figure 63: Length distribution of antidiabetic dataset.

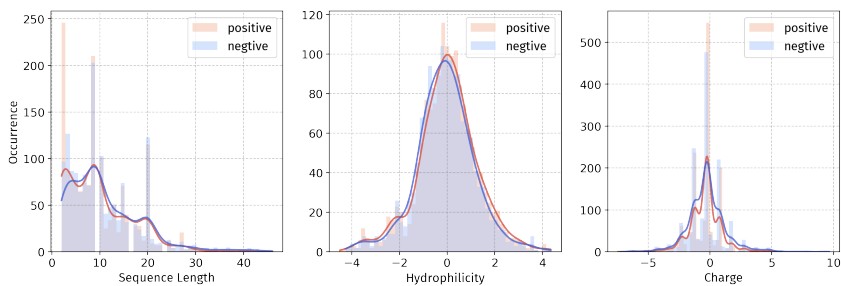

Figure 64: Property comparison between positive and negative samples for antidiabetic dataset.

## B.5 OTHERS GROUP

**Definition.** Datasets in this group include a variety of peptide datasets with biological activities relevant to chronic disease management, which do not fall into the major categories described above.

**Impact.** Peptides targeting chronic diseases beyond major categories offer therapeutic options for conditions that require long-term management. Their high target specificity, favorable biocompatibility, and reduced off-target effects make them particularly suitable for prolonged treatment, potentially improving patient adherence and minimizing cumulative side effects.

**Pipeline.** Activity Modeling

### B.5.1 ANTIAGING

• **Property and Application**: The antiaging dataset contains peptides that promote healthy lifespan by mitigating adverse health consequences commonly observed in the elderly population. These peptides may target various aging-related pathways and cellular processes.

• **Data Source**: The positive samples are all collected from Peptipedia.

• **Dataset Statistics**: The dataset contains 558 datapoints with sequences ranging from 2 to 80 amino acids (average length 10.93) in length.

**Task: Classification; Split: Hybrid; Evaluation: ROC-AUC**

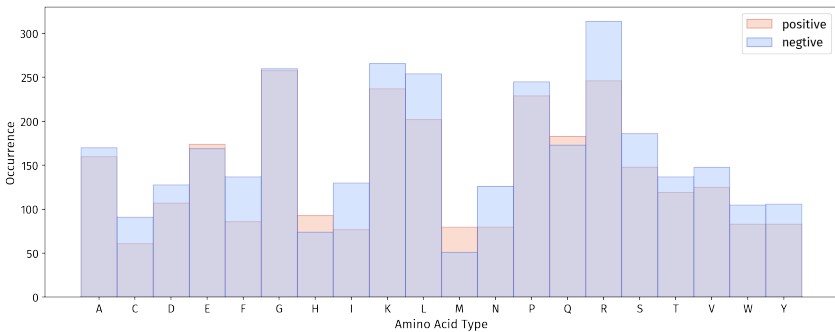

Figure 65: Amino acid distribution comparison between positive and negative samples for antiaging dataset.

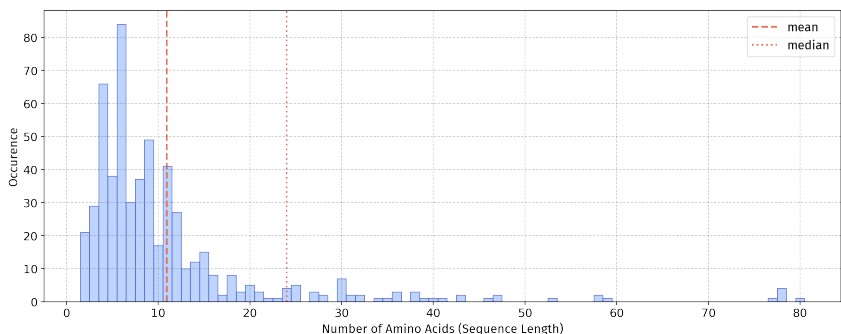

Figure 66: Length distribution of antiaging dataset.

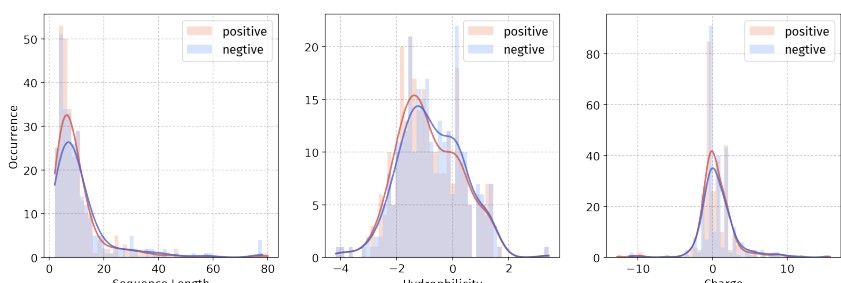

Figure 67: Property comparison between positive and negative samples for antiaging dataset.

### B.5.2 ANTI-INFLAMMATORY

• **Property and Application**: the anti-inflammatory dataset contains peptides that inhibit the activity of pro-inflammatory cytokines and signaling pathways, offering therapeutic potential for autoimmune disorders, chronic inflammatory conditions, and inflammatory tissue damage.

• **Data Source**: The positive samples are all collected from Peptipedia.

• **Dataset Statistics**: The dataset contains 7,750 datapoints with sequences ranging from 2 to 107 amino acids (average length 16.95) in length.

**Task: Classification; Split: Hybrid; Evaluation: ROC-AUC**

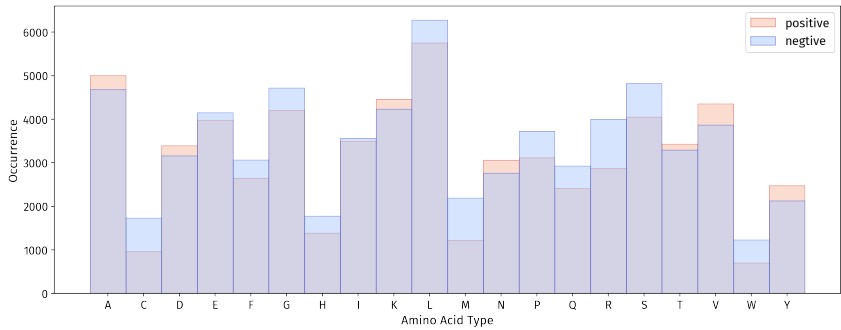

Figure 68: Amino acid distribution comparison between positive and negative samples for anti-inflammatory dataset.

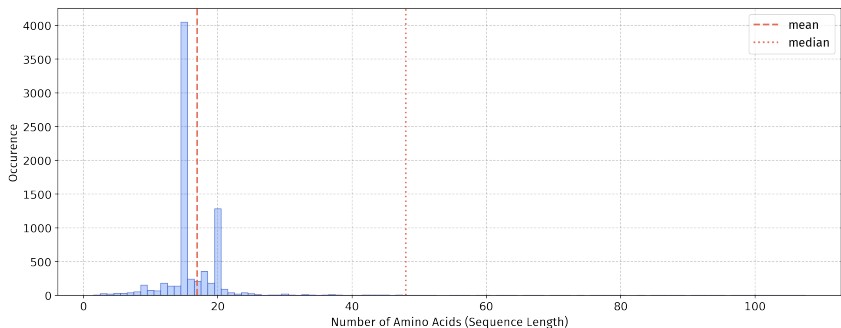

Figure 69: Length distribution of anti-inflammatory dataset.

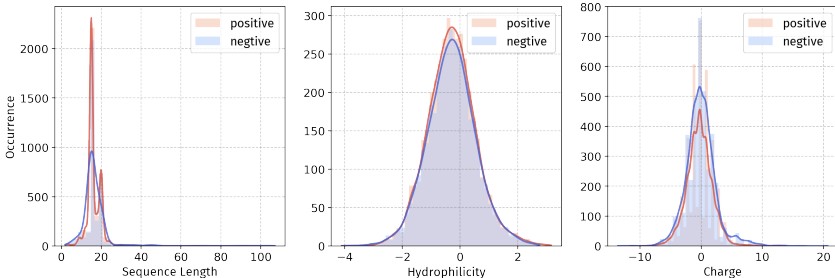

Figure 70: Property comparison between positive and negative samples for anti-inflammatory dataset.

### B.5.3  ANTIOXIDANT

• **Property and Application**:  The antioxidant dataset contains peptides that can scavenge reactive oxygen species and free radicals, protecting cells from oxidative damage. Such activity is critical for mitigating cellular injury associated with aging, chronic inflammation, and other oxidative stress-related diseases.

• **Data Source**:  The dataset is sourced from AnOxPePred (Olsen et al., 2020). The literature constructs a benchmark dataset by extracting data from published articles and the BIOPEP-UWM database (Iwaniak et al., 2024). Each peptide is binary labelled for two classes: free radical scavenger (FRS) and chelator. The classes are labeled 1 (positive) if their source have measured/indicated an activity and otherwise 0 (negative). Peptides with either class being positive are considered antioxidant peptides. To diminish homology bias while training, sequences are filtered using the Needleman–Wunsch algorithm so that no pair shared more than 90% identity, leaving 436 positive samples. In addition, 732 antioxidant peptides from Peptipedia are merged.

**Experiment Negative Samples**   Negative Samples are also collected from AnOxPePred (Olsen et al., 2020). Peptides with both classes negative are considered non-antioxidant. To reduce homology bias, negative peptides are filtered using the Needleman–Wunsch algorithm so that no pair shared more than 90% identity. After merging the two source datasets and removing 22 sequences that existed in both positive and negative samples, the final dataset contains 195 negative samples.

• **Dataset Statistics**:   The dataset contains 2,242 datapoints with sequences ranging from 2 to 11 amino acids (average length 4.10) in length.

**Task: Classification; Split: Hybrid; Evaluation: ROC-AUC**

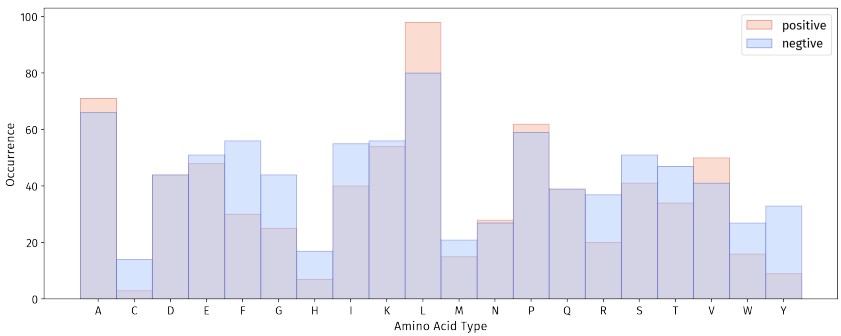

Figure 71: Amino acid distribution comparison between positive and negative samples for antioxidant dataset.

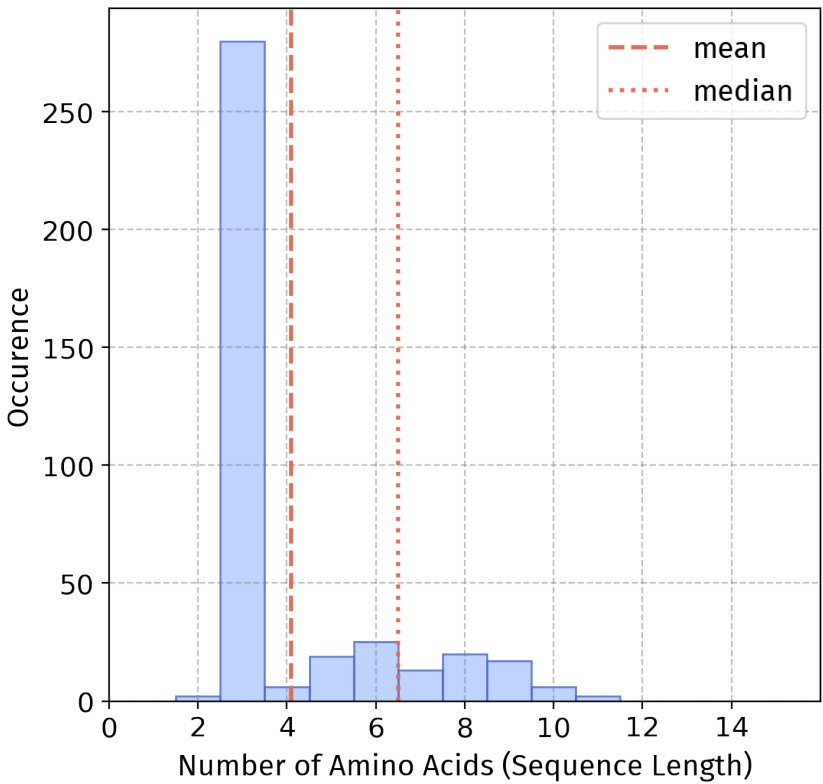

Figure 72: Length distribution of antioxidant dataset.

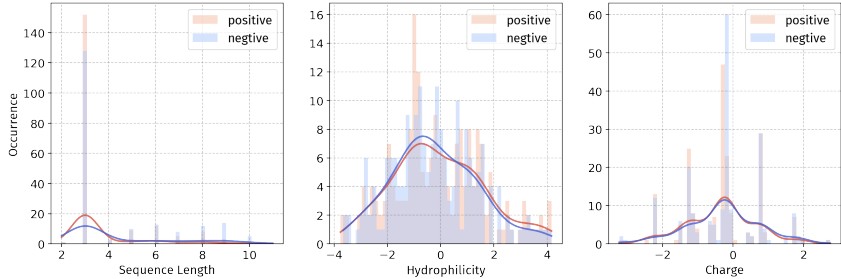

Figure 73: Property comparison between positive and negative samples for antioxidant dataset.

### B.5.4 NEUROPEPTIDE

• **Property and Application**: The neuropeptides dataset contains peptides that regulate neurotransmitter systems and neural signaling pathways, contributing to the treatment of neurological disorders including depression, epilepsy, and cognitive dysfunction.

• **Data Source**: NeuroPeptides (NPs) are sourced from Bin et al. (2020). The study derives experimentally validated NPs from the comprehensive resource of NeuroPep (Wang et al., 2024b). The positive dataset is created by three processes: (1) deleting NPs with more than 100 residues; (2) abandoning sequences with unnatural amino acids (B, J, O, U, X, and Z); and (3) removing samples with more than 90% sequence identity using CD-HIT, leaving 2425 positive samples. In addition, 2911 neuropeptides from Peptipedia are merged.

• **Dataset Statistics**: The dataset contains 9,254 datapoints with sequences ranging from 2 to 150 amino acids (average length 19.57) in length.

**Task: Classification; Split: Hybrid; Evaluation: ROC-AUC**

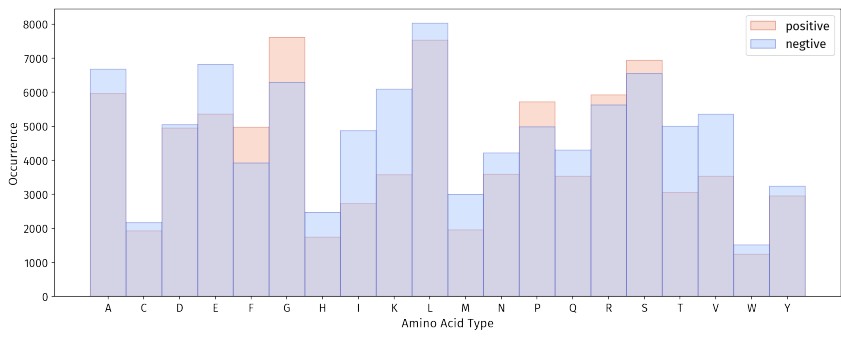

Figure 74: Amino acid distribution comparison between positive and negative samples for neuropeptide dataset.

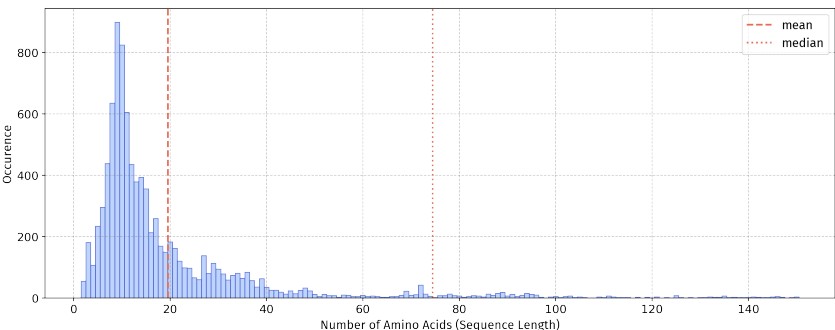

Figure 75: Length distribution of neuropeptide dataset.

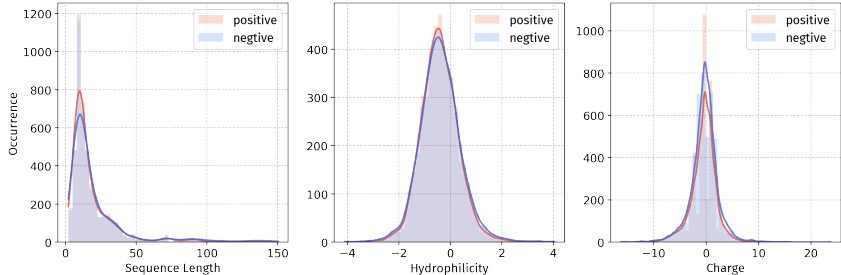

Figure 76: Property comparison between positive and negative samples for neuropeptide dataset.

### B.5.5 QUORUM SENSING

• **Property and Application**:   Quorum sensing peptides (QSPs) regulate communication between bacteria for colony formation, which may affect biofilm formation and host immune response. These peptides are important for understanding bacterial behavior and developing anti-biofilm strategies.

• **Data Source**:   QSPs are extracted from Rajput et al. (2015). The literature extractes 231 entries reported from 1955–2012 from the Quorumpeps database (Wynendaele et al., 2013). Additionally, PubMed is searched and 10 more entries are collected. 100% identical peptides are removed, leaving 218 positive samples. Additionally, 47 QS peptides from Peptipedia are merged.

• **Dataset Statistics**:   The dataset contains 490 datapoints with sequences ranging from 3 to 48 amino acids (average length 10.83) in length.

**Task: Classification; Split: Hybrid; Evaluation: ROC-AUC**

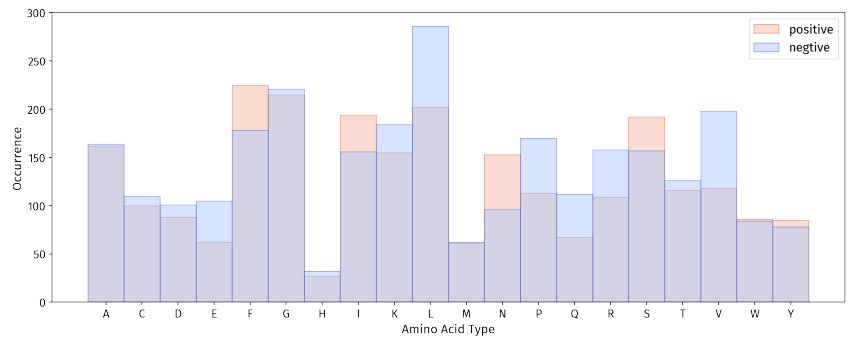

Figure 77: Amino acid distribution comparison between positive and negative samples for quorum sensing dataset.

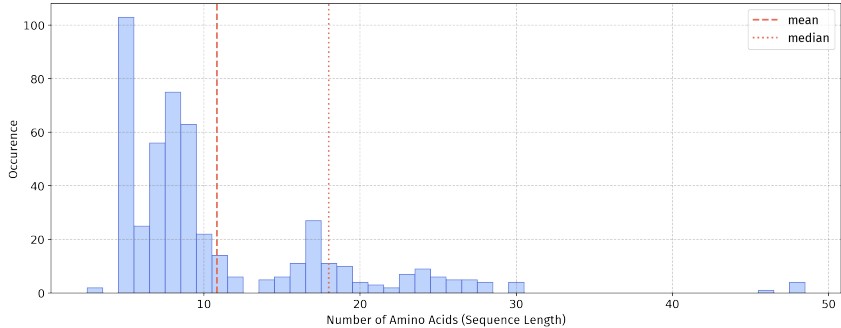

Figure 78: Length distribution of quorum sensing dataset.

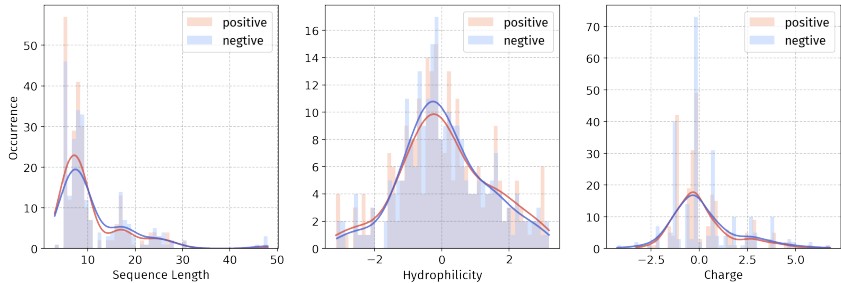

Figure 79: Property comparison between positive and negative samples for quorum sensing dataset.

## B.6 PEPPI GROUP

> **Definition.** The PepPI group comprises datasets of peptide–protein interactions.
> **Impact.** Peptides generally offer potential advantages over small-molecule drugs in target engagement, including higher binding specificity and affinity, due to their larger interaction surfaces and structural adaptability.
> **Pipeline.** Activity Modeling

### B.6.1 PPI

• **Property and Application**: The PpI dataset collects peptide-protein interaction pairs, providing a foundation for studying molecular recognition mechanisms. These data enable precision drug design by identifying peptide ligands capable of binding specific protein targets with therapeutic relevance.

• **Data Source**: The protein–peptide interaction pairs are extracted from Bhat et al. (2025) and PepNN (Abdin et al., 2022), which are both derived from the RCSB PDB by selecting peptide–protein complexes with a buried surface area $\geq 400 \ \mathring{A}^2$, peptides $\leq 25$ amino acids, and proteins $\geq 30$ amino acids.

• **Dataset Statistics**: The dataset contains 44,148 datapoints with peptide sequences ranging from 5 to 25 amino acids (average length 12.8) in length.

**Task: Classification; Split: Cold-start; Evaluation: ROC-AUC**

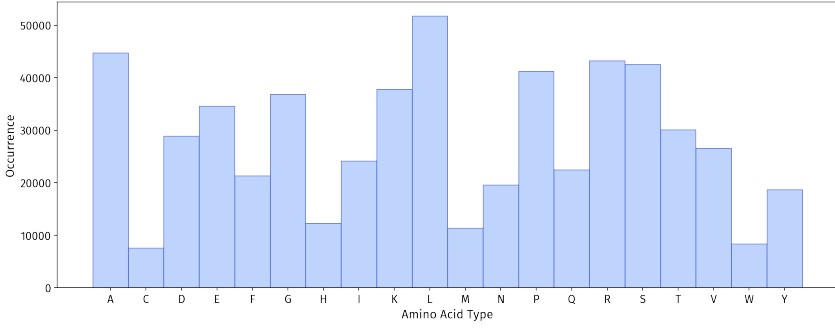

Figure 80: Amino acid distribution for PpI dataset.

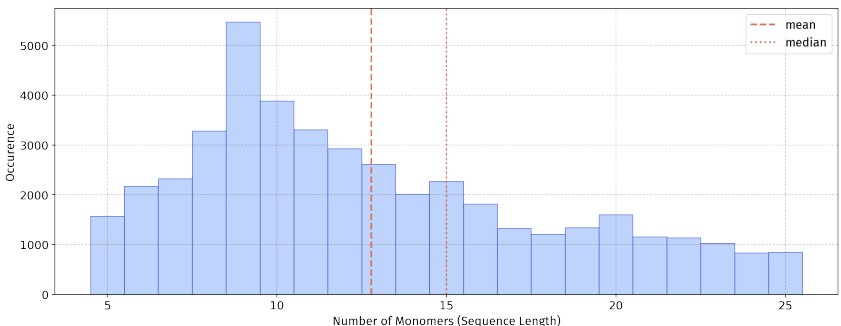

Figure 81: Length distribution of PpI dataset.

### B.6.2 PEPTIDE-PROTEIN BINDING AFFINITY (PPI_BA)

• **Property and Application**: The PpI_ba dataset provides binding affinity measurements for peptide-protein interactions, reported using metrics -lgKd(M). These quantitative values allow evaluation of interaction strength and support predictive modeling of high-affinity peptide ligands.

• **Data Source**: The dataset is sourced from Zhang et al. (2025), who constructs a benchmark for peptide–protein binding affinity prediction by following the workflow of Lei et al. (2021). The affinity data are derived from PDBbind v2019, which provides a high-quality collection of protein–ligand complex structures with experimentally measured binding affinities, all originating from the RCSB PDB.

• **Dataset Statistics**: The dataset contains 1,433 datapoints with peptide sequences ranging from 5 to 50 amino acids (average length 16.14) in length.

**Task: Regression; Split: Cold-start; Evaluation: MAE**

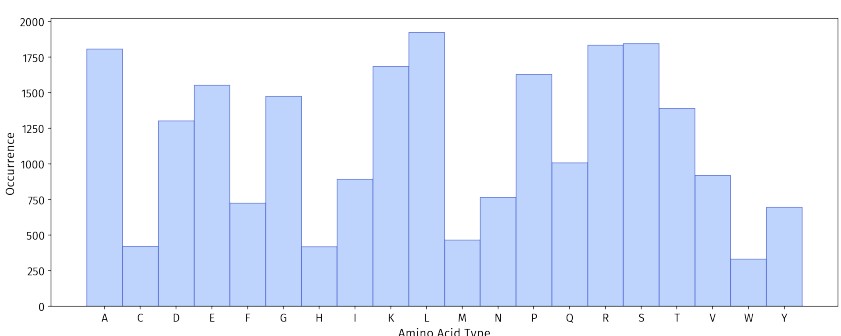

Figure 82: Amino acid distribution for PpI_ba dataset.

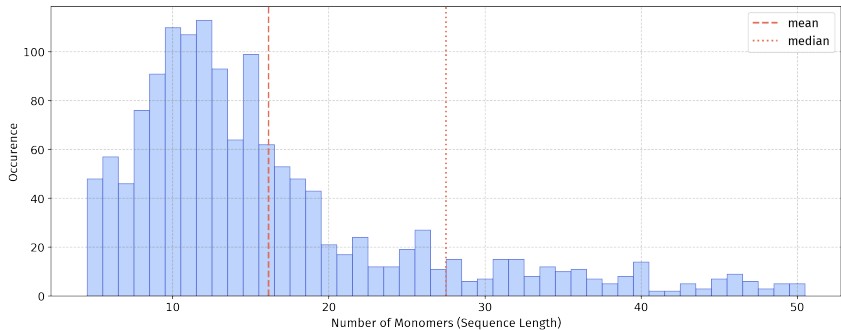

Figure 83: Length distribution of PpI_ba dataset.

### B.6.3 Non-canonical Peptide-Protein Binding Affinity (nc-PpI_ba)

• **Property and Application**: The nc-PpI_ba dataset extends binding affinity data to non-natural or modified peptides, reported using metrics -lgKd(M), enabling investigation of how chemical modifications affect peptide-protein recognition. It supports the design of synthetic ligands with improved binding properties and stability.

• **Data Source**: The dataset is sourced from Zhang et al. (2025). The construction of the non-canonical dataset follows the same workflow as the canonical dataset, but with a focus on peptides containing non-canonical amino acids.

• **Dataset Statistics**: The dataset contains 277 datapoints with peptide sequences ranging from 5 to 19 amino acids (average length 8.93) in length.

**Task: Regression; Split: Cold-start; Evaluation: MAE**

### B.7 Tox Group

> **Definition.** Datasets in this group collect datasets addressing safety-related properties of peptides. Peptide toxicity may manifest as unintended immune activation, neurotoxicity, or off-target cytotoxicity (e.g., hemolysis).
> **Impact.** This grouping is indispensable in peptide drug development, as safety liabilities often halt clinical translation even after efficacy has been demonstrated.
> **Pipeline.** Safety Assessment

### B.7.1 allergen

• **Property and Application**: The allergen dataset contains peptides that have been identified as immune-triggering epitopes, which are essential for understanding and preventing hypersensitivity reactions.

• **Data Source**: The dataset is sourced from Peptipedia.

• **Dataset Statistics**: The dataset contains 3,354 datapoints with sequences ranging from 4 to 150 amino acids (average length 39.97) in length.

**Task: Classification; Split: Hybrid; Evaluation: ROC-AUC**

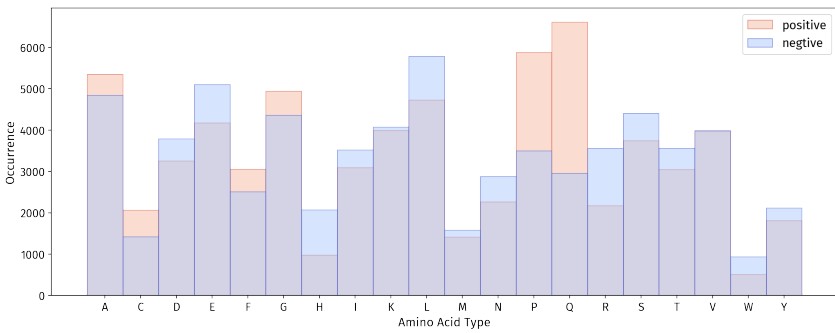

Figure 84: Amino acid distribution comparison between positive and negative samples for allergen dataset.

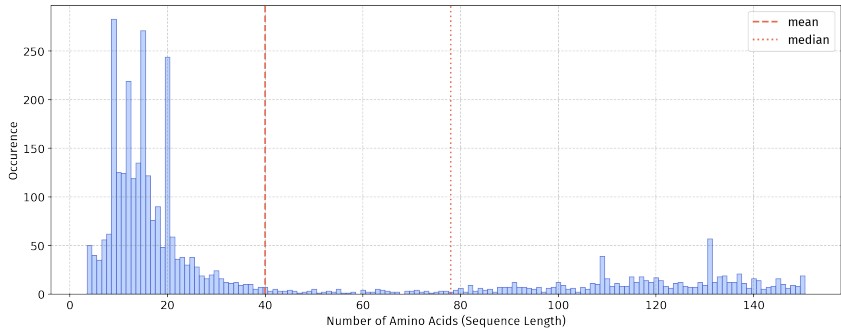

Figure 85: Length distribution of allergen dataset.

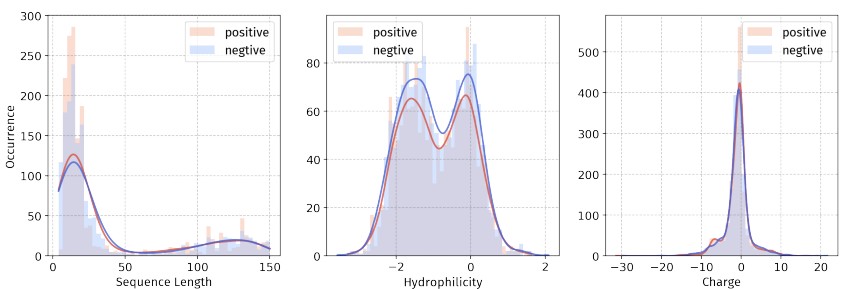

Figure 86: Property comparison between positive and negative samples for allergen dataset.

### B.7.2   HEMOLYTIC_HC50

• **Property and Application**:   The hemolytic_hc50 dataset reports hemolytic activity as HC50 values (the concentration causing 50% lysis of red blood cells). This quantitative dataset allows dose–response modeling and helps assess the severity of hemolytic potential in peptide candidates.

• **Data Source**:   The dataset is sourced from Rathore et al. (2025). The study collectes 3,147 peptides from DBAASP and 560 peptides from the Hemolytik database (Singh et al., 2025), whose HC50 values are available. Peptides containing non-natural amino acids and peptides containing less than six residues are removed. In cases where a peptide sequence has multiple HC50 values or a range of HC50 values, the average of these values is computed. The mean activity measure represents the overall hemolytic activity of the peptide under various experimental conditions. By averaging, the model captures the general behavior of the peptide's hemolytic activity rather than specific instances. HC50 values are standardized by converting them into a uniform measurement unit ($\mu$M). Following this, these HC50 values are transformed into pHC50 values.

• **Dataset Statistics**:   The dataset contains 1,926 datapoints with sequences ranging from 6 to 39 amino acids (average length 18.40) in length.

**Task: Regression; Split: Cold-start; Evaluation: MAE**

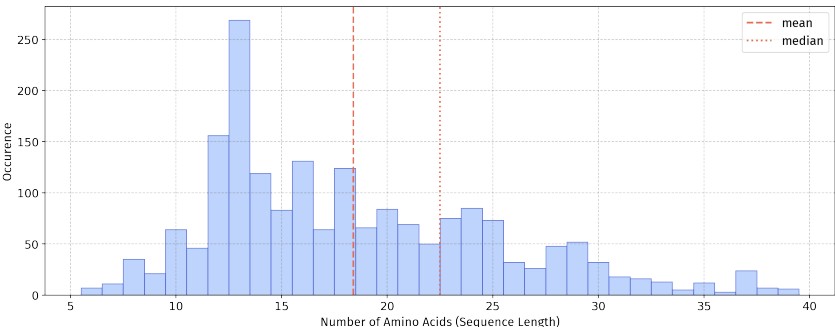

Figure 87: Length distribution of hemolytic hc50 dataset.

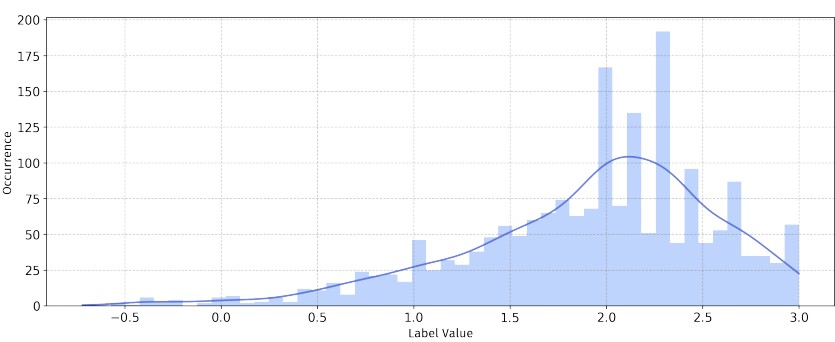

Figure 88: Label distribution of hemolytic HC50 dataset.

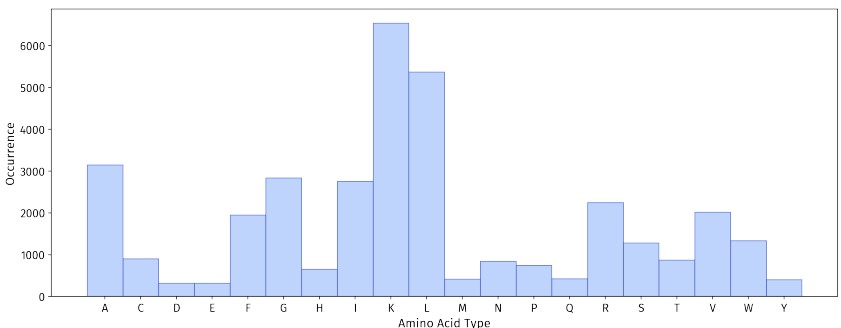

Figure 89: Amino acid distribution of hemolytic hc50 dataset.

### B.7.3 HEMOLYTIC

• **Property and Application**: The hemolytic dataset evaluates the ability of peptides to cause red blood cell lysis, serving as an indirect indicator of blood compatibility and systemic safety. It integrates sequences with hemolytic annotations from multiple sources, providing a reference for identifying peptides with undesirable cytotoxicity.

• **Data Source**: Data sources include: (1) Natural peptides stored in Hemolytik 2 (Singh et al., 2025): Data in Hemolytik 2 were manually collected from published literature and various databases. Hemolytic peptides are included in the database if those are found to be evaluated experimentally using hemolysis assay. The database gives each sequence a label (possibly "hemolytic", "low-hemolytic" and "Non-hemolytic"). Since we want to construct a relatively broad classification dataset for preliminary screening of sequences that may have hemolytic properties, "hemolytic" and "low-hemolytic" are defined as "hemolytic", and "Non-hemolytic" is defined as "non-hemolytic". Some sequences in the dataset have conflicting labels of "hemolytic", "low-hemolytic" and "Non-hemolytic". For sequences with conflicting labels, majority voting is used (the most frequent label is used as the final label; sequences with equal positive/negative label counts are removed); (2) Since 100 $\mu$M is usually used as a threshold for distinguishing compound biological activity in machine learning modeling, we additionally include sequences from hemolytic_hc50 with HC50 ¡100 $\mu$M as positive samples; (3) Hemolytic peptides recorded in Peptipedia. Duplicates are removed from different sources of positive samples.

**Experiment Negative Samples** Negative samples are samples from source (1) that are labeled as "non-hemolytic" after processing.

• **Dataset Statistics**: The dataset contains 4,512 datapoints with sequences ranging from 2 to 144 amino acids (average length 23.69) in length.

**Task: Classification; Split: Hybrid; Evaluation: ROC-AUC**

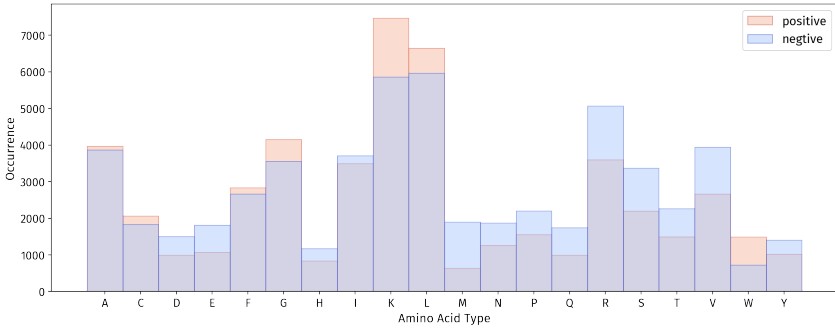

Figure 90: Amino acid distribution comparison between positive and negative samples for hemolytic dataset.

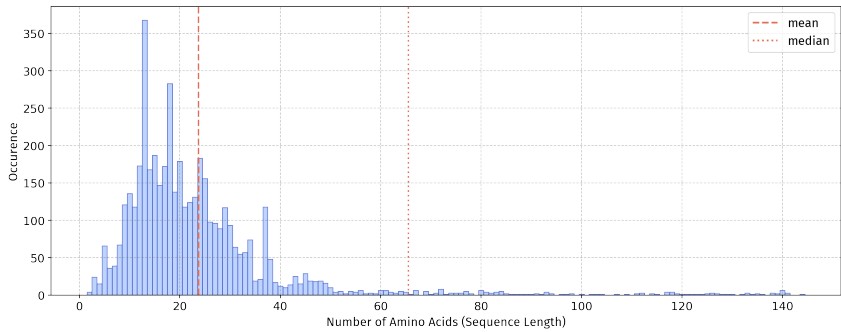

Figure 91: Length distribution of hemolytic dataset.

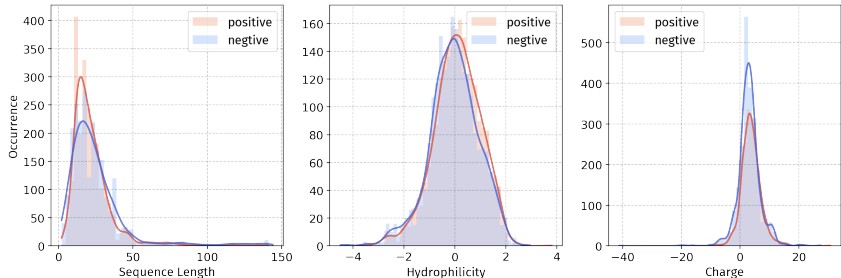

Figure 92: Property comparison between positive and negative samples for hemolytic dataset.

### B.7.4 NC-HEMOLYTIC

• **Property and Application**: The nc-hemolytic dataset extends the hemolytic dataset to include non-canonical amino acids, providing a comprehensive dataset for evaluating the hemolytic activity of non-canonical amino acids.

• **Data Source**: The dataset sourced from Hemolytik 2 (Singh et al., 2025), The determination of data labels refers to the hemolytic dataset.

• **Dataset Statistics**:

**Task: Classification; Split: ECFP-based; Evaluation: ROC-AUC**

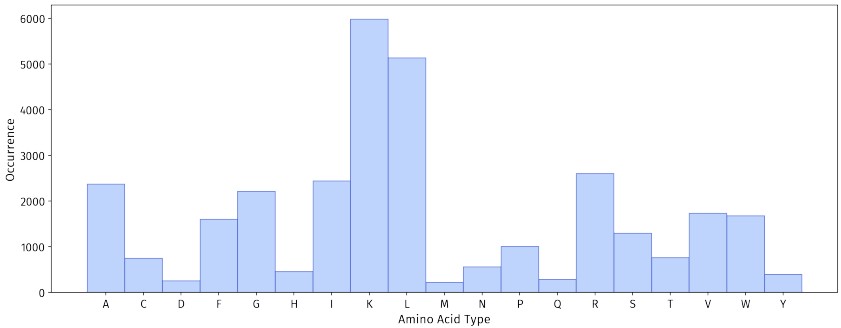

Figure 93: Canonical Amino acid distribution comparison between positive and negative samples for nc-hemolytic dataset.

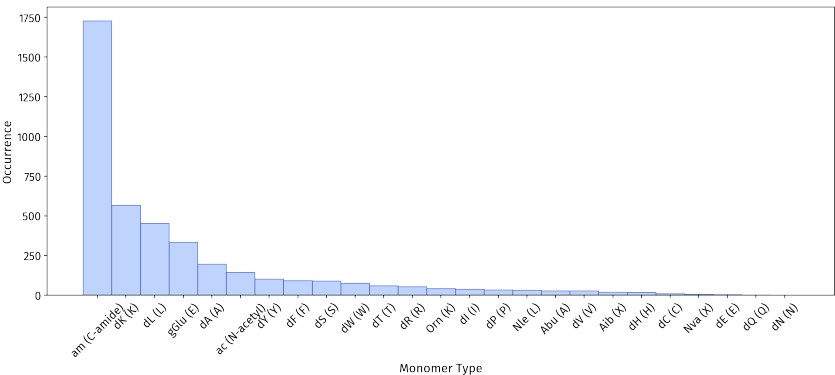

Figure 94: non-canonical Amino acid distribution comparison between positive and negative samples for nc-hemolytic dataset.

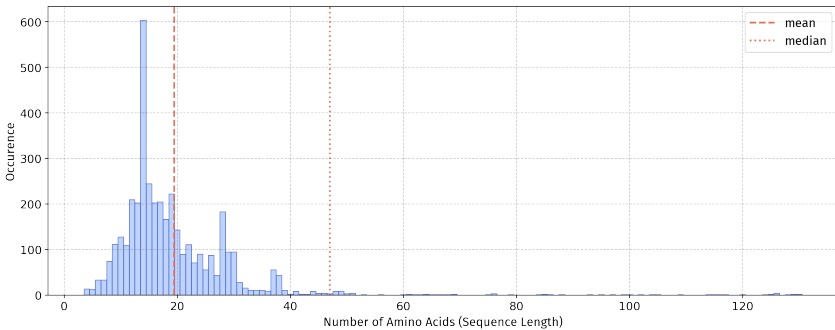

Figure 95: Length distribution of nc-hemolytic dataset.

### B.7.5 NEUROTOXIN

• **Property and Application**: The neurotoxin dataset consists of peptides with neurotoxic activity, which disrupt nervous system function by blocking ion channels or interfering with neurotransmitter release. These peptides can impair neuronal signaling and cause severe adverse effects, making them critical safety liabilities in therapeutic development.

• **Data Source**: The dataset is sourced from Peptipedia.

• **Dataset Statistics**: The dataset contains 3,506 datapoints with sequences ranging from 7 to 138 amino acids (average length 39.24) in length.

**Task: Classification; Split: Hybrid; Evaluation: ROC-AUC**

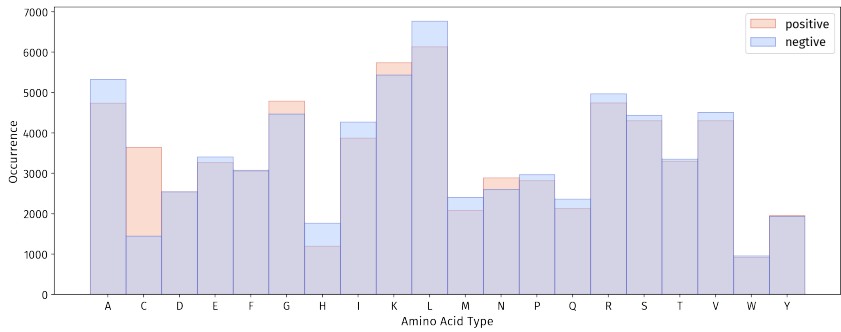

Figure 96: Amino acid distribution comparison between positive and negative samples for neurotoxin dataset.

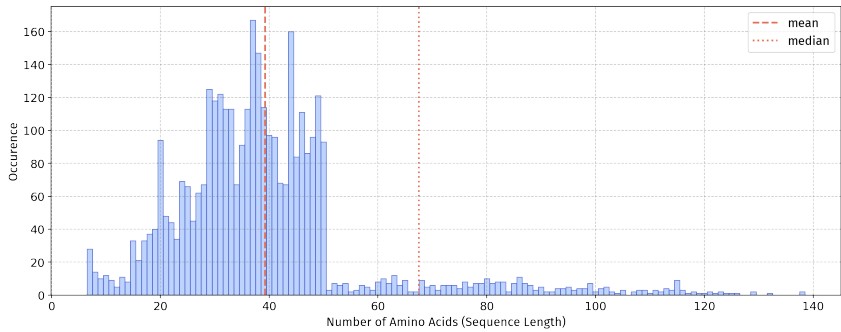

Figure 97: Length distribution of neurotoxin dataset.

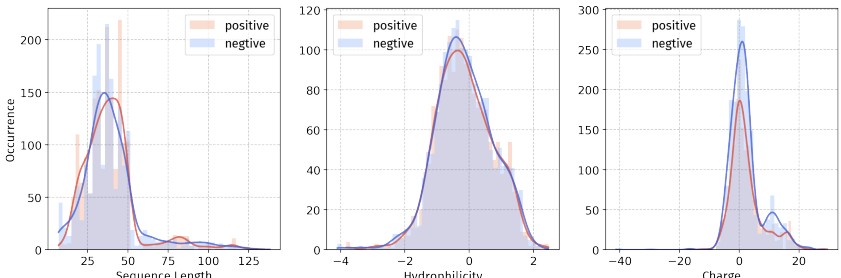

Figure 98: Property comparison between positive and negative samples for neurotoxin dataset.

### B.7.6 TOXICITY

• **Property and Application**: This dataset specifically refers to UniProt keyword 0800 annotations for peptides. These toxins are produced by animals (e.g., snakes, scorpions, spiders, cone snails), plants, fungi, and pathogenic bacteria. They act through diverse mechanisms—including neurotoxicity and ion channel disruption—and represent a broad spectrum of natural defense and predation strategies.

• **Data Source**: The dataset is sourced from Wang et al. (2025b). Positive samples consist of experimentally validated toxic peptides collected from three publicly available databases: ConoServer Kaas et al. (2008), ArachnoServer Pineda et al. (2018), and SwissProt. In the SwissProt database, toxic peptides are identified using the keyword "KW-0800". These toxic peptides range in length from 10 to 50 amino acids. After removing duplicate sequences across the three databases, a total of 3,992 toxic peptides are obtained. To further reduce model bias caused by high sequence similarity, CD-HIT is employed to remove sequences with more than 90% similarity within both the toxic and nontoxic peptide sets. This process results in a final set of 1,932 toxic peptides as positive

samples. Since neurotoxic peptides are mechanistically a type of toxicity, neurotoxin peptides from Peptipedia are merged.

• **Dataset Statistics**: The dataset contains 4,408 datapoints with sequences ranging from 7 to 138 amino acids (average length 38.41) in length.

**Task: Classification; Split: Hybrid; Evaluation: ROC-AUC**

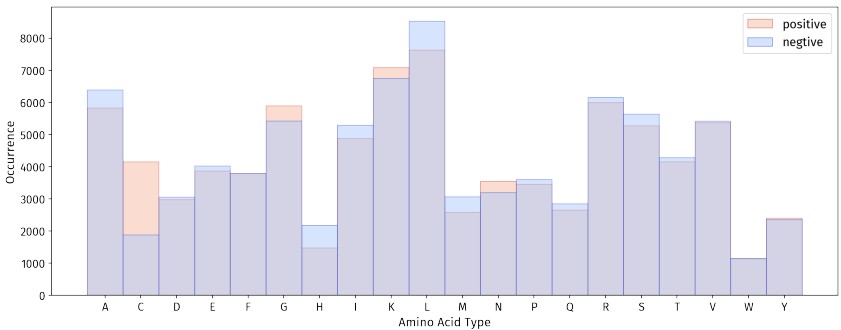

Figure 99: Amino acid distribution comparison between positive and negative samples for toxicity dataset.

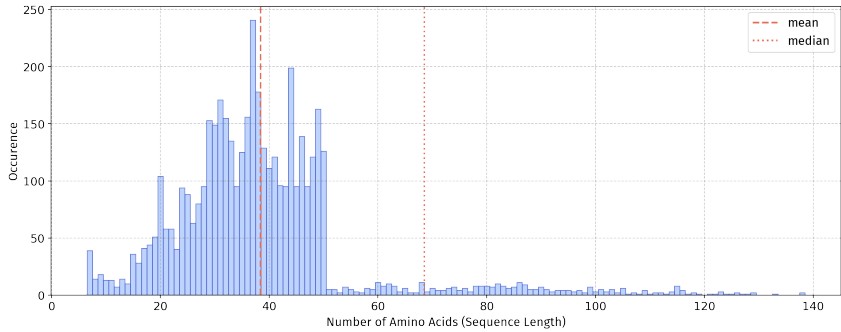

Figure 100: Length distribution of toxicity dataset.

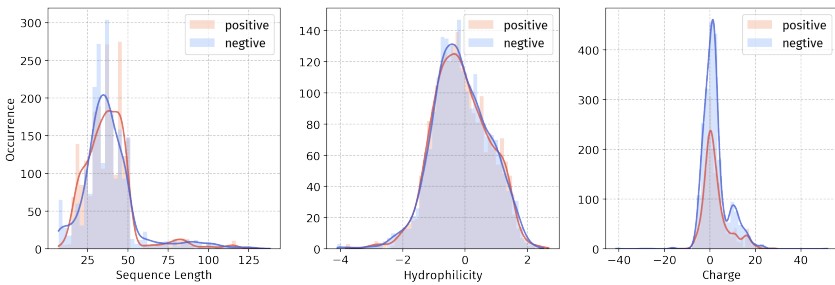

Figure 101: Property comparison between positive and negative samples for toxicity dataset.

### B.8   LENGTH LIMIT

In general, peptides are defined as sequences with fewer than 50 amino acids. However, in the literature, some sequences longer than 50 residues are also ambiguously referred to as peptides. Our analysis of the collected datasets revealed the following:

1. In 12 datasets, all sequences have lengths below 50;
2. In 18 datasets, fewer than 10% of the sequences exceed a length of 50;

3. In 5 datasets, at least 10% of the sequences are longer than 50, with maximum lengths reaching up to 150 (including datasets on antifungal, antimicrobial, antiparasitic, anticancer, allergen, neurotoxin, and toxicity).

For the benchmark, we believe it is necessary to define a unified length criterion. From the perspective of drug development experts, different research scenarios may focus on peptides of different lengths. For example, peptides with certain properties tend to be relatively short (e.g. ddpiv_inhibitors), whereas others are generally longer.

From a model development perspective, sequence length also influences model design. If the focus is only on very short sequences (e.g., fewer than 20 amino acids), methods originally developed for small molecules (e.g., graph neural networks on SMILES at the atomic level) may be applicable. Conversely, for longer sequences, their properties are more protein-like, and pretrained protein models can already capture such features effectively.

1. `PepBenchData-150`: with a maximum length set to 150, including datasets in which at least 10% of the sequences are longer than 50;
2. `PepBenchData-50`: with a maximum length set to 50.

## C  DATA PROCESSING OVERVIEW

Normalized data processing is crucial for establishing a reliable benchmark. While specific strategies may vary depending on research objectives, the goal of our study is to develop a benchmark that ensures fair and consistent model comparisons and maximizes the distinguishability between different models.

To achieve this goal, we have analyzed potential issues in the data processing procedures of previous studies that result in an oversimplified final dataset. Based on these findings, we systematically design comprehensive processing procedures for each stage. To clearly differentiate the datasets used in previous studies and the official datasets we release, we append the suffix "*raw*" to their names. Table 9 provides a summary of the raw datasets considered in this study.

Table 9: Summary of the datasets constructed in prior studies employed in analysis.

| Dataset Name | Positive Samples | Positive Sample Source | Negative Sampling Method | Used in Articles |
|---|---|---|---|---|
| ace_inhibitory_raw | 1053 | Manavalan et al. (2019) | Bioactive peptides | APML Fernández-Díaz et al. (2024) |
| antibacterial_raw | 6010 | Pinacho-Castellanos et al. (2021) | Bioactive peptides | APML Fernández-Díaz et al. (2024) |
| anticancer_raw | 861 | Agrawal et al. (2021) | Bioactive peptides | APML Fernández-Díaz et al. (2024) |
| antidiabetic_raw | 418 | Yue et al. (2024) | Random peptides | Yue et al. (2024) |
| antifungal_raw | 993 | Pinacho-Castellanos et al. (2021) | Bioactive peptides | APML Fernández-Díaz et al. (2024) |
| antimicrobial_raw | 22176 | Wang et al. (2025c) | Inactive peptides | Wang et al. (2025c) |
| antioxidant_raw | 436 | Olsen et al. (2020) | Bioactive peptides | APML Fernández-Díaz et al. (2024) |
| antiparasitic_raw | 301 | Zhang et al. (2022) | Bioactive peptides | APML Fernández-Díaz et al. (2024) |
| antiviral_raw | 2944 | Pinacho-Castellanos et al. (2021) | Bioactive peptides | APML Fernández-Díaz et al. (2024) |
| bbp_raw | 119 | Dai et al. (2021) | Bioactive peptides | APML Fernández-Díaz et al. (2024) |
| cpp_raw | 1162 | Zhang et al. (2025) | Random peptides | Pepland Zhang et al. (2025), PepTune Tang et al. (2025) |
| dppiv_inhibitors_raw | 664 | Charoenkwan et al. (2020a) | Bioactive peptides | APML Fernández-Díaz et al. (2024) |
| hemolytic_raw | 1826 | Guntuboina et al. (2023) | Experimental data | PepDoRA Wang et al. (2024a), PepTune Tang et al. (2025) |
| neuropeptide_raw | 2425 | Bin et al. (2020) | Bioactive peptides | APML Fernández-Díaz et al. (2024) |
| nonfouling_raw | 3600 | Guntuboina et al. (2023) | Insoluble and hemolytic peptides | PepDoRA Wang et al. (2024a), PepTune Tang et al. (2025) |
| quorum_sensing_raw | 218 | Rajput et al. (2015) | Bioactive peptides | APML Fernández-Díaz et al. (2024) |
| toxicity_raw | 1932 | Wei et al. (2021) | Bioactive peptides | APML Fernández-Díaz et al. (2024) |
| ttca_raw | 592 | Charoenkwan et al. (2020b) | Bioactive peptides | APML Fernández-Díaz et al. (2024) |

## D  CONSTRUCTION PIPELINE OF CLASSIFICATION DATASETS

As illustrated in Figure 102, we outline the pipeline for constructing classification datasets. In particular, we emphasize three key stages: *sequence redundancy removal*, *negative sampling*, and *dataset splitting*.

### D.1  SEQUENCE REDUNDANCY REMOVAL

In typical drug discovery pipelines, researchers often begin with a lead peptide of known bioactivity and systematically generate derivatives through point mutations, truncations, or elongations to probe critical residues and evaluate structural stability. This process inevitably results in **near-duplicate**

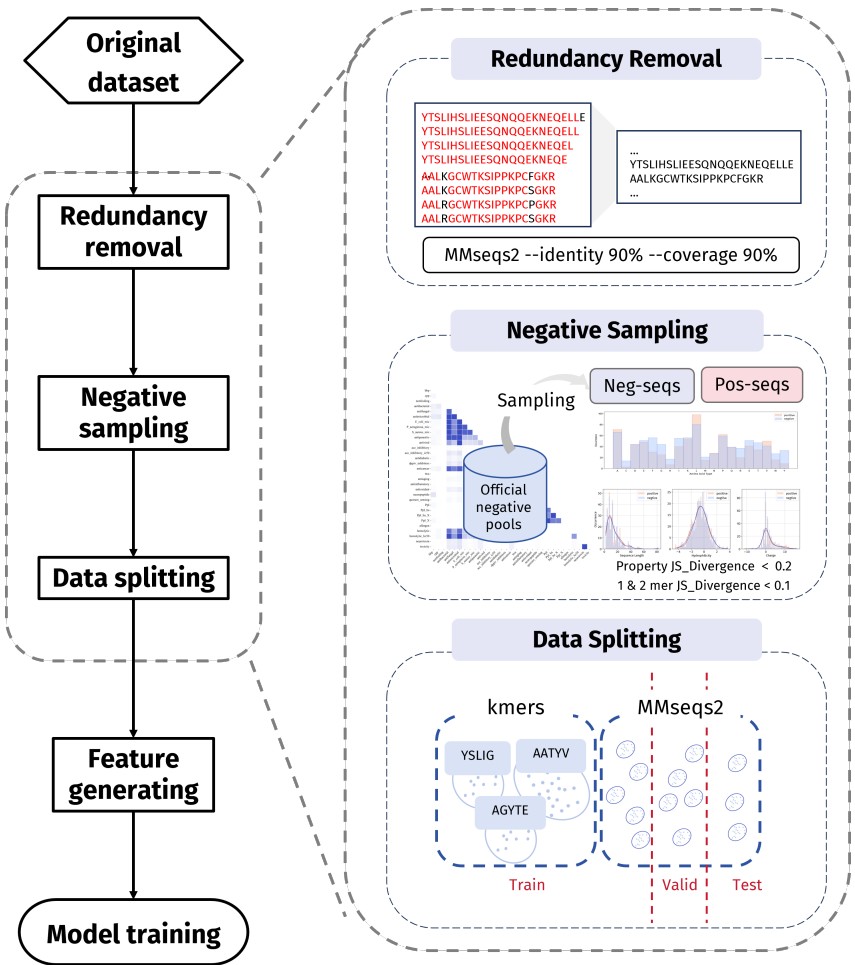

Figure 102: Overview of the construction pipeline for peptide classification datasets

**peptides**, where candidates differ by only a few amino acids. For example, within the `antifungal` dataset, several positive samples are nearly identical:

```
CIKNGNGCQPDGSQGNCCSRYCHKEPGWVAGYCR,1
CIANRNGCQPDGSQGNCCSGYCHKEPGWVAGYCR,1
CIKNGNGCQPNGSQGNCCSGCHKQPGWVAGYCRRK,1
CIKNGNGCQPNGSQGNCCSGYCHKQPGWVAGYCRRK,1
```

Near-duplicate peptides introduce a substantial source of bias in machine learning models: they promote memorization, exacerbate overfitting to trivial sequence variations, and ultimately compromise generalization. Consequently, redundancy removal constitutes a critical preprocessing step for ensuring reliable performance estimation and for improving the robustness of downstream models. Nevertheless, this issue is often overlooked in existing studies, thereby leading to inflated and potentially misleading performance. As summarized in Table 10, there are 9 benchmark datasets analyzed in this work exhibit considerable redundancy, with rates exceeding 5% at a 90% sequence identity threshold as determined by MMseqs2. To quantify the impact of such redundancy, we trained and evaluated a fingerprint-based Random Forest (RF) model on both the original and deduplicated datasets. The consistent decline in performance following deduplication demonstrates that the previously reported high performances were, at least in part, driven by model memorization of redundant sequences. This finding underscores the necessity of incorporating redundancy removal into peptide modeling pipelines. Notably, the hemolytic dataset displays an exceptionally high redundancy rate of 47%. After deduplication, its ROC–AUC decreases by 17.39%, highlighting the pronounced and detrimental influence of redundant sequences on model performance.

Table 10:  Dataset redundancy and model performance before and after deduplication. Deduped_ratio: fraction of redundant positive sequences removed; ROC–AUC: model performance on the original dataset; de_ROC–AUC: performance on the deduplicated dataset; Perf_drop: relative decrease in ROC–AUC after deduplication.

| Dataset | Deduped_ratio | ROC–AUC | de_ROC–AUC | Perf_drop (%) |
|---|---|---|---|---|
| hemolytic_raw | 0.470 | $0.805_{(0.022)}$ | $0.665_{(0.037)}$ | 17.391 |
| dppiv_inhibitors_raw | 0.050 | $0.830_{(0.049)}$ | $0.799_{(0.024)}$ | 3.735 |
| antibacterial_raw | 0.290 | $0.889_{(0.006)}$ | $0.858_{(0.017)}$ | 3.487 |
| cpp_raw | 0.270 | $0.929_{(0.005)}$ | $0.904_{(0.018)}$ | 2.691 |
| quorum_sensing_raw | 0.140 | $0.930_{(0.024)}$ | $0.913_{(0.026)}$ | 1.828 |
| anticancer_raw | 0.190 | $0.938_{(0.017)}$ | $0.918_{(0.010)}$ | 2.132 |
| antifungal_raw | 0.280 | $0.955_{(0.019)}$ | $0.944_{(0.019)}$ | 1.151 |
| antiviral_raw | 0.280 | $0.956_{(0.004)}$ | $0.942_{(0.007)}$ | 1.464 |
| antimicrobial_raw | 0.370 | $0.977_{(0.003)}$ | $0.962_{(0.002)}$ | 1.535 |

Here, we applied `MMseqs2` clustering with a set of stringent parameters specifically tailored for short peptides (sequence length $\leq$ 50 amino acids). The minimum sequence identity was set to `0.90` (`--min-seq-id 0.9`), thereby excluding peptide pairs that differed by more than $\sim$10% of residues. To ensure reliable alignments, we imposed a coverage threshold of `0.9` (`-c 0.9`) in combination with `--cov-mode 0`, which enforced that the alignment spanned at least 90% of the longer sequence in each pair and thus prevented spurious clustering of short fragments into longer peptides. Low-complexity masking was disabled (`--mask 0`) to preserve functionally relevant Cys- and Lys-rich motifs. For alignment, we employed the most informative mode (`--alignment-mode 3`) and defined sequence identity according to the aligned region without terminal gaps (`--seq-id-mode 2`), which provides robustness to local insertions and deletions. Furthermore, we enabled high sensitivity (`-s 8`) and set the number of sampled kmers per sequence to `50` (`--kmer-per-seq 50`) to maximize the detection of near-duplicate peptides. This configuration effectively clustered highly similar variants (e.g., sequences differing by single-residue substitutions, insertions, or deletions) while retaining genuinely distinct peptides.

In our sequence filtering strategy, the use of `--cov-mode 0` together with `-c 0.9` required that at least 90% of both the query and target sequences be aligned. Since the aligned region cannot exceed the length of the shorter sequence, this condition implicitly ensures that the shorter sequence spans at least 90% of the longer one, thereby preventing spurious alignments of short fragments to only a small portion of longer peptides. In addition, by combining `--min-seq-id 0.9` with `--seq-id-mode 2`, sequence identity was defined as the number of identical residues divided by the length of the shorter sequence. Under these parameters, an alignment must exhibit at least 90% identity across the shorter sequence, permitting at most $\lfloor 0.1 \times L_b \rfloor$ mismatches or gaps for a sequence of length $L_b$ (see Table 11). Taken together, these stringent criteria ensure that only sequence pairs of comparable length and high similarity were retained for downstream analysis.

Table 11: MMseqs2 clustering parameters for redundancy removal in peptide datasets.

| Parameter | Value | Meaning / Rationale |
|---|---|---|
| `--min-seq-id` | 0.90 | Minimum sequence identity (90%); removes peptides differing by >10%. |
| `-c` | 0.9 | Coverage threshold (90%). |
| `--cov-mode` | 0 | Coverage relative to longer sequence; avoids clustering of short fragments into longer peptides. |
| `--mask` | 0 | Disable low-complexity masking; retain functional motifs (e.g., Cys- and Lys-rich). |
| `--alignment-mode` | 3 | Full alignment with start, end, and identity reported. |
| `--seq-id-mode` | 2 | Identity based on aligned region without terminal gaps; robust to local indels. |
| `-s` | 8 | High sensitivity search; minimizes missed near-duplicates. |
| `--kmer-per-seq` | 50 | Extract up to 50 kmers per sequence; maximizes sensitivity for short peptides. |

We apply redundancy removal to the canonical classification datasets using `MMseqs2` with the parameters described above. Table 12 summarizes the dataset sizes before and after redundancy filtering.

Table 12: Dataset Statistics: Original and Filtered Sample Counts

| Dataset Name | Origin_pos | New_pos | Filt_ratio | Exp_neg | New_exp_neg | Filt_ratio |
|---|---|---|---|---|---|---|
| ace_inhibitory | 1833 | 1780 | 0.029 | – | – | – |
| allergen | 2405 | 1677 | 0.303 | – | – | – |
| antiaging | 282 | 279 | 0.011 | – | – | – |
| antibacterial | 21220 | 15838 | 0.254 | – | – | – |
| anticancer | 9022 | 6926 | 0.232 | – | – | – |
| antidiabetic | 1599 | 1514 | 0.053 | 75 | 75 | 0.000 |
| antifungal | 11087 | 8349 | 0.247 | – | – | – |
| antiinflamatory | 3902 | 3875 | 0.007 | – | – | – |
| antimicrobial | 42800 | 30752 | 0.281 | – | – | – |
| antioxidant | 1146 | 1121 | 0.022 | 195 | 195 | 0.000 |
| antiparasitic | 6041 | 4316 | 0.286 | – | – | – |
| antiviral | 5210 | 4134 | 0.207 | – | – | – |
| bbp | 358 | 336 | 0.061 | 13 | 13 | 0.000 |
| cpp | 1318 | 1162 | 0.118 | – | – | – |
| dppiv_inhibitors | 650 | 634 | 0.025 | 85 | 85 | 0.000 |
| hemolytic | 3096 | 2256 | 0.271 | 544 | 475 | 0.127 |
| neuropeptide | 5336 | 4627 | 0.133 | – | – | – |
| neurotoxin | 2509 | 1753 | 0.301 | – | – | – |
| nonfouling | 3600 | 3600 | 0.000 | – | – | – |
| quorum_sensing | 265 | 245 | 0.075 | – | – | – |
| toxicity | 2509 | 2204 | 0.122 | – | – | – |
| ttca | 592 | 591 | 0.002 | – | – | – |

## D.2 NEGATIVE SAMPLING

### D.2.1 LIMITATIONS OF EXISTING NEGATIVE SAMPLING METHODS

In peptide classification tasks, most datasets lack experimentally validated negative samples, and there is currently no consensus on negative sampling strategies. Existing approaches for negative sampling often suffer from inherent limitations that bias the training process:

1. **Random sampling or truncation:** Negative samples are constructed by randomly selecting peptide fragments from proteins or databases such as UniProt or SwissProt (Agrawal et al., 2021; Pinacho-Castellanos et al., 2021; Bin et al., 2020). However, the sampled sequences often differ substantially from biologically active peptides, which may lead models to learn the trivial distinction between *active peptides vs. random sequences* rather than features truly associated with the target property.

2. **Sampling from other bioactive peptides:** Some studies treat peptides with unrelated activities as negative samples. For example, antimicrobial peptides have been used as negatives for anticancer peptides (Agrawal et al., 2021), or antifungal/antiviral peptides as negatives for antibacterial peptides (Fernández-Díaz et al., 2025). The potential issue is that models may learn activity-specific differences between the two classes instead of generalizable rules relevant to the target property. APML (Fernández-Díaz et al., 2024) proposed drawing negative peptides from a database containing multiple bioactivities, but no explicit rule was defined to exclude all activities potentially overlapping with the property of interest. Even if the target property itself is excluded, mislabeling of peptides as negatives (i.e., false negatives) is likely for mechanistically related properties (e.g., antimicrobial vs. anticancer), increasing the risk of biased learning.

Another important source of bias introduced by negative sampling is uncontrolled differences between positive and negative samples. In some studies, overall distribution differences (e.g., peptide length, physicochemical properties) between positive and negative samples are not controlled. In the *nonfouling_raw* and *ttca_raw* datasets, the length distributions of positive and negative samples differ markedly, while in the *antimicrobial* dataset, the isoelectric points of positive and negative samples show significant differences ( 104). Such disparities allow models to rely on shortcut features rather than true biological signals.

### D.2.2 BIOLOGICALLY INFORMED AND DISTRIBUTION-CONTROLLED NEGATIVE SAMPLING STRATEGY

Based on the aforementioned issues, we propose two guiding principles for negative sampling:

1. Principle 1: The differences between positive and negative samples should reflect generalizable biological properties rather than dataset-specific artifacts (e.g., clear distributional differences, inherent contrasts between active and inactive peptides).

2. Principle 2·: False negatives should be avoided as much as possible.

Based on these principles, we introduce **Biologically Informed and Distribution-Controlled Negative Sampling (BDNegSamp)**. The procedure involves three key steps:

**Step 1: Construction of the Biologically Informed Negative Sample Pool**  The initial negative pool is defined as all collected bioactive peptides. Using bioactive sequences as candidate negatives avoids the trivial distinction between *active peptides vs. random sequences*, which is a major limitation of random sampling. However, not all bioactive datasets are suitable for inclusion. Many peptide classes are mechanistically related, which raises the risk of introducing false negatives if they are sampled as negatives. For example, antimicrobial peptides (AMPs) and anticancer peptides (ACPs) are strongly correlated: some ACPs have been directly identified from antimicrobial datasets. Therefore, using AMPs as negatives for ACPs would violate Principle 2 by introducing mislabeled samples. To mitigate this risk, we exclude datasets with high biological relatedness to the target activity when constructing the negative pool. This strategy ensures that negative samples are still bioactive peptides, thereby enhancing generalizability, while minimizing the chance of sampling potential positives.

To identify task-relatedness between datasets, we adopt two complementary approaches:

1. **Expert knowledge.**  Based on mechanistic insights and existing literature, we grouped peptide datasets into three clusters of highly related activities:
   - *Membrane-interaction-related activities:* antimicrobial, anticancer, hemolytic, and transmembrane peptides. Their biological functions are all closely associated with interactions at the cell membrane interface. Specifically: (i) AMPs with strong membrane-disruptive ability often also display hemolytic activity, with up to 70% of AMPs reported as toxic to human red blood cells (Qiu et al., 2025), largely due to their cationic and amphipathic structures that insert into and perturb membranes. (ii) AMPs and ACPs overlap substantially, as many ACPs share physicochemical features with AMPs, enabling them to disrupt cancer cell membranes (Roudi et al., 2017). (iii) AMPs also resemble transmembrane peptides (TMPs) in their ability to insert into lipid bilayers. At sufficient concentrations, AMPs may adopt TMP-like orientations to form transient pores, although TMPs typically form stable transport channels rather than inducing direct lysis (Pirtskhalava et al., 2013).
   - *Glucose-regulating peptides:* including DPP-IV inhibitors, antidiabetic, antioxidant, and anti-inflammatory peptides. These activities are mechanistically interrelated. For instance, DPP-IV inhibition represents a primary mechanism underlying antidiabetic peptides. Antioxidant and anti-inflammatory peptides can also contribute to glucose homeostasis by reducing oxidative stress and suppressing pro-inflammatory pathways that impair insulin secretion and action (Leo et al., 2016).
   - *Neuroactive peptides:* peptides that penetrate the blood-brain barrier or interact with neural receptors/ion channels. Their structural features (e.g., high positive charge, amphipathic motifs) promote transmembrane transport and often result in overlapping neuro-related activities.

2. **Sequence overlap statistics.** To complement expert knowledge, we compute the overlap ratio between datasets, defined as the number of shared sequences divided by the size of the smaller dataset. Datasets with an overlap ratio $> 0.05$ are considered related. The overlap analysis is visualized in Figure 103. In most cases, this statistical assessment is consistent with expert-driven groupings. In cases where datasets show high overlap but lack clear mechanistic explanation, we conservatively treat them as "highly related," thereby excluding them from negative sampling to avoid false negatives.

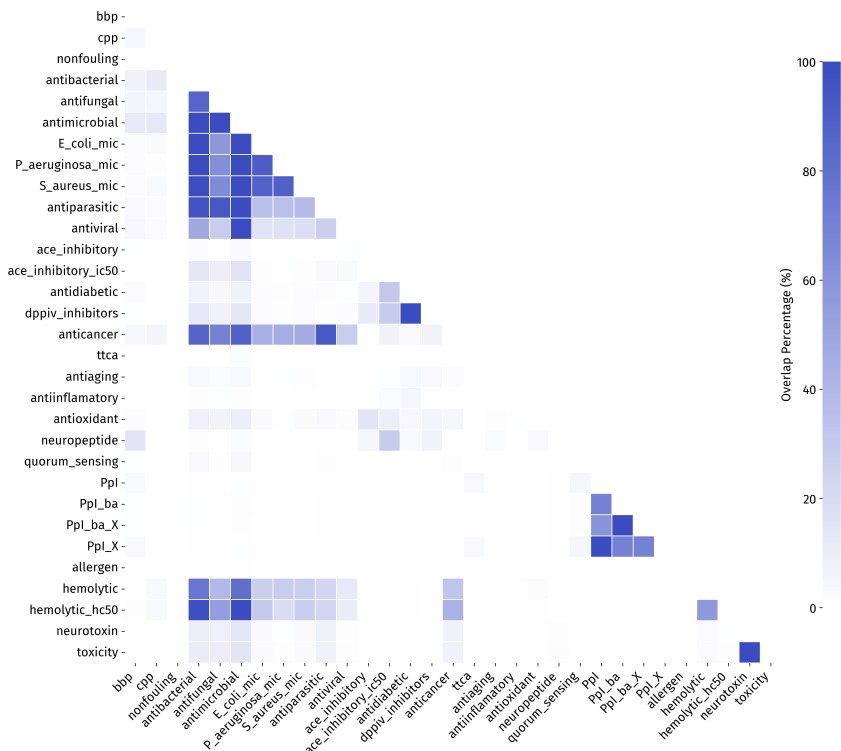

Figure 103: Dataset sequence overlap heatmap. The color intensity represents the proportion of shared sequences relative to the smaller of the two datasets.

**Step 2: Redundancy Removal and Similarity Filtering** The biologically informed negative pool may still contain a high level of redundancy, where many sequences are highly similar variants of one another. Excessive redundancy reduces the effective diversity of sampled negatives and can bias the training process. To address this, we remove redundant sequences using MMseqs2, applying clustering parameters identical to those summarized in Table 11. This ensures that the remaining pool preserves broad sequence diversity and avoids over-representing particular motifs or sequence families.

In addition to redundancy reduction, we further filter the negative pool to reduce the risk of false negatives. Based on the general principle that *similar sequences tend to share similar biological properties*, sequences in the negative pool that are highly similar to the positive set are excluded. To achieve this, we apply MMseqs2 with the same clustering parameters, except that the sequence identity threshold is lowered to 0.6 using the `--min-seq-id` parameter. This threshold is stringent enough to exclude close homologs of positive peptides while retaining more distantly related sequences that are less likely to exhibit the target activity.

**Step 3: Distribution-Controlled Sampling and Validation** After redundancy removal and similarity filtering, we obtain a refined negative pool. However, if negatives are sampled directly from this pool, they may still differ substantially from positives in overall distributional properties (e.g., sequence length, physicochemical features). Such mismatches would violate Principle 1 by allowing models to exploit trivial dataset-specific artifacts rather than learning biologically meaningful distinctions. Here, we implement a distribution-controlled sampling procedure that explicitly balances positive and negative sets across five key properties: length, net charge, hydrophobicity, amino acid composition, and dipeptide composition. For length, charge, and hydrophobicity, values are discretized into 30 bins and Jensen–Shannon (JS) divergence is constrained below 0.2. For amino acid (1-mer) and dipeptide (2-mer) distributions, stricter thresholds are applied, with JS divergence constrained below 0.05 and 0.15, respectively. This procedure ensures that the sampled negatives

are closely matched to the positives in multiple dimensions, reducing the risk of models exploiting trivial distributional artifacts. The results are shown in Table 13.

Table 13: Jensen-Shannon divergence between positive and negative samples after negative sampling for each dataset. **Length_js** measures differences in sequence length distributions; **Charge_js** quantifies differences in net charge distributions; **Hydrophobicity_js** evaluates differences in hydrophobicity distributions; **1mers_js** assesses differences in amino acid composition; and **2mers_js** measures differences in dipeptide composition.

| Dataset | Length_js | Charge_js | Hydrophobicity_js | 1mers_js | 2mers_js |
|---|---|---|---|---|---|
| ace_inhibitory | 0.115 | 0.077 | 0.075 | 0.026 | 0.069 |
| allergen | 0.249 | 0.075 | 0.054 | 0.014 | 0.039 |
| antiaging | 0.122 | 0.111 | 0.074 | 0.004 | 0.072 |
| antibacterial | 0.082 | 0.100 | 0.041 | 0.012 | 0.025 |
| anticancer | 0.082 | 0.049 | 0.041 | 0.011 | 0.025 |
| antidiabetic | 0.104 | 0.095 | 0.063 | 0.004 | 0.031 |
| antifungal | 0.059 | 0.046 | 0.040 | 0.010 | 0.023 |
| antiinflamatory | 0.123 | 0.081 | 0.037 | 0.004 | 0.010 |
| antimicrobial | 0.096 | 0.039 | 0.034 | 0.008 | 0.018 |
| antioxidant | 0.065 | 0.045 | 0.073 | 0.005 | 0.031 |
| antiparasitic | 0.081 | 0.067 | 0.040 | 0.010 | 0.024 |
| antiviral | 0.102 | 0.038 | 0.061 | 0.006 | 0.016 |
| bbp | 0.088 | 0.106 | 0.086 | 0.006 | 0.047 |
| cpp | 0.184 | 0.199 | 0.145 | 0.016 | 0.060 |
| dppiv_inhibitors | 0.167 | 0.175 | 0.122 | 0.023 | 0.115 |
| hemolytic | 0.144 | 0.106 | 0.074 | 0.013 | 0.034 |
| neuropeptide | 0.074 | 0.045 | 0.033 | 0.007 | 0.022 |
| neurotoxin | 0.167 | 0.104 | 0.061 | 0.005 | 0.016 |
| nonfouling | 0.069 | 0.056 | 0.047 | 0.028 | 0.061 |
| quorum_sensing | 0.107 | 0.140 | 0.082 | 0.009 | 0.101 |
| toxicity | 0.178 | 0.093 | 0.077 | 0.004 | 0.013 |
| ttca | 0.130 | 0.193 | 0.107 | 0.006 | 0.040 |

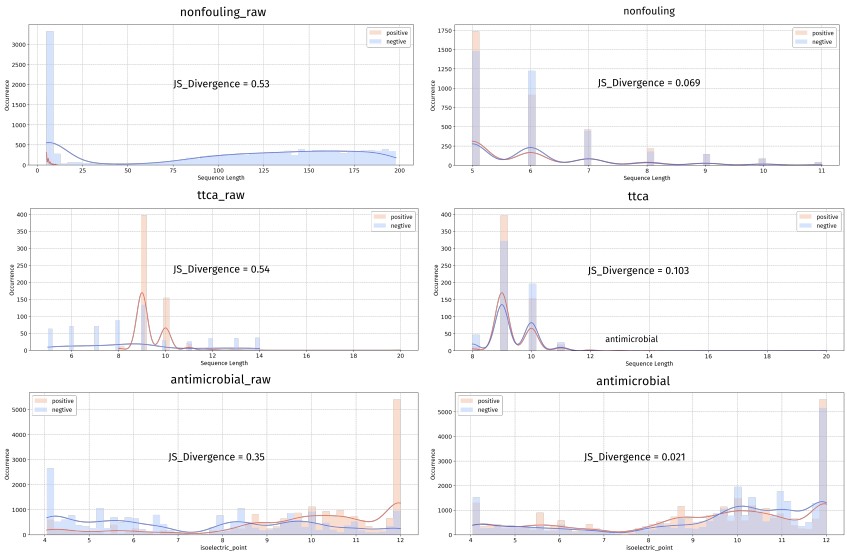

Figure 104: Distributional comparison of positive and negative samples before and after sampling.

By enforcing these constraints, the negative set is guaranteed to remain statistically comparable to the positive set across multiple dimensions. Results in Table 13 demonstrate that BDNegSamp substantially improves distributional balance compared to the original datasets, particularly for the `nonfouling`, `ttca`, and `antimicrobial` datasets (see Figure 104).

In practice, to realize distribution-controlled sampling we implemented a modular framework that supports multiple distribution-matching strategies. These include random sampling, kernel density estimation (KDE) importance sampling, maximum mean discrepancy (MMD) herding, nearest-neighbor matching, entropy-regularized optimal transport (OT), and moment or histogram-based matching. In addition, hybrid strategies that jointly balance physicochemical properties and k-mer statistics were introduced to further constrain sequence-level distributions. To ensure robustness, BDNegSamp executes multiple strategies in parallel and selects the negative set that achieves the best trade-off between distributional alignment and sample diversity. This flexible design allows BD-NegSamp to adaptively select negative samples that not only avoid false negatives but also achieve close alignment to the positive set across multiple distributional dimensions.

In some cases, such as the `antimicrobial` dataset (30,752 sequences), the bioactive negative pool ($\sim$90,000 sequences in total) is insufficient to satisfy distributional balance constraints. To address this limitation, we adopt a hybrid strategy: additional peptide sequences are drawn from UniProt to expand the candidate pool, and for each sequence length bin in the positive dataset we ensure at least a $10\times$ coverage of candidate negatives, thereby providing sufficient diversity to achieve balanced sampling. All UniProt-derived sequences are subjected to the same redundancy removal, similarity filtering, and distribution-controlled validation procedures as bioactive negatives, ensuring methodological consistency. This compromise allows BDNegSamp to generate a sufficiently diverse and balanced negative set even for very large datasets, without violating its guiding principles.

## D.3 DATA SPLITTING

### D.3.1 LIMITATIONS OF EXISTING SPLITTING PROTOCOLS

Prior peptide classification studies typically adopt either random partitioning or homology-based partitioning with `MMseqs2`. To quantify the impact of these choices, we have evaluated a broad panel of models on several widely used peptide datasets under an 8:1:1 train/validation/test split. The models include three traditional learners have been trained on 1024-bit ECFP6 fingerprints (LightGBM, Random Forest, and XGBoost) and four protein sequence models (ESM2 with 150M and 650M parameters; DPLM with 150M and 650M parameters).

Table 14: Model Performance Comparison

| Dataset | Models | | | | | | | | | | | | | |
| --- | --- | --- | --- | --- | --- | --- | --- | --- | --- | --- | --- | --- | --- | --- |
| | Random Split | | | | | | | MMseqs2 Split | | | | | | |
| | LightGBM | RF | XGBoost | ESM2_150M | ESM2_650M | DPLM_150M | DPLM_650M | LGB | RF | XGB | ESM2_150M | ESM2_650M | DPLM_150M | DPLM_650M |
| anticancer | 0.930 | 0.938 | 0.931 | 0.919 | 0.919 | 0.927 | 0.929 | 0.827 | 0.825 | 0.815 | 0.795 | 0.815 | 0.795 | 0.815 |
| antifungal | 0.937 | 0.955 | 0.933 | 0.960 | 0.974 | 0.959 | 0.975 | 0.686 | 0.670 | 0.705 | 0.798 | 0.807 | 0.759 | 0.788 |
| antimicrobial | 0.971 | 0.977 | 0.974 | 0.986 | 0.987 | 0.984 | 0.986 | 0.951 | 0.953 | 0.952 | 0.962 | 0.965 | 0.961 | 0.967 |
| antiviral | 0.923 | 0.956 | 0.929 | 0.955 | 0.955 | 0.956 | 0.959 | 0.785 | 0.814 | 0.775 | 0.838 | 0.857 | 0.843 | 0.864 |
| cpp | 0.933 | 0.929 | 0.927 | 0.917 | 0.933 | 0.936 | 0.944 | 0.862 | 0.848 | 0.855 | 0.874 | 0.884 | 0.866 | 0.888 |
| neuropeptide | 0.925 | 0.918 | 0.923 | 0.920 | 0.930 | 0.930 | 0.938 | 0.922 | **0.923** | 0.919 | 0.915 | **0.932** | 0.919 | 0.934 |
| quorum_sensing | 0.885 | 0.930 | 0.893 | 0.922 | 0.949 | 0.946 | 0.934 | 0.822 | 0.862 | 0.844 | 0.870 | 0.889 | 0.888 | **0.903** |
| toxicity | 0.910 | 0.913 | 0.908 | 0.920 | 0.920 | 0.924 | 0.923 | 0.780 | 0.769 | 0.770 | 0.843 | 0.839 | 0.846 | 0.842 |
| ttca | 0.949 | 0.941 | 0.947 | 0.969 | 0.976 | 0.971 | 0.967 | 0.868 | 0.858 | 0.861 | 0.915 | 0.938 | 0.911 | 0.940 |
| nonfouling | 0.930 | 0.929 | 0.926 | 0.924 | 0.923 | 0.921 | 0.921 | 0.903 | 0.903 | 0.900 | 0.903 | 0.898 | 0.893 | 0.893 |
| dppiv_inhibitory | 0.844 | 0.830 | 0.839 | 0.824 | 0.836 | 0.854 | 0.847 | **0.857** | **0.843** | **0.846** | **0.846** | **0.858** | **0.862** | **0.860** |
| ace_inhibitory_raw | 0.844 | 0.837 | 0.834 | 0.856 | 0.857 | 0.868 | 0.864 | **0.853** | **0.852** | **0.851** | **0.863** | **0.873** | **0.879** | **0.892** |
| antioxidant | 0.653 | 0.645 | 0.647 | 0.691 | 0.680 | 0.693 | 0.672 | **0.693** | **0.689** | **0.680** | **0.708** | **0.711** | **0.705** | **0.696** |
| bbp | 0.549 | 0.620 | 0.572 | 0.650 | 0.635 | 0.697 | 0.646 | **0.696** | **0.684** | **0.718** | **0.689** | 0.591 | 0.620 | **0.673** |

As shown in Table 14, under random splitting many datasets exhibit very high ROC–AUC scores (often exceeding 0.9), where different models show almost indistinguishable performance (e.g., *antimicrobial_raw*, *anticancer_raw*, *antiviral_raw*). MMseqs2-based homology splitting alleviates this issue to some extent, as it is generally considered a more challenging evaluation protocol that reduces the similarity between training and test sets. However, we observed that in some datasets (e.g., *ace_inhibitory_raw*, *dppiv_inhibitory_raw*), performance under MMseqs2 splitting is even higher than under random splitting. We attribute this counter-intuitive phenomenon to an overlooked issue in previous studies, namely *k*-mer leakage.

Specifically, we observes that in most peptide datasets, certain k-mers appear with very high frequency among positive samples, and these k-mers are often dataset-specific. We hypothesize that such k-mers may correspond to activity-related motifs that experts intentionally introduce during peptide design. Unfortunately, this poses a serious problem for machine learning: models can achieve inflated scores by memorizing these local shortcuts rather than learning biologically meaningful patterns, resulting in poor generalization to new active motifs. Importantly, `MMseqs2` cannot

resolve this issue, since two sequences may share the same k-mer while exhibiting low overall sequence similarity, and thus will not be clustered together.

### D.4 DETECTION OF K-MER LEAKAGE VIA ENRICHMENT ANALYSIS

We rigorously assess potential k-mer leakage by developing an enrichment analysis framework grounded in Fisher's exact test. For each candidate k-mer, a $2 \times 2$ contingency table have been constructed to quantify its distribution across positive and negative samples, and the statistical significance of its enrichment in the positive class have been evaluated. Multiple hypothesis testing was addressed using the Benjamini–Hochberg procedure for false discovery rate (FDR) control. Additional stringency criteria were imposed to ensure that only robust and biologically meaningful signals were retained. Each motif was required to surpass a minimum sample support threshold, exhibit a sufficient number of occurrences within positive sequences, and simultaneously satisfy both a stringent $p$-value cutoff and an odds ratio requirement. This multi-layered filtering strategy prioritized k-mers with strong statistical association to the activity label, while reducing the likelihood of confounding artifacts arising from dataset-specific peptide design biases rather than genuine biological patterns. We set $k = 5$ for datasets with mean sequence length $> 15$ and $k = 3$ otherwise.

#### D.4.1 A K-MER–AWARE PARTITIONING STRATEGY

Motivated by these findings, we introduce a partitioning protocol that explicitly suppresses k-mer leakage by allocating *motif clusters*, rather than individual sequences, to data splits. The procedure is:

1. Run the enrichment analysis to identify significantly enriched k-mers (henceforth *motifs*).

2. Group sequences into clusters such that any two sequences sharing at least one enriched motif are placed in the same cluster.

3. Assign clusters (not sequences) to train/validation/test according to the target ratio (8:1:1 in our experiments).

By construction, this protocol prevents any enriched motif from appearing across partitions, thereby forcing models to generalize beyond dataset-specific shortcuts and yielding a more faithful estimate of cross-motif generalization difficulty.

#### D.4.2 HYBRID-SPLIT: COMBINING K-MER AND HOMOLOGY CONSTRAINTS

While the kmer–aware partitioning addresses motif leakage, it does not exploit information about broader sequence homology among motif-free sequences. We therefore propose **hybrid-split**, a hybrid protocol that enforces both constraints:

1. Apply the kmer–aware procedure in subsubsection D.4.1 to identify motif clusters and allocate them to splits.

2. For sequences that do not contain any enriched motif, apply `MMseqs2` to form homology clusters and allocate these clusters to the existing splits, maintaining the desired proportions.

For MMseqs2-based homology splitting, parameter settings also need to be standardized. By examining the changes in the number of isolated sequences under different identity thresholds (see Figure 105), we find that when the threshold is set to 0.3, the sequences in the dataset can be clustered to the greatest extent. Therefore, in our study, 0.3 is selected as the clustering threshold for MMseqs2.

Hybrid-split simultaneously blocks kmer leakage and reduces global homology between splits, providing a more stringent and biologically grounded evaluation. In practice, we find that hybrid-split mitigates shortcut exploitation and yields more discriminative performance estimates than random or homology-only partitioning, particularly on datasets where enriched motifs are prevalent.

**Practical Considerations.** Stricter partitioning can reduce effective training size—especially in small datasets—and may lower absolute scores. However, this trade-off reflects a closer alignment

with real-world generalization, where future peptides may lack the same dataset-specific motifs or share low global similarity with training examples. We therefore recommend hybrid-split as a default evaluation protocol for peptide classification tasks, and we report all results under both conventional splits and hybrid-split to facilitate transparent comparison. The experimental results under different partitioning methods can be found in H.

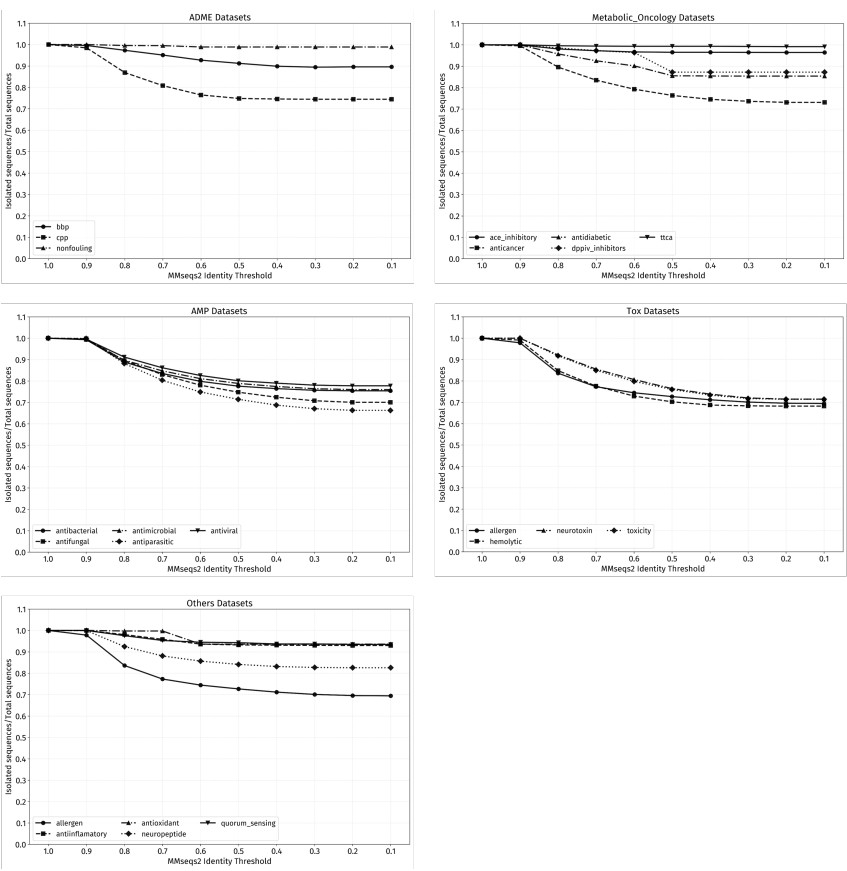

Figure 105: Isolated Sequences Ratio vs MMseqs2 Identity Threshold (from 0.1 to 1.0) Across classification Datasets.

### D.5 PIPELINE OF NON-CANONICAL DATASET CONSTRUCTION

The workflow described above applies to canonical peptide datasets. For non-canonical peptides, however, several important distinctions must be considered. We analyzed the non-canonical datasets with respect to the same three key issues as in canonical datasets.

**Sequence Redundancy**  Non-canonical peptides are usually generated from canonical peptides through subtle chemical modifications (e.g., N-terminal acetylation, C-terminal amidation), resulting in sequences that are highly similar to their canonical counterparts. Removing redundancy in this case would discard a substantial number of meaningful samples. Therefore, we do not apply redundancy removal to non-canonical peptides. Instead, to mitigate potential data leakage, we relied on carefully designed data-splitting strategies described below.

**Negative Sampling**  Similar to canonical datasets, non-canonical peptide classification datasets lack experimentally validated negative samples. Unfortunately, the number of available non-canonical sequences is too limited to serve as a sufficiently large negative sampling pool. To address this challenge, we have trained a generative model using 9,512 known non-canonical peptides (see Appendix I). This model takes a canonical peptide as input and outputs a chemically modi-

fied non-canonical peptide, allowing us to transform a canonical peptide–based negative pool into a non-canonical negative pool.

Concretely, we first mapped non-canonical peptides to their canonical equivalents by replacing each non-canonical residue with its closest natural homolog (e.g., N-methyl-L-alanine → Alanine), and then applied the negative sampling strategy established for canonical peptides. Using the more than 9,500 non-canonical sequences in our benchmark, we trained a converter capable of mapping canonical sequences back into their non-canonical forms. This converter was subsequently used to transform the sampled canonical negatives into non-canonical negative samples. Since non-canonical amino acids generally share similar physicochemical properties with their canonical homologs, we argue that controlling the distribution of homologous canonical sequences effectively ensures that the property distribution of positive and negative non-canonical peptides is also well controlled.

**Data Splitting**   Non-canonical peptides (with unnatural residues, stereochemical inversions, covalent modifications, cyclizations, capping, etc.) are not faithfully handled by sequence-only formats and tools that assume the standard 20-letter alphabet. To prevent near-duplicate leakage under this setting, we perform a ECFP-based split. Concretely, we compute ECFP fingerpring for each peptide and use a Tanimoto similarity $s_{ij}$ between every pair $(i,j)$. We then construct a similarity graph $G = (V, E_\tau)$ where $(i,j) \in E_\tau$ iff $s_{ij} \geq \tau$. The connected components $\{C_k\}$ of $G$ are treated as clusters, and we assign clusters—rather than individual samples—to the training, validation, and test sets. The similarity threshold is set to $\tau = 0.95$. This cluster-level assignment preserves distributional balance while minimizing cross-split redundancy. Given our dataset size, the $O(N^2)$ similarity computation is tractable.

For **larger datasets**, a scalable alternative is to map each non-canonical site to an ambiguity token "X" and then apply standard sequence-based partitioning pipelines to the *X-collapsed* sequences (e.g., MMseqs2-based splitting or hybrid-style cluster splitting). This leverages mature sequence tools for efficiency, albeit at the cost of potentially conflating chemistry that is distinct but collapsed by "X". We provide a quantitative comparison between the ECFP-based split and the X-collapsed, sequence-based splits (MMseqs2-split, hybrid-split) in Appendix H.

# E   CONSTRUCTION PIPELINE OF REGRESSION DATASETS

As illustrated in Figure 106, we outline the pipeline for constructing regression datasets, focusing on three critical issues: *outlier removal*, *redundancy*, and *data splitting*.

.

## E.1   OUTLIER REMOVAL

Many datasets are collected from heterogeneous literature sources, where the same peptide sequence may correspond to multiple experimental measurements due to variations in experiment conditions. For regression tasks, outlier detection is often neglected, and existing studies usually compute the mean of all available measurements (Huang et al., 2023). We argue that this approach is inappropriate.

For example, in the dataset `S.aureus_mic_raw`, the sequence `GILSSIKGVAKGVAKNVAAQLLDTLKCKITGC` has five records: 0.778, 0.602, 0.778, 1.792, and 0.602. The value 1.792 is clearly an outlier; therefore, it should be removed before averaging the remaining values. Without such filtering, outliers may degrade data quality and compromise model reliability. This issue is particularly prevalent in **antimicrobial peptide (AMP) datasets** and **Peptide–protein interaction datasets**, where minimum inhibitory concentration (MIC) and peptide–protein binding affinity (Kd) are often repeatedly measured.

To address this, we apply an **interquartile range (IQR)-based method** for regression datasets. Specifically, values outside the range

$$[Q1 - 1.5 \times IQR, Q3 + 1.5 \times IQR]$$

are removed, and the mean of the remaining values is taken as the final label. We apply this procedure to all regression datasets containing duplicate samples, which exclude many potentially confounding values (Table 15).

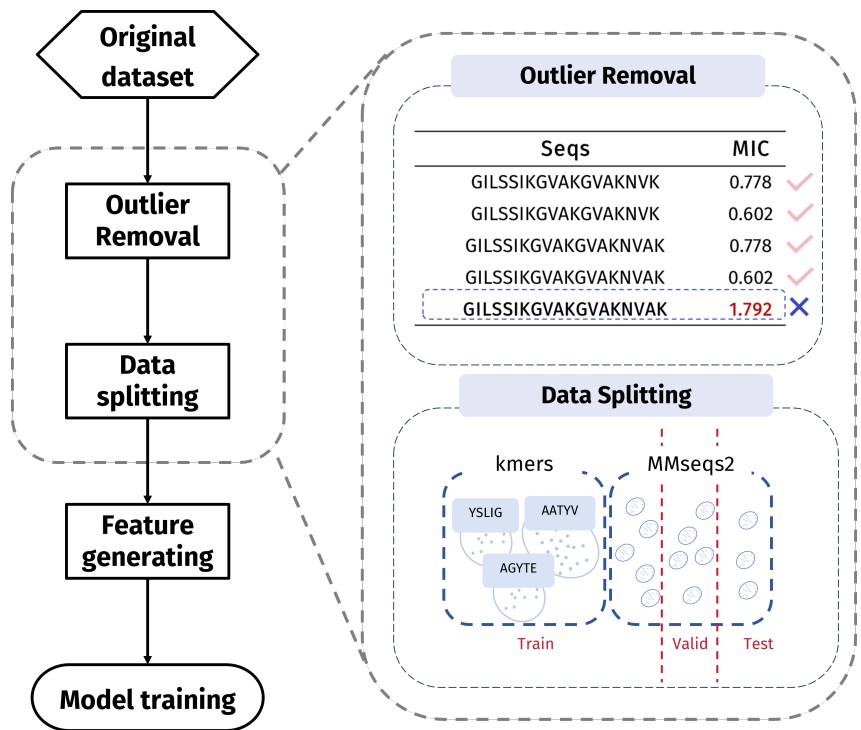

Figure 106: Overview of the construction pipeline for peptide regression datasets

Table 15: Outlier removal statistics for regression datasets. Outlier_remove: number of data points identified as outliers.

| Dataset | Sample_count | Outlier_remove | Unique_sample_count |
|---|---|---|---|
| S.aureus_mic | 5,070 | 1,167 | 2,900 |
| E.coli_mic | 5,465 | 1,231 | 3,312 |
| P.aeruginosa_mic | 2,523 | 500 | 1,618 |
| PpI_ba_X | 1,806 | 0 | 1,709 |
| nc-PpI_ba_X | 286 | 0 | 277 |

### E.2 REDUNDANCY IN REGRESSION DATASETS

Regression datasets also contain a large number of nearly identical peptide sequences. However, unlike classification tasks, we argue that **redundancy removal is not necessary for regression tasks**. This is because although highly similar sequences often show different measured values, these differences provide informative variations that benefit model learning. In other words, retaining similar sequences helps the model to capture subtle sequence–activity relationships rather than discarding them.

### E.3 DATA SPLITTING

In our collected datasets, the labels cover a very large number of values, so there is no obvious risk of k-mer leakage as in typical classification tasks. However, high-frequency k-mers can still carry meaningful information. For this reason, using a k-mer–aware hybrid-split remains valuable. In our experiments, we also adopt the hybrid-split strategy. Similar to the classification datasets, we find that when the MMseqs2 identity threshold was set to 0.3, the sequences in the regression datasets can be clustered to the greatest extent ( 107). Therefore, 0.3 was selected as the clustering threshold for MMseqs2.

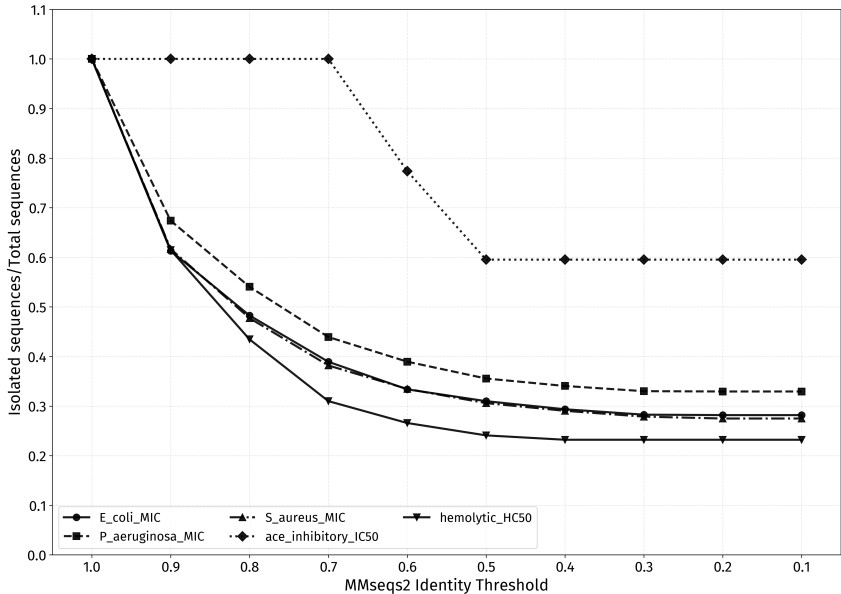

Figure 107: Isolated Sequences Ratio vs MMseqs2 Identity Threshold (from 0.1 to 1.0) Across Regression Datasets.

Table 16: Statistics of datasets with special amino acids

| Dataset | Size |
|---------|------|
| PpI_X | 8570 |
| PpI | 7358 |
| PpI_ba | 1433 |
| PpI_ba_X | 1709 |

## F    CONSTRUCTION PIPELINE OF OF PEPTIDE-PROTEIN INTERACTION DATASETS

We describe here the data processing procedures for the Peptide–Protein Interaction (PPI) datasets. Before processing, we recognized that the datasets contained special amino acids that do not belong to the 20 canonical amino acids, such as X (unknown amino acid) and U (selenocysteine). For example, `PpI_X` and `PpI_ba_X` contain 1,211 and 276 samples with "X," respectively. In addition, `PpI_X` includes one peptide sequence containing "U" (GIVEQCUASVCSLYQLENYCNM). While `lm-base` models can accept such special symbols (treating them as "unknown" tokens), methods that rely on molecular fingerprints or descriptors computed from canonical amino acids cannot handle this type of input.

To address this issue, we provide two versions of the datasets. Since only `PpI_X` contains a single sequence with "U" we directly removed this sequence. For "X" we generated paired datasets: `PpI_X` and `PpI`, as well as `PpI_ba_X` and `PpI_ba`, where the suffix "X" indicates the inclusion of sequences containing "X" (Table 16). As the proportion of sequences with "X" is relatively small, we recommend using the versions without "X" i.e., `PpI` and `PpI_ba`, in most cases.

.

Both canonical and non-canonical Peptide-Protein Interaction Datasets were the processed using the following steps(illustrated in Figure 108).

### F.1    NEGATIVE SAMPLING

For the collected datasets, only experimentally validated binding pairs were used to construct positive samples. Following the approach of CAMP (Lei et al., 2021), negative samples were generated

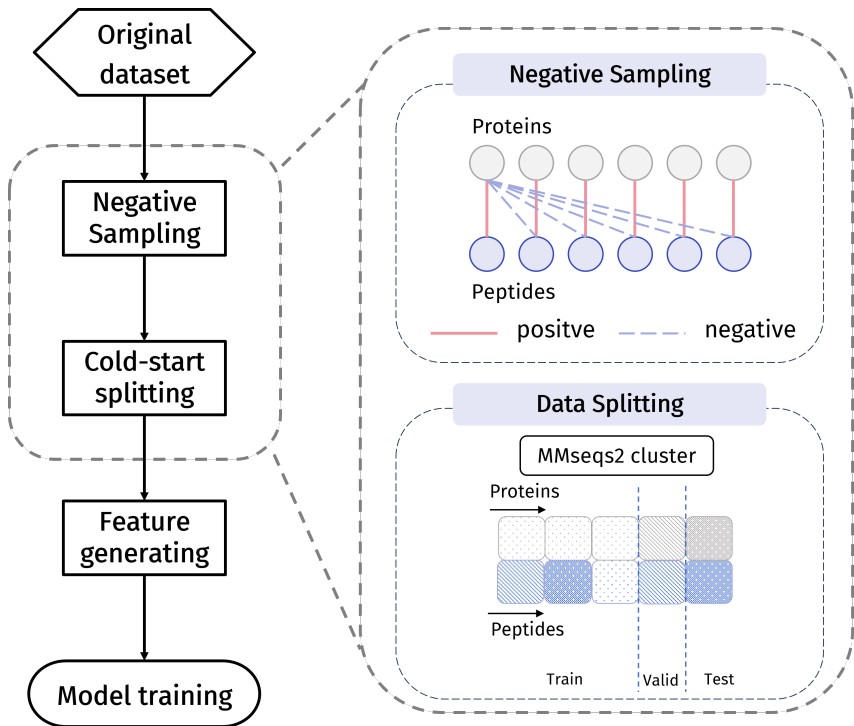

Figure 108: Schematic diagram of the classification dataset construction.

by randomly shuffling the protein–peptide pairs. Each positive sample was paired with five corresponding negative samples.

## F.2 COLD-START SPLITTING

For the `PepPI` task, we follow the standard paradigm in prior studies, where proteins serve as the basis for data splitting. Specifically, proteins are clustered based on sequence homology using `MMseqs2`, and a cold-start setting is enforced: proteins appearing in the training set are excluded from the test set.

## G EXPERIMENTAL DETAILS

### G.1 HYPERPARAMETER SETTINGS FOR FOUR MODEL FAMILIES

Given the scale of our experiments—35 datasets, 20 models, and 5 repetitions per experiment—exhaustive hyperparameter optimization for every dataset–model pair is computationally infeasible. Instead, we adopt a pragmatic strategy: for each family of models, we select a subset of representative datasets and conduct hyperparameter optimization using `Optuna`, which combines TPE-based probabilistic modeling with pruning to efficiently explore configurations under limited budgets. The resulting hyperparameters are then applied uniformly across all datasets.

**PLM-based and SMILES-based Models.** For PLM-based and SMILES-based models, we adopt the unified configuration in Table 17.

**GNN-based Models.** The hyperparameter settings for GNN-based models are shown in Table 18.

**Traditional Machine Learning Models.** For traditional models (Random Forest, XGBoost, and LightGBM), the default hyperparameters already provided competitive performance, so we did not perform additional tuning. The software versions are:

Table 17: Hyperparameters for PLM-based and SMILES-based models

| Parameter | Value | Description |
| --- | --- | --- |
| epochs | 50 | Maximum number of training epochs |
| batch_size | 64 | Training batch size |
| learning_rate | 5e-5 | Learning rate |
| early_stopping_patience | 5 | Stop if validation loss stagnates for 5 epochs |
| weight_decay | 0.0 | No regularization |

Table 18: Hyperparameters for GNN-based models

| Parameter | Value | Description |
| --- | --- | --- |
| epochs | 50 | Maximum number of training epochs |
| batch_size | 64 | Training batch size |
| num_layers | 3 | Number of GNN layers |
| emb_dim | 300 | Dimension of hidden embeddings |
| learning_rate | 0.001 | Learning rate |

- LightGBM 4.6.0

- XGBoost 3.0.2

- scikit-learn 1.7.0

**PPI Task Settings.** For the PPI prediction task, we apply a unified configuration across all models, summarized in Table 19.

Table 19: Hyperparameters for PPI task

| Parameter | Value | Description |
| --- | --- | --- |
| epochs | 50 | Maximum number of training epochs |
| batch_size | 16 | Smaller batch size due to memory usage |
| learning_rate | 1e-4 | Learning rate |

### G.2 FINE-TUNING ESM-150M-F

The ESM2 model is pretrained on UniRef Suzek et al. (2015), spanning $\sim$43M UniRef50 clusters and $\sim$138M UniRef90 sequences, totaling about 65M unique proteins. However, short peptides are severely underrepresented: in UniRef50, only **2.8%** of sequences are shorter than 50 residues, while most fall between 51–600. Consequently, model updates are dominated by medium- and long-length peptides, leading to poor modeling of short peptides.

**Dataset Preparation.** To address this, we constructed a short-peptide dataset by truncating UniRef50 (as of April 2025) to sequences of $\leq$50 residues, removing invalid entries, and obtaining 1,932,360 sequences. This dataset, denoted uniref50_50, was split into training and validation sets at a 9:1 ratio. Figure 109 and Figure 110 illustrate the sequence length and amino acid distribution.

**Training Setup.** We fine-tuned ESM-150M using DeepSpeed for distributed training and BF16 mixed precision for efficiency. Training was conducted on 8 A800 GPUs with a per-device batch size of 512 (effective batch size: 4096). The hyperparameters are summarized in Table 20.

**Evaluation Metric.** We employed **pseudo-perplexity (PseudoPPL)** to assess model performance on peptides. Unlike autoregressive models, where perplexity is computed as

$$\text{PPL}(x) = \exp\left(-\frac{1}{L}\sum_{i=1}^{L}\log p(x_i \mid x_{<i})\right),$$

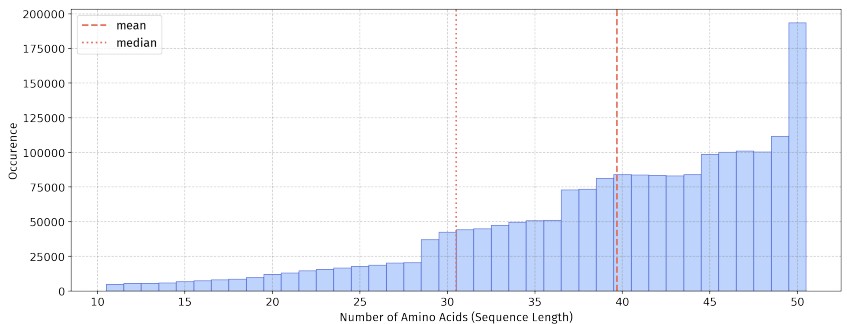

Figure 109: Length distribution of the `uniref50_50` dataset.

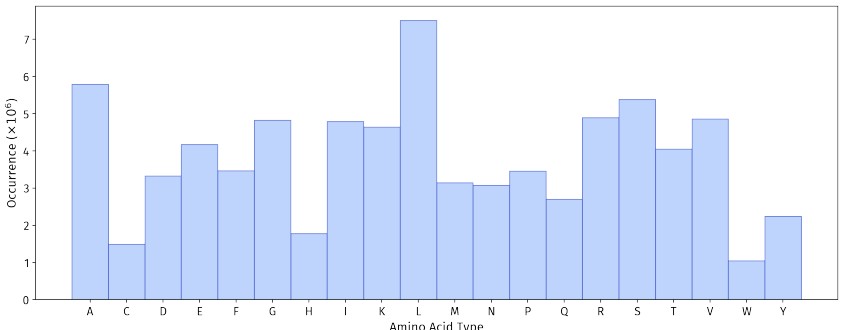

Figure 110: Amino acid distribution of the `uniref50_50` dataset.

masked language models require the following variant:

$$\text{PseudoPPL}(x) = \exp\left(-\frac{1}{L}\sum_{i=1}^{L}\log p(x_i \mid x_{j\neq i})\right).$$

We used 85,113 unique peptide sequences absent from UniRef100 for evaluation. As shown in Fig. 111, ESM-150M suffers from high perplexity on short peptides, with performance deteriorating as length decreases. In contrast, ESM-150M-F consistently achieves lower perplexity, confirming the effectiveness of fine-tuning.

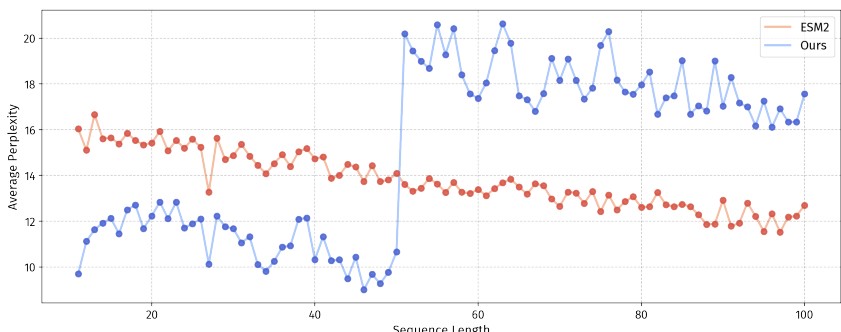

Figure 111: Perplexity of ESM-150M (ESM2) and fine-tuned ESM-150M-F (ours) across peptide lengths.

**Summary.** Fine-tuning on short peptides substantially improves ESM-150M's ability to model short sequences, as evidenced by reduced perplexity and overall better performance.

Table 20: Hyperparameters for fine-tuning ESM-150M (ESM-150M-F)

| Parameter | Value | Description |
|---|---|---|
| learning_rate | 4e-4 | Learning rate |
| num_train_epochs | 500 | Training epochs |
| per_device_train_batch_size | 512 | Batch size per device |
| gradient_accumulation_steps | 1 | Gradient accumulation steps |
| warmup_steps | 2000 | Learning rate warm-up steps |
| weight_decay | 0.01 | Weight decay |
| max_grad_norm | 5.0 | Gradient clipping threshold |
| optimizer | AdamW | Optimizer type |
| adam_beta1 | 0.9 | Adam $\beta_1$ parameter |
| adam_beta2 | 0.98 | Adam $\beta_2$ parameter |
| adam_epsilon | 1e-8 | Adam $\epsilon$ parameter |
| lr_scheduler_type | linear | Learning rate scheduler |
| eval_steps | 500 | Evaluation frequency |
| save_steps | 500 | Checkpoint saving frequency |
| distributed_type | DeepSpeed | Distributed backend |
| num_processes | 8 | Number of processes (GPUs) |
| num_machines | 1 | Number of machines |
| zero_stage | 2 | ZeRO optimization stage |
| bf16 | true | Use BF16 mixed precision |
| seed | 42 | Random seed |

## H  ADDITIONAL RESULTS

In this section, we provide detailed experimental results, including:

1. Results on 22 canonical peptide classification datasets under four splitting strategies: *hybrid-split* (Table 21), *MMseqs2-split* (Table 23), *k-mer–split* (Table 22), and *random-split* (Table 24).

2. Results on 4 non-canonical peptide regression datasets under four splitting strategies: *hybrid-split* (Table 25), *MMseqs2-split* (Table 27), *k-mer–split* (Table 26), and *random-split* (Table 28).

3. Results on 5 non-canonical peptide datasets under five splitting strategies: *ECFP-split* (Table 29), *hybrid-split* (Table 30), *MMseqs2-split* (Table 32), *k-mer–split* (Table 31), and *random-split* (Table 33).

4. Results on 3 PepPI datasets without freezing the protein encoder (Table 34).

5. Results on the *PepBenchData-150* benchmark (Table 36). For most datasets in *PepBenchData-150*, the proportion of sequences longer than 50 residues is below 10% (see Table 7 and Table 8). Therefore, we consider these datasets to be largely comparable to the *PepBenchData-50* version. Only five datasets contain more than 50% sequences longer than 50 residues, and we conducted additional experiments on these datasets.

By analyzing these results, we can draw the following conclusions:

1. The conclusions in the main text hold across all partitioning strategies.

2. The performance of kmer-split is significantly lower than that of random-split, indicating that the issue of kmer leakage is severe. The hybrid partition combines the advantages of MMseqs2-split and kmer-split, making it a more challenging partitioning strategy.

3. In the PepBenchData-150 version, the fine-tuned ESM2-150-F performs worse than ESM2-150M. This result is expected, as ESM2-150-F is fine-tuned only on peptide data and thus forgets the protein pre-training knowledge. As shown in Figure 111, ESM2-150M-F exhibits higher perplexity than ESM2-150M on sequences longer than 50.

4. For the PepPI task, it remains uncertain whether freezing the protein encoder is necessary.

Table 21: Performance of models on canonical peptide classification (ROC-AUC↑, %) with hybrid-split. Dataset sizes are shown separately; results are mean$_{\pm std}$. Best and second-best scores per row are in **bold** and gray shadow.

| Dataset | Size | FP-based models | | | | | GNN-based models | | | | SMILES-based models | | |
|---|---|---|---|---|---|---|---|---|---|---|---|---|---|
| | | RF | XGBoost | LightGBM | GradBoost | AdaBoost | GCN | GAT | GIN | Pepland | ChemBERTa | PeptideCLM | PepDoRA |
| nonfouling | 7200 | 76.4±1.5 | 75.4±2.1 | 76.4±1.4 | 77.0±1.4 | 76.3±1.5 | 76.8±1.0 | 76.1±0.9 | 76.7±1.4 | 75.3±2.3 | 76.2±1.0 | 75.6±2.1 | 70.7±2.1 |
| cpp | 2296 | 82.1±2.9 | 82.3±4.9 | 82.1±3.4 | 79.3±3.8 | 74.7±3.4 | 59.7±5.4 | 53.6±5.2 | 57.7±5.5 | 68.2±7.2 | 72.2±8.2 | 58.5±3.8 | 58.9±4.3 |
| bbp | 665 | 69.3±6.4 | 68.2±8.4 | 69.2±7.5 | 65.5±10.5 | 60.9±5.0 | 62.5±5.6 | 62.8±5.0 | 61.2±3.4 | 56.1±9.3 | 61.2±7.1 | 61.9±6.9 | 54.7±7.2 |
| antimicrobial | 52941 | 87.6±0.4 | 87.8±0.5 | 87.5±0.6 | 85.2±0.6 | 82.8±0.8 | 81.7±0.4 | 73.2±1.1 | 80.5±1.9 | 67.5±1.6 | 89.9±0.8 | 83.3±1.6 | 75.4±0.9 |
| antibacterial | 28591 | 88.4±0.9 | 88.3±0.8 | 87.6±0.7 | 85.6±0.8 | 84.1±0.9 | 83.2±1.0 | 75.9±1.3 | 82.2±1.8 | 62.7±3.4 | 88.9±2.2 | 77.0±5.4 | 73.8±2.6 |
| antifungal | 12887 | 87.3±0.8 | 87.3±0.8 | 87.1±0.8 | 85.3±1.0 | 83.3±1.2 | 76.0±2.5 | 75.2±1.1 | 78.5±3.6 | 53.5±3.5 | 87.0±1.1 | 59.0±2.2 | 70.3±1.2 |
| antiparasitic | 6755 | 86.8±1.6 | 87.3±1.0 | 87.3±0.9 | 84.8±1.0 | 83.2±1.5 | 75.2±3.4 | 74.2±2.8 | 75.6±2.2 | 58.7±2.5 | 85.0±1.3 | 63.6±4.6 | 67.2±3.5 |
| antiviral | 7785 | 84.2±0.9 | 83.6±1.0 | 84.5±1.3 | 82.7±1.5 | 80.3±0.9 | 74.4±4.6 | 64.6±7.4 | 74.6±4.9 | 71.0±3.1 | 81.2±2.9 | 57.2±2.5 | 59.7±2.2 |
| ace inhibitory | 3537 | **82.2±1.3** | 80.4±1.8 | 81.1±1.8 | 79.7±1.8 | 77.6±1.3 | 78.2±1.4 | 76.1±1.1 | 78.8±1.4 | 72.4±3.6 | 77.3±2.5 | 73.4±2.6 | 70.3±2.0 |
| antidiabetic | 3028 | 73.8±3.4 | 71.0±1.6 | 71.5±3.7 | 70.0±3.9 | 67.6±4.6 | 66.3±2.3 | 63.9±2.9 | 66.2±1.3 | 62.2±2.6 | 66.3±3.6 | 64.0±5.5 | 62.9±2.0 |
| dppiv inhibitors | 1268 | 82.4±3.0 | 79.8±1.5 | 81.2±3.4 | 79.9±1.7 | 75.8±2.0 | 71.8±4.3 | 69.1±1.0 | 71.3±4.8 | 65.6±4.6 | 76.0±3.1 | 71.0±2.9 | 63.4±8.3 |
| anticancer | 12013 | 87.7±1.0 | 87.6±0.9 | 87.1±0.6 | 84.4±0.4 | 82.4±0.8 | 78.4±3.6 | 75.0±1.1 | 78.9±2.6 | 69.5±4.6 | 87.5±1.2 | 66.7±10.4 | 71.4±2.8 |
| ttca | 1182 | 80.6±4.5 | 78.1±8.6 | 78.8±7.0 | 77.1±8.8 | 74.0±5.6 | 66.6±6.7 | 62.2±6.9 | 70.5±8.1 | 71.4±4.1 | 67.0±4.9 | 55.7±5.0 | 54.8±4.7 |
| neuropeptide | 8687 | 84.1±1.7 | 84.0±1.8 | 84.3±1.7 | 81.5±2.3 | 77.7±1.8 | 71.7±2.1 | 70.8±1.5 | 72.5±1.2 | 67.8±3.2 | 78.8±2.6 | 65.5±3.6 | 66.8±1.8 |
| antiinflamatory | 7665 | 74.3±1.4 | 72.4±1.0 | 73.7±1.2 | 74.0±1.0 | 71.3±1.6 | 68.7±4.2 | 65.5±2.4 | 71.1±3.0 | 57.7±2.7 | 73.5±2.4 | 63.5±1.9 | 62.4±1.8 |
| antioxidant | 390 | **68.1±2.7** | 65.9±3.2 | 65.6±2.8 | 63.6±3.2 | 61.7±2.9 | 60.5±4.3 | 57.7±3.7 | 59.7±4.3 | 61.3±3.0 | 58.9±3.3 | 55.6±2.9 | 53.8±3.7 |
| antiaging | 548 | 66.5±1.9 | 63.3±5.6 | 65.3±2.4 | 65.1±3.8 | 63.6±1.1 | 51.6±10.2 | 54.3±6.6 | 53.2±11.7 | 64.0±12.2 | 50.6±6.2 | 57.2±9.2 | 55.6±3.5 |
| quorum sensing | 490 | 85.2±5.1 | 81.3±2.9 | 81.4±3.1 | 76.4±4.5 | 67.0±3.4 | 59.0±7.1 | 61.7±5.8 | 60.5±9.6 | 67.9±3.4 | 55.3±5.2 | 58.4±5.7 | 50.7±5.7 |
| hemolytic | 4306 | 82.7±2.7 | 81.8±2.6 | 82.8±2.0 | 81.6±2.6 | 78.6±2.0 | 70.3±2.1 | 64.0±5.8 | 70.3±4.2 | 67.8±3.8 | 80.2±2.0 | 64.3±3.4 | 58.3±2.9 |
| toxicity | 4056 | 64.3±1.4 | 63.2±2.1 | 64.1±1.2 | 60.6±1.6 | 57.1±1.8 | 55.7±2.6 | 56.0±2.8 | 55.8±2.5 | | 59.5±5.8 | 49.6±4.2 | 53.1±2.6 |
| neurotoxin | 3159 | 63.7±4.1 | 60.7±3.0 | 61.9±1.7 | 61.1±3.0 | 57.0±3.4 | 57.9±3.2 | 57.9±3.7 | 56.9±2.7 | 53.2±2.0 | 56.1±5.6 | 49.3±4.6 | 51.0±5.0 |
| allergen | 2538 | 84.0±4.2 | 85.6±3.4 | 86.1±3.6 | 83.5±3.6 | 75.9±4.5 | 73.6±2.6 | 63.8±7.4 | 77.5±1.7 | 58.7±3.6 | 81.5±3.8 | 54.3±6.2 | 61.8±1.5 |
| Average | - | 79.2 | 78.0 | 78.5 | 76.5 | 73.3 | 69.1 | 66.1 | 69.5 | 63.9 | 73.2 | 62.9 | 62.1 |

| Dataset | Size | PLM-based models | | | | | | | | |
|---|---|---|---|---|---|---|---|---|---|---|
| | | ESM2-8M | ESM2-35M | ESM2-150M | ESM2-650M | DPLM-150M | DPLM-650M | ProtBERT | ESM2-8M-S | ESM2-150M-F |
| nonfouling | 7200 | 77.3±1.3 | 77.6±1.1 | 77.9±1.5 | 77.4±1.0 | 78.0±0.8 | 78.0±1.0 | 59.8±7.3 | 77.0±1.2 | **78.2±1.0** |
| cpp | 2296 | 80.5±3.2 | 82.2±2.3 | 83.2±3.1 | 80.1±6.0 | 81.9±2.8 | **83.8±5.8** | 75.6±2.1 | 79.6±8.0 | 81.9±3.1 |
| bbp | 665 | 66.8±9.5 | 61.8±15.7 | 62.4±12.2 | 64.6±9.0 | 64.6±8.9 | 66.8±6.2 | 61.1±7.1 | **71.6±5.7** | 64.6±12.5 |
| antimicrobial | 52941 | 92.3±0.3 | 92.4±0.4 | 92.3±0.6 | 92.6±1.1 | 92.4±0.2 | 92.5±0.4 | 78.0±22.1 | 92.0±0.6 | **93.2±0.3** |
| antibacterial | 28591 | 92.0±0.9 | 92.0±0.9 | 93.0±1.2 | 93.0±0.6 | 92.3±1.2 | 92.3±1.2 | 78.7±16.9 | 90.8±0.6 | **93.2±0.5** |
| antifungal | 12887 | 90.6±0.6 | 90.7±1.1 | 91.0±1.0 | 90.6±0.8 | 90.7±1.4 | 90.6±0.9 | 86.2±4.1 | 87.8±2.2 | **91.1±0.7** |
| antiparasitic | 6755 | 90.7±1.4 | 90.8±1.1 | 91.1±0.9 | 91.4±1.0 | **91.8±0.8** | 91.1±1.3 | 84.8±7.1 | 88.0±1.4 | 91.6±1.5 |
| antiviral | 7785 | 86.0±1.1 | 86.2±1.5 | 86.2±1.6 | 86.8±0.4 | 86.3±2.2 | 86.0±2.0 | 83.3±4.0 | 84.1±2.8 | **87.1±1.1** |
| ace inhibitory | 3537 | 79.2±1.4 | 80.7±1.9 | 80.2±0.5 | 79.6±1.9 | 79.7±1.0 | 80.3±1.1 | 64.7±15.0 | 79.1±1.6 | 80.2±2.3 |
| antidiabetic | 3028 | 72.9±2.0 | 74.1±4.1 | 70.5±4.1 | **74.6±3.4** | 71.9±4.5 | 72.8±3.1 | 56.4±10.1 | 69.8±2.5 | 72.7±2.5 |
| dppiv inhibitors | 1268 | 81.5±2.2 | 80.5±3.8 | 79.9±2.3 | 80.5±0.7 | **84.4±2.4** | 79.3±3.9 | 68.2±8.6 | 73.2±4.0 | 81.8±2.9 |
| anticancer | 12013 | 92.3±0.9 | 92.3±0.9 | 92.3±0.8 | 92.4±0.6 | 92.2±0.9 | **92.5±1.1** | 72.6±15.0 | 90.7±0.8 | 92.2±0.6 |
| ttca | 1182 | 79.7±6.2 | 79.0±5.7 | 79.2±4.4 | 75.7±7.3 | 79.2±7.1 | **82.9±6.3** | 64.3±15.3 | 66.8±6.9 | 81.7±5.8 |
| neuropeptide | 8687 | 85.2±3.6 | 86.7±2.3 | 86.3±2.3 | 88.0±1.1 | 87.3±3.2 | **88.0±1.5** | 67.6±15.1 | 84.0±1.9 | 85.6±1.8 |
| antiinflamatory | 7665 | 77.3±1.9 | 78.0±2.0 | 76.8±2.3 | 77.3±1.9 | 77.4±2.3 | 77.3±3.0 | 61.9±8.0 | 74.6±1.7 | **79.9±2.0** |
| antioxidant | 390 | 67.4±4.0 | 66.5±5.2 | 66.3±2.7 | 65.1±5.0 | 64.8±6.1 | 67.0±4.3 | 47.5±6.8 | 61.5±4.3 | **68.0±3.5** |
| antiaging | 548 | 61.4±4.6 | **67.2±5.2** | 62.5±3.3 | 63.6±5.8 | 64.0±7.1 | 63.3±7.3 | 54.4±7.9 | 58.1±2.2 | 61.9±3.9 |
| quorum sensing | 490 | 79.4±5.5 | 85.4±5.8 | 85.4±5.8 | 84.8±6.2 | 83.7±6.5 | 85.5±10.7 | 71.3±10.8 | 74.5±11.1 | **86.6±6.4** |
| hemolytic | 4306 | 84.5±2.4 | 85.1±1.9 | 85.3±1.4 | 85.7±1.7 | 85.2±1.6 | **86.0±1.8** | 73.1±11.1 | 82.9±2.9 | 84.3±2.7 |
| toxicity | 4056 | 69.6±3.0 | 71.6±2.7 | 72.3±2.6 | 72.8±3.3 | 72.9±2.0 | 73.0±3.3 | 66.1±4.7 | 65.8±1.8 | **78.0±3.0** |
| neurotoxin | 3159 | 66.5±3.7 | 67.7±4.8 | 69.9±3.4 | 69.4±3.4 | 71.0±4.1 | 69.5±3.6 | 65.9±3.3 | 62.4±3.0 | **73.0±4.7** |
| allergen | 2538 | 88.3±2.6 | 87.2±3.6 | 90.2±1.2 | 87.4±3.8 | 88.3±1.4 | 87.7±1.4 | 77.3±14.4 | 83.2±2.5 | 86.8±1.8 |
| Average | - | 80.3 | 80.6 | 80.6 | 80.6 | 80.9 | 81.2 | 69.0 | 77.2 | **81.5** |

Table 22: Performance of models on canonical peptide classification (ROC-AUC↑, %) with kmer-split. Dataset sizes are shown separately; results are mean$_{\pm std}$. Best and second-best scores per row are in **bold** and gray shadow.

| Dataset | Size | FP-based models | | | | | GNN-based models | | | | SMILES-based models | | |
|---|---|---|---|---|---|---|---|---|---|---|---|---|---|
| | | RF | XGBoost | LightGBM | GradBoost | AdaBoost | GCN | GAT | GIN | Pepland | ChemBERTa | PeptideCLM | PepDoRA |
| nonfouling | 7200 | 77.0$_{\pm1.4}$ | 76.9$_{\pm1.6}$ | 77.4$_{\pm1.2}$ | 77.6$_{\pm1.4}$ | 77.0$_{\pm1.9}$ | 77.7$_{\pm1.7}$ | 76.6$_{\pm2.5}$ | 78.4$_{\pm1.5}$ | 77.0$_{\pm2.5}$ | 77.2$_{\pm1.9}$ | 75.6$_{\pm2.6}$ | 70.9$_{\pm2.0}$ |
| cpp | 2296 | **85.8$_{\pm3.1}$** | 83.9$_{\pm3.2}$ | 84.0$_{\pm1.9}$ | 81.2$_{\pm1.7}$ | 76.5$_{\pm2.4}$ | 60.8$_{\pm7.1}$ | 54.2$_{\pm3.4}$ | 64.5$_{\pm2.4}$ | 70.3$_{\pm2.9}$ | 74.5$_{\pm4.1}$ | 60.1$_{\pm3.4}$ | 59.5$_{\pm4.5}$ |
| bbp | 665 | 69.8$_{\pm11.1}$ | 67.4$_{\pm9.0}$ | **69.8$_{\pm9.2}$** | 62.8$_{\pm4.8}$ | 59.3$_{\pm5.4}$ | 59.3$_{\pm8.3}$ | 59.3$_{\pm5.8}$ | 58.6$_{\pm10.7}$ | 69.1$_{\pm7.0}$ | 51.5$_{\pm11.4}$ | 59.3$_{\pm6.9}$ | 47.5$_{\pm5.0}$ |
| antimicrobial | 52941 | 87.8$_{\pm0.1}$ | 87.9$_{\pm0.3}$ | 87.5$_{\pm0.2}$ | 85.3$_{\pm0.3}$ | 82.8$_{\pm0.5}$ | 81.9$_{\pm0.7}$ | 73.5$_{\pm1.3}$ | 81.8$_{\pm0.3}$ | 68.8$_{\pm2.2}$ | 90.2$_{\pm0.6}$ | 83.9$_{\pm1.1}$ | 76.1$_{\pm2.0}$ |
| antibacterial | 28591 | 89.6$_{\pm1.0}$ | 89.3$_{\pm1.1}$ | 88.7$_{\pm0.9}$ | 86.3$_{\pm0.9}$ | 84.5$_{\pm0.8}$ | 82.6$_{\pm1.9}$ | 77.3$_{\pm1.2}$ | 83.8$_{\pm0.8}$ | 63.5$_{\pm3.7}$ | 89.9$_{\pm0.7}$ | 83.2$_{\pm0.9}$ | 75.6$_{\pm1.7}$ |
| antifungal | 12887 | 90.2$_{\pm0.8}$ | 89.8$_{\pm0.9}$ | 89.9$_{\pm1.0}$ | 87.4$_{\pm0.7}$ | 85.0$_{\pm1.0}$ | 78.1$_{\pm1.9}$ | 77.7$_{\pm1.4}$ | 82.2$_{\pm3.3}$ | 54.1$_{\pm5.9}$ | 88.9$_{\pm1.2}$ | 66.1$_{\pm12.9}$ | 69.0$_{\pm2.1}$ |
| antiparasitic | 6755 | 91.1$_{\pm1.5}$ | 91.2$_{\pm1.3}$ | 90.8$_{\pm1.8}$ | 89.1$_{\pm1.8}$ | 87.0$_{\pm1.8}$ | 80.2$_{\pm1.9}$ | 78.8$_{\pm2.4}$ | 83.1$_{\pm1.2}$ | 64.3$_{\pm2.9}$ | 88.7$_{\pm1.2}$ | 63.3$_{\pm9.6}$ | 69.2$_{\pm1.8}$ |
| antiviral | 7785 | 85.9$_{\pm1.7}$ | 84.5$_{\pm1.7}$ | 84.9$_{\pm1.2}$ | 82.8$_{\pm1.8}$ | 80.4$_{\pm1.8}$ | 72.3$_{\pm2.3}$ | 71.4$_{\pm2.4}$ | 76.6$_{\pm1.6}$ | 70.4$_{\pm2.4}$ | 81.9$_{\pm1.6}$ | 59.0$_{\pm1.9}$ | 60.5$_{\pm1.9}$ |
| ace inhibitory | 3537 | 81.3$_{\pm1.1}$ | 80.7$_{\pm2.0}$ | 81.5$_{\pm1.6}$ | 79.0$_{\pm1.2}$ | 76.4$_{\pm1.6}$ | 76.9$_{\pm1.0}$ | 74.4$_{\pm1.1}$ | 76.4$_{\pm1.3}$ | 73.3$_{\pm1.0}$ | 77.6$_{\pm1.9}$ | 73.9$_{\pm4.1}$ | 71.5$_{\pm2.9}$ |
| antidiabetic | 3028 | **75.2$_{\pm2.6}$** | 73.8$_{\pm2.9}$ | 74.5$_{\pm2.8}$ | 70.1$_{\pm3.9}$ | 65.9$_{\pm4.2}$ | 63.6$_{\pm3.7}$ | 59.2$_{\pm1.7}$ | 65.8$_{\pm2.9}$ | 64.8$_{\pm3.1}$ | 63.1$_{\pm1.8}$ | 60.5$_{\pm3.9}$ | 60.6$_{\pm2.0}$ |
| dppiv inhibitors | 1268 | 81.8$_{\pm2.4}$ | 81.6$_{\pm3.7}$ | **83.0$_{\pm2.6}$** | 79.9$_{\pm1.7}$ | 73.6$_{\pm3.6}$ | 75.0$_{\pm3.8}$ | 70.6$_{\pm4.8}$ | 73.7$_{\pm2.9}$ | 71.5$_{\pm1.7}$ | 75.3$_{\pm1.5}$ | 73.1$_{\pm3.1}$ | 69.1$_{\pm4.1}$ |
| anticancer | 12013 | 89.9$_{\pm0.5}$ | 89.6$_{\pm0.5}$ | 89.4$_{\pm0.8}$ | 86.7$_{\pm1.0}$ | 83.9$_{\pm0.9}$ | 75.5$_{\pm1.3}$ | 74.9$_{\pm1.0}$ | 82.0$_{\pm2.9}$ | 72.0$_{\pm2.4}$ | 88.9$_{\pm2.6}$ | 73.5$_{\pm10.4}$ | 68.1$_{\pm1.7}$ |
| ttca | 1182 | 77.0$_{\pm3.2}$ | 77.6$_{\pm1.5}$ | **78.6$_{\pm3.7}$** | 75.3$_{\pm4.4}$ | 73.2$_{\pm4.0}$ | 56.9$_{\pm2.9}$ | 57.2$_{\pm4.8}$ | 59.8$_{\pm4.6}$ | 73.3$_{\pm1.1}$ | 63.0$_{\pm3.5}$ | 56.8$_{\pm2.9}$ | 53.7$_{\pm3.5}$ |
| neuropeptide | 8687 | 83.9$_{\pm1.7}$ | 84.0$_{\pm1.5}$ | 84.0$_{\pm1.6}$ | 79.8$_{\pm2.1}$ | 75.4$_{\pm2.7}$ | 69.3$_{\pm2.3}$ | 68.7$_{\pm2.2}$ | 69.5$_{\pm3.5}$ | 67.2$_{\pm2.1}$ | 76.8$_{\pm1.8}$ | 65.2$_{\pm5.2}$ | 66.2$_{\pm1.5}$ |
| antiinflamatory | 7665 | 74.5$_{\pm1.3}$ | 74.1$_{\pm1.1}$ | 74.2$_{\pm1.0}$ | 73.4$_{\pm1.3}$ | 70.1$_{\pm1.7}$ | 66.1$_{\pm3.5}$ | 63.4$_{\pm0.9}$ | 66.3$_{\pm2.7}$ | 55.9$_{\pm3.8}$ | 71.0$_{\pm2.6}$ | 59.2$_{\pm2.4}$ | 60.8$_{\pm2.3}$ |
| antioxidant | 390 | 73.4$_{\pm3.1}$ | 72.6$_{\pm2.1}$ | 74.6$_{\pm2.9}$ | 71.3$_{\pm3.5}$ | 68.3$_{\pm4.3}$ | 62.4$_{\pm3.4}$ | 59.4$_{\pm2.1}$ | 60.7$_{\pm3.3}$ | 60.6$_{\pm4.4}$ | 66.8$_{\pm3.5}$ | 56.8$_{\pm5.6}$ | 56.5$_{\pm3.7}$ |
| antiaging | 548 | 72.4$_{\pm5.9}$ | 71.9$_{\pm2.7}$ | 72.3$_{\pm7.3}$ | 73.2$_{\pm3.9}$ | 67.3$_{\pm7.3}$ | 52.4$_{\pm12.6}$ | 58.9$_{\pm7.9}$ | 54.0$_{\pm11.9}$ | 62.7$_{\pm8.6}$ | 53.0$_{\pm5.2}$ | 51.8$_{\pm5.3}$ | 57.1$_{\pm7.1}$ |
| quorum sensing | 490 | 86.6$_{\pm5.0}$ | 84.9$_{\pm6.1}$ | 84.8$_{\pm5.5}$ | 82.4$_{\pm8.3}$ | 71.3$_{\pm4.7}$ | 63.1$_{\pm11.6}$ | 60.8$_{\pm12.3}$ | 66.3$_{\pm5.3}$ | 69.6$_{\pm2.5}$ | 61.6$_{\pm6.9}$ | 64.3$_{\pm7.7}$ | 55.1$_{\pm10.8}$ |
| hemolytic | 4306 | 85.2$_{\pm2.7}$ | 85.0$_{\pm2.1}$ | 85.3$_{\pm2.0}$ | 82.8$_{\pm1.9}$ | 80.0$_{\pm2.5}$ | 75.4$_{\pm4.5}$ | 69.3$_{\pm4.2}$ | 75.8$_{\pm3.0}$ | 70.5$_{\pm2.6}$ | 82.7$_{\pm4.0}$ | 69.9$_{\pm2.4}$ | 59.9$_{\pm4.0}$ |
| toxicity | 4056 | 70.6$_{\pm3.0}$ | 70.9$_{\pm2.1}$ | 71.3$_{\pm2.6}$ | 66.0$_{\pm2.7}$ | 59.4$_{\pm3.1}$ | 62.6$_{\pm2.5}$ | 60.7$_{\pm2.1}$ | 61.0$_{\pm2.3}$ | 60.4$_{\pm2.9}$ | 67.8$_{\pm3.5}$ | 50.7$_{\pm3.9}$ | 53.6$_{\pm3.8}$ |
| neurotoxin | 3159 | 71.4$_{\pm4.4}$ | 70.3$_{\pm3.2}$ | 69.0$_{\pm4.2}$ | 67.5$_{\pm2.5}$ | 60.9$_{\pm2.0}$ | 64.5$_{\pm1.1}$ | 63.9$_{\pm1.5}$ | 63.5$_{\pm2.3}$ | 59.5$_{\pm4.9}$ | 66.1$_{\pm2.8}$ | 52.8$_{\pm3.9}$ | 54.6$_{\pm1.7}$ |
| allergen | 2538 | 89.0$_{\pm1.3}$ | 90.6$_{\pm2.0}$ | **91.6$_{\pm1.8}$** | 88.1$_{\pm2.2}$ | 78.1$_{\pm3.4}$ | 74.7$_{\pm4.9}$ | 67.2$_{\pm6.8}$ | 77.1$_{\pm3.8}$ | 66.2$_{\pm1.3}$ | 82.6$_{\pm1.5}$ | 58.7$_{\pm6.5}$ | 63.9$_{\pm2.6}$ |
| Average | - | 81.4 | 80.8 | 81.2 | 78.5 | 74.4 | 69.6 | 67.2 | 71.0 | 66.6 | 74.5 | 64.4 | 62.7 |

| Dataset | Size | PLM-based models | | | | | | | | |
|---|---|---|---|---|---|---|---|---|---|---|
| | | ESM2-8M | ESM2-35M | ESM2-150M | ESM2-650M | DPLM-150M | DPLM-650M | ProtBERT | ESM2-8M-S | ESM2-150M-F |
| nonfouling | 7200 | **79.2$_{\pm1.2}$** | 78.8$_{\pm1.1}$ | 79.1$_{\pm1.6}$ | 78.8$_{\pm1.6}$ | 78.8$_{\pm1.1}$ | 78.6$_{\pm1.6}$ | 62.4$_{\pm17.5}$ | 78.2$_{\pm1.6}$ | 79.1$_{\pm1.2}$ |
| cpp | 2296 | 84.1$_{\pm1.5}$ | 83.1$_{\pm3.2}$ | 84.8$_{\pm2.4}$ | 83.8$_{\pm1.5}$ | 81.7$_{\pm2.9}$ | 85.1$_{\pm1.3}$ | 79.7$_{\pm2.3}$ | 82.0$_{\pm2.4}$ | 85.2$_{\pm2.2}$ |
| bbp | 665 | 64.7$_{\pm10.4}$ | 60.7$_{\pm13.4}$ | 62.0$_{\pm14.8}$ | 65.4$_{\pm10.6}$ | 64.7$_{\pm12.6}$ | 63.5$_{\pm12.6}$ | 64.7$_{\pm9.7}$ | 63.9$_{\pm12.4}$ | 68.3$_{\pm12.4}$ |
| antimicrobial | 52941 | 92.3$_{\pm0.3}$ | 92.4$_{\pm0.8}$ | 92.8$_{\pm0.2}$ | 92.7$_{\pm0.6}$ | 92.4$_{\pm0.6}$ | 92.9$_{\pm1.0}$ | - | 92.3$_{\pm0.3}$ | **93.5$_{\pm0.2}$** |
| antibacterial | 28591 | 92.8$_{\pm0.4}$ | 93.5$_{\pm0.3}$ | 93.6$_{\pm0.7}$ | **93.9$_{\pm0.7}$** | 93.8$_{\pm0.4}$ | 93.4$_{\pm0.7}$ | 82.8$_{\pm7.4}$ | 91.8$_{\pm1.1}$ | 93.9$_{\pm0.5}$ |
| antifungal | 12887 | 92.9$_{\pm0.4}$ | 93.2$_{\pm0.7}$ | 93.5$_{\pm0.5}$ | 93.1$_{\pm0.8}$ | 93.1$_{\pm1.1}$ | **93.7$_{\pm0.9}$** | 82.4$_{\pm15.5}$ | 90.3$_{\pm1.0}$ | 93.6$_{\pm1.2}$ |
| antiparasitic | 6755 | 94.2$_{\pm1.5}$ | 93.9$_{\pm1.4}$ | 94.2$_{\pm1.3}$ | 94.2$_{\pm0.9}$ | 94.5$_{\pm1.6}$ | 94.5$_{\pm1.0}$ | 91.3$_{\pm1.7}$ | 91.2$_{\pm1.3}$ | **94.6$_{\pm1.1}$** |
| antiviral | 7785 | 87.0$_{\pm1.8}$ | 87.0$_{\pm2.2}$ | 87.9$_{\pm1.9}$ | 88.0$_{\pm1.2}$ | **89.0$_{\pm0.9}$** | 87.8$_{\pm1.8}$ | 79.2$_{\pm6.5}$ | 82.9$_{\pm1.5}$ | 88.5$_{\pm1.2}$ |
| ace inhibitory | 3537 | 79.7$_{\pm1.6}$ | 78.0$_{\pm1.8}$ | 80.4$_{\pm2.1}$ | 79.4$_{\pm0.9}$ | 80.3$_{\pm2.9}$ | 80.2$_{\pm0.8}$ | 69.3$_{\pm11.7}$ | 76.6$_{\pm1.8}$ | 80.2$_{\pm1.8}$ |
| antidiabetic | 3028 | 71.6$_{\pm3.8}$ | 73.6$_{\pm3.9}$ | 70.5$_{\pm4.0}$ | 71.5$_{\pm2.9}$ | 73.7$_{\pm3.5}$ | 72.3$_{\pm2.9}$ | 56.7$_{\pm7.1}$ | 68.0$_{\pm4.5}$ | 74.0$_{\pm2.3}$ |
| dppiv inhibitors | 1268 | 82.6$_{\pm3.5}$ | 81.5$_{\pm4.1}$ | 80.9$_{\pm3.0}$ | 79.9$_{\pm2.3}$ | 81.6$_{\pm2.2}$ | 80.4$_{\pm3.6}$ | 68.3$_{\pm10.7}$ | 75.6$_{\pm6.2}$ | 81.3$_{\pm4.0}$ |
| anticancer | 12013 | 93.6$_{\pm1.0}$ | 93.9$_{\pm0.9}$ | 94.1$_{\pm0.9}$ | **94.1$_{\pm0.8}$** | 94.0$_{\pm0.7}$ | 93.9$_{\pm1.2}$ | 80.8$_{\pm10.5}$ | 91.7$_{\pm0.7}$ | 93.9$_{\pm0.5}$ |
| ttca | 1182 | 75.4$_{\pm2.0}$ | 78.0$_{\pm2.9}$ | 76.6$_{\pm2.2}$ | 75.5$_{\pm5.7}$ | 73.8$_{\pm3.3}$ | 77.1$_{\pm1.4}$ | 68.4$_{\pm5.1}$ | 65.6$_{\pm1.6}$ | 77.3$_{\pm3.6}$ |
| neuropeptide | 8687 | 85.7$_{\pm1.0}$ | 86.5$_{\pm1.3}$ | 85.5$_{\pm3.6}$ | 88.0$_{\pm2.4}$ | **88.7$_{\pm1.4}$** | 87.2$_{\pm1.0}$ | 69.2$_{\pm12.6}$ | 83.5$_{\pm1.5}$ | 86.9$_{\pm1.1}$ |
| antiinflamatory | 7665 | 76.1$_{\pm1.2}$ | 77.3$_{\pm1.1}$ | 75.7$_{\pm1.2}$ | 77.8$_{\pm1.1}$ | 77.1$_{\pm1.5}$ | 77.8$_{\pm1.7}$ | 65.6$_{\pm9.3}$ | 73.2$_{\pm0.8}$ | **79.6$_{\pm1.5}$** |
| antioxidant | 390 | 70.9$_{\pm2.5}$ | 71.6$_{\pm2.1}$ | 74.6$_{\pm3.6}$ | 73.4$_{\pm4.1}$ | 70.7$_{\pm1.9}$ | 73.1$_{\pm2.7}$ | 59.8$_{\pm8.0}$ | 65.6$_{\pm3.0}$ | **75.7$_{\pm3.3}$** |
| antiaging | 548 | 64.0$_{\pm4.9}$ | 66.0$_{\pm11.3}$ | 59.8$_{\pm4.5}$ | 64.9$_{\pm8.3}$ | 66.2$_{\pm6.1}$ | 60.3$_{\pm10.1}$ | 57.8$_{\pm11.3}$ | 59.7$_{\pm7.6}$ | 70.7$_{\pm8.4}$ |
| quorum sensing | 490 | 85.3$_{\pm6.5}$ | 84.9$_{\pm4.4}$ | 78.9$_{\pm3.3}$ | 81.6$_{\pm10.6}$ | 80.8$_{\pm5.9}$ | **87.6$_{\pm4.5}$** | 73.8$_{\pm8.4}$ | 79.7$_{\pm4.9}$ | 84.5$_{\pm4.4}$ |
| hemolytic | 4306 | 87.2$_{\pm1.7}$ | 86.5$_{\pm2.1}$ | 87.3$_{\pm1.6}$ | 87.6$_{\pm1.8}$ | 87.5$_{\pm1.9}$ | 88.2$_{\pm2.1}$ | 83.5$_{\pm2.0}$ | 84.8$_{\pm3.1}$ | 88.0$_{\pm1.5}$ |
| toxicity | 4056 | 75.8$_{\pm2.5}$ | 75.4$_{\pm1.6}$ | 77.8$_{\pm3.1}$ | 78.5$_{\pm3.0}$ | 77.9$_{\pm1.9}$ | 78.7$_{\pm3.1}$ | 73.8$_{\pm5.9}$ | 72.8$_{\pm2.9}$ | **81.1$_{\pm1.7}$** |
| neurotoxin | 3159 | 74.8$_{\pm2.4}$ | 75.6$_{\pm3.6}$ | 79.1$_{\pm2.9}$ | 78.4$_{\pm1.2}$ | 79.4$_{\pm2.5}$ | 80.1$_{\pm2.6}$ | 71.7$_{\pm5.8}$ | 70.2$_{\pm2.8}$ | **84.2$_{\pm1.8}$** |
| allergen | 2538 | 90.1$_{\pm2.3}$ | 89.2$_{\pm1.9}$ | 89.5$_{\pm1.2}$ | 90.8$_{\pm1.8}$ | 89.4$_{\pm1.8}$ | 90.4$_{\pm3.0}$ | 82.2$_{\pm2.2}$ | 86.0$_{\pm1.3}$ | 89.3$_{\pm2.1}$ |
| Average | - | 81.8 | 81.7 | 81.8 | 82.3 | 82.2 | 82.6 | 72.5 | 78.4 | **83.8** |

Table 23: Performance of models on canonical peptide classification (ROC-AUC↑, %) with MMseqs2-split. Dataset sizes are shown separately; results are mean$_{\pm std}$. Best and second-best scores per row are in **bold** and gray shadow.

| Dataset | Size | FP-based models | | | | | GNN-based models | | | | SMILES-based models | | |
|---|---|---|---|---|---|---|---|---|---|---|---|---|---|
| | | RF | XGBoost | LightGBM | GradBoost | AdaBoost | GCN | GAT | GIN | Pepland | ChemBERTa | PeptideCLM | PepDoRA |
| nonfouling | 7200 | 77.7±1.4 | 76.4±2.4 | 77.3±2.0 | 77.7±1.8 | 76.8±1.8 | 77.9±1.9 | 76.7±1.5 | 77.8±1.8 | 77.8±1.6 | 77.6±1.8 | 75.4±1.9 | 71.5±2.7 |
| cpp | 2296 | 85.8±2.2 | 84.3±2.8 | 84.2±2.4 | 81.2±1.3 | 78.1±1.3 | 69.0±3.0 | 63.5±2.7 | 67.2±5.8 | 69.7±9.8 | 80.1±2.8 | 68.3±2.1 | 68.0±1.8 |
| bbp | 665 | **73.5±5.8** | 70.7±5.4 | 71.6±5.5 | 71.0±7.0 | 63.9±3.9 | 65.4±4.5 | 63.4±4.4 | 65.9±6.4 | 67.6±4.5 | 58.2±7.8 | 64.1±6.8 | 58.9±7.4 |
| antimicrobial | 52941 | 91.9±0.4 | 91.6±0.5 | 91.1±0.5 | 88.8±0.5 | 86.0±0.7 | 85.9±0.9 | 78.8±1.1 | 85.6±0.8 | 72.0±5.5 | 92.9±0.5 | 87.1±1.8 | 83.7±1.2 |
| antibacterial | 28591 | 93.3±0.3 | 92.6±0.3 | 92.2±0.3 | 90.2±0.4 | 88.2±0.5 | 88.3±0.4 | 83.6±0.6 | 88.3±0.6 | 70.1±4.2 | 93.6±0.5 | 87.0±0.9 | 82.4±2.1 |
| antifungal | 12887 | 91.6±0.7 | 90.8±0.6 | 90.8±0.6 | 88.7±1.0 | 86.2±0.5 | 81.6±0.9 | 80.2±1.3 | 85.6±1.9 | 58.5±3.8 | 89.8±0.7 | 61.4±11.6 | 73.0±1.0 |
| antiparasitic | 6755 | 89.6±1.3 | 89.9±0.7 | 89.6±1.5 | 87.3±1.2 | 85.8±1.5 | 79.0±2.5 | 77.7±2.8 | 80.5±3.6 | 63.0±3.5 | 86.8±0.9 | 64.3±7.6 | 68.4±2.0 |
| antiviral | 7785 | 86.6±0.9 | 85.2±0.6 | 85.4±0.8 | 83.1±1.6 | 80.9±1.3 | 73.7±2.9 | 69.6±2.6 | 76.5±2.6 | 71.7±1.7 | 82.2±0.8 | 55.0±1.7 | 62.5±2.8 |
| ace inhibitory | 3537 | **83.0±1.1** | 82.5±1.7 | 82.8±1.5 | 81.6±0.7 | 79.1±0.9 | 79.0±1.3 | 78.0±2.1 | 78.3±1.2 | 77.9±2.5 | 78.9±1.8 | 77.7±1.8 | 73.6±2.7 |
| antidiabetic | 3028 | 75.9±4.2 | 74.3±4.2 | 73.6±3.4 | 69.5±4.8 | 64.5±3.0 | 63.9±3.1 | 64.7±2.7 | 64.5±3.4 | 60.4±4.5 | 66.3±1.6 | 61.7±4.7 | 61.1±1.6 |
| dppiv inhibitors | 1268 | 82.7±3.4 | 82.7±1.8 | 82.3±3.1 | 81.8±3.2 | 77.0±4.5 | 73.3±4.3 | 73.3±3.6 | 73.9±4.8 | 72.5±2.1 | 76.5±5.7 | 73.6±1.1 | 66.7±6.8 |
| anticancer | 12013 | 90.3±0.5 | 90.1±0.4 | 89.8±0.3 | 86.6±0.6 | 83.6±0.7 | 79.7±3.0 | 77.6±1.3 | 82.2±2.3 | 71.2±6.8 | 90.4±0.6 | 69.0±13.2 | 73.4±1.8 |
| ttca | 1182 | 81.2±3.2 | 81.2±5.1 | **81.9±5.0** | 81.6±4.7 | 78.0±2.8 | 65.3±6.7 | 61.8±5.1 | 65.6±5.4 | 78.0±3.0 | 63.8±10.1 | 56.3±4.9 | 57.9±6.9 |
| neuropeptide | 8687 | 85.7±0.9 | 86.7±0.9 | 86.6±1.0 | 83.2±1.3 | 79.0±1.7 | 74.1±1.3 | 73.2±1.2 | 75.0±1.3 | 74.7±2.8 | 81.5±2.3 | 67.3±4.1 | 69.4±1.4 |
| antiinflamatory | 7665 | 74.3±0.6 | 71.7±1.0 | 72.6±1.2 | 73.2±0.9 | 71.0±1.2 | 69.7±3.3 | 65.7±1.4 | 71.0±2.8 | 56.1±4.6 | 73.3±1.2 | 63.5±2.2 | 62.8±0.6 |
| antioxidant | 390 | **71.2±1.2** | 69.7±1.9 | 70.7±1.0 | 68.8±3.3 | 66.4±1.9 | 61.2±5.5 | 57.6±5.2 | 60.3±5.6 | 60.1±7.2 | 60.3±3.7 | 56.0±3.5 | 56.3±2.8 |
| antiaging | 548 | **66.8±6.5** | 64.6±7.8 | 65.0±8.1 | 66.7±5.9 | 59.1±1.9 | 46.0±5.3 | 51.2±4.8 | 47.3±5.5 | 53.2±6.6 | 52.0±10.9 | 51.6±8.1 | 52.5±9.3 |
| quorum sensing | 490 | 86.4±3.7 | **87.1±5.5** | 85.8±4.5 | 81.4±3.4 | 71.9±7.0 | 62.7±8.3 | 64.8±4.9 | 65.3±6.5 | 65.5±2.2 | 60.1±10.3 | 60.3±4.9 | 55.8±10.1 |
| hemolytic | 4306 | 87.8±1.7 | 86.7±0.9 | 86.9±1.2 | 86.1±1.4 | 82.4±1.2 | 78.0±2.3 | 74.7±2.7 | 80.3±1.6 | 76.8±3.0 | 85.3±1.8 | 66.5±3.9 | 66.4±2.7 |
| toxicity | 4056 | 62.4±1.2 | 61.9±1.5 | 61.5±2.5 | 58.1±2.1 | 54.5±1.5 | 55.4±2.5 | 54.6±2.8 | 54.7±2.8 | 56.2±4.4 | 53.9±3.5 | 47.9±2.4 | 50.2±2.9 |
| neurotoxin | 3159 | 64.0±3.4 | 64.6±2.1 | 63.8±2.3 | 62.7±2.8 | 55.8±2.5 | 57.5±4.2 | 57.3±3.5 | 56.6±3.9 | 50.1±5.7 | 59.3±3.8 | 49.1±4.3 | 53.2±4.2 |
| allergen | 2538 | 87.7±2.4 | 89.9±1.7 | 89.6±1.5 | 88.7±1.4 | 82.3±2.8 | 79.4±3.0 | 75.9±4.0 | 82.2±4.2 | 72.3±3.8 | 85.8±3.7 | 71.3±2.7 | 71.3±3.0 |
| Average | - | 81.3 | 80.7 | 80.7 | 79.0 | 75.0 | 71.2 | 69.3 | 72.0 | 67.1 | 74.9 | 65.2 | 65.4 |

| Dataset | Size | PLM-based models | | | | | | | | |
|---|---|---|---|---|---|---|---|---|---|---|
| | | ESM2-8M | ESM2-35M | ESM2-150M | ESM2-650M | DPLM-150M | DPLM-650M | ProtBERT | ESM2-8M-S | ESM2-150M-F |
| nonfouling | 7200 | 78.6±2.4 | 78.8±1.7 | 78.9±2.5 | 78.9±1.5 | 79.1±1.7 | 78.6±1.6 | 62.5±11.5 | 77.8±1.8 | **79.4±2.4** |
| cpp | 2296 | 85.8±1.8 | 85.4±2.5 | 85.5±2.4 | 85.4±2.3 | 85.3±3.0 | **87.4±3.0** | 78.9±7.0 | 84.7±1.4 | 85.6±1.3 |
| bbp | 665 | 70.5±7.5 | 70.3±3.9 | 73.1±2.5 | 67.2±7.5 | 70.2±3.8 | 69.9±5.4 | 64.2±5.9 | 68.1±4.7 | 68.4±3.4 |
| antimicrobial | 52941 | 94.8±0.4 | 94.8±0.4 | 94.7±0.4 | 95.1±0.4 | 95.0±0.4 | 95.2±0.4 | 72.4±15.8 | 94.1±0.3 | **95.6±0.5** |
| antibacterial | 28591 | 95.0±0.6 | 95.5±0.4 | 95.6±0.6 | 95.6±0.8 | 95.6±0.3 | 95.5±0.6 | 89.8±7.2 | 94.3±0.5 | **95.8±0.4** |
| antifungal | 12887 | 93.7±0.7 | 94.2±0.5 | **94.3±0.6** | 94.3±0.4 | 94.2±0.6 | 94.3±0.4 | 90.1±3.0 | 91.0±0.9 | 93.8±1.0 |
| antiparasitic | 6755 | 92.2±1.5 | 92.1±1.7 | 92.7±0.9 | 92.6±1.6 | 92.6±1.0 | **93.0±1.4** | 86.9±3.3 | 89.6±1.8 | 92.3±1.8 |
| antiviral | 7785 | 86.8±0.7 | 87.6±1.4 | 88.0±1.1 | **88.3±1.0** | 88.1±1.1 | 87.8±2.0 | 74.7±14.1 | 84.3±1.2 | 88.0±0.7 |
| ace inhibitory | 3537 | 81.5±1.8 | 80.9±2.8 | 82.0±1.8 | 81.1±1.9 | 81.4±1.7 | 82.4±1.4 | 61.2±7.8 | 79.3±1.7 | 81.3±1.9 |
| antidiabetic | 3028 | 74.3±4.0 | **76.0±1.9** | 73.3±1.6 | 75.5±1.8 | 73.8±5.9 | 74.1±2.4 | 50.5±7.6 | 69.9±2.7 | 73.7±1.8 |
| dppiv inhibitors | 1268 | **84.1±2.1** | 82.4±1.6 | 82.4±2.4 | 83.6±5.2 | 83.0±4.6 | 82.4±1.8 | 78.8±5.1 | 75.8±5.0 | 80.8±3.2 |
| anticancer | 12013 | 93.5±0.6 | 93.9±1.0 | 94.2±0.6 | 94.3±0.6 | 93.5±0.8 | **94.4±0.6** | 85.7±6.9 | 92.8±0.8 | 94.1±1.1 |
| ttca | 1182 | 78.4±3.2 | 77.6±3.9 | 78.9±3.6 | 80.4±2.2 | 77.0±5.3 | 79.2±3.8 | 70.7±10.2 | 69.6±4.3 | 80.5±3.3 |
| neuropeptide | 8687 | 87.7±1.1 | 87.1±2.1 | 87.4±2.1 | 89.0±1.8 | **89.1±1.1** | 88.6±1.7 | 71.7±18.9 | 85.3±0.9 | 87.3±1.5 |
| antiinflamatory | 7665 | 76.8±1.7 | 76.6±1.0 | 77.3±1.2 | 78.6±1.5 | 76.9±1.0 | 78.5±2.1 | 60.4±8.0 | 74.2±1.2 | **80.1±1.5** |
| antioxidant | 390 | 65.9±3.6 | 69.5±2.5 | 69.1±4.2 | 65.5±5.5 | 65.3±3.4 | 67.8±1.9 | 52.8±2.1 | 62.8±2.9 | 69.1±5.0 |
| antiaging | 548 | 60.6±8.1 | 64.8±11.4 | 54.8±8.5 | 60.6±9.0 | 65.1±6.7 | 57.8±10.4 | 55.3±6.6 | 56.2±8.0 | 59.5±6.9 |
| quorum sensing | 490 | 81.6±6.0 | 80.9±8.0 | 78.6±12.6 | 80.1±7.9 | 80.1±7.0 | 86.1±6.9 | 77.3±6.3 | 76.1±5.0 | 83.7±2.0 |
| hemolytic | 4306 | 89.9±2.0 | 89.6±1.5 | 89.2±2.9 | 89.4±1.5 | **90.2±1.5** | 90.1±1.7 | 82.8±2.9 | 86.8±1.9 | 90.1±1.1 |
| toxicity | 4056 | 70.2±0.8 | 68.9±2.3 | 71.4±3.6 | 71.5±4.8 | 69.8±5.0 | 69.7±2.3 | 68.5±2.1 | 65.2±3.9 | **75.4±2.9** |
| neurotoxin | 3159 | 69.5±3.4 | 71.8±4.5 | 71.8±4.5 | 72.3±2.5 | 72.3±4.6 | 72.9±2.1 | 67.0±4.0 | 63.5±3.8 | **76.4±2.4** |
| allergen | 2538 | 91.4±0.9 | 90.4±2.5 | 91.1±2.6 | 92.5±2.0 | 91.3±0.8 | **92.5±1.4** | 79.8±13.6 | 88.7±1.1 | 91.4±1.4 |
| Average | - | 81.9 | 82.1 | 82.0 | 82.4 | 82.5 | 82.3 | 71.9 | 78.6 | **82.8** |

Table 24: Performance of models on canonical peptide classification (ROC-AUC↑, %) with random-split. Dataset sizes are shown separately; results are mean$_{\pm std}$. Best and second-best scores per row are in **bold** and gray shadow.

| Dataset | Size | FP-based models | | | | | GNN-based models | | | | SMILES-based models | | |
|---|---|---|---|---|---|---|---|---|---|---|---|---|---|
| | | RF | XGBoost | LightGBM | GradBoost | AdaBoost | GCN | GAT | GIN | Pepland | ChemBERTa | PeptideCLM | PepDoRA |
| nonfouling | 7200 | 77.1±1.5 | 76.9±1.6 | 77.4±1.2 | 77.6±1.5 | 77.0±1.9 | 78.1±1.7 | 76.7±1.7 | 78.4±1.6 | 77.5±2.7 | 77.3±1.7 | 75.6±2.5 | 70.9±2.0 |
| cpp | 2296 | 91.9±2.8 | 91.7±3.0 | 91.7±2.9 | 88.4±2.4 | 81.8±3.2 | 78.6±3.4 | 71.3±5.0 | 73.7±7.8 | 77.0±5.4 | 85.4±2.6 | 66.4±5.3 | 71.8±2.6 |
| bbp | 665 | 76.5±4.3 | 75.5±4.4 | 75.0±3.1 | 71.2±7.1 | 59.7±6.2 | 59.7±6.8 | 58.8±7.3 | 60.0±7.3 | 62.3±7.5 | 54.9±5.3 | 60.8±7.7 | 47.1±6.9 |
| antimicrobial | 52941 | 93.9±0.4 | 93.2±0.3 | 92.6±0.4 | 90.4±0.5 | 87.5±0.7 | 88.1±0.6 | 80.5±0.9 | 88.0±0.7 | 76.2±0.7 | 94.3±0.4 | 88.5±0.8 | 85.0±0.6 |
| antibacterial | 28591 | 95.0±0.4 | 94.5±0.4 | 93.9±0.4 | 91.7±0.4 | 89.3±0.7 | 89.5±0.5 | 83.8±1.3 | 89.7±0.6 | 70.2±3.1 | 95.2±0.2 | 88.8±0.6 | 84.0±1.0 |
| antifungal | 12887 | 94.9±0.6 | 94.1±0.7 | 94.1±0.7 | 91.7±0.8 | 89.2±0.8 | 87.9±2.9 | 83.0±1.1 | 89.1±1.2 | 55.2±4.4 | 94.0±0.6 | 78.1±11.5 | 79.1±1.5 |
| antiparasitic | 6755 | 94.7±0.7 | 94.5±0.5 | 94.4±0.5 | 92.4±0.9 | 89.8±0.9 | 87.4±2.2 | 83.5±2.3 | 88.9±1.1 | 73.0±1.9 | 92.3±0.7 | 79.5±2.5 | 71.5±5.6 |
| antiviral | 7785 | 90.5±0.7 | 89.2±1.1 | 89.0±1.3 | 85.4±0.7 | 81.5±0.5 | 82.2±0.8 | 73.5±1.8 | 81.8±0.9 | 74.7±1.2 | 85.1±0.3 | 58.8±4.6 | 66.7±1.8 |
| ace inhibitory | 3537 | 84.0±2.1 | 82.4±2.2 | 83.3±1.9 | 81.5±2.4 | 80.3±2.9 | 80.2±2.9 | 79.4±3.4 | 80.5±3.3 | 76.0±7.0 | 80.2±2.7 | 78.5±2.7 | 74.3±1.8 |
| antidiabetic | 3028 | 81.0±2.4 | 77.0±1.8 | 79.1±1.4 | 75.3±3.3 | 69.7±3.5 | 64.8±4.2 | 63.2±5.1 | 64.2±5.1 | 59.8±3.3 | 70.5±3.3 | 66.0±5.0 | 63.9±2.1 |
| dppiv inhibitors | 1268 | 85.4±3.0 | 85.9±4.4 | 87.0±3.6 | 83.9±2.9 | 79.5±2.4 | 74.6±3.5 | 70.2±4.1 | 75.0±4.2 | 74.3±2.3 | 79.6±6.2 | 70.3±5.3 | 66.8±5.7 |
| anticancer | 12013 | 94.7±0.6 | 94.2±0.6 | 93.8±0.8 | 91.6±0.9 | 88.7±1.0 | 88.6±0.8 | 83.0±1.3 | 88.3±0.5 | 76.3±2.3 | 94.0±0.4 | 80.3±12.2 | 77.3±1.8 |
| ttca | 1182 | 81.4±4.0 | 82.6±1.6 | 79.9±2.2 | 80.3±3.6 | 77.3±2.9 | 68.1±5.6 | 60.7±6.2 | 68.8±4.8 | 77.0±4.3 | 70.2±5.8 | 55.7±1.4 | 55.6±3.2 |
| neuropeptide | 8687 | 89.8±0.7 | 89.4±0.7 | 89.4±1.0 | 85.6±0.9 | 80.7±1.1 | 75.6±1.6 | 74.6±1.8 | 75.2±3.6 | 69.2±1.9 | 85.1±0.8 | 74.4±3.9 | 71.1±0.5 |
| antiinflamatory | 7665 | 75.1±2.4 | 73.9±2.1 | 74.7±1.6 | 73.3±2.0 | 70.8±3.2 | 67.8±2.2 | 65.8±2.1 | 70.0±2.0 | 63.5±2.1 | 69.9±4.4 | 65.3±3.8 | 61.5±4.9 |
| antioxidant | 390 | 75.2±2.2 | 73.1±2.2 | 74.3±3.6 | 72.9±3.1 | 69.9±3.6 | 66.1±2.6 | 65.1±3.2 | 67.7±2.1 | 63.5±2.1 | 69.9±4.4 | 65.3±3.8 | 61.5±4.9 |
| antiaging | 548 | 72.9±2.8 | 70.7±6.2 | 70.0±6.8 | 73.4±5.7 | 68.0±5.1 | 53.8±12.2 | 55.7±4.6 | 56.5±10.5 | 55.8±6.7 | 51.8±3.1 | 57.4±6.6 | 44.3±6.1 |
| quorum sensing | 490 | 88.7±3.7 | 87.7±2.6 | 87.8±3.7 | 86.4±4.1 | 79.9±5.3 | 65.3±5.8 | 60.6±10.1 | 63.0±3.9 | 64.9±2.1 | 58.3±11.6 | 64.3±4.6 | 55.6±8.3 |
| hemolytic | 4306 | 90.0±0.8 | 89.3±1.3 | 89.6±0.9 | 87.0±1.5 | 84.7±1.6 | 81.1±2.2 | 76.1±6.2 | 82.5±2.0 | 78.2±3.3 | 87.6±1.8 | 76.7±4.7 | 70.7±2.7 |
| toxicity | 4056 | 78.8±2.7 | 77.1±2.7 | 77.7±2.5 | 73.1±4.4 | 65.9±4.3 | 63.7±2.7 | 60.2±4.5 | 63.1±2.5 | 58.8±2.1 | 72.4±2.5 | 55.9±6.7 | 57.0±3.0 |
| neurotoxin | 3159 | 80.8±1.7 | 78.8±2.4 | 79.4±1.5 | 75.2±1.9 | 67.8±2.4 | 59.1±2.6 | 56.7±2.8 | 60.0±1.9 | 54.3±2.5 | 71.9±3.7 | 54.4±2.9 | 56.0±3.1 |
| allergen | 2538 | 94.1±1.4 | 95.1±1.3 | 95.4±1.3 | 93.7±1.4 | 89.4±2.6 | 86.8±1.5 | 84.6±2.6 | 89.3±2.5 | 81.7±3.0 | 92.3±1.2 | 83.8±2.9 | 80.7±3.5 |
| Average | - | 85.7 | 84.9 | 85.0 | 82.8 | 78.6 | 74.9 | 71.2 | 75.2 | 68.7 | 78.9 | 70.2 | 66.9 |

| Dataset | Size | PLM-based models | | | | | | | | |
|---|---|---|---|---|---|---|---|---|---|---|
| | | ESM2-8M | ESM2-35M | ESM2-150M | ESM2-650M | DPLM-150M | DPLM-650M | ProtBERT | ESM2-8M-S | ESM2-150M-F |
| nonfouling | 7200 | 79.0±1.4 | 78.6±1.0 | 78.8±1.2 | 78.8±1.3 | 78.9±0.6 | 78.7±1.4 | 59.5±12.9 | 78.7±1.2 | 79.3±0.9 |
| cpp | 2296 | 92.2±2.6 | 93.0±2.0 | 91.9±3.3 | 92.5±3.5 | 91.6±2.8 | 93.2±2.3 | 86.9±6.7 | 90.8±2.0 | 91.5±2.8 |
| bbp | 665 | 70.8±7.1 | 70.1±6.5 | 71.9±5.6 | 67.5±3.7 | 73.9±4.8 | 70.4±5.5 | 61.0±12.1 | 69.0±3.8 | 73.5±6.5 |
| antimicrobial | 52941 | 96.1±0.4 | 96.3±0.2 | 96.5±0.2 | 96.8±0.2 | 96.4±0.2 | 96.3±0.4 | 83.4±3.3 | 95.2±0.2 | 96.7±0.1 |
| antibacterial | 28591 | 96.7±0.2 | 96.9±0.2 | 97.2±0.1 | 97.3±0.3 | 97.1±0.3 | 97.2±0.2 | 91.3±6.3 | 96.0±0.2 | 97.2±0.3 |
| antifungal | 12887 | 96.6±0.7 | 96.9±0.6 | 97.0±0.5 | 97.0±0.6 | 97.1±0.4 | 96.9±0.5 | 88.3±5.0 | 94.9±1.0 | 97.1±0.4 |
| antiparasitic | 6755 | 96.8±0.4 | 96.9±0.6 | 96.9±0.6 | 96.5±0.4 | 96.8±0.8 | 96.8±0.8 | 88.7±8.1 | 94.8±0.5 | 96.9±0.6 |
| antiviral | 7785 | 91.3±0.8 | 91.6±1.0 | 91.6±0.8 | 92.0±0.9 | 92.2±1.3 | 92.8±1.3 | 84.8±5.6 | 89.0±1.4 | 91.8±0.5 |
| ace inhibitory | 3537 | 82.8±1.7 | 83.4±1.7 | 82.8±2.0 | 82.4±2.3 | 83.9±1.2 | 84.3±0.8 | 73.8±14.6 | 81.3±2.2 | 83.0±2.1 |
| antidiabetic | 3028 | 77.4±2.4 | 78.5±2.6 | 77.2±1.3 | 81.1±3.7 | 80.2±2.9 | 78.1±3.3 | 57.2±5.3 | 73.7±3.6 | 77.9±0.4 |
| dppiv inhibitors | 1268 | 86.7±3.2 | 85.8±1.7 | 86.0±2.9 | 85.9±3.1 | 83.6±7.1 | 86.3±3.0 | 67.8±14.2 | 77.8±7.6 | 85.4±4.6 |
| anticancer | 12013 | 96.9±0.3 | 96.9±0.2 | 97.0±0.6 | 97.3±0.4 | 97.0±0.2 | 97.1±0.0 | 92.9±4.3 | 95.0±0.4 | 96.7±0.4 |
| ttca | 1182 | 77.8±3.6 | 79.1±3.2 | 79.4±3.0 | 78.6±3.7 | 77.8±4.5 | 81.4±3.4 | 59.0±14.5 | 68.9±3.8 | 80.4±3.4 |
| neuropeptide | 8687 | 91.3±1.1 | 91.3±0.9 | 92.2±0.9 | 93.6±1.0 | 92.2±0.6 | 93.4±0.5 | 75.4±22.6 | 87.8±0.6 | 91.9±0.7 |
| antiinflamatory | 7665 | 77.2±2.5 | 78.6±1.9 | 78.3±1.7 | 79.9±1.6 | 77.9±2.4 | 79.9±1.6 | 68.1±3.3 | 74.8±2.5 | 81.4±1.3 |
| antioxidant | 390 | 72.8±2.8 | 74.2±2.1 | 73.8±3.3 | 73.8±4.1 | 72.3±2.5 | 72.2±3.5 | 51.5±5.1 | 68.5±2.5 | 74.6±3.7 |
| antiaging | 548 | 66.5±3.0 | 67.1±5.9 | 68.8±3.0 | 71.7±8.9 | 69.7±8.1 | 60.8±7.6 | 58.9±10.8 | 61.4±2.8 | 71.9±3.4 |
| quorum sensing | 490 | 88.2±6.1 | 88.7±5.9 | 85.8±7.3 | 84.4±5.1 | 87.7±4.4 | 84.0±13.4 | 70.0±15.5 | 81.4±4.9 | 86.2±4.8 |
| hemolytic | 4306 | 90.6±0.9 | 91.2±1.4 | 91.3±0.7 | 91.6±1.2 | 90.8±1.5 | 91.0±1.1 | 83.1±5.7 | 89.5±1.1 | 91.3±0.8 |
| toxicity | 4056 | 83.7±3.5 | 84.8±1.2 | 86.5±1.9 | 86.5±2.9 | 85.4±1.5 | 85.4±2.0 | 81.7±4.0 | 79.4±2.2 | 90.0±1.3 |
| neurotoxin | 3159 | 82.6±2.7 | 83.5±3.2 | 83.7±2.3 | 83.8±4.3 | 84.5±3.4 | 84.8±4.1 | 81.0±2.6 | 76.7±3.0 | 88.9±3.1 |
| allergen | 2538 | 94.8±1.6 | 94.8±1.6 | 95.4±1.7 | 94.3±2.4 | 95.6±1.6 | 95.7±1.1 | 92.8±1.5 | 93.4±1.4 | 94.7±1.3 |
| Average | - | 85.9 | 86.2 | 86.3 | 86.5 | 86.5 | 86.2 | 75.3 | 82.6 | 87.2 |

Table 25: Performance of models on canonical peptide regression (MAE↓, %) with hybrid-split. Dataset sizes are shown separately; results are mean$_{\pm std}$. Best and second-best scores per row are in **bold** and gray shadow.

| Dataset | Size | FP-based models | | | | | GNN-based models | | | | SMILES-based models | | |
|---|---|---|---|---|---|---|---|---|---|---|---|---|---|
| | | RF | XGBoost | LightGBM | GradBoost | AdaBoost | GCN | GAT | GIN | Pepland | ChemBERTa | PeptideCLM | PepDoRA |
| E.coli_mic | 3204 | 0.593±0.015 | 0.604±0.019 | 0.584±0.019 | 0.629±0.010 | 0.677±0.013 | 0.670±0.039 | 0.665±0.023 | 0.631±0.015 | 0.650±0.062 | 0.603±0.007 | 0.621±0.012 | 0.650±0.013 |
| S.aureus_mic | 2822 | 0.569±0.035 | 0.588±0.022 | 0.572±0.025 | 0.613±0.029 | 0.651±0.022 | 0.643±0.017 | 0.639±0.028 | 0.635±0.032 | 0.648±0.016 | 0.627±0.011 | 0.659±0.045 | 0.657±0.029 |
| P.aeruginosa_mic | 1490 | 0.540±0.030 | 0.559±0.025 | 0.550±0.026 | 0.555±0.019 | 0.586±0.006 | 0.612±0.047 | 0.579±0.027 | 0.537±0.032 | 0.566±0.037 | 0.520±0.041 | 0.529±0.035 | 0.546±0.029 |
| hemolytic_hc50 | 1926 | 0.517±0.042 | 0.527±0.038 | 0.523±0.035 | 0.546±0.036 | 0.619±0.048 | 0.603±0.077 | 0.539±0.023 | 0.528±0.036 | 0.516±0.033 | 0.498±0.040 | 0.539±0.020 | 0.535±0.027 |
| Average | - | 0.555 | 0.569 | 0.557 | 0.586 | 0.633 | 0.632 | 0.605 | 0.583 | 0.595 | 0.562 | 0.587 | 0.597 |

| Dataset | Size | PLM-based models | | | | | | | | |
|---|---|---|---|---|---|---|---|---|---|---|
| | | ESM2-8M | ESM2-35M | ESM2-150M | ESM2-650M | DPLM-150M | DPLM-650M | ProtBERT | ESM2-8M-S | ESM2-150M-F |
| E.coli_mic | 3204 | 0.539±0.023 | 0.527±0.004 | 0.509±0.010 | 0.488±0.008 | 0.525±0.006 | 0.528±0.010 | 0.602±0.087 | 0.548±0.024 | 0.517±0.023 |
| S.aureus_mic | 2822 | 0.545±0.030 | 0.549±0.022 | 0.527±0.032 | 0.522±0.022 | 0.563±0.023 | 0.552±0.018 | 0.649±0.071 | 0.585±0.016 | 0.533±0.028 |
| P.aeruginosa_mic | 1490 | 0.506±0.046 | 0.497±0.042 | 0.496±0.042 | 0.471±0.043 | 0.524±0.036 | 0.493±0.042 | 0.586±0.012 | 0.516±0.031 | 0.483±0.024 |
| hemolytic_hc50 | 1926 | 0.422±0.042 | 0.400±0.021 | 0.413±0.047 | 0.394±0.017 | 0.404±0.028 | 0.419±0.014 | 0.497±0.071 | 0.463±0.011 | 0.412±0.020 |
| Average | - | 0.503 | 0.493 | 0.486 | 0.469 | 0.504 | 0.498 | 0.584 | 0.528 | 0.486 |

Table 26: Performance of models on canonical peptide regression (MAE↓, %) with kmer-split. Dataset sizes are shown separately; results are mean$_{\pm std}$. Best and second-best scores per row are in **bold** and gray shadow.

| Dataset | Size | FP-based models | | | | | GNN-based models | | | | SMILES-based models | | |
|---|---|---|---|---|---|---|---|---|---|---|---|---|---|
| | | RF | XGBoost | LightGBM | GradBoost | AdaBoost | GCN | GAT | GIN | Pepland | ChemBERTa | PeptideCLM | PepDoRA |
| E.coli_mic | 3204 | $0.465_{\pm0.022}$ | $0.484_{\pm0.014}$ | $0.476_{\pm0.016}$ | $0.532_{\pm0.023}$ | $0.597_{\pm0.027}$ | $0.593_{\pm0.038}$ | $0.607_{\pm0.025}$ | $0.577_{\pm0.011}$ | $0.557_{\pm0.026}$ | $0.577_{\pm0.026}$ | $0.582_{\pm0.021}$ | $0.635_{\pm0.021}$ |
| S.aureus_mic | 2822 | $0.450_{\pm0.025}$ | $0.467_{\pm0.022}$ | $0.450_{\pm0.014}$ | $0.513_{\pm0.029}$ | $0.575_{\pm0.036}$ | $0.572_{\pm0.026}$ | $0.573_{\pm0.029}$ | $0.559_{\pm0.020}$ | $0.576_{\pm0.027}$ | $0.551_{\pm0.029}$ | $0.554_{\pm0.022}$ | $0.586_{\pm0.033}$ |
| P.aeruginosa_mic | 1490 | $0.435_{\pm0.018}$ | $0.440_{\pm0.032}$ | $0.439_{\pm0.024}$ | $0.466_{\pm0.020}$ | $0.510_{\pm0.015}$ | $0.517_{\pm0.024}$ | $0.539_{\pm0.024}$ | $0.504_{\pm0.014}$ | $0.521_{\pm0.017}$ | $0.493_{\pm0.032}$ | $0.511_{\pm0.016}$ | $0.533_{\pm0.029}$ |
| hemolytic_hc50 | 1926 | $0.383_{\pm0.024}$ | $0.392_{\pm0.029}$ | $0.391_{\pm0.029}$ | $0.428_{\pm0.018}$ | $0.512_{\pm0.035}$ | $0.561_{\pm0.098}$ | $0.550$ | $0.475_{\pm0.009}$ | $0.454_{\pm0.021}$ | $0.439_{\pm0.025}$ | $0.501_{\pm0.039}$ | $0.490_{\pm0.025}$ |
| Average | - | 0.433 | 0.446 | 0.439 | 0.485 | 0.548 | 0.561 | 0.550 | 0.529 | 0.527 | 0.515 | 0.537 | 0.561 |

| Dataset | Size | PLM-based models | | | | | | | | |
|---|---|---|---|---|---|---|---|---|---|---|
| | | ESM2-8M | ESM2-35M | ESM2-150M | ESM2-650M | DPLM-150M | DPLM-650M | ProtBERT | ESM2-8M-S | ESM2-150M-F |
| E.coli_mic | 3204 | $0.430_{\pm0.027}$ | $0.418_{\pm0.029}$ | $0.431_{\pm0.043}$ | $0.403_{\pm0.026}$ | $0.430_{\pm0.021}$ | $0.423_{\pm0.027}$ | $0.478_{\pm0.111}$ | $0.494_{\pm0.035}$ | $\mathbf{0.403}_{\pm0.022}$ |
| S.aureus_mic | 2822 | $0.454_{\pm0.021}$ | $0.438_{\pm0.032}$ | $\mathbf{0.401}_{\pm0.018}$ | $0.409_{\pm0.022}$ | $0.431_{\pm0.017}$ | $0.428_{\pm0.009}$ | $0.510_{\pm0.066}$ | $0.496_{\pm0.015}$ | $0.410_{\pm0.018}$ |
| P.aeruginosa_mic | 1490 | $0.426_{\pm0.020}$ | $0.400_{\pm0.031}$ | $0.415_{\pm0.031}$ | $0.401_{\pm0.031}$ | $0.438_{\pm0.029}$ | $0.416_{\pm0.029}$ | $0.481_{\pm0.059}$ | $0.480_{\pm0.016}$ | $\mathbf{0.395}_{\pm0.023}$ |
| hemolytic_hc50 | 1926 | $0.336_{\pm0.025}$ | $0.326_{\pm0.026}$ | $\mathbf{0.309}_{\pm0.026}$ | $0.318_{\pm0.015}$ | $0.347_{\pm0.016}$ | $0.324_{\pm0.011}$ | $0.399_{\pm0.062}$ | $0.407_{\pm0.016}$ | $0.316_{\pm0.010}$ |
| Average | | 0.412 | 0.396 | 0.389 | 0.383 | 0.412 | 0.398 | 0.467 | 0.469 | **0.381** |

Table 27: Performance of models on canonical peptide regression (MAE↓, %) with MMseqs2-split. Dataset sizes are shown separately; results are mean$_{\pm std}$. Best and second-best scores per row are in **bold** and gray shadow.

| Dataset | Size | FP-based models | | | | | GNN-based models | | | | SMILES-based models | | |
|---|---|---|---|---|---|---|---|---|---|---|---|---|---|
| | | RF | XGBoost | LightGBM | GradBoost | AdaBoost | GCN | GAT | GIN | Pepland | ChemBERTa | PeptideCLM | PepDoRA |
| E.coli_mic | 3204 | $0.573_{\pm0.035}$ | $0.583_{\pm0.036}$ | $0.578_{\pm0.034}$ | $0.620_{\pm0.027}$ | $0.676_{\pm0.031}$ | $0.662_{\pm0.040}$ | $0.639_{\pm0.027}$ | $0.618_{\pm0.043}$ | $0.639_{\pm0.042}$ | $0.583_{\pm0.013}$ | $0.608_{\pm0.027}$ | $0.630_{\pm0.021}$ |
| S.aureus_mic | 2822 | $0.568_{\pm0.026}$ | $0.592_{\pm0.024}$ | $0.577_{\pm0.026}$ | $0.620_{\pm0.022}$ | $0.644_{\pm0.026}$ | $0.631_{\pm0.012}$ | $0.624_{\pm0.025}$ | $0.616_{\pm0.022}$ | $0.638_{\pm0.043}$ | $0.615_{\pm0.023}$ | $0.636_{\pm0.027}$ | $0.644_{\pm0.033}$ |
| P.aeruginosa_mic | 1490 | $0.518_{\pm0.030}$ | $0.523_{\pm0.032}$ | $0.526_{\pm0.023}$ | $0.532_{\pm0.020}$ | $0.577_{\pm0.023}$ | $0.575_{\pm0.049}$ | $0.565_{\pm0.019}$ | $0.558_{\pm0.036}$ | $0.561_{\pm0.013}$ | $0.547_{\pm0.036}$ | $0.561_{\pm0.042}$ | $0.568_{\pm0.028}$ |
| hemolytic_hc50 | 1926 | $0.547_{\pm0.034}$ | $0.554_{\pm0.047}$ | $0.543_{\pm0.046}$ | $0.561_{\pm0.038}$ | $0.627_{\pm0.032}$ | $0.617_{\pm0.075}$ | $0.548_{\pm0.051}$ | $0.532_{\pm0.028}$ | $0.540_{\pm0.017}$ | $0.532_{\pm0.045}$ | $0.556_{\pm0.023}$ | $0.538_{\pm0.031}$ |
| Average | - | 0.552 | 0.563 | 0.556 | 0.583 | 0.631 | 0.621 | 0.594 | 0.581 | 0.594 | 0.569 | 0.590 | 0.595 |

| Dataset | Size | PLM-based models | | | | | | | | |
|---|---|---|---|---|---|---|---|---|---|---|
| | | ESM2-8M | ESM2-35M | ESM2-150M | ESM2-650M | DPLM-150M | DPLM-650M | ProtBERT | ESM2-8M-S | ESM2-150M-F |
| E.coli_mic | 3204 | $0.517_{\pm0.018}$ | $0.497_{\pm0.022}$ | $0.489_{\pm0.022}$ | $\mathbf{0.472}_{\pm0.018}$ | $0.523_{\pm0.012}$ | $0.500_{\pm0.031}$ | $0.607_{\pm0.097}$ | $0.543_{\pm0.018}$ | $0.488_{\pm0.022}$ |
| S.aureus_mic | 2822 | $0.569_{\pm0.007}$ | $0.548_{\pm0.036}$ | $0.539_{\pm0.021}$ | $0.526_{\pm0.034}$ | $0.566_{\pm0.051}$ | $0.536_{\pm0.021}$ | $0.622_{\pm0.021}$ | $0.578_{\pm0.023}$ | $\mathbf{0.523}_{\pm0.018}$ |
| P.aeruginosa_mic | 1490 | $0.523_{\pm0.026}$ | $0.514_{\pm0.040}$ | $0.495_{\pm0.044}$ | $\mathbf{0.471}_{\pm0.023}$ | $0.528_{\pm0.044}$ | $0.497_{\pm0.053}$ | $0.557_{\pm0.054}$ | $0.529_{\pm0.022}$ | $0.476_{\pm0.030}$ |
| hemolytic_hc50 | 1926 | $0.431_{\pm0.029}$ | $0.432_{\pm0.038}$ | $\mathbf{0.414}_{\pm0.013}$ | $0.424_{\pm0.039}$ | $0.422_{\pm0.025}$ | $0.421_{\pm0.023}$ | $0.491_{\pm0.045}$ | $0.468_{\pm0.022}$ | $0.432_{\pm0.020}$ |
| Average | | 0.510 | 0.498 | 0.484 | **0.473** | 0.510 | 0.488 | 0.569 | 0.530 | 0.480 |

Table 28: Performance of models on canonical peptide regression (MAE↓, %) with random-split. Dataset sizes are shown separately; results are mean$_{\pm std}$. Best and second-best scores per row are in **bold** and gray shadow.

| Dataset | Size | FP-based models | | | | | GNN-based models | | | | SMILES-based models | | |
|---|---|---|---|---|---|---|---|---|---|---|---|---|---|
| | | RF | XGBoost | LightGBM | GradBoost | AdaBoost | GCN | GAT | GIN | Pepland | ChemBERTa | PeptideCLM | PepDoRA |
| E.coli_mic | 3204 | $0.468_{\pm0.022}$ | $0.484_{\pm0.014}$ | $0.476_{\pm0.016}$ | $0.532_{\pm0.023}$ | $0.593_{\pm0.025}$ | $0.587_{\pm0.028}$ | $0.613_{\pm0.025}$ | $0.570_{\pm0.025}$ | $0.565_{\pm0.017}$ | $0.551_{\pm0.045}$ | $0.577_{\pm0.028}$ | $0.640_{\pm0.021}$ |
| S.aureus_mic | 2822 | $0.448_{\pm0.020}$ | $0.467_{\pm0.022}$ | $0.450_{\pm0.014}$ | $0.513_{\pm0.029}$ | $0.577_{\pm0.036}$ | $0.572_{\pm0.028}$ | $0.573_{\pm0.036}$ | $0.554_{\pm0.021}$ | $0.513_{\pm0.021}$ | $0.558_{\pm0.040}$ | $0.552_{\pm0.018}$ | $0.587_{\pm0.033}$ |
| P.aeruginosa_mic | 1490 | $0.433_{\pm0.021}$ | $0.440_{\pm0.032}$ | $0.439_{\pm0.024}$ | $0.465_{\pm0.020}$ | $0.508_{\pm0.020}$ | $0.532_{\pm0.037}$ | $0.529_{\pm0.021}$ | $0.499_{\pm0.015}$ | $0.513_{\pm0.021}$ | $0.491_{\pm0.033}$ | $0.513_{\pm0.032}$ | $0.534_{\pm0.032}$ |
| hemolytic_hc50 | 1926 | $0.377_{\pm0.026}$ | $0.392_{\pm0.029}$ | $0.391_{\pm0.029}$ | $0.428_{\pm0.018}$ | $0.519_{\pm0.022}$ | $0.515_{\pm0.058}$ | $0.551$ | $0.468_{\pm0.021}$ | $0.454_{\pm0.023}$ | $0.433_{\pm0.019}$ | $0.494_{\pm0.034}$ | $0.490_{\pm0.025}$ |
| Average | - | 0.432 | 0.446 | 0.439 | 0.485 | 0.549 | 0.551 | 0.551 | 0.523 | 0.528 | 0.508 | 0.534 | 0.563 |

| Dataset | Size | PLM-based models | | | | | | | | |
|---|---|---|---|---|---|---|---|---|---|---|
| | | ESM2-8M | ESM2-35M | ESM2-150M | ESM2-650M | DPLM-150M | DPLM-650M | ProtBERT | ESM2-8M-S | ESM2-150M-F |
| E.coli_mic | 3204 | $0.441_{\pm0.026}$ | $0.445_{\pm0.021}$ | $0.406_{\pm0.027}$ | $\mathbf{0.402}_{\pm0.024}$ | $0.423_{\pm0.039}$ | $0.428_{\pm0.038}$ | $0.534_{\pm0.113}$ | $0.477_{\pm0.022}$ | $0.411_{\pm0.024}$ |
| S.aureus_mic | 2822 | $0.438_{\pm0.011}$ | $0.418_{\pm0.014}$ | $0.414_{\pm0.014}$ | $\mathbf{0.403}_{\pm0.019}$ | $0.444_{\pm0.035}$ | $0.437_{\pm0.037}$ | $0.579_{\pm0.084}$ | $0.495_{\pm0.020}$ | $0.412_{\pm0.018}$ |
| P.aeruginosa_mic | 1490 | $0.425_{\pm0.039}$ | $0.405_{\pm0.020}$ | $0.411_{\pm0.020}$ | $\mathbf{0.377}_{\pm0.012}$ | $0.430_{\pm0.038}$ | $0.419_{\pm0.027}$ | $0.485_{\pm0.053}$ | $0.476_{\pm0.021}$ | $0.395_{\pm0.021}$ |
| hemolytic_hc50 | 1926 | $0.342_{\pm0.020}$ | $0.341_{\pm0.022}$ | $0.322_{\pm0.031}$ | $0.317_{\pm0.031}$ | $0.324_{\pm0.035}$ | $0.348_{\pm0.016}$ | $0.418_{\pm0.108}$ | $0.395_{\pm0.019}$ | $0.329_{\pm0.016}$ |
| Average | | 0.411 | 0.402 | 0.388 | **0.375** | 0.405 | 0.408 | 0.504 | 0.461 | 0.387 |

Table 29: Performance of models on non-canonical peptide classification (ROC-AUC↑, %) and regression (MAE↓, %) with ECFP-split. Dataset sizes are shown separately; results are mean$_{\pm std}$. Best and second-best scores per row are in **bold** and gray shadow.

| Task | Metric | Dataset | Size | FP-based models | | | | | GNN-based models | | | | SMILES-based models | | |
|---|---|---|---|---|---|---|---|---|---|---|---|---|---|---|---|
| | | | | RF | XGBoost | LightGBM | GradBoost | AdaBoost | GCN | GAT | GIN | Pepland | ChemBERTa | PeptideCLM | PepDoRA |
| cls | AUC ROC | nc-antibacterial | 1668 | $\mathbf{94.4}_{\pm1.4}$ | $93.6_{\pm2.2}$ | $93.8_{\pm1.8}$ | $92.5_{\pm2.8}$ | $90.9_{\pm3.4}$ | $93.0_{\pm2.6}$ | $81.5_{\pm3.5}$ | $90.7_{\pm4.9}$ | $84.9_{\pm3.9}$ | $91.9_{\pm2.7}$ | $71.4_{\pm1.8}$ | $72.4_{\pm7.4}$ |
| | | nc-antifungal | 407 | $95.4_{\pm3.2}$ | $\mathbf{96.5}_{\pm2.9}$ | $94.8_{\pm3.1}$ | $95.2_{\pm3.8}$ | $95.4_{\pm2.4}$ | $70.1_{\pm12.0}$ | $78.5_{\pm6.2}$ | $86.1_{\pm2.8}$ | $83.5_{\pm4.4}$ | $78.0_{\pm18.3}$ | $65.4_{\pm7.0}$ | $65.9_{\pm12.0}$ |
| | | nc-antimicrobial | 2465 | $97.6_{\pm0.9}$ | $97.7_{\pm0.9}$ | $\mathbf{97.8}_{\pm0.6}$ | $97.3_{\pm0.7}$ | $95.3_{\pm1.8}$ | $94.9_{\pm1.6}$ | $79.2_{\pm4.4}$ | $91.5_{\pm2.3}$ | $88.0_{\pm0.9}$ | $95.2_{\pm1.9}$ | $68.3_{\pm4.7}$ | $80.0_{\pm1.9}$ |
| | | nc-hemolytic | 3971 | $96.1_{\pm1.1}$ | $96.2_{\pm0.5}$ | $\mathbf{96.3}_{\pm0.7}$ | $95.3_{\pm0.8}$ | $93.7_{\pm1.6}$ | $89.5_{\pm3.7}$ | $85.7_{\pm2.6}$ | $89.0_{\pm4.1}$ | $82.8_{\pm2.1}$ | $91.4_{\pm1.5}$ | $76.1_{\pm4.5}$ | $72.1_{\pm7.3}$ |
| | | avg | - | $95.9$ | **96.0** | 95.7 | 95.1 | 93.8 | 86.9 | 81.2 | 89.3 | 84.8 | 89.1 | 70.3 | 72.6 |
| reg | MAE | nc-cpp | 6970 | $\mathbf{0.649}_{\pm0.006}$ | $0.705_{\pm0.026}$ | $0.651_{\pm0.016}$ | $0.683_{\pm0.007}$ | $0.829_{\pm0.019}$ | $0.754_{\pm0.027}$ | $0.767_{\pm0.025}$ | $0.701_{\pm0.027}$ | $0.736_{\pm0.026}$ | $0.712_{\pm0.033}$ | $0.822_{\pm0.035}$ | $0.879_{\pm0.013}$ |

Table 30: Performance of models on non-canonical peptide classification (ROC-AUC↑, %) and regression (MAE↓, %) with hybrid-split. Dataset sizes are shown separately; results are mean$_{\pm std}$. Best and second-best scores per row are in **bold** and gray shadow.

| Task | Metric | Dataset | Size | FP-based models | | | | | GNN-based models | | | | SMILES-based models | | |
|---|---|---|---|---|---|---|---|---|---|---|---|---|---|---|---|
| | | | | RF | XGBoost | LightGBM | GradBoost | AdaBoost | GCN | GAT | GIN | Pepland | ChemBERTa | PeptideCLM | PepDoRA |
| cls | AUC ROC | nc-antibacterial | 1668 | **94.1**$_{\pm1.4}$ | 93.8$_{\pm1.9}$ | 94.0$_{\pm2.2}$ | 94.0$_{\pm2.4}$ | 91.8$_{\pm2.6}$ | 88.1$_{\pm6.6}$ | 82.0$_{\pm4.0}$ | 89.7$_{\pm1.9}$ | 82.9$_{\pm1.6}$ | 92.2$_{\pm1.8}$ | 71.9$_{\pm4.5}$ | 75.1$_{\pm8.8}$ |
| | | nc-antifungal | 407 | **97.3**$_{\pm1.1}$ | 96.0$_{\pm1.9}$ | 96.0$_{\pm1.6}$ | 94.9$_{\pm1.1}$ | 96.0$_{\pm1.9}$ | 70.5$_{\pm2.9}$ | 77.4$_{\pm9.8}$ | 81.5$_{\pm8.5}$ | 84.5$_{\pm2.4}$ | 87.3$_{\pm6.9}$ | 69.8$_{\pm8.2}$ | 61.8$_{\pm14.1}$ |
| | | nc-antimicrobial | 2465 | 96.8$_{\pm1.3}$ | 96.9$_{\pm0.7}$ | **97.0**$_{\pm0.9}$ | 96.7$_{\pm0.4}$ | 95.6$_{\pm1.2}$ | 91.8$_{\pm1.3}$ | 83.5$_{\pm2.7}$ | 89.2$_{\pm3.1}$ | 86.9$_{\pm2.0}$ | 95.2$_{\pm0.8}$ | 75.4$_{\pm2.6}$ | 80.2$_{\pm4.6}$ |
| | | nc-hemolytic | 3971 | 96.5$_{\pm1.3}$ | 96.0$_{\pm1.4}$ | **96.8**$_{\pm1.2}$ | 95.7$_{\pm1.7}$ | 93.8$_{\pm2.2}$ | 90.9$_{\pm3.8}$ | 83.9$_{\pm5.4}$ | 91.2$_{\pm2.8}$ | 84.3$_{\pm4.8}$ | 92.0$_{\pm2.5}$ | 76.1$_{\pm9.5}$ | 76.8$_{\pm7.3}$ |
| | | avg | - | **96.2** | 95.7 | 95.9 | 95.3 | 94.3 | 85.3 | 81.7 | 87.9 | 84.7 | 91.7 | 73.3 | 73.5 |
| reg | MAE | nc-cpp | 6970 | **0.736**$_{\pm0.022}$ | 0.778$_{\pm0.038}$ | 0.740$_{\pm0.030}$ | 0.886$_{\pm0.030}$ | 0.892$_{\pm0.028}$ | 0.771$_{\pm0.018}$ | 0.791$_{\pm0.031}$ | 0.772$_{\pm0.049}$ | 0.785$_{\pm0.033}$ | 0.841$_{\pm0.057}$ | 0.937$_{\pm0.102}$ | 0.937$_{\pm0.026}$ |

Table 31: Performance of models on non-canonical peptide classification (ROC-AUC↑, %) and regression (MAE↓, %) with kmer-split. Dataset sizes are shown separately; results are mean$_{\pm std}$. Best and second-best scores per row are in **bold** and gray shadow.

| Task | Metric | Dataset | Size | FP-based models | | | | | GNN-based models | | | | SMILES-based models | | |
|---|---|---|---|---|---|---|---|---|---|---|---|---|---|---|---|
| | | | | RF | XGBoost | LightGBM | GradBoost | AdaBoost | GCN | GAT | GIN | Pepland | ChemBERTa | PeptideCLM | PepDoRA |
| cls | AUC ROC | nc-antibacterial | 1668 | 94.7$_{\pm0.9}$ | 94.8$_{\pm0.9}$ | **95.2**$_{\pm1.3}$ | 94.4$_{\pm0.8}$ | 91.6$_{\pm2.3}$ | 92.2$_{\pm3.0}$ | 80.6$_{\pm7.4}$ | 85.6$_{\pm12.0}$ | 83.6$_{\pm4.9}$ | 91.9$_{\pm2.8}$ | 72.2$_{\pm5.7}$ | 73.2$_{\pm5.6}$ |
| | | nc-antifungal | 407 | **98.9**$_{\pm1.1}$ | 98.7$_{\pm1.0}$ | 98.9$_{\pm0.6}$ | 98.3$_{\pm0.9}$ | 98.1$_{\pm0.9}$ | 71.8$_{\pm12.8}$ | 77.7$_{\pm9.8}$ | 89.6$_{\pm5.3}$ | 77.3$_{\pm20.1}$ | 91.4$_{\pm14.6}$ | 65.9$_{\pm3.9}$ | 57.2$_{\pm10.7}$ |
| | | nc-antimicrobial | 2465 | **97.9**$_{\pm0.9}$ | 97.4$_{\pm0.9}$ | 97.8$_{\pm0.8}$ | 97.6$_{\pm0.5}$ | 95.7$_{\pm1.4}$ | 93.5$_{\pm1.8}$ | 82.6$_{\pm5.8}$ | 92.2$_{\pm3.5}$ | 87.8$_{\pm1.5}$ | 96.1$_{\pm0.8}$ | 77.9$_{\pm3.0}$ | 81.1$_{\pm1.7}$ |
| | | nc-hemolytic | 3971 | 97.1$_{\pm0.9}$ | 97.4$_{\pm0.9}$ | **97.8**$_{\pm0.7}$ | 96.7$_{\pm0.9}$ | 94.0$_{\pm0.8}$ | 86.9$_{\pm4.2}$ | 81.8$_{\pm0.9}$ | 88.3$_{\pm3.7}$ | 83.2$_{\pm2.7}$ | 93.2$_{\pm1.2}$ | 75.1$_{\pm3.8}$ | 77.9$_{\pm3.1}$ |
| | | avg | - | 97.2 | 97.1 | **97.4** | 96.7 | 94.9 | 86.1 | 80.7 | 88.9 | 83.0 | 93.1 | 72.8 | 72.4 |
| reg | MAE | nc-cpp | 6970 | **0.358**$_{\pm0.007}$ | 0.360$_{\pm0.011}$ | 0.361$_{\pm0.005}$ | 0.408$_{\pm0.009}$ | 0.586$_{\pm0.009}$ | 0.460$_{\pm0.015}$ | 0.490$_{\pm0.018}$ | 0.459$_{\pm0.028}$ | 0.483$_{\pm0.021}$ | 0.400$_{\pm0.024}$ | 0.437$_{\pm0.022}$ | 0.591$_{\pm0.018}$ |

Table 32: Performance of models on non-canonical peptide classification (ROC-AUC↑, %) and regression (MAE↓, %) with MMseqs2-split. Dataset sizes are shown separately; results are mean$_{\pm std}$. Best and second-best scores per row are in **bold** and gray shadow.

| Task | Metric | Dataset | Size | FP-based models | | | | | GNN-based models | | | | SMILES-based models | | |
|---|---|---|---|---|---|---|---|---|---|---|---|---|---|---|---|
| | | | | RF | XGBoost | LightGBM | GradBoost | AdaBoost | GCN | GAT | GIN | Pepland | ChemBERTa | PeptideCLM | PepDoRA |
| cls | AUC ROC | nc-antibacterial | 1668 | **97.9**$_{\pm1.2}$ | 97.7$_{\pm0.7}$ | 97.8$_{\pm1.1}$ | 97.1$_{\pm1.1}$ | 95.7$_{\pm1.4}$ | 96.2$_{\pm1.6}$ | 89.9$_{\pm3.9}$ | 95.8$_{\pm3.5}$ | 90.7$_{\pm2.3}$ | 96.9$_{\pm1.6}$ | 84.5$_{\pm2.0}$ | 86.4$_{\pm3.5}$ |
| | | nc-antifungal | 407 | 95.4$_{\pm3.4}$ | 93.5$_{\pm2.9}$ | 93.6$_{\pm3.4}$ | 93.2$_{\pm4.6}$ | **95.4**$_{\pm2.5}$ | 73.8$_{\pm1.7}$ | 79.8$_{\pm5.6}$ | 85.2$_{\pm4.3}$ | 84.3$_{\pm5.9}$ | 92.2$_{\pm7.0}$ | 75.3$_{\pm5.0}$ | 61.5$_{\pm5.1}$ |
| | | nc-antimicrobial | 2465 | **98.0**$_{\pm0.4}$ | 97.8$_{\pm0.4}$ | 97.9$_{\pm0.5}$ | 97.8$_{\pm0.5}$ | 96.4$_{\pm0.6}$ | 95.8$_{\pm1.1}$ | 87.4$_{\pm3.6}$ | 91.6$_{\pm1.9}$ | 89.4$_{\pm1.1}$ | 96.7$_{\pm1.1}$ | 82.1$_{\pm1.3}$ | 87.5$_{\pm2.1}$ |
| | | nc-hemolytic | 3971 | 98.2$_{\pm0.4}$ | 98.1$_{\pm0.6}$ | **98.3**$_{\pm0.5}$ | 97.5$_{\pm0.7}$ | 95.7$_{\pm0.7}$ | 94.4$_{\pm1.1}$ | 89.8$_{\pm2.9}$ | 94.9$_{\pm1.1}$ | 90.8$_{\pm0.7}$ | 96.2$_{\pm1.1}$ | 85.4$_{\pm2.5}$ | 87.9$_{\pm1.2}$ |
| | | avg | - | **97.4** | 96.8 | 96.9 | 96.4 | 95.8 | 90.1 | 86.7 | 92.9 | 88.8 | 95.5 | 81.8 | 80.8 |
| reg | MAE | nc-cpp | 6970 | 0.743$_{\pm0.015}$ | 0.780$_{\pm0.012}$ | **0.722**$_{\pm0.021}$ | 0.846$_{\pm0.018}$ | 0.901$_{\pm0.011}$ | 0.768$_{\pm0.013}$ | 0.803$_{\pm0.030}$ | 0.792$_{\pm0.035}$ | 0.783$_{\pm0.025}$ | 0.875$_{\pm0.027}$ | 0.931$_{\pm0.096}$ | 0.936$_{\pm0.017}$ |

Table 33: Performance of models on non-canonical peptide classification (ROC-AUC↑, %) and regression (MAE↓, %) with random-split. Dataset sizes are shown separately; results are mean$_{\pm std}$. Best and second-best scores per row are in **bold** and gray shadow.

| Task | Metric | Dataset | Size | FP-based models | | | | | GNN-based models | | | | SMILES-based models | | |
|---|---|---|---|---|---|---|---|---|---|---|---|---|---|---|---|
| | | | | RF | XGBoost | LightGBM | GradBoost | AdaBoost | GCN | GAT | GIN | Pepland | ChemBERTa | PeptideCLM | PepDoRA |
| cls | AUC ROC | nc-antibacterial | 1668 | 98.4$_{\pm1.1}$ | 98.4$_{\pm1.0}$ | **98.7**$_{\pm0.9}$ | 98.1$_{\pm1.0}$ | 97.0$_{\pm1.8}$ | 97.1$_{\pm0.6}$ | 94.1$_{\pm1.5}$ | 96.6$_{\pm2.0}$ | 92.3$_{\pm2.8}$ | 97.8$_{\pm1.1}$ | 85.5$_{\pm4.3}$ | 86.0$_{\pm3.7}$ |
| | | nc-antifungal | 407 | 97.5$_{\pm2.5}$ | 96.6$_{\pm2.6}$ | 96.2$_{\pm3.2}$ | 96.2$_{\pm3.1}$ | 95.1$_{\pm3.6}$ | 85.1$_{\pm11.6}$ | 83.4$_{\pm5.8}$ | 85.1$_{\pm8.5}$ | 85.8$_{\pm9.7}$ | 95.2$_{\pm4.9}$ | 79.1$_{\pm6.4}$ | 72.2$_{\pm11.6}$ |
| | | nc-antimicrobial | 2465 | **99.1**$_{\pm0.4}$ | 98.9$_{\pm0.3}$ | 99.0$_{\pm0.3}$ | 98.6$_{\pm0.3}$ | 97.7$_{\pm0.6}$ | 96.2$_{\pm1.7}$ | 91.6$_{\pm1.9}$ | 97.0$_{\pm1.0}$ | 92.7$_{\pm1.7}$ | 97.9$_{\pm0.8}$ | 88.2$_{\pm1.5}$ | 85.9$_{\pm2.7}$ |
| | | -hemolytic | 3971 | **99.0**$_{\pm0.2}$ | 98.7$_{\pm0.2}$ | 98.9$_{\pm0.2}$ | 98.2$_{\pm0.5}$ | 96.6$_{\pm0.8}$ | 95.7$_{\pm0.9}$ | 89.1$_{\pm1.9}$ | 94.3$_{\pm3.0}$ | 89.1$_{\pm1.5}$ | 97.6$_{\pm0.8}$ | 83.6$_{\pm2.2}$ | 87.2$_{\pm2.1}$ |
| | | avg | - | **98.5** | 98.1 | 98.2 | 97.8 | 96.6 | 93.5 | 89.5 | 93.3 | 90.0 | 97.1 | 84.1 | 82.9 |
| reg | MAE | nc-cpp | 6970 | **0.357**$_{\pm0.007}$ | 0.360$_{\pm0.011}$ | 0.361$_{\pm0.005}$ | 0.408$_{\pm0.009}$ | 0.570$_{\pm0.022}$ | 0.468$_{\pm0.018}$ | 0.504$_{\pm0.022}$ | 0.466$_{\pm0.022}$ | 0.468$_{\pm0.021}$ | 0.404$_{\pm0.019}$ | 0.446$_{\pm0.046}$ | 0.586$_{\pm0.013}$ |

Table 34: Performance of models on PepPI classification (ROC-AUC↑, %) and regression (MAE↓, %) with cold-start-split. Dataset sizes are shown separately; results are mean$_{\pm std}$. Best and second-best scores per row are in **bold** and gray shadow.

| Task | Metric | Dataset | Size | FP-based models | | GNN-based models | | | | SMILES-based models | | | PLM-based models | | | | |
|---|---|---|---|---|---|---|---|---|---|---|---|---|---|---|---|---|---|
| | | | | ECFP4 | ECFP6 | GCN | GAT | GIN | Pepland | ChemBERTa | PeptideCLM | PepDoRA | ESM2-8M | ESM2-35M | ESM2-150M | ESM2-650M | ESM2-150M-F |
| cls | AUC ROC | ppi | 44148 | 48.7$_{\pm3.2}$ | 46.7$_{\pm3.9}$ | 47.5$_{\pm2.3}$ | 49.5$_{\pm4.3}$ | 51.1$_{\pm3.9}$ | 50.5$_{\pm3.5}$ | 51.9$_{\pm4.4}$ | 52.4$_{\pm2.9}$ | 51.0$_{\pm2.5}$ | 50.6$_{\pm6.6}$ | 51.8$_{\pm4.4}$ | 52.6$_{\pm2.3}$ | **56.0**$_{\pm3.1}$ | 55.0$_{\pm2.9}$ |
| reg | MAE | ppi_ba | 1433 | 1.040$_{\pm0.052}$ | 1.052$_{\pm0.063}$ | 1.103$_{\pm0.032}$ | 1.128$_{\pm0.030}$ | 1.112$_{\pm0.009}$ | 1.091$_{\pm0.048}$ | 1.053$_{\pm0.025}$ | 1.124$_{\pm0.104}$ | 1.125$_{\pm0.031}$ | 1.059$_{\pm0.043}$ | 1.052$_{\pm0.062}$ | 1.062$_{\pm0.060}$ | 1.082$_{\pm0.013}$ | **1.026**$_{\pm0.043}$ |
| | | nc_ppi_ba | 278 | 1.425$_{\pm0.249}$ | 1.462$_{\pm0.177}$ | 1.468$_{\pm0.183}$ | 1.451$_{\pm0.218}$ | 1.489$_{\pm0.208}$ | 1.584$_{\pm0.144}$ | 1.473$_{\pm0.217}$ | 1.474$_{\pm0.129}$ | **1.422**$_{\pm0.100}$ | - | - | - | - | - |

Table 35: Performance of models on PepPI classification (ROC-AUC↑, %) and regression (MAE↓, %) with cold-start-split. Dataset sizes are shown separately; results are mean$_{\pm std}$. Best and second-best scores per row are in **bold** and gray shadow.

| Task | Metric | Dataset | Size | FP-based models | | GNN-based models | | SMILES-based models | | | PLM-based models | | | | |
|---|---|---|---|---|---|---|---|---|---|---|---|---|---|---|---|
| | | | | ECFP4 | ECFP6 | GIN | Pepland | ChemBERTa | PeptideCLM | PepDoRA | ESM2-8M | ESM2-35M | ESM2-150M | ESM2-650M | ESM2-150M-F |
| cls | AUC ROC | ppi | 44148 | 54.4$_{\pm2.3}$ | 53.7$_{\pm2.3}$ | 61.3$_{\pm7.2}$ | 59.6$_{\pm2.7}$ | 52.0$_{\pm4.4}$ | 51.4$_{\pm3.6}$ | 59.4$_{\pm2.8}$ | **60.2**$_{\pm6.3}$ | 57.6$_{\pm7.2}$ | 55.4$_{\pm3.5}$ | 51.9$_{\pm2.4}$ | 56.0$_{\pm3.1}$ |
| reg | MAE | ppi_ba | 1433 | 1.043$_{\pm0.050}$ | 1.043$_{\pm0.060}$ | 1.189$_{\pm0.140}$ | 1.176$_{\pm0.152}$ | **1.034**$_{\pm0.013}$ | 1.128$_{\pm0.042}$ | 1.084$_{\pm0.044}$ | 1.061$_{\pm0.043}$ | 1.056$_{\pm0.092}$ | 1.079$_{\pm0.038}$ | 1.051$_{\pm0.072}$ | 1.038$_{\pm0.030}$ |
| | | nc_ppi_ba | 278 | 1.665$_{\pm0.260}$ | 1.647$_{\pm0.286}$ | 1.741$_{\pm0.359}$ | 1.705$_{\pm0.366}$ | 1.613$_{\pm0.189}$ | 1.580$_{\pm0.142}$ | **1.465**$_{\pm0.234}$ | - | - | - | - | - |

Table 36: Models' classification (ROC-AUC↑, %) performance on the canonical peptide dataset (length $> 150$) using hybrid-split. Dataset sizes are shown separately; results are mean$_{\pm std}$. Best and second-best scores per row are in **bold** and gray shadow.

| Dataset | Size | FP-based models | | | | | GNN-based models | | | |
|---|---|---|---|---|---|---|---|---|---|---|
| | | RF | XGBoost | LightGBM | GradBoost | AdaBoost | GCN | GAT | GIN | Pepland |
| antimicrobial | 52941 | $88.0_{\pm0.3}$ | $87.5_{\pm0.5}$ | $87.3_{\pm0.4}$ | $85.1_{\pm0.3}$ | $82.5_{\pm0.4}$ | $81.4_{\pm0.7}$ | $72.3_{\pm1.5}$ | $80.2_{\pm1.6}$ | $80.2_{\pm1.3}$ |
| antifungal | 12887 | $87.0_{\pm1.0}$ | $86.0_{\pm0.7}$ | $85.6_{\pm0.8}$ | $83.8_{\pm0.6}$ | $82.0_{\pm0.8}$ | $76.0_{\pm4.4}$ | $73.5_{\pm1.6}$ | $78.9_{\pm2.9}$ | $75.2_{\pm2.4}$ |
| antiparasitic | 6755 | $86.8_{\pm1.2}$ | $86.2_{\pm1.7}$ | $86.5_{\pm0.8}$ | $84.5_{\pm1.1}$ | $82.8_{\pm1.6}$ | $75.6_{\pm1.5}$ | $75.8_{\pm1.4}$ | $74.4_{\pm2.6}$ | $74.8_{\pm2.3}$ |
| anticancer | 12013 | $86.9_{\pm0.8}$ | $86.4_{\pm0.6}$ | $86.0_{\pm0.5}$ | $83.3_{\pm0.8}$ | $81.0_{\pm0.8}$ | $75.0_{\pm4.1}$ | $73.7_{\pm1.0}$ | $77.3_{\pm4.3}$ | $78.1_{\pm1.2}$ |
| allergen | 2538 | $76.5_{\pm2.9}$ | $82.2_{\pm2.0}$ | $82.2_{\pm1.4}$ | $79.4_{\pm1.7}$ | $66.8_{\pm2.8}$ | $63.3_{\pm1.6}$ | $59.9_{\pm4.7}$ | $67.2_{\pm2.7}$ | $68.8_{\pm2.5}$ |
| Average | - | 85.0 | 85.6 | 85.5 | 83.2 | 79.0 | 74.3 | 71.0 | 75.6 | 75.4 |

| Dataset | Size | SMILES-based models | | | PLM-based models | | | |
|---|---|---|---|---|---|---|---|---|
| | | ChemBERTa | PeptideCLM | PepDoRA | ESM2-8M | ESM2-35M | ESM2-150M | ESM2-150M-F |
| antimicrobial | 52941 | $88.0_{\pm1.1}$ | $52.7_{\pm10.4}$ | $77.9_{\pm3.4}$ | $91.8_{\pm0.8}$ | $91.8_{\pm0.4}$ | $92.4_{\pm0.4}$ | $\mathbf{93.3_{\pm1.2}}$ |
| antifungal | 12887 | $84.1_{\pm1.5}$ | $50.7_{\pm6.7}$ | $70.3_{\pm2.2}$ | $90.9_{\pm0.5}$ | $91.5_{\pm1.6}$ | $\mathbf{92.6_{\pm0.9}}$ | $91.8_{\pm0.7}$ |
| antiparasitic | 6755 | $82.0_{\pm1.6}$ | $59.2_{\pm4.7}$ | $69.2_{\pm1.8}$ | $91.6_{\pm0.7}$ | $91.9_{\pm1.2}$ | $\mathbf{92.8_{\pm1.2}}$ | $92.6_{\pm0.8}$ |
| anticancer | 12013 | $87.0_{\pm1.6}$ | $55.5_{\pm1.3}$ | $70.9_{\pm2.3}$ | $92.4_{\pm0.9}$ | $93.1_{\pm0.6}$ | $\mathbf{93.3_{\pm0.5}}$ | $92.8_{\pm0.7}$ |
| allergen | 2538 | $74.8_{\pm3.1}$ | $53.8_{\pm2.3}$ | $55.7_{\pm1.6}$ | $84.0_{\pm1.1}$ | $83.4_{\pm3.1}$ | $\mathbf{87.3_{\pm2.6}}$ | $85.9_{\pm1.8}$ |
| Average | - | 83.2 | 54.4 | 68.8 | 90.1 | 90.4 | **91.7** | 91.3 |

# I  NON-NATURAL PEPTIDE GENERATION MODEL

## I.1  MOTIVATION FOR A SEQUENCE-TO-SEQUENCE GENERATION MODEL

Non-natural peptides, which incorporate non-canonical amino acids or modified backbones, represent a significant extension of their natural counterparts. They exhibit enhanced pharmacokinetic properties, such as improved stability and oral bioavailability, which directly address the principal limitations of traditional peptide therapeutics, including rapid *in vivo* degradation. These attributes have expanded their utility in diverse therapeutic areas, from antimicrobial agents to anticancer treatments and membrane translocation.

The systematic design of non-natural peptides, however, faces substantial challenges. The primary obstacle is the pronounced data scarcity in public repositories. This scarcity, compounded by the vast and heterogeneous chemical space of non-canonical monomers, severely constrains the applicability of mainstream deep learning models. Consequently, a methodological gap persists, where prior methods for sequence generation, predominantly tailored for natural peptides or small molecules, lack effective mechanisms to model the complex sequence-property relationships in the non-natural domain. Applying these models without appropriate biophysical and synthetic constraints often yields chemically implausible structures.

This work introduces a sequence-to-sequence framework for the generation of non-natural peptides from natural peptide templates. We employ a large language model, specifically a GPT-2 architecture, to capture the implicit grammatical and semantic correspondences between natural and non-natural sequences. Our approach facilitates the generation of a virtual library of non-natural peptides whose designs are guided by existing natural sequences, thereby enhancing their potential chemical and pharmacological relevance. The framework is designed to improve the controllability and diversity of sequence generation, providing a structured methodology for exploring the non-natural peptide space.

## I.2  MODEL ARCHITECTURE, DATA, AND TRAINING PROCEDURE

**Sequence-to-Sequence Formulation.**   We formulate the generation of non-natural peptides as a conditional sequence-to-sequence conversion task, where a natural peptide sequence serves as the input condition from which a corresponding non-natural sequence is generated. We select the GPT-2 architecture as the backbone model, given its established capacity for contextual modeling in analogous translation and generation tasks.

**Data Representation and Preprocessing.**   To support this formulation, a custom vocabulary was constructed which explicitly defines all canonical amino acids and a set of task-specific symbols, including `[NATURAL]`, `[BILN]`, and `[EOS]`, as discrete tokens. The model's token embedding layer was resized to accommodate this expanded vocabulary, where the `[EOS]` token also serves as the padding token. The training corpus consists of paired sequences, each structured as `[NATURAL]` < natural_seq > `[BILN]` <non-natural_seq> `[EOS]`. All sequences were tokenized and processed to a maximum length of 768, where shorter inputs were padded while longer ones were truncated. Data handling was managed using the Hugging Face `datasets` library.

**Model Training and Optimization.**   The model was fine-tuned using a causal language modeling (CLM) objective within the Hugging Face `Trainer` framework. Optimization was performed with the AdamW algorithm, which was configured with a learning rate of $1 \times 10^{-4}$ and a weight decay coefficient of 0.01. A linear learning rate warm-up schedule was applied over the initial 500 steps. The model was trained for 500 epochs with a batch size of 32, leveraging mixed-precision (FP16) computation to ensure resource efficiency on a single NVIDIA A800 GPU.

**Inference and Post-processing.**   For sequence generation, the model checkpoint exhibiting the optimal validation performance was utilized. Inference was initiated with a prompt containing the natural sequence and the `[BILN]` tag, after which the model autoregressively completed the non-natural sequence via greedy decoding. A rule-based post-processing procedure was subsequently applied to extract the sequence segment between the `[BILN]` and `[EOS]` tokens and to remove

Table 37: Comprehensive evaluation of generated molecules across multiple dimensions.

| Category | Metric | Value |
|---|---|---|
| Validity | BLIN→SMILES Success Rate | 89.7% |
| | Chemical Validity Rate | 100% |
| | Overall Success Rate | 89.7% |
| Closeness to Ground Truth | Exact Match Rate | 21.9% |
| | Avg. Tanimoto Similarity | 87.9% |
| Diversity and Novelty | Internal Diversity | 66.7% |
| | Novelty Ratio | 76.5% |
| Language Modeling | Perplexity (PPL) | 1232.0 |

any residual special symbols, thereby yielding the final non-natural peptide sequence. All implementations were based on the PyTorch and Hugging Face Transformers libraries.

## I.3 EVALUATION METRICS AND RESULTS FOR GENERATION QUALITY

**Chemical Validity**   The chemical validity of the generated sequences was assessed based on the success rates of their conversion to SMILES representations and the subsequent validation thereof by RDKit. The model achieved high rates for both metrics (Table 37), which confirms its capacity to generate chemically plausible molecular structures from the target sequence representation.

**Fidelity to Reference Structures**   Fidelity was quantified using two complementary metrics, which were the exact match rate against reference sequences and the average Tanimoto similarity of Morgan fingerprints (radius=2, 1024 bits). The model yielded a low exact match rate, whereas the structural similarity remained high. This outcome indicates that the model generalizes beyond simple sequence replication, enabling the generation of structurally analogous yet novel molecules.

**Physicochemical Property Distribution**   A comparison of physicochemical property distributions was conducted between the generated molecular ensemble and the training set. The properties analyzed included molecular weight (MolWt), lipophilicity (LogP), hydrogen bond donor and acceptor counts (NumHDonors, NumHAcceptors), and topological polar surface area (TPSA). The generated molecules exhibit distributions closely aligned with the training set (Figure 112), confirming that the model preserves the underlying statistical characteristics of the reference data.

**Diversity and Novelty**   Diversity was measured by internal pairwise Tanimoto similarity, while novelty was defined as the fraction of generated molecules absent from the training set. The model produced a generated set with high internal diversity and a substantial novelty ratio (Table 37). These two results, taken together, demonstrate the model's ability to explore the chemical space beyond the training data.

**Sequence Modeling Performance**   The model's sequence modeling capability was evaluated using perplexity (PPL) on the validation set. A final perplexity of 1.23e3 was achieved, which indicates a robust capture of the statistical patterns inherent in the non-natural peptide sequence representation.

## I.4 INFERENCE PIPELINE FOR LARGE-SCALE DATASET GENERATION

**Inference Configuration**   For generation, the trained model and tokenizer were loaded, with the model set to evaluation mode. The padding token was mapped to the end-of-sequence token to ensure consistent autoregressive decoding. Generation was conditioned on prompts which follow a unified template, `[NATURAL] natural_peptide_sequence [BILN]`, thereby maintaining structural consistency with the training format. The maximum number of new tokens to be generated was dynamically adjusted relative to the input sequence length within each batch, a process which, combined with batch processing, improves computational efficiency. The generation process accommodates both greedy decoding for deterministic output and temperature-controlled sampling, where the temperature parameter modulates output stochasticity.

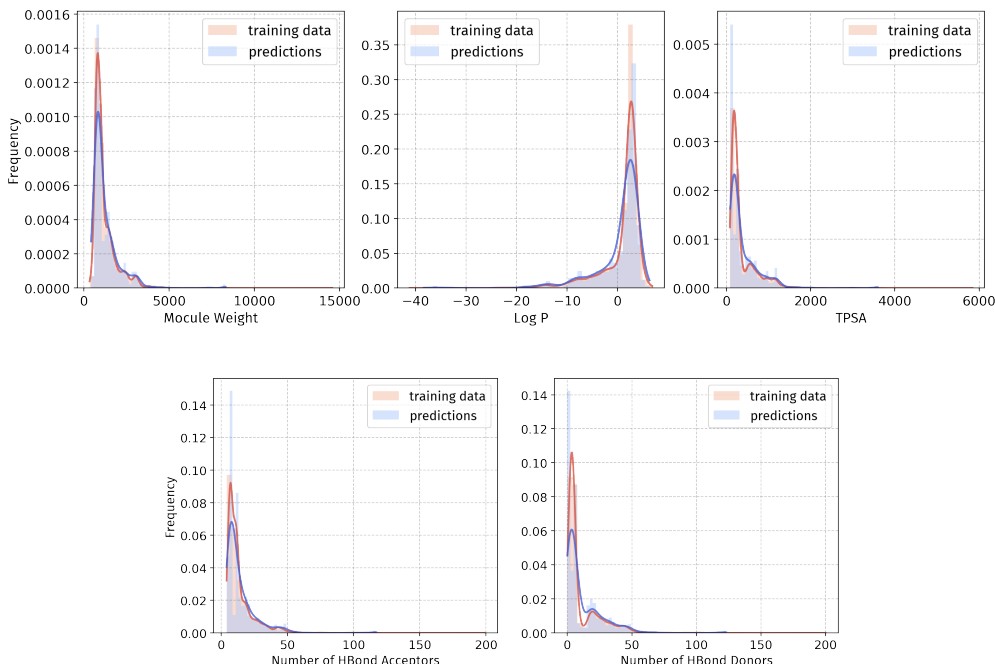

Figure 112: Comparison of physicochemical property distributions between generated molecules and the training set.

**Application and Performance**    The described protocol was applied to convert a corpus of 4,323 natural peptide sequences derived from four distinct datasets. We conducted eight independent generation runs using a range of temperature parameters, from which up to five chemically valid non-natural sequences were retained per input. This procedure successfully yielded corresponding non-natural peptide sequences for 4,221 of the inputs, which represents a conversion success rate of 97.6

## J    TOOLKIT SUPPORT

To facilitate reproducible and standardized dataset construction for the research community, we introduce `PepBenchmark`, a Python package designed with a modular and extensible architecture. The framework allows users to compose, customize, and extend pipeline components to meet diverse experimental requirements. It integrates peptide-specific utilities for dataset analysis and visualization, and provides six splitting strategies—including $k$-mer, MMseqs2, ECFP, random, yydra, and protein cold-start—to ensure rigorous control of partitioning under sequence similarity and distributional constraints. In addition, `PepBenchmark` offers negative-sampling management tools that support the construction of dynamic negative pools with adjustable similarity thresholds and physicochemical property matching. A sequence similarity analysis module is also included for redundancy removal and identity-based filtering, thereby guaranteeing strict benchmarking conditions.

`PepBenchmark` is under active development, and we plan to continuously integrate additional functionalities that are broadly applicable to peptide-related machine learning tasks. The source code is provided in the supplementary material and will be made publicly available.

Below we illustrate typical usage patterns.

**Basic Usage** For users who only need standardized datasets, `PepBenchmark` offers ready-to-use processed datasets that can be loaded, split, trained, and evaluated with just a few lines of code.

```python
# 1. Load a standardized dataset
dataset = SinglePeptideDatasetManager(
    dataset_name="bbp",
    official_feature_names=["fasta", "label"]
)

# 2. Set the split strategy (reproducible with a fixed seed)
dataset.set_official_split_indices(split_type="mmseqs2_split", fold_seed
    =0)

# 3. Initialize a baseline model
model = PLM(model="facebook/esm2_t30_150M_UR50D", dataset=dataset)

# 4. Train and evaluate
model.run(epochs=50, metrics=["roc-auc"])

# Users may also export processed features to train their own models
train_features, val_features, test_features = dataset.
    get_train_val_test_features(format="dict")
```

**Advanced Usage** For advanced users who wish to build customized datasets, the package exposes modular utilities for redundancy removal, negative sampling, splitting, and feature conversion.

```python
# 1. Remove redundancy using MMseqs2 (sequence identity threshold = 0.9)
dedup_seqs, result = remove_redundancy(pos_seq, method="mmseqs2",
    identity=0.9)

# 2. Negative sampling pipeline
# 2.1 Construct a negative pool from existing datasets
pool_manager = SamplingPoolManager(include_datasets=["cpp", "bbp"])

# 2.2 Remove sequences in the pool that are too similar to the positive
    set (identity threshold = 0.6)
pool_manager.remove_similar_to_positives(dedup_seqs, method="mmseqs2",
    threshold=0.6)

# 2.3 Initialize a sampler that generates negatives with distributional
    control
neg_sampler = NegSampler(pool_manager.get_sampling_pool(), dedup_seqs)

# 2.4 Sample negatives while matching key physicochemical properties
neg_seqs = neg_sampler.sample_negatives(
    ratio=1, # 1:1 ratio of negatives to positives
    method="bin", # property-matching method
    properties=["length", "charge"] # distributions to preserve
)

# 2.5 Combine positive and negative sequences into a single dataset
fasta_list = pos_seq + neg_seqs
labels = [1] * len(pos_seq) + [0] * len(neg_seqs)

# 3. Dataset splitting with hybridSplitter
# - Splits into train/validation/test while controlling redundancy and
    motif enrichment
hybrid_splitter = HybridSplitter(
    fasta_list, labels,
    frac_train=0.8,
    frac_valid=0.1,
    frac_test=0.1,
    cluster_distribution_strategy="sort_cluster_size", # distribute
        clusters by size
```

```
    preserve_cluster_integrity=True, # ensure clusters remain intact
    seed=42, # reproducibility
    ks=5, # number of random splits
    test_method="fisher", # statistical test for motif enrichment
    alternative="greater",
    min_cluster_size=3, # motifs must cover at least 3 samples
    min_pos=3, # motif must appear at 3 times
    mode="all",
    pval_threshold=0.05, # significance cutoff
    min_score=4.0, # enrichment score threshold
    fdr_correct=True, # apply FDR correction
    min_support=5, # minimum supporting sequences
    min_jaccard=0.6 # Jaccard similarity cutoff
)
split_res = hybrid_splitter.get_split_indices()

# 4. Feature transformation: convert FASTA sequences into SMILES strings
converter = Fasta2Smiles()
smiles_list = converter(fasta_list)
```

