# MORE RELATED WORK

## 1 CANONICAL PEPTIDE PREDICTION

Machine learning models have been widely applied to predict therapeutically relevant properties of peptides, particularly canonical peptides. Existing approaches span classical machine-learning algorithms, graph neural networks (GNNs), recurrent neural networks (RNNs), Transformer architectures, and large protein language models (PLMs). Representative works include:

- **amPEPpy** (Lawrence et al., 2021) trains a Random Forest (RF) classifier on a diverse set of physicochemical descriptors to identify antimicrobial peptides, and employs a drop-column strategy to quantify feature importance.
- **HemoPI2** (Rathore et al., 2025) develops both classification and regression models to predict hemolytic peptides and their hemolytic concentration ($HC_{50}$). The classifier uses a hybrid RF-based approach incorporating motif features, whereas the regression model predicts $HC_{50}$ based on protein language model embeddings.
- **PepExplainer** (Zhai et al., 2024) proposes an explainable graph neural network (GNN) framework for predicting and optimizing the bioactivity of macrocyclic peptides derived from selection-based libraries. It introduces a substructure-masking explanation (SME) mechanism to estimate atomic-level contributions and leverages transfer learning to improve $IC_{50}$ prediction on enriched libraries.
- **AMPainter** (Dong et al., 2025) incorporates a high-accuracy antimicrobial activity predictor—HyperAMP, a hypergraph neural network (HGNN) model in which: nodes represent individual amino-acid residues, hyperedges represent 2-gram, 3-gram, or 4-gram contiguous fragments that correlate with AMP activity, and node features are 768-dimensional embeddings from the pretrained PLM Ankh, which outperforms alternatives such as ESM-1b and ProtT5.
- **APEX** (Wan et al., 2024) adopts a sequence-encoder + dual-decoder multi-task architecture. The encoder uses an RNN with attention to extract hidden sequence features—such as hydrophobic patterns or charge distribution—yielding a 566-dimensional representation. Its two decoders perform: regression (predicting MIC values against 34 bacterial species, with specific optimization for ESKAPE pathogens), and classification (distinguishing AMPs from non-AMPs using 5,093 AMPs and 5,500 non-AMPs curated from the DBAASP database).
- **AMPDesigner** (Wang et al., 2025b) trains a GPT-based large language model on peptide sequences, and shows that the resulting embeddings enable highly accurate MIC prediction and guide LLM-based peptide design.
- **PepBERT** (Du and Li, 2025) presents lightweight Transformer-based peptide language models trained from scratch on peptide-specific corpora. Available in small (1.86M) and large (4.9M) versions, PepBERT yields compact yet powerful peptide representations and outperforms or matches ESM-2 on eight of nine downstream peptide-property prediction tasks.
- **PeptideBERT** (Guntuboina et al., 2023) fine-tunes ProtBERT (Elnaggar et al., 2020) on peptide-only datasets and achieves state-of-the-art performance in predicting hemolysis and non-fouling properties.

## 2 NON-CANONICAL PEPTIDE PREDICTION

While the advances in ML approaches for canonical peptides highlight their effectiveness, modeling non-canonical peptides remains less explored, primarily due to the limited availability of data. Sev-

eral methods based on GNNs, pre-trained small-molecule language models, and transfer learning have been applied to address this challenge:

- **TransSAFP** (Liu et al., 2025) constructs a balanced antimicrobial activity prediction dataset across all 11 N-terminal types. The model undergoes pre-training on a public natural peptide dataset, followed by fine-tuning with the SAFP dataset for improved predictive performance.

- **PepLand** (Zhang et al., 2025) introduces a novel pre-training architecture using a multi-view heterogeneous graph neural network to analyze peptide stability, degradation resistance, and biomolecular interactions. It effectively predicts properties such as protein–protein interactions, permeability, solubility, and synthesizability.

- **PepDoRA** (Wang et al., 2024) is a unified peptide representation model that applies Weight-Decomposed Low-Rank Adaptation (DoRA) to fine-tune ChemBERTa-77M-MLM for peptide property prediction. It captures functional properties of both modified and unmodified peptides, enabling accurate predictions of membrane permeability, non-fouling behavior, hemolysis propensity, and target-specific binding.

- **PeptideCLM** (Feller and Wilke, 2025) introduces a peptide-specific chemical language model capable of encoding peptides with various chemical modifications, including non-canonical amino acids and cyclic structures. It outperforms models such as ChemBERTa and ChemBERTa-2, particularly in predicting the transmembrane diffusion of cyclic peptides.

## 3 CANONICAL SEQUENCE GENERATION

Peptide generation tasks can be broadly categorized into sequence generation and structure generation. The frameworks are typically unconditional, conditional, or reinforcement learning (RL)-based. Key high-quality works include:

- **Peptide-GPT** (Shah et al., 2024) is based on the ProtGPT2 architecture and is trained to generate peptides with specific bioactivities such as hemolytic activity, solubility, and anti-fouling behavior.

- **AMPDesigner** (Wang et al., 2025b) proposes a generative framework that integrates GPT, prompt tuning, knowledge distillation, and RL to ensure diversity and reduce computational cost, focusing on antimicrobial peptide generation.

- **PepDiffusion** (Wang et al., 2025d) uses a latent diffusion model combined with molecular dynamics to generate antimicrobial peptides optimized for activity and structural properties.

- **Evolgradient** (Wang et al., 2025a) treats activity as an optimizable objective and iteratively refines amino-acid sequences using gradient-guided optimization, similar to in-silico directed evolution.

- **AMPainter** (Dong et al., 2025) unifies peptide sequence optimization and generation within a deep reinforcement learning framework.

- **EvoPlay** (Wang et al., 2023) draws inspiration from self-play in Go, exploring the vast mutation space and using multiple structural/functional simulators to train an agent that enhances protein function through mutation.

- **PepMLM** (Chen et al., 2025) fine-tunes ESM-2 with a masked language modeling strategy by masking C-terminal peptide regions bound to target proteins, enabling de novo binder generation.

- **EvoBind2** (Li et al., 2025) designs linear and cyclic peptide binders using only target-protein sequences. A modified AlphaFold2 predicts complex structures, and the peptide is mutated iteratively to minimize interface distance while maximizing structural confidence (pLDDT).

## 4 NON-CANONICAL PEPTIDE GENERATION

Recent models extend beyond canonical amino acids to incorporate non-canonical residues and diverse chemical modifications:

- **PepTune** (Tang et al., 2025) is a masked diffusion language model trained to generate peptide SMILES sequences from scratch. It is pretrained on large datasets of non-canonical amino acids and cyclic peptides, and uses classifier-guided Monte Carlo tree search for multi-objective optimization (e.g., dual-target binding, solubility, membrane permeability).

- **GPepT** (Oikawa et al., 2025) creates a vocabulary of over 17,000 non-canonical building blocks mined from ChEMBL. A language model trained on this vocabulary generates peptidomimetic sequences with rich structural diversity.

- **HELM-GPT** (Zhang et al., 2012) is a GPT-based cyclic peptide generator using HELM notation. It incorporates reinforcement learning for property-directed optimization (e.g., transmembrane diffusion, KRAS binding), comparing ten predictive models and selecting RF/XGB for key properties.

- **PepINVENT** (Geylan et al., 2025) extends the REINVENT platform to non-natural amino-acid design, enabling de novo exploration of modified peptides via RL-based property optimization.

- **PepThink R1** (Wang et al., 2025c) combines LLMs, chain-of-thought supervision, and RL to directly reason over monomer-level substitutions with pharmacology-aware reward shaping.