# OpenReview forum: "PepBenchmark: A Standardized Benchmark for Peptide Machine Learning"
_ICLR.cc/2026/Conference — ICLR 2026 Poster_

### Official Review · Reviewer_XKRu · 2025-10-28

**Soundness:** 3
**Presentation:** 3
**Contribution:** 2
**Rating:** 4
**Confidence:** 5

**Summary:**

The paper provides a collection of standard benchmarks and leaderboards for peptide representation learning. The authors collected and clean a large set of peptides including both canonical and non-canonical peptides. Negative sampling was proposed to extend the peptide datasets for machine learning.  Different splitting strategies were considered, including spliting by kmers and then spltting by sequence similarity using clustering methods. The authors consider different representations of the peptides including simple fingerprints, pretrained smiles representation, pretrained PLM representation (for canonical sequences) and difefrent learning approaches including tree-based ML methods like RF and LightGBM, graph neural networks and finetuning pretrained smiles or PLM models. The datasets, the splits and the benchmark results are reported on the rich datasets. Some conclusions were drawn from the experimental results based on the benchmark experiments.

**Strengths:**

In peptide representation, there is lack of standard leaderboards with fixed splits so that research in the field can rely on whenever there is a requirement for comparison between different methods. The proposed benchmark and frameworks are useful for that purpose.

The dataset cover broad types of tasks, even the datasets were collected elsewhere, the collection and bringing them to the same place will trigger standard benchmarking for the research field.

The conclusion and benchmarking results are interesting even it is known in the field that current not only limitted to non-canonical peptide in general for small molecules, fingerprint-based approaches are the best representation. For canonical representation PLM is the best. This is not new as previous work that the authors cited also had such similar results.

**Weaknesses:**

Although the datasets and the benchmark is useful, I think the technical contribution of the work is limited for the machine learning community. The paper may fits better a specific dataset and benchmark tracks rather a research main track.

Regarding the methods for collecting negative samples, the authors criticizes related work regarding false negatives possibility but their own method neither resolve that issue, I don't see how the proposed approach in the paper can help resolving the false negative may happen in the collected data.

**Questions:**

Could you please explain how the proposed methods of negative sampling would be better in related approach in resolving the false negative issues in the benchmark data?

---

> ### Author Response · Authors · 2025-11-22
> **Thank You and Responses to Reviewer’s Comments**
>
> We sincerely thank you for the careful and constructive evaluation of our work. We have provided detailed, point-by-point responses to all the **weaknesses** and **questions** raised. We hope that our clarifications adequately address your concerns, and **we would be very happy to engage in further discussion should any additional questions arise.**

---

> ### Author Response · Authors · 2025-11-22
> **Response to Weakness 1 （Part I — Clarification on Track Selection and Relevance to ML Community）**
>
> > **Weakness 1** Although the datasets and the benchmark is useful, I think the technical contribution of the work is limited for the machine learning community. The paper may fits better a specific dataset and benchmark tracks rather a research main track.
>
> We sincerely thank the reviewer for the constructive and detailed feedback. We are glad that you find our benchmark useful for establishing standardized evaluation protocols in peptide representation learning. Below, we provide a three-part response addressing:
> + **(1) the track selection and relevance to the ML community,**
> + **(2) the technical contributions of our work, and**
> + **(3) future directions enabled by our benchmark.**
>
> ## **1. Choice of Track**
>
> ### **(1) About benchmark papers and ICLR’s scope**
> Thank you for raising this point. We completely understand the concern. While some conferences—such as NeurIPS—explicitly separate the “main track” from a dedicated “dataset/benchmark track,” ICLR does **not** adopt such a division. Instead, ICLR officially includes **“datasets and benchmarks”** as one of its **Primary Areas**.
> We therefore selected ***“datasets and benchmarks”**  as our Primary Area during submission (visible directly below the “abstract” on this page). **We hope this clarifies that the intended positioning of our paper aligns with ICLR’s track structure.**
>
> ### **(2) Why we submit to the ML community rather than a biology-specific venue**
>
> We understand and appreciate the reviewer’s perspective regarding domain suitability. Compared with proteins, peptide machine learning remains relatively underexplored, and peptide-specific **foundation models** are still very rare. Many existing works rely on protein PLMs such as ESM2 or ProtBert, which, while effective in protein modeling, are not optimized for peptide-specific characteristics.
>
> By presenting our benchmark to the machine-learning community, we hope to **encourage ML researchers to engage in peptide modeling** and to lower the entry barrier by providing curated data, standardized evaluation pipelines, and strong baselines. In this way, PepBenchMark aims to facilitate the development of more effective peptide-focused models within the ML community.

---

> ### Author Response · Authors · 2025-11-22
> **Response to Weakness 1 （Part II — Technical Contributions of This Work）**
>
> ### **2. Technical contribution**
> Although our paper does not introduce a new model architecture, our **technical contributions lie in dataset construction, pipeline standardization, systematic evaluation, and empirical insights**. Below, we summarize them in a structured form.
>
> ####  **A. PepBenchData: A comprehensive and unified peptide database**
> We construct the largest AI-ready peptide dataset to date:
> - 35 datasets in total, including
>     - 29 canonical datasets (68,588 sequences)
>     - 6 non-canonical datasets (9,512 sequences)
> - covering 7 pharmacologically relevant task groups.
> - includes classification, regression, and PepPI tasks
> - provides BILN–HELM–SMILES conversion tools to unify non-canonical peptide representations
>
> All datasets are preprocessed and split, and **can be loaded in a few lines of Python**. **Appendix J** provides usage examples.
>
> #### **B. PepBenchPipeline: Identifying overlooked issues and providing practical solutions**
> During dataset construction, we **discovered several challenges that have been under-addressed in prior studies**, such as:
>
> - outliers in regression tasks,
> - k-mer leakage in data splitting,
> - distribution mismatch in negative sampling,
> - distribution mismatch and false negatives in negative sampling
> - special challenges for non-canonical peptides
>
> We provide standardized tools to help researchers avoid these issues. We believe this will meaningfully improve reproducibility and fairness in future studies.
>
> #### **C. PepBenchLeaderboard:  strong and accessible  baselines**
> Although we do not propose a new architecture, we release **ESM2-150M-F**, a peptide-finetuned version of ESM2-150M. This model has two important values:
> 1. ESM2-150M-F achieves SOTA performance across our benchmark, demonstrating that directly applying protein PLMs to peptides is suboptimal and highlighting the need for peptide-focused models.
> 2. We will release ESM2-150M-F on HuggingFace, providing the community with a strong and ready-to-use baseline that future peptide ML studies can build upon.
>
> We also conduct a **large-scale evaluation** covering:
> - **6 splitting strategies**:
>     hybrid-split, kmer-split, ecfp-split, cold-split, MMseqs2-split, random-split
> - **35 datasets**:
>     - classification (27 datasets) vs regression (8 datasets)
>     - canonical peptides (29 datasets) vs non-canonical peptides (6 datasets)
>     - single-peptide tasks (32 datasets) vs peptide–protein interaction (3 PepPI datasets)
>     - 7 pharmacological groups** (ADME, AMP, Metabolic, Tox, Others), corresponding to multiple stages of the drug-discovery pipeline.
> - **21 models** across four major architecture families:
>     PLM-based, SMILES-based, FP-based, and GNN-based.
>
> #### **D. Novel empirical findings enabled by our benchmark**
> Our large-scale experiments provide several new findings. **While a small subset of results aligns with earlier observations, our broader model coverage, larger dataset collection, and richer splitting strategies provide **stronger validation** and yield **new insights** that previous studies did not report**, such as:
> - performance patterns of different model architectures across task categories,
> - differences between PLM-based models on classification vs regression tasks,
> - impact of different splitting strategies,
> - influence of maximum peptide length constraints,
> - effects of pretraining,
> - benefits of peptide-aware finetuning for PLMs,
> - strengths of FP-based methods in low-data regimes.

---

> ### Author Response · Authors · 2025-11-22
> **Response to Weakness 1 （Part III — Future Directions Enabled by Our Benchmark）**
>
> ### **3.Future Directions Enabled by Our Benchmark**
> Through systematic dataset exploration and extensive experiments—although without introducing a new method—we also identify several promising research directions that future work may explore:
> 1. **Hybrid FP–PLM architectures.**
>     Since FP-based methods perform particularly well in low-data regimes, developing hybrid models that integrate FP-based and PLM-based representations may yield improved performance.
> 2. **Designing peptide-specific molecular fingerprints.**
>     Given the strong potential of FP-based methods, it would be valuable to systematically study different types of molecular fingerprints and design new fingerprints tailored specifically for peptides.
> 3. **Incorporating k-mer priors into model architectures.**
>     k-mer information is particularly important for peptides. Embedding explicit k-mer priors into model design could further enhance performance.
> 4. **Developing peptide-specific pretraining models and addressing catastrophic forgetting.**
>     The strong performance of ESM2-150M-F—achieved by simply finetuning ESM2-150M on peptide data—suggests that peptide-focused pretraining could be highly beneficial.
> 5. **Multi-granularity modeling combining atom-level and residue-level representations.**
>     In PepPI tasks, SMILES-based and GNN-based models perform well, suggesting that combining atom-level (SMILES/GNN) and residue-level (sequence/PLM) information may lead to multi-scale models with improved generalization.
> 6. **Pretraining language models on non-canonical peptide representations.**
>     Leveraging BILN and HELM representations to pretrain language models for non-canonical peptides is a promising direction to better support complex peptide chemistries.
> 7. **Hierarchical multi-label modeling for antimicrobial peptides.**
>     AMP datasets naturally exhibit a multi-label structure. Constructing hierarchical multi-label classification tasks and corresponding models may better capture the underlying biological relationships.
> 8. **Addressing activity cliffs in regression tasks.**
>     Tasks such as _E. coli MIC_ and _P. aeruginosa MIC_ exhibit strong activity-cliff phenomena, where highly similar peptides have drastically different labels. Developing models that are more robust to such cliffs is another valuable direction.
> ### **4. Conclusion**
> While our work does not introduce a new architecture, we believe the contributions presented here—comprehensive dataset construction, standardized and reproducible pipelines, large-scale and carefully controlled evaluations, a strong peptide-aware PLM baseline, and extensive empirical analyses—collectively provide meaningful value to the machine learning community. By addressing long-standing issues in peptide representation learning and offering a unified platform for benchmarking, we hope that PepBenchMark can serve as a solid foundation for future methodological developments and help catalyze further research in this emerging area.g.

---

### Official Review · Reviewer_26PZ · 2025-10-31

**Soundness:** 3
**Presentation:** 3
**Contribution:** 3
**Rating:** 6
**Confidence:** 4

**Summary:**

The paper presents PepBenchmark, a standardized benchmark for peptide machine learning. It integrates three components: PepBenchData, a curated collection of 35 datasets (29 canonical, 6 non-canonical) across 7 pharmacological tasks; PepBenchPipeline, a preprocessing framework featuring biologically informed negative sampling and hybrid data splitting to prevent leakage; and PepBenchLeaderboard, a unified evaluation across four model families: fingerprint, GNN, SMILES, and PLM. Experiments show that Protein Language Models (PLMs) outperform others, with peptide-specific finetuning and scaling improving performance. Fingerprint models remain strong for small and non-canonical datasets, while GNN and SMILES models excel in peptide–protein interaction tasks. PepBenchmark provides the first reproducible foundation for systematic peptide ML evaluation and model comparison.

**Strengths:**

1. Comprehensive scale: 35 datasets covering over 78k sequences across canonical, non-canonical, and interaction tasks enable unified evaluation.

2. Rigorous preprocessing: New negative sampling and hybrid-split methods reduce leakage and overestimation of model performance.

3. Empirical insight: Systematic benchmarking across 4 model types and 30 tasks reveals clear scaling laws and practical model guidance for peptide ML.

**Weaknesses:**

1. Authors might want to inlcude these works in the related works section
- https://www.nature.com/articles/s42256-023-00691-9
- https://www.nature.com/articles/s42004-025-01601-3
- https://www.nature.com/articles/s41587-025-02761-2
- https://arxiv.org/abs/2410.19222?

2. For data splitting, the authors use sequence-level splits. However, since some evaluated methods are graph-based or structure-based, a more rigorous approach might consider structural or graph similarity during splitting. At minimum, the authors could quantify the graph or structural similarity within and across splits or clusters to assess potential data leakage.

**Questions:**

1. In Lines 185–186, the authors state that all non-canonical peptides are converted to SMILES. How exactly was this conversion performed, and does it result in any **loss of structural or chemical information**?

2. In Lines 234–245, regarding **MMSeq**, what **exact command** and **sensitivity parameter** were used during clustering or filtering?

3. For **“Task correlation filtering,”** the paper mentions using expert prior knowledge and statistical analysis to estimate correlations among tasks and exclude closely related ones. Could the authors clarify **what specific expert knowledge** and **which statistical methods** were applied in this process?

4. The paper notes that the model takes a canonical peptide as input and generates a chemically modified non-canonical peptide, allowing transformation of a canonical negative set into a non-canonical one. Please provide **more details about this generative model**, including its architecture, training data, and how the modifications are controlled or validated.

---

> ### Author Response · Authors · 2025-11-22
> **Thank You and Responses to Reviewer’s Comments**
>
> We sincerely thank you for the careful and constructive evaluation of our work. We have provided detailed, point-by-point responses to all the **weaknesses** and **questions** raised. We hope that our clarifications adequately address your concerns, and **we would be very happy to engage in further discussion should any additional questions arise.**

---

> ### Author Response · Authors · 2025-11-22
> **Response to Weakness 1**
>
> > **Weekness 1.** _Authors might want to include some works in the related works section._
>
> We carefully reviewed the references you provided, including Peptide-GPT [1], PepMLM [2], EvoBind2 [3], and EvoPlay [4]. These works make valuable contributions to conditional and unconditional peptide/protein generation, binder design, and optimization, and are indeed highly relevant to the broader landscape of peptide machine learning.
>
> However, the primary focus of our paper is on benchmarking peptide prediction models. Accordingly, **the related work section in the main text concentrates on peptide property prediction methods, and existing standardized benchmarks in the life sciences** (e.g., TDC for small molecules, ProteinGym for proteins), with the goal of highlighting the current lack of a standardized and reproducible benchmark for peptide prediction and motivating the need for developing PepBenchmark.
>
> That said, we fully agree that a broader overview of peptide ML—including generative and non-canonical peptide modeling—would help readers better contextualize our work. Therefore, in the revised manuscript, we will **add a “More Related Work” section in the Appendix**, summarizing four important categories of related literature:
> 1. Canonical peptide prediction
> 2. Canonical peptide generation
> 3. Non-canonical peptide prediction
> 4. Non-canonical peptide generation
> We have also included a supplementary document (**“related_work.pdf”**) providing a concise survey of these areas. We sincerely appreciate the reviewer’s suggestion, which improves the completeness and clarity of our manuscript.
>
> ### **References**
>
> [1] Shah A, Guntuboina C, Farimani A B. _Peptide-GPT: Generative Design of Peptides using Generative Pre-trained Transformers and Bio-informatic Supervision._ arXiv:2410.19222, 2024.
> [2] Chen L T, Quinn Z, Dumas M, et al. _Target sequence-conditioned design of peptide binders using masked language modeling._ Nature Biotechnology, 2025.
> [3] Li Q, Vlachos E N, Bryant P. _Design of linear and cyclic peptide binders from protein sequence information._ Communications Chemistry, 2025, 8(1): 211.
> [4] Wang Y, Tang H, Huang L, et al. _Self-play reinforcement learning guides protein engineering._ Nature Machine Intelligence, 2023, 5(8): 845–860.

---

> ### Author Response · Authors · 2025-11-22
> **Response to Weakness 2**
>
> > **Weakness 2.** _For data splitting, the authors use sequence-level splits. However, since some evaluated methods are graph-based or structure-based, a more rigorous approach might consider structural or graph similarity during splitting. At minimum, the authors could quantify the graph or structural similarity within and across splits or clusters to assess potential data leakage._
>
> Thank you for this valuable suggestion. We fully agree that data-splitting strategies play a critical role in preventing leakage and ensuring rigorous evaluation. Below, we clarify why we adopt **sequence-level splitting** in the current version of PepBenchmark, and why **graph- or structure-based splitting is not directly applicable at this stage**.
> ### **1. Graph-based splitting is not necessary in our setting**
>
> In PepBenchmark, molecular graphs are derived directly from **SMILES strings**, not from experimentally determined or reliably predicted 3D structures.
>
> For peptides composed of the 20 standard amino acids, the topology of the graph representation is uniquely determined by the amino-acid sequence. As a result, **graph-level similarity is effectively equivalent to sequence similarity**, which is already controlled under sequence-level splitting. Even for peptides containing non-canonical residues, the structural scaffold remains predominantly governed by canonical backbones, meaning that graph topology still closely follows sequence patterns.
>
> In contrast to small molecules, peptides do not exhibit well-defined or diverse chemical scaffolds. **Scaffold-based strategies**—commonly used in small-molecule benchmarks such as TDC or MoleculeNet—therefore have limited applicability in peptide space.
>
> Therefore, a graph-based splitting protocol would provide **no additional leakage control** beyond what is already achieved by sequence-level splitting.
>
> ### **2. Structure-based splitting is conceptually valuable but currently infeasible**
>
> We acknowledge that structure-based splits could, in principle, offer insights into generalization across novel 3D conformations. However, implementing such a strategy in a peptide benchmark faces several substantial challenges:
>
> - **Lack of reliable structural data, especially for non-canconical peptides.**  Most peptides in our dataset do not have experimentally resolved structures. Although recent structure predictors (e.g., AlphaFold3) can generate candidate conformations, their accuracy for **short, flexible peptides** is substantially lower than for proteins due to the absence of stable folds and limited evolutionary information. Moreover, for **non-canonical peptides**, there are currently **no reliable prediction tools** capable of handling arbitrary chemical modifications with satisfactory accuracy.
> - **High intrinsic flexibility and conformational ensembles.**  Short peptides rarely adopt stable structures and instead exist as **conformational ensembles**. Unlike proteins, they lack persistent secondary-structure motifs or domain features that can anchor a stable structural similarity metric. Additionally, peptide conformations are **highly environment-dependent**, meaning the same sequence may adopt different folds under different conditions. This raises open questions such as:
> 	- How to select a representative structure?
> 	- Should structural similarity be computed over entire ensembles?
> 	- How to define meaningful thresholds for splitting?
> + **A structure-aware benchmark requires a fundamentally different design.**  We agree that structural information is valuable and worth incorporating. However, doing so would require substantial new infrastructure, including:
> 	1. collecting or generating reliable peptide structural ensembles,
> 	2. defining robust structure-based dataset splitting criteria, and
> 	3. evaluating a wide range of structure-aware or multimodal models.
>
> 	Such an undertaking constitutes a significantly larger effort that extends beyond the scope of our current **sequence-focused** benchmark.
>
> 	Therefore, in this work, we deliberately focus on **sequence-based evaluation**, and we regard the development of a comprehensive **structure-aware peptide benchmark** as an important direction for future work.
> ### **Conclusion**
>
> In summary, we adopt **sequence-level splitting** because graph-based splitting is effectively redundant for peptide graphs, and structure-based splitting is not yet feasible due to the lack of reliable, consistent structural information. We agree that integrating structure into future benchmark designs would be valuable, and we plan to explore structure-aware splitting strategies in subsequent versions of PepBenchmark.

---

> ### Author Response · Authors · 2025-11-22
> **Response to Question 1**
>
> > **Q1.** _In Lines 185–186, the authors state that all non-canonical peptides are converted to SMILES. How exactly was this conversion performed, and does it result in any loss of structural or chemical information?_
>
> Thank you for this question. In our work, all non-canonical peptides are converted to SMILES through a **deterministic and fully reproducible parsing-and-assembly pipeline**. This procedure relies on a complete monomer library derived from the original dataset and does **not introduce any structural simplification**, ensuring that the resulting SMILES retain the full chemical detail of the input representation.
>
> ### **1. Conversion Procedure**
> The conversion follows a deterministic multi-step pipeline:
> 1. **Monomer Library Construction**：We first collect all monomers appearing in non-canonical peptides from the original data sources—CycPeptMPDB [1] and Hemolytik [2]—and merge them with the BILN monomer library provided by pyPept [3]. This results in a local monomer library containing **621 monomers**.  For each monomer, the library stores:
> 	- the exact molecular structure (RDKit `Mol` object),
> 	- all R-groups / connection points,
> 	- leaving-group definitions for polymerization.
> 	All structural information was obtained from the original data sources and subsequently consolidated into a unified SDF file.
> 2. **Parsing BILN/HELM Sequences**
>     We implement a rule-based parser for BILN/HELM formats. The parser:
>     - identifies all monomers (including nested or parenthesized ones),
>     - reads explicit connectivity encoded in the notation,
>     - inserts implicit peptide bonds where appropriate,
>     - verifies validity of all bond identifiers.
>     The parsed output is stored in an intermediate `ParsedData` structure that explicitly records which monomers constitute the peptide and how they are connected to each other.
>
> 3. **Molecular Assembly (MolBuilder)**
>     For each monomer, we instantiate its RDKit structure, remove leaving groups at its R-group positions, and form covalent bonds strictly according to the connectivity rules captured in `ParsedData`.
>     All monomers are then merged into a single RDKit `Mol` object representing the complete peptide structure.
>
> 4. **SMILES Generation**
>     The final RDKit `Mol` object is serialized into SMILES using RDKit’s `MolToSmiles()` function, which produces a canonical and reproducible SMILES string.
>
> We will release our entire codebase, including the complete implementation of the conversion procedure, **to ensure full transparency and reproducibility**.
>
>
> ### **2. Does the Conversion Lose Structural or Chemical Information?**
> The conversion is **lossless**, for the following reasons:
> - All monomer structures and annotations (R-groups, connection points, stereochemistry, etc.) come **directly** from the SDF files of the original datasets and are **not modified** at any stage.
> - During parsing and assembly, all explicit and implicit bonding information is preserved and mapped to chemically valid covalent bonds in RDKit.
> - SMILES serves solely as a textual serialization of the fully constructed molecular graph; it does **not** remove or simplify any chemical features, including stereochemistry, bond order.
> Therefore, the transformation from BILN/HELM to SMILES retains the **complete structural and chemical fidelity** defined in the source data.
> In addition, **all parsing and conversion code used in this work is publicly released**, ensuring full transparency and reproducibility.
>
> ### Reference
> [1] Shah A, Guntuboina C, Farimani A B. Peptide-GPT: Generative Design of Peptides using Generative Pre-trained Transformers and Bio-informatic Supervision[J]. arXiv preprint arXiv:2410.19222, 2024.
>
> [2] Chen L T, Quinn Z, Dumas M, et al. Target sequence-conditioned design of peptide binders using masked language modeling[J]. Nature Biotechnology, 2025: 1-9.
>
> [3] Li Q, Vlachos E N, Bryant P. Design of linear and cyclic peptide binders from protein sequence information[J]. Communications Chemistry, 2025, 8(1): 211.
>
> [4] Wang Y, Tang H, Huang L, et al. Self-play reinforcement learning guides protein engineering[J]. Nature Machine Intelligence, 2023, 5(8): 845-860.

---

> ### Author Response · Authors · 2025-11-22
> **Response to Question 2**
>
> > **Q2.** _In Lines 234–245, regarding MMSeq, what exact command and sensitivity parameter were used during clustering or filtering?_
>
> Thank you for the question. We provide the full MMseqs2 command and parameter configurations in **Appendix D.1**. For redundancy removal and clustering of short peptides (≤50 aa), we adopted a **strict and high-sensitivity configuration** to maximally detect near-duplicate sequences (identity > 0.9). These settings ensure high sensitivity and accuracy for short-peptide clustering, allowing us to distinguish even subtle sequence variations.
>
> The **key parameters and their rationale** are summarized below:
>
> |Parameter|Value|Meaning / Rationale|
> |---|---|---|
> |–min-seq-id|0.90|Minimum sequence identity (90%); removes peptides differing by >10%.|
> |-c|0.9|Coverage threshold (90%).|
> |–cov-mode|0|Coverage relative to longer sequence; avoids clustering of short fragments into longer peptides.|
> |–mask|0|Disable low-complexity masking; retain functional motifs (e.g., Cys- and Lys-rich).|
> |–alignment-mode|3|Full alignment with start, end, and identity reported.|
> |–seq-id-mode|2|Identity based on aligned region without terminal gaps; robust to local indels.|
> |-s|8|High sensitivity search; minimizes missed near-duplicates.|
> |–kmer-per-seq|50|Extract up to 50 kmers per sequence; maximizes sensitivity for short peptides.|

---

> ### Author Response · Authors · 2025-11-22
> **Response to Question 3**
>
> > **Q3.** _For “Task correlation filtering,” the paper mentions using expert prior knowledge and statistical analysis to estimate correlations among tasks and exclude closely related ones. Could the authors clarify what specific expert knowledge and which statistical methods were applied in this process?_
> Thank you for the question. The full procedure is already outlined in Appendix D.2.2, and we summarize here the key expert knowledge and statistical analyses used for task-correlation filtering.
>
> ### **1. Expert Knowledge**
>
> Our assessments rely on **established biological mechanisms**, **published mechanistic literature**, and **consultation with peptide-domain experts**. A classical example is the relationship between antimicrobial peptides (AMPs) and anticancer peptides (ACPs): numerous studies [1–2] have shown that many AMPs also display anticancer activity due to shared physicochemical properties, similar membrane-disruptive mechanisms, and overlapping biological pathways.
>
> Extending this reasoning, we grouped all tasks into **three major mechanistic categories**, with a few datasets treated individually when no clear grouping applied. Within each category, tasks are expected to be biologically correlated due to similar mechanisms or physicochemical determinants:
> - **Membrane-interaction–related activities:** includes AMPs, ACPs, hemolytic peptides, and transmembrane peptides (TMPs). These activities are unified by peptide–membrane interactions. For example, most AMPs disrupt microbial membranes, ~70% also show hemolytic activity [3], AMPs and ACPs share strong physicochemical similarity [4], and AMPs can form TMP-like transient pores [5].
> - **Glucose-regulation–related activities:** includes DPP-IV inhibitory peptides, antidiabetic peptides, antioxidant peptides, and anti-inflammatory peptides. DPP-IV inhibition is a primary antidiabetic mechanism [6], while antioxidant and anti-inflammatory peptides improve insulin secretion/action by reducing oxidative stress and inflammation [7].
> - **Neuroactive peptides:** includes peptides capable of crossing the blood–brain barrier or interacting with neuronal receptors/ion channels. These peptides often share features such as high positive charge and amphipathic motifs that facilitate membrane penetration and correspond to neuroactive functions [8].
> ### **2. Statistical Sequence-Overlap Analysis**
>
> To complement expert knowledge, we performed a **quantitative analysis of sequence overlap** between datasets. The underlying intuition is:
>
> > If two datasets share a large number of identical positive sequences, the corresponding tasks are likely to be highly correlated.
>
> For example, the anticancer and antimicrobial datasets contain:
> - 6,926 (ACP) positive sequences,
> - 30,752 (AMP) positive sequences,
> - with **4,877 overlapping sequences**.
> This substantial overlap strongly indicates relatedness. In our benchmark, we label two tasks as “correlated” if their **positive-sequence overlap exceeds 5%**, a conservative threshold to avoid including redundant or highly similar tasks. Full statistics are shown in **Appendix Figure 103**.
>
>
>
>
> ### **3. Cross-Validation Between Expert Prior and Statistical Evidence**
> As illustrated in Figure 103 (Appendix D.2.2), the overlap heatmap is broadly consistent with expert-derived groups. In most cases, expert assessments and statistical evidence reinforce one another, providing a robust basis for identifying correlated tasks.
>
> ### **Conclusion**
> Expert-derived mechanistic grouping, together with sequence-overlap statistics, provides a principled and mutually corroborating foundation for filtering correlated tasks. This approach substantially reduces the risk of sampling false negatives during negative-sample construction.
> ### References
> [1] Ma Y. Efficient Mining of Anticancer Peptides from Gut Metagenome (Adv. Sci. 25/2023). _Adv Sci (Weinh)_. 2023;10(25):2370172.
>
> [2] Tolos A.M. Anticancer Potential of Antimicrobial Peptides: Focus on Buforins. _Polymers_. 2024, 16, 728.
>
> [3] Peng Qiu. Amplyze: A deep learning model for predicting the hemolytic concentration. _arXiv preprint_ arXiv:2507.08162, 2025.
>
> [4] Raheleh Roudi. Antimicrobial peptides as biologic and immunotherapeutic agents against cancer. _Frontiers in Immunology_. 2017.
>
> [5] Pirtskhalava M. Transmembrane and antimicrobial peptides. _arXiv preprint_ arXiv:1307.6160, 2013.
>
> [6] Tuersun A. Safety and Efficiency of Dipeptidyl Peptidase IV Inhibitors in Patients with Diabetic Kidney Disease. _Current Therapeutic Research_. 2024.
>
> [7] Leo E E M. Biopeptides with antioxidant and anti-inflammatory potential in diabesity. _Biomedicine & Pharmacotherapy_. 2016.
>
> [8] Egleton R D. Development of neuropeptide drugs that cross the blood–brain barrier. _NeuroRx_. 2005.

---

> ### Author Response · Authors · 2025-11-22
> **Response to Question 4**
>
> Thank you for the question. A detailed description of the generative pipeline is presented in **Appendix I (around Line 4374)**. Below, we summarize the model architecture, training data construction, and validation procedures. We refer readers to the appendix for the complete methodology.
>
> ### **1. Model Architecture**
>
> We formulate non-canonical peptide generation as a **conditional sequence-to-sequence translation task**, where a **canonical peptide** serves as the input condition and the model generates a corresponding **non-canonical peptide**.
>
> We use a **GPT-2–based autoregressive transformer** as the backbone due to its strong contextual modeling ability in sequence translation and generation tasks. The model is conditioned using a prompting scheme of the form:
>
> ```
> [NATURAL] <canonical sequence (FASTA Format)> [BILN] <non-canonical sequence (BILN Format)> [EOS]
> ```
> A custom vocabulary was constructed that includes:
> - all 20 canonical amino acids,
> - all BILN symbols used by non-canonical peptides,
> - task-specific tokens such as `[NATURAL]`, `[BILN]`, and `[EOS]`.
> The token embedding layer was expanded accordingly. All sequences were tokenized to a maximum length of 768 tokens (padding shorter sequences and truncating longer ones). Data processing was implemented using the _Hugging Face Datasets_ library.
>
> ### **2. Training Data Construction**
>
> Training data was derived from our experimentally validated **non-canonical peptide datasets**:
> - `nc-antibacterial`
> - `nc-antifungal`
> - `nc-antimicrobial`
> - `nc-hemolytic`
> - `nc-cpp`
> - `nc-ppi_ba`
>
> Each non-canonical peptide was paired with a synthetic “canonical analogue” constructed as follows:
> 1. Remove cyclization and modification annotations from the BILN representation.
> 2. Replace non-canonical monomers with randomly sampled canonical amino acids, keeping length approximately consistent.
>
> The result is a **paired dataset**:
> - **Input:** canonical-like peptide in **FASTA** format
> - **Output:** original non-canonical peptide in **BILN** format
> Thus, the model learns to translate a natural-like sequence into its non-natural BILN counterpart.
> ### **3. Evaluation of Generated Peptides**
>
> We assess the model along four major dimensions: **chemical validity**, **closeness to ground truth**, **diversity/novelty**, and **language-modeling quality**. The main results are summarized below:
>
> |**Categoty**|**Metric**|**Value**|
> |---|---|---|
> |Validity|BLIN→SMILES Success Rate|89.7%|
> ||Chemical Validity Rate|100%|
> ||Overall Success Rate|89.7%|
> |Closeness to Ground Truth|Exact Match Rate|21.9%|
> ||Avg.TanimotoSimilarity|87.9%|
> |Diversity and Novelty|Internal Diversity|66.7%|
> ||Novelty Ratio|76.5%|
> |Language Modeling|Perplexity (PPL)|1232.0|
>
>
> These metrics demonstrate that the model generates peptides that are chemically valid, structurally diverse, and reasonably close to ground truth while still producing novel sequences.
>
>
> ### **4. Inference and Post-processing**
>
> During inference, the model receives prompts of the form:
>
> ```
> [NATURAL] <canonical sequence> [BILN]
> ```
> The GPT-2 decoder then autoregressively generates a non-canonical peptide sequence using greedy decoding. A rule-based post-processing module extracts the sequence between the `[BILN]` and `[EOS]` tokens, removes any remaining special symbols, and verifies the result through:
> - **format validation** (BILN syntax checking),
> - **sequence-to-structure conversion** (ensuring the BILN string can be parsed into a chemically valid molecule).
> Only peptides passing these checks are retained as valid non-canonical outputs.

---

### Official Review · Reviewer_HNKA · 2025-11-01

**Soundness:** 4
**Presentation:** 4
**Contribution:** 3
**Rating:** 8
**Confidence:** 4

**Summary:**

This paper introduces PepBenchmark, a standardized benchmark for peptide machine learning that unifies diverse datasets(PepBenchData), standardized preprocessing(PepBenchPipeline), and unified evaluation protocols(PepBenchLeaderboard) across diverse peptide-related tasks.

**Strengths:**

- The paper presents the largest AI-ready peptide database, integrating 23 datasets across 7 pharmacological tasks, which supports a wide range of predictive applications in peptide therapeutics.
- The biologically-informed, distribution-controlled negative sampling strategy ensures that decoy peptides are generated in a realistic and un-biased manner.
- The benchmark systematically evaluates multiple model families and provide detailed insights into how each type of model performs across different peptide-related tasks.
- The benchmark explicitly addresses k-mer leakage and sequence redundancy

**Weaknesses:**

1. The benchmark is sequence-centric, which can understate GNN baselines that need reliable 3D/contact graphs.
2. More strong PLM embedders should be benchmarked (for example, ESM-2, ProtT5, and MSA Transformer), since they output fixed embeddings for prediction tasks. [1]
3. Add language models that can process non-canonical peptides (for example, GPepT) to cover tasks with modified residues. [2]
4. An ablation of the biologically informed, distribution-controlled negative sampling would show how much this strategy drives the gains.


[1] Zhang, R., et al. “Evaluating the advancements in protein language models for encoding protein sequences.” *Frontiers in Bioengineering and Biotechnology*, 2025.

[2] Oikawa, Y., et al. “GPepT: A Foundation Language Model for Peptidomimetics Incorporating Non-canonical Amino Acids.” *ACS Medicinal Chemistry Letters*, 2025.

**Questions:**

1. How are graphs built for the GNN-based models?
2. How does distribution-controlled sampling change training and test results? Does matching five key properties between positive and negative sets (length, charge, hydrophobicity, molecular weight, isoelectric point) improve early enrichment and calibration?

---

> ### Author Response · Authors · 2025-11-21
> **Response to Weakness 1**
>
> > **Weakness 1** The benchmark is sequence-centric, which can understate GNN baselines that need reliable 3D/contact graphs.
>
> We greatly appreciate this insightful comment. We fully agree that incorporating 3D structural information would be beneficial not only for GNN-based baselines that explicitly use geometric graphs, but also for a broader range of models. However, this work intentionally focuses on sequence-based prediction, as extending the benchmark to a structure-aware setting introduces several substantial challenges.
>
> ## **1. Difficulty in Obtaining Reliable Peptide Structural Data**
>
> 1. **Limited availability of experimental peptide structures.** Most datasets collected in our benchmark do not include experimentally determined structures.
> 2. **Unreliability of predicted peptide structures, especially for non-canonical peptides.**  Although recent works (e.g., [1–3]) have explored using AlphaFold3 and related predictors to generate peptide structures, their accuracy for peptides is still limited. This challenge becomes even more pronounced for **non-canonical peptides**: existing structure predictors provide neither reliable nor generalizable support for arbitrary chemical modifications, and current tools fall well short of producing satisfactory structural predictions in this setting.
>
> 3. **High intrinsic flexibility makes a single conformation insufficient to represent a peptide.**  Peptides often exist as highly dynamic conformational ensembles rather than adopting a single stable structure in solution. As a result, any single static 3D conformation provides only a partial and potentially biased representation of the true structural behavior. Selecting one “representative” structure from the ensemble—or deciding whether models should instead operate directly on the full 4D conformational landscape—remains an open and nontrivial challenge.
>
> ### **2. Challenges in Performing a Structure-Based Dataset Split**
> Once peptide 3D structures or 4D ensembles are obtained, a **structure-based dataset split** would be required to assess a model’s generalization to novel structural states. For proteins, structural splitting is relatively straightforward using established metrics such as TM-score or DALI. In contrast, peptides present additional challenges:
> - their conformational flexibility,
> - the ensemble nature of their structural states, and
> - the difficulty of defining meaningful **structural similarity metrics** for short sequences without stable folds.
>
> These factors make structure-based splitting substantially more complex and far less well-defined in the peptide setting.
> ### **3. Broadening the Benchmark to Multimodal Sequence–Structure Models (not limited to GNNs)**
>
>
> While the reviewer highlights GNN-based baselines, we note that **many model families**—including SMILES-based models and PLM-based sequence models—may also benefit from 3D structural information. Incorporating structure would convert the benchmark into a **multimodal sequence–structure** setting and require:
> - defining structural representations for all peptides,
> - evaluating sequence-based, structure-based, and multimodal baselines across diverse model classes,
> - ensuring fair comparison across heterogeneous architectures.
> This would significantly expand the scope of the current study.
> ### **Conclusion**
>
> We agree that incorporating structural information is an important and valuable direction. However, doing so requires a **fundamentally different benchmark design**, including:
> 1. collecting or generating reliable peptide structural ensembles,
> 2. defining robust structure-based dataset splitting criteria, and
> 3. evaluating a wide range of structure-aware or multimodal models.
> Such an undertaking constitutes a significantly larger effort that extends beyond the scope of our current **sequence-focused** benchmark.
>
> Therefore, in this work, we deliberately focus on **sequence-based evaluation**, and we regard the development of a comprehensive **structure-aware peptide benchmark** as an important direction for future work.
>
> ### Reference
> **[1]** H. Ebrahimikondori et al. _Structure-aware deep learning model for peptide toxicity prediction._ **Protein Science**, 33(7):e5076, 2024.
> **[2]** Z. Sun et al. _Multimodal geometric learning for antimicrobial peptide identification by leveraging AlphaFold2-predicted structures and surface features._ **Briefings in Bioinformatics**, 26(3): bbaf261, 2025.
> **[3]** R. Zhang et al. _PepHarmony: A multi-view contrastive learning framework for integrated sequence and structure-based peptide encoding._ **arXiv**, 2024.

---

> > ### Author Response · Authors · 2025-11-22
> > **Response to Weakness 3**
> >
> > > **Weakness 3** Add language models that can process non-canonical peptides (for example, GPepT) to cover tasks with modified residues.
> >
> > We sincerely appreciate the reviewer’s thoughtful suggestion. Below we clarify how PepBenchmark already supports modeling non-canonical peptides, and why certain models (e.g., GPepT) are not currently included.
> >
> > ### **1. PepBenchmark already includes language models capable of handling non-canonical peptides via SMILES representations**
> >
> > As discussed in **Sections 5.2 and 5.3**, our benchmark evaluates classification, regression, and peptide–protein interaction tasks that involve non-canonical peptides. The reviewer recommends incorporating models that explicitly support modified residues. We would like to clarify that **ChemBERTa** [1], **PeptideCLM** [2], and **PepDoRA** [3], all included in our study, are **SMILES-based language models**. Since SMILES provides an all-atom, chemically faithful representation, it inherently supports non-canonical monomers and a wide variety of chemical modifications.
> >
> > Therefore, our benchmark already incorporates language models that are well-suited for non-canonical peptide representation and predictive modeling.
> > ### **2. Rationale for not including GPepT**
> >
> > We appreciate the reviewer’s mention of **GPepT** [4], which is indeed an interesting contribution to peptidomimetic modeling. However, GPepT is designed primarily as a **generative model** (a GPT-2-based sequence generator), rather than as an embedding model optimized for downstream predictive tasks.
> >
> > Since our benchmark emphasizes **prediction-oriented evaluation**, models that serve primarily as generative frameworks fall outside the current scope.
> > Nonetheless, we recognize the importance of generative peptide models and plan to include GPepT and related methods in **future benchmark extensions focused on peptide generation tasks**.
> >
> > ### **3. Perspectives on modeling non-canonical peptides**
> >
> > Peptide sequences can be represented in several formats:
> > 1. **FASTA** – simple but insufficient for non-canonical residues
> > 2. **SMILES** – atom-level, chemically faithful representation
> > 3. **HELM** – hierarchical polymer representation
> > 4. **BILN** – linearized notation supporting monomer definitions
> >
> > FASTA cannot encode arbitrary modified monomers, making it unsuitable for many non-canonical settings. In contrast, **SMILES is currently the most practical and widely adopted representation for non-canonical and chemically modified peptides**, especially in recent machine learning literature.
> >
> > Alternative representations (e.g., HELM, BILN) have been explored, such as **HELM-GPT** [5], but the field still lacks a general-purpose, community-standard foundation model specialized for non-canonical peptides. Given these considerations, incorporating **strong, SMILES-based language models**—as PepBenchmark already does—currently provides the most robust and representative approach for modeling non-canonical peptides.
> >
> > ### **Conclusion**
> >
> > In summary, PepBenchmark already includes **powerful SMILES-based language models** that inherently support non-canonical peptide representations. While GPepT is a valuable model, its generative design places it outside the predictive scope of our current benchmark. We sincerely appreciate the reviewer’s suggestion and will consider GPepT and other non-canonical–aware generative models in future benchmark expansions focused on peptide design and generation.
> >
> > ### **References**
> >
> > [1] Chithrananda, S., Grand, G., & Ramsundar, B. (2020). _ChemBERTa: Large-Scale Self-Supervised Pretraining for Molecular Property Prediction._ arXiv:2010.09885.
> > [2] Feller, A. L., & Wilke, C. O. (2024). _Peptide-aware chemical language model successfully predicts membrane diffusion of cyclic peptides._ bioRxiv.
> > [3] Wang, L. et al. (2024). _PepDoRA: A Unified Peptide Language Model via Weight-Decomposed Low-Rank Adaptation._
> > [4] Oikawa, Y. et al. (2025). _GPepT: A foundation language model for peptidomimetics incorporating non-canonical amino acids._ ChemRxiv.
> > [5] Xu, X. et al. (2024). _HELM-GPT: de novo macrocyclic peptide design using generative pre-trained transformer._ Bioinformatics, 40(6), btae364.

---

> ### Author Response · Authors · 2025-11-22
> **Response to Weakness 2**
>
> > **Weaknesses 2** More strong PLM embedders should be benchmarked (for example, ESM-2, ProtT5, and MSA Transformer), since they output fixed embeddings for prediction tasks. [1]
>
> Thank you very much for this constructive suggestion. We fully agree that evaluating strong PLM embedders is essential for a comprehensive peptide benchmark. Below, we clarify our model selection criteria, describe our efforts to incorporate even more advanced PLMs, and explain why the current selection is appropriate for the scope of this work.
>
> ### **1. The PLM embedders we selected are already strong and widely adopted**
>
> Our current benchmark already includes several powerful and widely adopted PLM embedders:
> - The **ESM-2 series [1] (2023)** and **DPLM series [2] (2024)** represent the latest generation of _sequence-only_ protein language models and are among the strongest publicly available PLMs.
> - Although **ProtBert (2020)** [3] is older, it remains one of the most commonly used backbones in peptide-related literature. Including it ensures compatibility with prior work and supports fair, broad comparisons.
> Thus, our selected PLM baselines are **both strong and highly representative of current practice** in peptide modeling.
>
> ### **2. We actively track stronger PLMs and have attempted to integrate them**
>
> We fully recognize the value of including even more advanced PLMs. PepBenchmark is designed to be **continuously extensible**, and we plan to update its baselines as stronger models become stable and reproducible. Our early investigations include:
> - **ESM-3 (2025)** [4] : Although promising, recent foundation models such as ESM-3 are **multimodal** (sequence + structure), making them incompatible with our **sequence-only evaluation setup**.
> - **xTrimoPGLM** (2025) [5] : We attempted reproduction and integration of this very recent sequence-based PLM. However, the publicly released checkpoints currently suffer from reproducibility issues (see discussion: [https://huggingface.co/biomap-research/proteinglm-1b-mlm/discussions/1](https://huggingface.co/biomap-research/proteinglm-1b-mlm/discussions/1)), preventing reliable benchmarking despite substantial effort.
>
> We very much appreciate the reviewer’s suggestion and would be grateful for additional recommendations of strong, reproducible, and sequence-only PLM embedders suitable for integration. PepBenchmark is intended as a community resource, and we plan to keep expanding it in future releases.
>
> ### **Conclusion**
>
> In summary, while we agree that continuously incorporating stronger PLM embedders is valuable, the set of PLM baselines included in our benchmark already reflects **recent, powerful, and widely adopted models** that are representative of current peptide modeling practice. Our results—derived from ESM-2, DPLM, ProtBert, and other competitive baselines—are therefore robust and sufficiently strong to support the conclusions of this work. We view the inclusion of additional PLMs as an exciting future extension rather than a limitation of the current benchmark.
>
> ### Reference
> [1] Z. Lin et al. _Evolutionary-scale prediction of atomic-level protein structure with a language model._ **Science**, 379, 1123–1130 (2023).
> [2] X. Wang et al. _Diffusion Language Models Are Versatile Protein Learners._ **arXiv**, 2402.18567 (2024).
> [3] A. Elnaggar et al. _ProtTrans: Toward Understanding the Language of Life Through Self-Supervised Learning._ **TPAMI**, 44(10): 7112–7127 (2022).
> [4] T. Hayes et al. _Simulating 500 million years of evolution with a language model._ **Science**, 387, 850–858 (2025).
> [5] B. Chen et al. _xTrimoPGLM: Unified 100-billion-parameter pretrained transformer for deciphering the language of proteins._ **Nat Methods**, 22, 1028–1039 (2025).

---

> > ### Author Response · Authors · 2025-11-22
> > **Thank You and Responses to Reviewer’s Comments**
> >
> > We sincerely thank you for the careful and constructive evaluation of our work. We have provided detailed, point-by-point responses to all the **weaknesses** and **questions** raised. We hope that our clarifications adequately address your concerns, and **we would be very happy to engage in further discussion should any additional questions arise.**

---

> ### Author Response · Authors · 2025-11-22
> **Response to Weakness 4 — Part I (Distribution-Controlled Sampling)**
>
> > **Weakness 4** An ablation of the biologically informed, distribution-controlled negative sampling would show how much this strategy drives the gains.
>
> We sincerely appreciate the reviewer’s insightful question. Our **Biologically Informed and Distribution-Controlled Negative Sampling (BDNegSamp)** strategy was designed to construct **more realistic and more challenging** peptide activity prediction tasks. Before presenting our quantitative results, we would like to emphasize two key points:
> - **Negative sampling design depends on the goal of dataset construction.**
>     There is no single “correct” way to generate negative samples—the appropriate strategy depends on the intended purpose of the dataset. Our goal is to evaluate models in a way that more faithfully reflects their true predictive ability. For this reason, we introduce distribution control to avoid low-level statistical shortcuts that can make the task artificially easy and inflate performance.
> - **False negatives are inevitable but can be reduced.**
>     Because peptide activity datasets lack experimentally validated negatives, false negatives are an inherent challenge. Prior work has largely overlooked this issue. Our biologically informed filtering specifically aims to reduce the likelihood of selecting peptides that are biologically plausible positives, thereby mitigating (though not eliminating) the false-negative risk.
>
> ### **1. Effect of distribution-controlled sampling**
> Prior work typically samples negatives either (1) randomly or (2) from unrelated activity datasets. These approaches often lead to **distributional biases** in properties such as charge, length, and hydrophobicity. BDNegSamp explicitly matches these distributions, reducing the JS divergence between positive and negative sets to **below 0.2**. By reducing these biases, BDNegSamp produces **more challenging and biologically realistic prediction tasks**, encouraging models to focus on high-level biological signals rather than shortcut features.
>
> To illustrate this effect, we compared BDNegSamp with a random negative-sampling strategy using hybrid splits and Random Forest models:
>
> |           | bbp  | allergen | ace_inhibitory | antidiabetic | antifungal | antioxidant | cpp  |
> | --------- | ---- | -------- | -------------- | ------------ | ---------- | ----------- | ---- |
> | Random    | 75.2 | 89.8     | 85.1           | 78.2         | 91.3       | 73.2        | 86.3 |
> | BDNegSamp | 69.3 | 84.0     | 82.2           | 73.8         | 87.3       | 68.1        | 82.1 |
>
> The consistently higher ROC-AUC under random negatives reflects **inflated performance** due to artificial separability, rather than genuine generalization. BDNegSamp yields **more realistic evaluations**.

---

> ### Author Response · Authors · 2025-11-22
> **Response to Question 1～2**
>
> ## Question 1
> > **Question 1** How are graphs built for the GNN-based models?
>
> Thank you for this question.. For GNN-based models, we construct molecular graphs using the `smiles2graph` function provided in the **Open Graph Benchmark (OGB)** library ([https://ogb.stanford.edu/](https://ogb.stanford.edu/)).
> In this representation:
> - **Nodes** correspond to atoms.
> - **Edges** correspond to chemical bonds.
> Node features include:  atomic number, degree, chirality, aromaticity, and other structural descriptors.
> Edge features include:  bond type, stereochemistry, conjugation, among others.
>
> We follow the implementation provided by OGB, and detailed definitions of node and edge features can be found in the official source code:
> - [https://github.com/snap-stanford/ogb/blob/master/ogb/utils/features.py](https://github.com/snap-stanford/ogb/blob/master/ogb/utils/features.py)
> - [https://github.com/snap-stanford/ogb/blob/master/ogb/utils/mol.py](https://github.com/snap-stanford/ogb/blob/master/ogb/utils/mol.py)
>
> ## Question 2
> >**Question 2**. How does distribution-controlled sampling change training and test results? Does matching five key properties between positive and negative sets (length, charge, hydrophobicity, molecular weight, isoelectric point) improve early enrichment and calibration?
>
> Thank you for this question. As discussed in our response to **Weakness 4**, PepBenchmark includes several distribution-controlled negative sampling strategies, enabling users to match any chosen subset of physicochemical properties. In this work, we apply distribution control on **three core properties**: _length_, _charge_, and _hydrophobicity_.
>
> We chose these three properties for two reasons:
> 1. **Practical considerations:** Matching more dimensions requires a substantially larger negative sampling pool and significantly increases computational cost.
> 2. **Empirical observations:** In practice, we find that matching five properties yields only marginal improvements over matching these three core properties. Thus, controlling length, charge, and hydrophobicity already captures the major distributional biases present in peptide datasets.
> After applying distribution control, models can no longer rely on low-level statistical differences between positive and negative samples.
>
> Consistent with our explanation in "Respond to Weakness 4", we emphasize that the purpose of distribution-controlled sampling is **not** to improve metrics such as early enrichment or calibration. Instead, its goal is to:
> - prevent models from exploiting trivial physicochemical shortcuts,
> - increase task difficulty, and
> - ensure a fair and capability-revealing benchmark.

---

### Meta-Review · Area_Chair_edop · 2025-12-31

**Summary:**

this paper introduces a new benchmark for petitde ML with curated datasets, data pipeline and evaluation. The reviewers agree that the contribution for this standardization is critical.

The major concern is around the scope (not the correctness), although the overall sentiment is still mixed to positive (more on the positive side).

There is no ethical issues reaided by the reviewers.

**Reviewer Concerns:**

Reviewer HNKA (8)'s main concern is adding more PLM embedders and this benchmark may be less fair for structure based baselines like GNNs. The authors explains why structure based benchmarking may be out of scope at the moment and clarifiy for some other points.

Reviewer 26PZ gives 6 in the first place. The reviewer also supplies more related works to consider, and question on the graph baselines/design as well. The authors saud they will provide more clarification and add more related works.

R XKRu's major concern I think is benchmark paper vs method paper and some questions around negative sampling (whether can handle false negatives). The rebuttal is fair here. They clarify the intended track (datasets and benchmarks); they also clearly state that false negatives cannot be eliminated, but can be reduced. Which is ok to my view.

**Reviewer Scores:**

Reviewer HNKA (8): likely stays 8, maybe slightly higher confidence, given the original score is already high.

Reviewer 26PZ (6): likely 6 or 8 given the rebuttal answers it in details.

Reviewer XKRu (4): likely 4 -> 6. I do not think it will become strong accept given false negative is not addressed.

---

### Decision · Program_Chairs · 2026-01-26

Accept (Poster)